# Disturbances of paraventricular thalamic nucleus neurons in bipolar disorder revealed by single-nucleus analysis

Masaki Nishioka [1,2] ✉, Mie Sakashita-Kubota[1,2], Kouichirou Iijima [1], Yukako Hasegawa[1], Mizuho Ishiwata[1], Kaito Takase[1], Ryuya Ichikawa[1], Naguib Mechawar [3], Gustavo Turecki [3] & Tadafumi Kato [1,2] ✉

Bipolar disorder (BD) is a major global health burden, and its treatment challenges highlight the need for pathology-based therapeutic development. Emerging evidence suggests that the thalamus, particularly the paraventricular thalamic nucleus (PVT), is a key region in mood regulation. We performed single-nucleus RNA sequencing on 82 thalamic and cortical samples from 21 patients with BD and 20 controls to compare transcriptional pathology. PVT neurons showed the most striking abnormalities, including the largest number of differentially expressed genes and ~50% fewer cells in BD, whereas cortical alterations were comparatively modest. PVT neurons exhibited marked downregulation of synaptic and ion channel-related genes such as *SHISA9*, *CACNA1C*, and *KCNQ3*, which are linked to BD risk and serve as central nodes in downregulated networks. We also observed disrupted interactions between thalamic excitatory neurons and microglia. Overall, PVT neurons emerge as a central pathological substrate and a promising diagnostic and therapeutic target in BD.

BD is a psychiatric condition that affects approximately 1% of the global population. Patients with BD experience recurrent episodes of mania and depression, often accompanied by suicide attempts. Although existing treatments can be effective for certain patients, potential adverse effects and treatment resistance highlight urgent needs for novel therapeutic approaches. Advancements in treatment will hinge on a deeper understanding of the biological mechanisms underlying BD; however, pathological processes within the brain remain elusive. Previous genetic studies have pointed to dysregulation of synaptic and ion-channel-related pathways in BD[1–4]. Yet, the investigation of detailed mechanisms has been hampered due to a lack of analytical methods to observe cellular and molecular changes in the human brain extensively.

The advancement of single-nucleus RNA sequencing (snRNA-seq) has provided opportunities to comprehensively analyze gene expression in the human brain at single-cell resolution[5–7]. Emerging studies using snRNA-seq have illuminated cellular and molecular pathologies underlying schizophrenia[8–10], autism spectrum disorder[11,12], and major depressive disorder[13,14]. For instance, Ling et al. identified disruptions in the synaptic neuron-astrocyte program in schizophrenia[10]. However, the pathological mechanisms of BD remain underexplored despite the previous bulk RNA-seq studies[15–18]. Furthermore, prior snRNA-seq investigations into psychiatric disorders have predominantly focused on cortical regions, particularly the frontal cortex. Subcortical regions warrant further investigation to elucidate the mechanisms underlying psychiatric disorders.

Comprehensive volumetric studies using MRI have revealed significant volume reductions in subcortical regions, particularly the thalamus and hippocampus, in patients with BD[19], as well as in cortical regions[20]. Notably, lithium treatment has been reported to restore

[1]Department of Psychiatry and Behavioral Science, Juntendo University Graduate School of Medicine, Bunkyo-Ku, Tokyo, Japan. [2]Department of Molecular Pathology of Mood Disorders, Juntendo University Graduate School of Medicine, Bunkyo-Ku, Tokyo, Japan. [3]McGill Group for Suicide Studies, Douglas Institute, Department of Psychiatry, McGill University, Verdun, QC, Canada. ✉e-mail: m.nishioka@juntendo.ac.jp; tadafumi.kato@juntendo.ac.jp

thalamic volume, highlighting the thalamus as a potential core pathological region and therapeutic target for BD[19]. Within the thalamus, the paraventricular thalamic nucleus (PVT) is particularly promising for BD research. PVT serves as a critical hub for emotional processing, with projections to the nucleus accumbens, amygdala, and medial prefrontal cortex (mPFC) in mice[21–23], and has been implicated as a key region associated with mood disorders in the *Polg* transgenic mouse model[24] and neural circuit manipulation models[24,25]. Additionally, the PVT is one of limited regions enriched with serotonin and noradrenaline in mice[26], suggesting its relevance to mood modulation. While Schulmann et al. have reported the physiological cytoarchitecture of the medial thalamus by snRNA-seq and spatial transcriptomics[27], the thalamic cellular and molecular pathologies in BD warrant further investigation.

Here, we examined 82 postmortem thalamus and cortex samples from 21 patients with BD and 20 controls by snRNA-seq to investigate the basic cytoarchitecture of the human PVT and its alterations in BD. We compared the pathological changes in the thalamus to those observed in the frontal cortex (BA10) to determine which brain region is more closely associated with BD pathology (the analytical workflow is depicted in Fig. 1a). By analyzing both brain regions from the same cohort, we minimized confounding factors related to individual background and specimen conditions, thus enabling a clearer comparison between the thalamus and cortex. Among various cell types in these brain regions, the PVT showed a prominent reduction and dysregulated synaptic/neuronal genes in BD, alongside disturbed synaptic neuron-microglia interactions. In PVT neurons, *SHISA9*, *CACNA1C*, and *KCNQ3* exhibited notable changes as putative BD risk genes and/or critical hubs in the downregulated synaptic/neuronal gene networks. In contrast, the cortex exhibits less transcriptomic and compositional changes than the thalamus. Our study offers insights into the pathological mechanisms of BD and highlights the PVT as a potential target for future drug development.

## Results

A total of 383,130 single-nucleus transcriptomic profiles were obtained from 21 patients with BD and 20 controls, encompassing 183,354 thalamic nuclei and 199,776 cortical nuclei before QC (Supplementary Data 1). The thalamic nuclei were specifically isolated from the medial region of the thalamus (Fig. 1a).

### Cytoarchitecture of the medial thalamus

Initially, we focused on the thalamic and cortical snRNA-seq data from 20 controls to delineate the physiological cytoarchitecture of the human medial thalamus. Our QC yielded 32 distinct cell clusters (Fig. 1b). One cluster, designated as NeuCyto, was removed from the subsequent analysis as non-nuclei neuronal cytoplasmic components (see Methods). Accordingly, 147,477 nuclei in 31 cell nuclei clusters underwent subsequent analysis. Excitatory and inhibitory neurons were distinctly segregated between the thalamus and cortex, while astrocytes and oligodendrocyte progenitor cells (OPCs) were continuously separated. This observation aligned with Siletti et al.[6].

The thalamic excitatory neurons (ExN_Tha) predominantly expressed *SLC17A6* (coding VGLUT2), whereas the cortical excitatory neurons (ExN_FrC) primarily expressed *SLC17A7* (coding VGLUT1) (Supplementary Fig. 1a). *RBFOX3*, encoding a well-established neuronal marker known as NeuN, exhibited robust expression in ExN_FrC, yet was minimally expressed in ExN_Tha. Hierarchical clustering of the 31 cell clusters, using all expressed genes, indicated that thalamic excitatory neurons were more closely aligned with cortical inhibitory neurons than with cortical excitatory neurons (Fig. 1c), corroborating the findings by Mathys et al.[28]. Furthermore, mapping through cortex reference[29] confirmed that thalamic excitatory neurons corresponded more closely with cortical inhibitory neurons than with their excitatory counterparts (Supplementary Fig. 1b). These observed similarities

were represented by specific subtype markers of cortical inhibitory neurons, particularly calcium-binding protein genes, including *PVALB*, *CALB1*, and *CALB2*, which encode parvalbumin, calbindin, and calretinin, respectively (Fig. 1d and Supplementary Fig. 1c).

When compared to ExN_FrC, ExN_Tha showed a preferential expression of genes associated with cardiac muscle development, such as *SHOX2*, *PROX1*, *HCN4*, and *CACNA1G*, as determined by gene set enrichment analysis (GSEA) (Fig. 1e, Supplementary Data 2, 3). *CACNA1G*, which encodes a subunit of T-type calcium channels involved in electrical pace-making activity, is specifically expressed in the thalamus and the heart[30], and has been identified as a core gene associated with schizophrenia[31]. Among the ExN_Tha-specific genes, *CASQ2* was highly characteristic to ExN_Tha, with virtually no expression in other cell clusters (Supplementary Fig. 2a). *CASQ2* encodes calsequestrin-2, a high-capacity calcium-binding protein particularly expressed in cardiac muscle cells[32], further underscoring the commonality between thalamic excitatory neurons and cardiac muscle cells.

We also assessed the expression of monoamine receptor genes as primary targets for antipsychotics (Supplementary Fig. 2b). Notably, *DRD2*, *HTR7*, and *HRH3* exhibited pronounced expressions in thalamic excitatory neurons compared to other cell types in the thalamus and cortex. *DRD2* is a key target of antipsychotic drugs and is implicated in schizophrenia, a genetically related disorder to BD[33]. *HTR7* represents the primary target of lurasidone, an antipsychotic used in the treatment of both BD and schizophrenia. The notable expression of *DRD2* in the thalamus, coupled with its minimal expression in the frontal cortex, underscores the thalamus's relative significance in psychotic disorders. Figure 1f illustrates the expression of representative genes characteristic to thalamic excitatory neurons.

Inhibitory neurons were classified into three distinct classes based on their source tissues and ganglionic eminence markers compiled from Schmitz et al.[34]. Thalamic inhibitory neurons expressed lateral ganglionic eminence (LGE) markers including *ZFHX4*, *FOXP2*, *MEIS2*, and *TSHZ1*[34], as well as midbrain-derived inhibitory markers such as *SOX14* and *OTX2*, consistent with the findings from Kim et al.[35] (Supplementary Fig. 1c). These neurons were annotated as InN_Tha without reference to their developmental origin, which, while intriguing, is beyond the scope of the present study. Compared to cortical inhibitory neurons, thalamic inhibitory neurons also demonstrated a preferential expression of genes associated with cardiac muscle, specifically in atrial cardiac muscle tissue development (Supplementary Data 2, 4). This enrichment, analogous to ExN_Tha Gene Ontology (GO) enrichment, was also driven by the expression of *CACNA1G* and *HCN4*.

### Cytoarchitecture of PVT neurons

Among the thalamic excitatory neurons, ExN_Tha_CALB cells in Fig. 1b were putatively supposed to be PVT neurons, as characterized by the expression of *CALB1* and *CALB2*. We used these genes as guidance for locating PVT neuron clusters because they have been employed in previous studies as indicators of PVT neurons in both humans[36] and rodents[37,38]. However, note that *CALB1* and *CALB2* expressions are not exclusive to PVT neurons, and additional heterogeneity likely exists within the human PVT beyond what can be defined by these genes. These putative PVT neurons were also robustly identified in the Human Brain Cell Atlas (HBCA) dataset[6], as demonstrated by the integration of our and HBCA datasets (Fig. 2a). ExN_CALB in Fig. 2a corresponded to ExN_Tha_CALB in Fig. 1b. The putative PVT neurons (ExN_CALB cells) were detected from all 20 our samples and 20 HBCA samples (see Methods). This integrated dataset was used for subsequent analyses to enhance generalizability.

As *CALB1* and *CALB2* expression alone were insufficient to confirm that the ExN_CALB clusters represented PVT neurons, we validated this classification using two approaches: (1) spatial transcriptomic deconvolution and (2) anchor-based label transfer using mouse PVT neuron

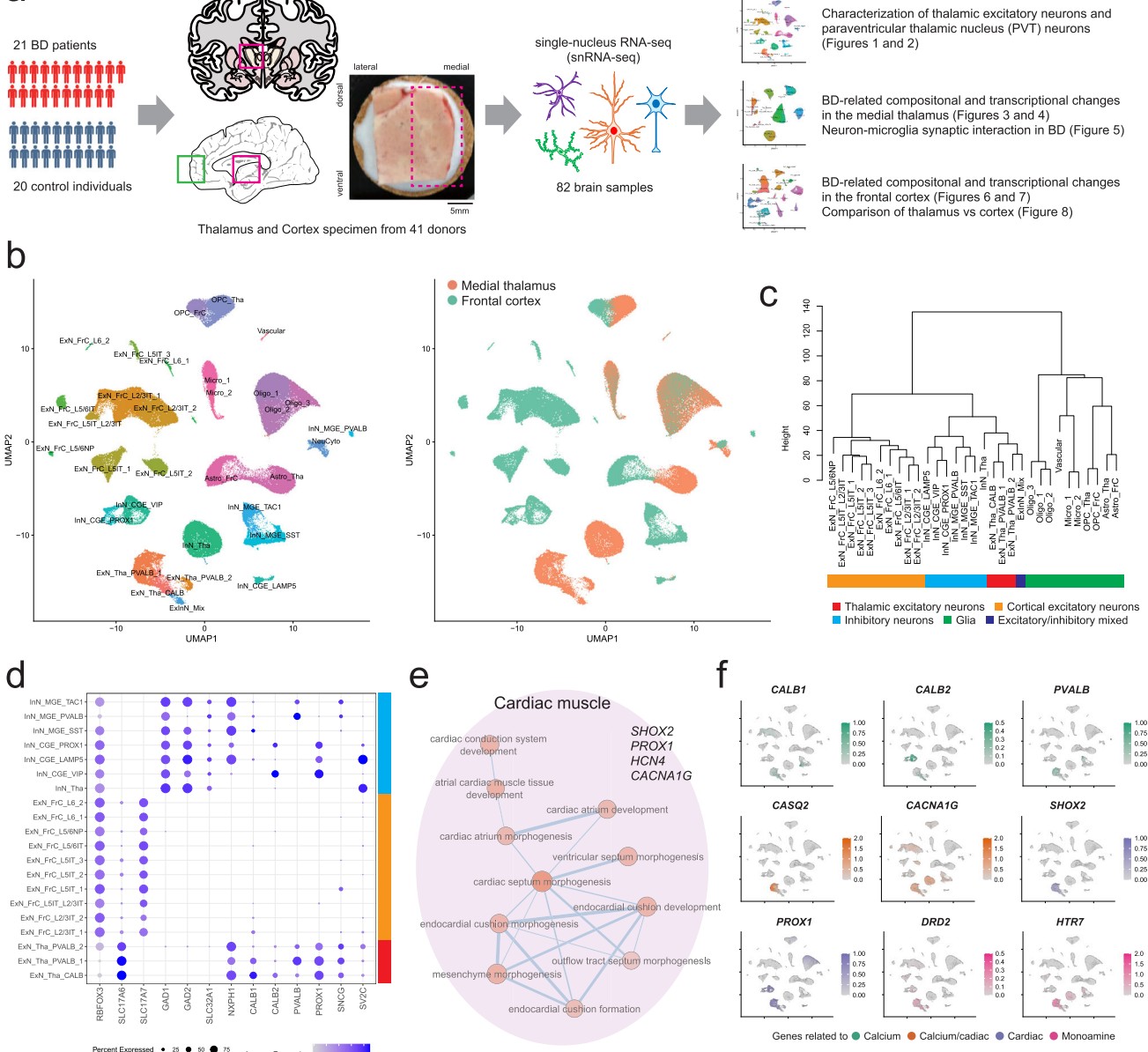

**Fig. 1 | Characterization of thalamic excitatory neurons. a** Schematic representation of the study design. Thalamic (magenta) and cortical (green) specimens from 21 BD cases and 20 controls underwent snRNA-seq to identify BD-associated cellular changes. A dashed line indicates the medial thalamic region analyzed in this study. **b** UMAP projection illustrating clusters identified in the thalamus and cortex (left), with corresponding brain region origins (right). Astro, astrocyte. ExN, excitatory neuron. FrC, frontal cortex. InN, inhibitory neuron. Micro, microglia. NeuCyto, neuronal cytoplasm. Oligo, oligodendrocyte. OPC, oligo precursor cell. Tha, thalamus. **c** Dendrogram representing hierarchical clustering of cell types based on transcriptomic profiles across all expressed genes. **d** Dot plot displaying selected neuronal markers across 20 neuronal cell clusters (see Supplementary Fig. 1 for details). Dot size represents the proportion of cells expressing the marker, and color intensity indicates average expression. Colored bars indicate major cell classes as in panel c. **e** Network visualization of GO terms enriched in the genes preferentially expressed in thalamic excitatory neurons, identified by GSEA. GO terms with enrichment scores >0.8 and networks with >3 nodes are visualized along their core genes. **f** UMAP projection illustrating expression patterns of representative genes characteristic of thalamic excitatory neurons, plotted on the same coordinates as in panel b. Source data are provided as a Source Data file.

reference data. We obtained spatial transcriptomic data of the medial thalamus from one control sample as a representative specimen (Fig. 2b). The PVT was characterized through unbiased clustering of spatial transcriptomic data and partially corresponded to the areas immunostained by calretinin, a protein coded by *CALB2*[36] (Fig. 2c). Deconvolution of thalamic spatial transcriptomic data using the integrated cell clusters by CytoSPACE[39] indicated that ExN_CALB cells were predominantly localized in the paraventricular regions of the thalamus with a clear demarcation from other excitatory neuron populations (Fig. 2d, e), aligning with the anatomical definition of the PVT by Uroz

et al.[36]. Oligodendrocytes were concentrated in the lateral side of the thalamus, while astrocytes were primarily found in the medial side. Oligodendrocyte precursor cells (OPCs) and microglia displayed a diffuse distribution throughout the thalamus. Expectedly, ExN_CALB corresponded most closely with mouse PVT neurons when employing anchor transfer with published single-cell datasets from mice[23,40] (Fig. 2f). These data confirmed the validity of ExN_CALB as PVT neurons. Upon closer examination of PVT neuron subtypes, the majority of human PVT neurons, particularly those from our dataset, corresponded to mouse posterior PVT neurons (Fig. 2g).

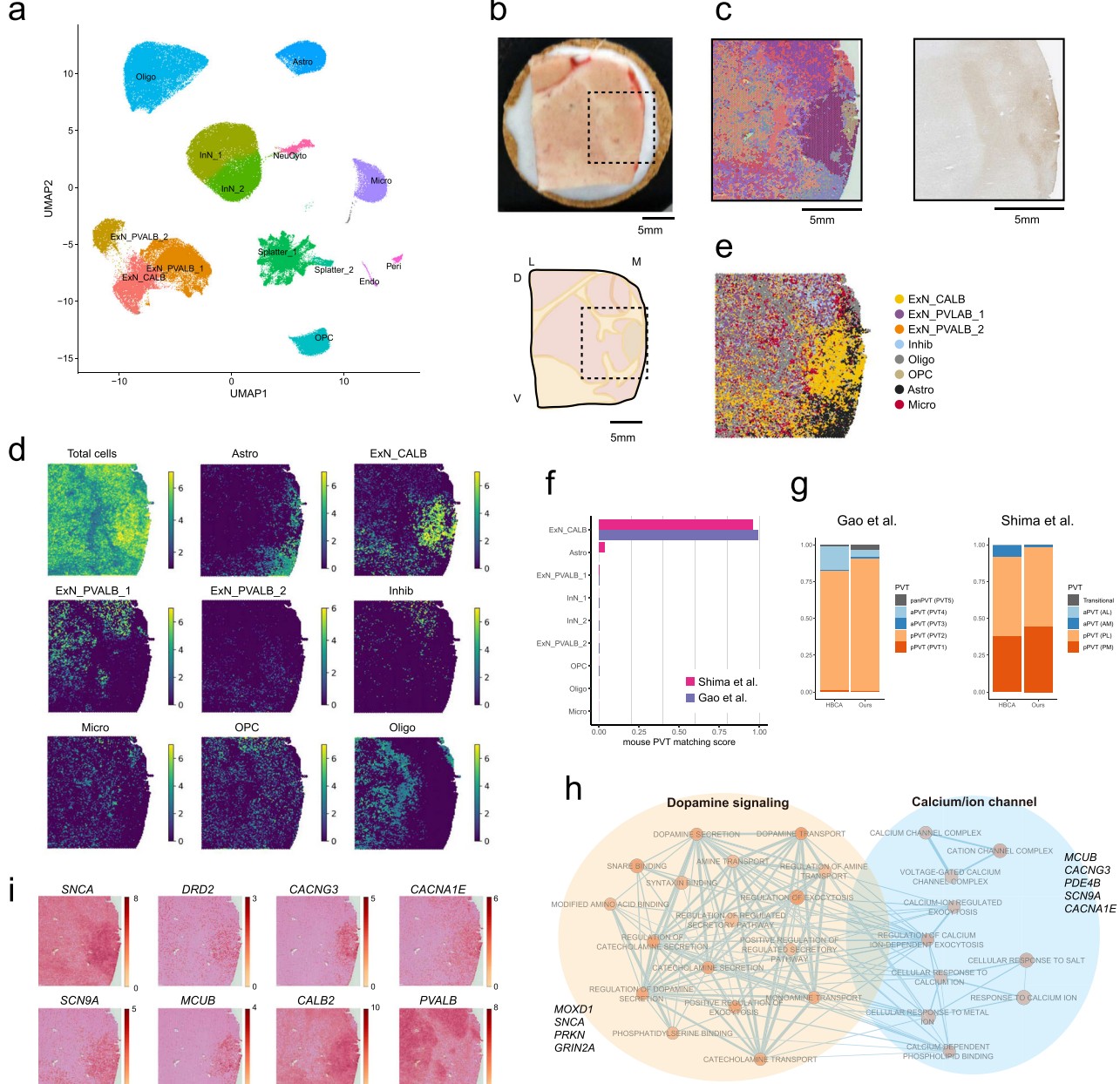

**Fig. 2 | Characterization of human PVT neurons. a** UMAP projection displaying cell clusters identified in the HBCA-integrated thalamic dataset. 88.6% of ExN_Tha_CALB cells in Fig. 1b are included within the ExN_CALB cluster here. **b** Top: Representative paraventricular thalamus with a dashed black rectangle indicating the target region for Visum analysis. Bottom: Illustrated version with topographical orientation. D, dorsal. V, ventral. M, medial. L, lateral. **c** Left: Spatial transcriptomic data delineating 15 distinct clusters, each represented by a different color, illustrating the spatial organization of gene expression across the paraventricular thalamus. Right: Calretinin immunohistochemistry staining of the adjacent section of Visium analysis. **d** Spatial localization of cell clusters as predicted by CytoSPACE deconvolution, illustrating their estimated anatomical distribution within the thalamus. Color intensity in the sidebars indicates the estimated number of cells per spot, providing a quantitative measure of cell density across regions. **e** Merged view of the spatial clusters from panel d for comparison with (**c**). **f** Matching scores of each cell cluster to mouse PVT neurons. **g** Proportions of mouse PVT neuron subtypes from Gao et al. and Shima et al. corresponding to human ExN_CALB clusters. All PVT subtypes refer to mouse PVT neurons; human PVT neurons were not subclassified in this study. aPVT, anterior PVT. The annotated PVT subtypes in pPVT, posterior PVT. AL, anterior lateral. AM, anterior medial.PL, posterior lateral. PM, posterior medial. PVT# in Gao et al. denotes the original subtype classification. **h** GO network visualization for PVT-enriched genes, illustrating major functional pathways related to dopamine metabolism and calcium/ion channels. GO terms with enrichment scores >0.5 and network connectivity >2 are shown. Core genes involved in these pathways are highlighted. **i** Spatial expression profiles of key PVT marker genes, with darker red indicating higher expression. *PVALB* serves as a negative control to demonstrate spatial specificity. Source data are provided as a Source Data file.

Compared to other thalamic excitatory neurons, PVT neurons exhibited a distinctive expression profile for genes (Supplementary Data 5) associated with dopamine signaling (e.g., *MOXD1*, *SNCA*, *PRKN*) and the regulation of calcium and ion channels (e.g., *MCUB*, *CACNG3*, *PDE4B*, *SCN9A*), as determined by GSEA (Fig. 2h, Supplementary Data 2). These PVT neuron markers were particularly expressed in the PVT regions identified through spatial transcriptomics (Fig. 2i). Notably, *DRD2*, highly expressed in the medial thalamus relative to the frontal cortex, was prominent in the PVT compared to other thalamic areas. This observation corroborates the enrichment of dopamine-

signaling-related genes in PVT neurons and aligns with Rieck et al.'s report of high D2 receptor density within paraventricular thalamic regions[41]. The higher expression of *SCNA* in PVT is consistent with Schulmann et al.[27]. Collectively, these findings define PVT neurons as a subset of excitatory neurons uniquely characterized by the expression of calcium-binding protein genes, with a specific emphasis on dopamine signaling and calcium ion regulation.

## Cellular compositional changes in the medial thalamus of BD

We subsequently examined the cellular compositional and transcriptional alterations in the medial thalamus and frontal cortex between 21 patients with BD and 20 controls. The thalamus and frontal cortex were analyzed separately to enhance precision. Cellular compositions refer to the relative proportions of cell clusters within the total population of cell nuclei.

We obtained 146,643 cell nuclei in 18 unique cell clusters and 2139 droplets in two NeuCyto clusters after QC (Fig. 3a and Supplementary Fig. 3a). Two NeuCyto clusters were removed as non-nuclei from the subsequent analysis. Tha_ExN_CALB in Fig. 3a corresponded to PVT neurons. All major cell classes were consistently detected across the 41 samples (Supplementary Fig. 3b). The Tha_ExN_CALB cluster was also robustly identified in all samples among thalamic excitatory neurons with two outliers, BD13 and CT10 (Supplementary Fig. 3c).

Significant compositional reductions were observed exclusively in thalamic excitatory neurons (FDR < 0.05, FC = 0.54–0.59), including PVT neurons (Tha_ExN_CALB, FDR = 0.00488, FC = 0.55), as determined by sccomp[42] (Fig. 3b, Supplementary Data 6). In contrast, five oligodendrocytes and astrocytes clusters showed compositional increases (FDR < 0.05, FC = 1.22–1.55). The reduction of Tha_ExN_CALB remained significant even after excluding the two outlier samples, BD13 and CT10 (FDR = 0.011, FC = 0.592). In examining six major cell classes, we observed a significant compositional reduction in excitatory neurons (FDR = 0.00525, FC = 0.58) and compositional increases in oligodendrocytes and astrocytes in BD (FDR = 0.0168 and 0.004, FC = 1.29 and 1.44, respectively) (Supplementary Fig. 3d). Complementary analysis using scCODA[43] also supported a consistent compositional reduction in thalamic excitatory neurons, including PVT neurons (Supplementary Data 7). In the scCODA analysis, Tha_ExN_CALB and Tha_ExN_PVALB_1 exhibited comparable compositional decreases (FC = 0.686 and 0.663).

Variability in tissue dissection may also influence cell-type composition; however, such variability is difficult to control during sampling and to explicitly model statistically. To address this potential confound, we randomly selected 10 BD and 10 control samples from the full dataset (21 BD vs. 20 controls) and used sccomp to evaluate BD-associated FCs across 1000 random subsamples. Notably, over 97.5% of these subsamples yielded diagnosis-associated FCs < 0.79 for both Tha_ExN_CALB and the broader thalamic excitatory neuron population (Supplementary Fig. 3e), suggesting that the observed compositional reductions are unlikely to be driven by a small number of anatomically atypical outlier samples.

To validate this finding, we compared the cell density of thalamic excitatory neurons and oligodendrocytes by immunohistochemical staining of VGLUT2 and SOX10, calculating cell density from 548,423 excitatory neurons and 1,819,912 oligodendrocytes in 36 thalamic samples (Supplementary Data 8). This immunohistochemical assessment revealed significant reductions of the thalamic excitatory neurons in BD (P = 0.00781, FC = 0.53) and no significant differences in the oligodendrocytes (P = 0.252, FC = 0.91) (Fig. 3c, d). If incorporating PMI and/or pH to the GLM models, the results remained consistent (Supplementary Data 9). Note that immunohistochemistry directly assessed cell density, while compositional analysis using snRNA-seq data assessed relative proportional changes. The immunohistochemical assessment indicated that the compositional increase in oligodendrocytes and astrocytes in BD should be mainly derived from

prominent reductions of excitatory neurons, not reflecting absolute cell density increases. If we set Tha_Oligo_1 as a reference cluster in scCODA analysis, assuming that the absolute counts of oligodendrocytes were comparable between BD and control from IHC results, we consistently observed compositional decreases in thalamic excitatory neurons (Supplementary Data 7).

## Differentially expressed genes in the medial thalamus of BD

Subsequently, we identified differentially expressed genes (DEGs) within each cluster using MAST[44], and categorized these DEGs into "probable" (robustness score >0.5) and "possible" (robustness score >0) DEGs, according to the confidence level of detection. To evaluate the functional implications, we conducted gene ontology (GO) enrichment analysis for the DEGs, defining GOs with robustness scores above 0.5 as "robust DEG-GOs" for each cluster (see Methods).

This analysis revealed 56 downregulated and 69 upregulated probable DEGs in the medial thalamus (Fig. 3e, Supplementary Data 10). Tha_ExN_CALB (PVT neurons) displayed the highest DEG count, with 22 downregulated and 43 upregulated probable DEGs (64 downregulated and 157 upregulated possible DEGs). Tha_ExN_PVALB_1 exhibited the second-largest number of upregulated DEGs, with 81.4% of these genes overlapping with those upregulated in Tha_ExN_CALB. There were 29 downregulated and 67 upregulated robust DEG-GOs in the thalamus, detected only in Tha_ExN_CALB, Tha_ExN_PVALB_1, and Tha_Micro_1 (Fig. 3f, Supplementary Data 11). Tha_ExN_CALB also demonstrated the highest number of DEG-GOs, 24 downregulated and 58 upregulated. The robust DEG-GOs were predominantly enriched in neuronal and synaptic processes, and several DEG-GOs demonstrated overlap between downregulated and upregulated categories in Tha_ExN_CALB, such as the regulation of trans-synaptic signaling (GO:0099177) and modulation of chemical synaptic transmission (GO:0050804). Given the enrichment of synaptic terms in the DEG-GOs, we also performed SynGO enrichment analysis[45]. The downregulated DEGs in the Tha_ExN_CALB cluster again showed the strongest SynGO enrichment, including three terms related to synaptic structure: integral component of postsynaptic membrane (GO:0099055), integral component of presynaptic membrane (GO:0099056), and integral component of postsynaptic density membrane (GO:0099061) (Supplementary Data 11).

Complementary GSEA based on fold changes supported the enrichment of neuronal and synaptic processes among downregulated genes in Tha_ExN_CALB (Supplementary Data 12). GSEA identified potassium channel activity, including *KCNQ3*, *KCND2*, and *KCNC2* as the top GO term for the downregulated genes in Tha_ExN_CALB. The overlap between DEG-GO and GSEA results (GSEA-GO) was more pronounced for downregulated than upregulated genes (9/24 vs. 6/58; two-sided Fisher's exact test, P = 0.00931, odds ratio = 5.07) (Supplementary Data 11). Notably, the top three DEG-GO terms among downregulated genes−transporter complex (GO:1990351), monatomic ion channel complex (GO:0034702), and main axon (GO:0044304)−were also identified by GSEA. The downregulated genes in Tha_ExN_CALB exhibit greater biological coherence and interpretability across enrichment methods. Supplementary Data 13 summarizes the downregulated probable DEGs indicated by both DEG-GO and GSEA as biologically interpretable genes for BD. Among them, *KCNQ3*, *CACNA1C*, *SHISA9*, *KCNC2*, *LRRC7*, *ROBO2*, and *CNTNAP4* were consistently enriched across all three analytical approaches (DEG-GO, GSEA, and SynGO).

Upon closer inspection of probable DEGs, MAGMA-based BD genomic risk[1,46] was most enriched in the downregulated DEGs in Tha_ExN_CALB (P = 0.00123). Additional fine-mapping by PolyFun +SuSiE/FINEMAP[47–49] further supported *CACNA1C* and *SHISA9* as putative BD risk genes within the downregulated set. Especially, *SHISA9* is one of the 36 credible genes for BD in the largest GWAS to date[1]. While *CACNA1C* is not listed in these 36 credible genes, *CACNA1C*

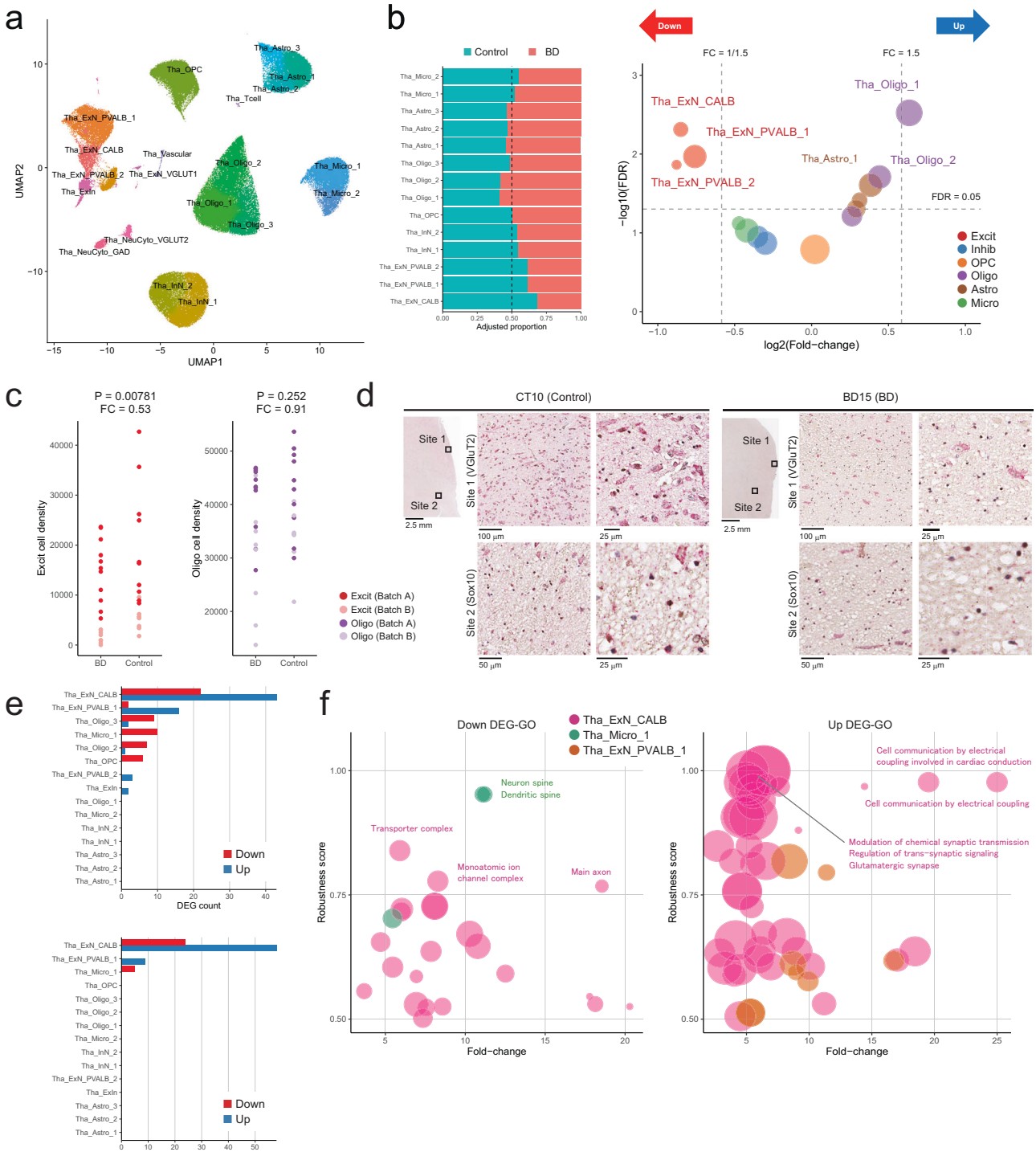

**Fig. 3 | Compositional and transcriptional alterations in the thalamus. a** UMAP projection displaying thalamic clusters. Tha_ExN_CALB population corresponds to PVT neurons, with 97.4% of cells classified within ExN_CALB in Fig. 2a. **b** Left: Proportional distribution of thalamic cell clusters, comparing BD and control. Total nuclei counts are normalized to 10,000, and the total nuclei count in BD and control is adjusted to an equivalent sum. Right: Compositional changes estimated by sccomp with log2(Fold-change) on the x-axis and -log10(FDR) on the y-axis. Circle size reflects the number of nuclei per cluster, and color denotes major cell classes. **c** Cell density (cell count per 100 mm²) of excitatory neurons and oligodendrocytes in BD and control by immunohistochemistry. The unadjusted two-

sided p-values and fold-changes estimated by GLM are indicated. **d** Representative images of immunohistochemistry of CT10 and BT15 using VGLUT2 and SOX10 for excitatory neurons and oligodendrocytes. Immunohistochemical analyses were performed on samples from 19 patients with BD and 17 control individuals. **e** DEG and DEG-GO count across thalamic clusters. **f** Downregulated (left) and upregulated (right) robust DEG-GOs. The x-axis denotes fold-change, and the y-axis reflects the robustness score. Circle size represents the number of DEGs within the GO term, and color indicates the respective cell cluster. Among robust DEGs, only enriched GO terms among possible DEGs are displayed for fold-change calculation. Source data are provided as a Source Data file.

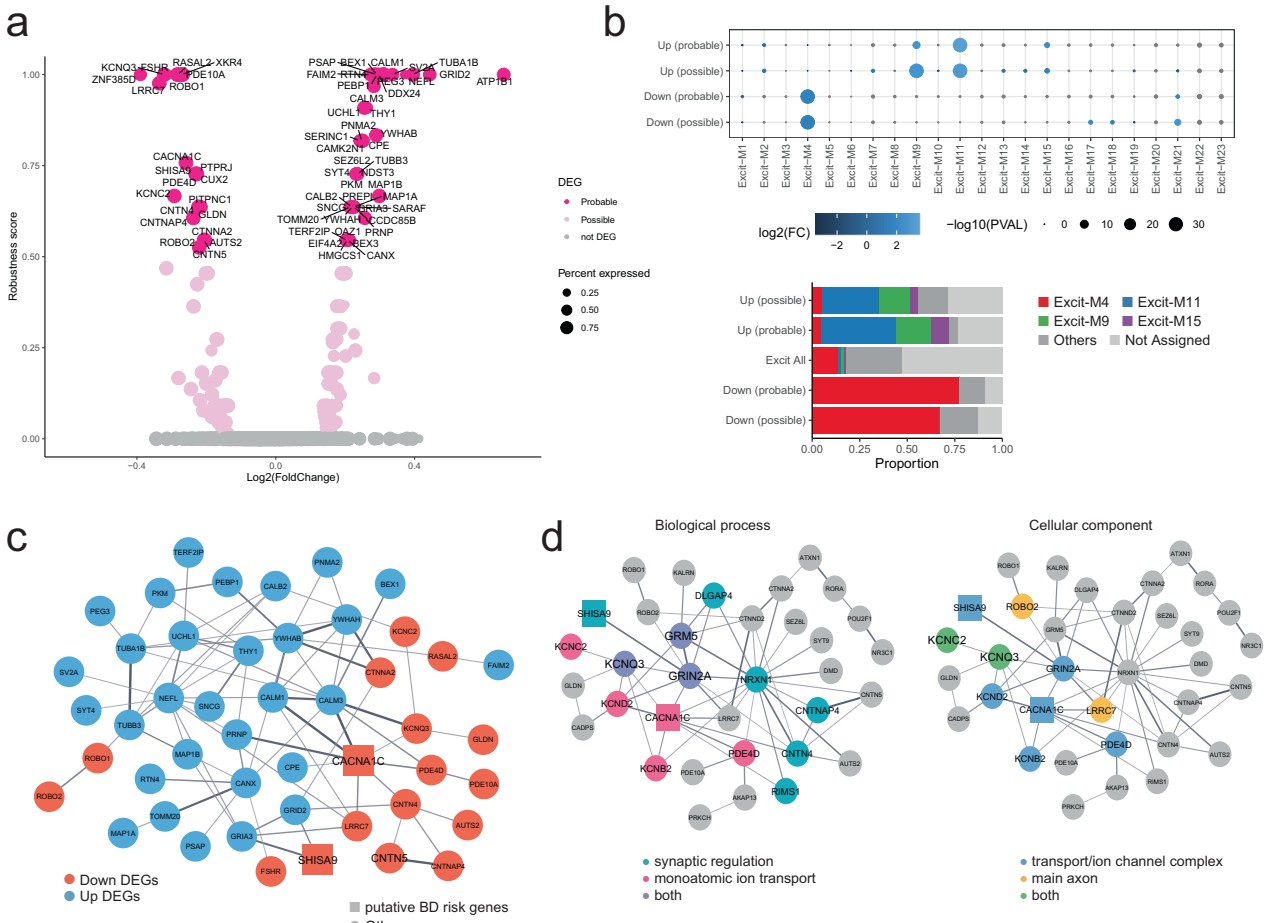

**Fig. 4 | Transcriptional disturbances in PVT neurons. a** Volcano plot of the genes expressed in Tha_ExN_CALB, displaying log2(Fold-change) on the x-axis and DEG robustness score on the y-axis. The probable DEGs were labeled by their gene names. **b** Top: Dot plot illustrating the enrichment of co-expression modules among DEGs in Tha_ExN_CALB (PVT neurons). Dot size indicates −log10 (unadjusted two-sided *P*-value) from hypergeometric tests, and color represents log2(-Fold-change). Bottom: Proportion of co-expression modules among downregulated and upregulated DEGs in Tha_ExN_CALB. FC fold-changes. PVAL *P*-values. **c** PPI networks highlighting downregulated and upregulated probable DEGs in PVT neurons. Node color indicates the direction of gene regulation (Red, down; blue, up). Rectangles indicate putative BD risk genes. The unadjusted one-sided PPI enrichment *P*-value calculated by STRING < $1.0 \times 10^{-16}$. **d** PPI networks highlighting downregulated possible DEGs in PVT neurons. Rectangles represent putative BD risk genes, and colors denote the genes in top DEG-GOs. Font size indicates DEG-GO enrichment. Note that the two PPI networks are identical but have different focuses on the genes. The unadjusted one-sided PPI enrichment *P*-value calculated by STRING < $1.0 \times 10^{-16}$. Source data are provided as a Source Data file.

has consistently ranked among the top genes by P-value in prior PGC1-4 studies[1,2,50,51], and was also implicated by our FUSION-based TWAS integration. In contrast, the upregulated probable DEGs in Tha_ExN_CALB exhibited weaker enrichment for MAGMA-based BD genomic risk (*P* = 0.0916), including no putative BD risk genes despite the greater number of upregulated genes (*n* = 22 for downregulated vs. 43 for upregulated, Supplementary Data 10).

**Transcriptional disturbances of PVT neurons in BD**
The notable enrichment of DEGs and DEG-GOs in Tha_ExN_CALB (PVT neurons) led us to focus on the DEGs in this cluster (Fig. 4a). When assigning the DEGs to co-expression network modules identified through hdWGCNA[52] (Supplementary Data 10), the downregulated DEGs in Tha_ExN_CALB were significantly enriched in Module-4 of excitatory neurons (Excit-M4), while the upregulated DEGs were primarily enriched in Excit-M11 and Excit-M9, with adjusted *p*-values < 0.05 and FCs > 2.5 for both probable and possible DEGs (Fig. 4b and Supplementary Fig. 4a). The downregulated genes exhibited strong aggregation within a single module (>67% in Excit-M4), whereas the

upregulated genes displayed more heterogeneous module enrichment. The clear delineation of modules between downregulated and upregulated genes suggests distinct molecular pathways.

Among the 1540 Excit-M4 genes in Tha_ExN_CALB, the module eigengene-based connectivity measure (defined as kME in the WGCNA package[53]) for downregulated DEGs were significantly higher than those for other Excit-M4 genes (Supplementary Fig. 4b), indicating the DEGs' centrality within the Excit-M4 co-expression network. Similarly, the kME for upregulated DEGs was elevated among the 236 genes in Excit-M11 and the 162 genes in Excit-M9. In addition to higher kME, the downregulated DEGs in Excit-M4 were significantly associated with core genes from the top 10 GSEA-derived GOs for this module (*P* = $1.54 \times 10^{-5}$ and $8.59 \times 10^{-5}$ for probable and possible DEGs, respectively, Supplementary Data 14). The upregulated DEGs in Excit-M11 also showed enrichment in core genes from the top 10 GSEA GOs for that module (*P* = $2.32 \times 10^{-4}$ and $1.43 \times 10^{-7}$ for probable and possible DEGs). We did not observe similar enrichment in Excit-M9. Therefore, the downregulated DEGs in Excit-M4 and the upregulated DEGs in Excit-M11 are likely core genes within their respective modules

that could exert influence on, or be influenced by, a broad array of genes within the same co-expression network. Excit-M4 genes are enriched in GOs related to axonal function and calcium ion regulation, while Excit-M11 genes are enriched in GOs related to calcium ion dynamics and synaptic activity (Supplementary Data 14). This consistency of module-based GO and DEG-GO in PVT neurons further supported the validity of the DEG-GOs in these neurons.

## Molecular insight into PVT neuron pathology

In addition to GO and module enrichment analyses, we examined protein-protein interaction (PPI) networks of DEGs for molecular insights into BD pathology. The downregulated and upregulated probable DEGs within PVT neurons (Tha_ExN_CALB) exhibited significant PPI network enrichment ($P = 3.43 \times 10^{-9}$ and $1.46 \times 10^{-12}$, respectively). When combined, these DEGs formed cohesive PPI networks (PPI enrichment $P < 1.0 \times 10^{-16}$, the lower limit of STRING[54], Fig. 4c). The cohesive PPI networks of downregulated and upregulated DEGs indicated their tight interactions at protein levels, despite the distinctive co-expression modules.

In PVT neurons, the downregulated genes showed consistent GO term enrichment in both GO analysis and GSEA, and the MAGMA-based BD genomic risk was relatively prominent for downregulated genes. Thus, we focused on the downregulated genes. When extending the PPI analysis to possible DEGs, the downregulated possible DEGs in PVT neurons also demonstrated robust PPI networks (enrichment $P < 1.0 \times 10^{-16}$), encompassing genes within the top three DEG-GOs as biological processes (BP) or cellular components (CC) (Fig. 4d). These key genes, including *CACNA1C* and *SHISA9* as putative BD risk genes, were identified as core DEGs based on their PPI and GO enrichment. Particularly, *KCNQ3, GRIN2A*, and *GRM5* intersected with the top three BP GOs—modulation of chemical synaptic transmission (GO:0050804), regulation of trans-synaptic signaling (GO:0099177), and regulation of monoatomic ion transmembrane transport (GO:0034765). *KCNQ3* and *KCNC2* intersected with the top three CC GOs—transporter complexes (GO:1990351), monoatomic ion channel complexes (GO:0034702), and main axon (GO:0044304). These synaptic and ion-channel-related genes as PPI network hubs aligned with the biological pathways indicated by BD genomic studies[1,2,4] and our DEG-GO/GSEA/SynGO results. Particularly, the overlap of *KCNQ3* in both BP and CC top GOs underscores its significant relevance to BD, besides putative BD risk genes.

## Disturbed interaction between PVT neurons and microglia in BD

In addition to Tha_ExN_CALB (PVT neurons), we identified downregulated robust DEG-GOs within Tha_Micro_1 (microglia). Notably, these two clusters were the only ones displaying downregulated robust DEG-GOs (Fig. 3e, f). Furthermore, all five DEG-GOs identified within Tha_Micro_1 were related to synaptic function (Supplementary Data 11). When we assigned the DEGs in Tha_Micro_1 to co-expression network modules, six of ten downregulated probable DEGs (60%) and 32 of the 55 downregulated possible DEGs (58.2%) were assigned to Module-11 of microglia (Micro-M11) ($P = 8.5 \times 10^{-4}$ and $1.20 \times 10^{-14}$, FC = 4.36 and 4.22, hypergeometric test) (Supplementary Data 10). Among the probable DEGs, *SYNDIG1*, *CYFIP1*, and *TIAM1* emerged as promising synaptic DEGs (Fig. 5a), each exhibiting robustness scores exceeding 0.95 and included within the five synaptic DEG-GOs. Additional GSEA identified synaptic enrichment among the downregulated DEGs in Tha_Micro_1 (Supplementary Data 12), again highlighting *SYNDIG1* and *CYFIP1*. SynGO enrichment analysis suggested potential enrichment of GO:1905274 (regulation of modification of the post-synaptic actin cytoskeleton) (Supplementary Data 11).

The simultaneous downregulation of synaptic genes in both microglia and PVT neurons implicates a disturbance in microglia-neuron interactions for synapses in BD. To investigate this possibility, we inferred potential cell-to-cell interactions as trans-cellular meta-

gene programs using non-negative matrix factorization (NMF) by GeneNMF[55]. While each NMF-based meta-gene program was generally confined to one major cell class (Supplementary Fig. 5a), we extracted one meta-gene program encompassing excitatory neurons and microglia (Excit-Micro meta-gene program, Fig. 5b), likely reflecting their physiological interactions. This meta-gene program (MP7), mainly driven by PVT neurons (Tha_ExN_CALB), was the only one showing significantly lower scores in BD (Bonferroni-adjusted $P = 0.0217$ by generalized linear mixed model [GLMM], Fig. 5c). This reinforced the possibility that interactions between excitatory neurons and microglia are disrupted in BD. The Excit-Micro meta-gene program with downregulation in BD was robustly detected in another down-sampling condition (Supplementary Fig. 5b), supporting the robustness of our method. Supplementary Data 13 summarizes the downregulated genes in Tha_ExN_CALB and Tha_Micro_1 that were enriched in one or more GO terms in both DEG-GO and GSEA analyses. Notably, these genes belonged to the same co-expression modules in each cluster.

At the gene level, *SYNDIG1* regulates excitatory synapse number and *SHISA9* encode a regulator of AMPA receptor (AMPAR) complex[56,57], with both genes' robustness scores exceeding 0.7 (Supplementary Data 10). Notably, *SYNDIG1* was prominently expressed in microglia, while *SHISA9* was primarily expressed in excitatory neurons (Fig. 5d). *SHISA9* consisted of core PPI networks of BD-associated downregulated genes in PVT neurons (Fig. 4d) and overlapped in the downregulated genes in Tha_ExN_PVALB_1, as one of the 36 putative BD-causal genes[1]. Beyond microglia and excitatory neurons, thalamic T cells were included in this meta-gene program. Figure 5e illustrates potential ligand-receptor interactions between neurons and microglia/T cells, as analyzed through LIANA[58]. Although we did not detect concurrently downregulated DEGs as ligand–receptor pairs, *KCND2* in Tha_ExN_CALB, *LAMB1* in Tha_ExN_PVALB_2, and *KCNQ1* in Tha_Micro_1 were identified as downregulated DEGs among the LIANA-based ligand-receptor pairs. These genes encode ion channels or extracellular matrix proteins that may potentially contribute to Excit–Micro interactions. However, as this analysis is an inference, the precise ligand–receptor mechanisms remain to be elucidated in future studies.

## Cellular compositional changes in the frontal cortex of BD

We performed a detailed analysis of frontal cortex samples to characterize compositional and transcriptional alterations associated with BD relative to those observed in the thalamic samples. We obtained 165,585 cell nuclei in 22 unique cell clusters and 2369 droplets in one NeuCyto cluster from the same cohort of 21 patients with BD and 20 controls (Fig. 6a and Supplementary Fig. 6a, b). NeuCyto cluster was removed from the subsequent analysis.

We observed a compositional decrease in FrC_ExN_L6ITCar3 (FDR = 0.00463, FC = 0.65) and compositional increases in FrC_Astro (FDR = 0.0416, FC = 1.36) and FrC_Oligo (FDR = 0.00450, FC = 1.41) by sccomp[42] (Fig. 6b, Supplementary Data 6). Within the major cell class analysis, we observed compositional reductions in excitatory and inhibitory neurons (FDR = 0.0283 and 0.0218, FC = 0.848 and 0.863, respectively) and compositional increases in oligodendrocytes (FDR = 0.0488, FC = 1.39) (Supplementary Fig. 6c). The scCODA analysis was not consistent to the sccomp analysis, exhibiting the compositional decreases of FrC_ExN_L2/3IT_1 (FC = 0.797), FrC_ExN_L5IT_4 (FC = 0.866), and FrC_InN_CGE_LAMP5 (FC = 0.848). In contrast to findings in the thalamic excitatory neurons, the cortical neurons showed inconsistent results and smaller reductions (Fig. 6b vs. Fig. 3b). Note that the absolute cell count in situ would be different from these compositional (proportional) changes. Given the immunohistochemical assessment for the thalamus, compositional increases in oligodendrocytes might be derived from more prominent reductions of neurons, not reflecting the absolute increases of oligodendrocytes.

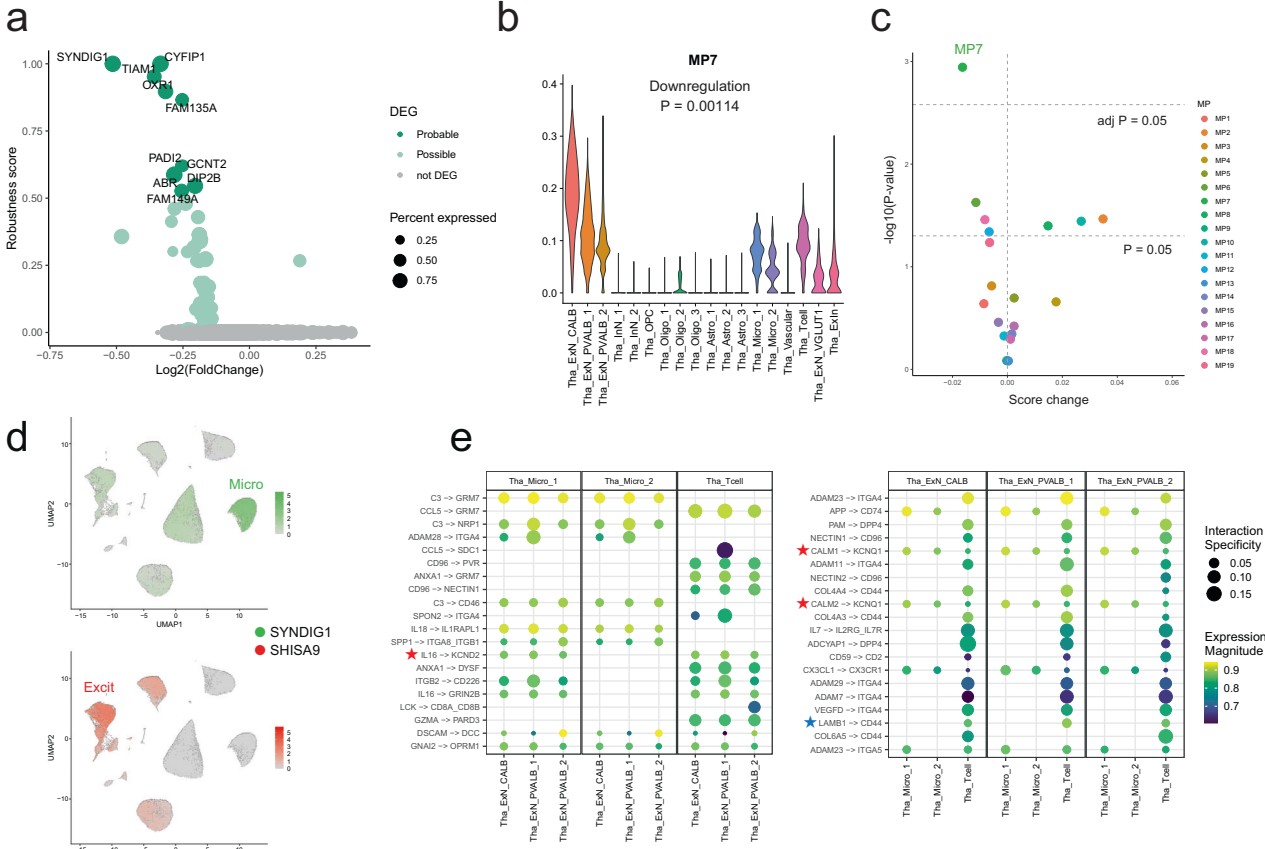

**Fig. 5 | Disturbed interaction between microglia and PVT neurons in the thalamus. a** Volcano plot of the genes expressed in Tha_Micro_1, displaying log2(Foldchange) on the x-axis and DEG robustness score on the y-axis. The probable DEGs were labeled by their gene names. **b** NMF-derived meta-gene program scores for MP7 across thalamic cell clusters. Complete descriptions of the meta-gene programs are provided in Supplementary Fig. 5a. The unadjusted two-sided P-values for downregulation assessed by GLMM are indicated. **c** Score changes estimated by GLMM for 19 meta-gene programs identified by NMF, with −log10(unadjusted two-sided $P$-value) on the y-axis. Colors represent different MPs. The dashed line with

"adj P" indicates the Bonferroni-corrected significance threshold for 19 MPs. **d** UMAP projection illustrating the expression patterns of *SYNDIG1* and *SHISA9*. **e** Predicted ligand-receptor interactions between excitatory neurons and microglia/T cells. Dot plots represent expression magnitudes and interaction specificities, with interactions labeled as [ligand gene] -> [receptor gene]. Blue or red stars indicate ligand–receptor pairs in which the ligand or receptor is a DEG. Left: ligands expressed in microglia or T cells targeting receptors in thalamic excitatory neurons. Right: ligands expressed in thalamic excitatory neurons targeting receptors in microglia or T cells. Source data are provided as a Source Data file.

## Differentially expressed genes in the frontal cortex of BD

Our DEG analysis revealed 55 downregulated and 73 upregulated probable DEGs in the frontal cortex (Fig. 6c, Supplementary Data 10). FrC_ExN_L5IT_4 and FrC_ExN_L6Car3 exhibited the highest numbers of probable DEGs (Supplementary Data 10) and 11 robust DEG-GOs (Supplementary Data 11). While the DEG-GOs primarily pertained to synaptic and neuronal functions, their number in the frontal cortex was lower than that observed in the thalamus (Fig. 6d vs. Fig. 3d). Among cortical DEG-GO, only GO:0045211 (postsynaptic membrane), an upregulated GO term in FrC_ExN_L6ITCar3, was identified by FC-based GSEA, and no SynGO enrichment was observed. These findings suggest that cortical DEGs exhibit lower biological interpretability compared to thalamic DEGs.

The enrichment of DEGs in FrC_ExN_L5IT_4 and FrC_ExN_L6Car3 prompted us to focus on the DEGs in these clusters (Fig. 7a). When assigning the DEGs to co-expression network modules identified through hdWGCNA, we observed specific module enrichments in FrC_ExN_L5IT_4 and FrC_ExN_L6Car3 (adjusted $P < 0.05$ and FC > 2.5 for both probable and possible DEGs). However, the enrichment for FrC_ExN_L6Car3 was less pronounced (Supplementary Fig. 7a, b). In FrC_ExN_L5IT_4, the upregulated DEGs were primarily enriched in modules ExcitDL-M3 and ExcitDL-M19 (ExcitDL: deep-layer excitatory neuron, Fig. 7b). The module scores of DEGs within FrC_ExN_L5IT_4

exceeded those of other genes in their respective modules, indicating more central roles of these DEGs within their co-expression networks (Supplementary Fig. 7c, d). In contrast to the findings in the thalamus, FrC_ExN_L5IT_4 and FrC_ExN_L6Car3 exhibited weaker protein-protein interaction (PPI) network enrichment, lacking a clear distinction in interactions between downregulated and upregulated DEGs (Fig. 7c vs. Fig 4c). Among the probable DEGs in these two clusters, only upregulated DEGs in FrC_ExN_L6Car3 were nominally enriched in MAGMA-based BD genomic risk ($P = 0.00454$), including *MACROD2* as a putative BD risk gene.

## Comparison of the thalamus and cortex

As described above, the medial thalamus exhibited more pronounced compositional and transcriptional alterations than the frontal cortex. Specifically, PVT neurons showed the most significant compositional and transcriptional shifts among the various cell types across both regions. Furthermore, downregulated genes in PVT neurons showed the most prominent enrichment of MAGMA-based BD genomic risk among all the probable DEG sets in the thalamus and cortex analysis (Fig. 8a) as the only one category surviving Bonferroni correction (adjusted $P = 0.0295$, $n$ of tests = 14). This enrichment pattern extended beyond BD, as downregulated DEGs in PVT neurons also showed the highest enrichment for MAGMA-based genomic risk associated

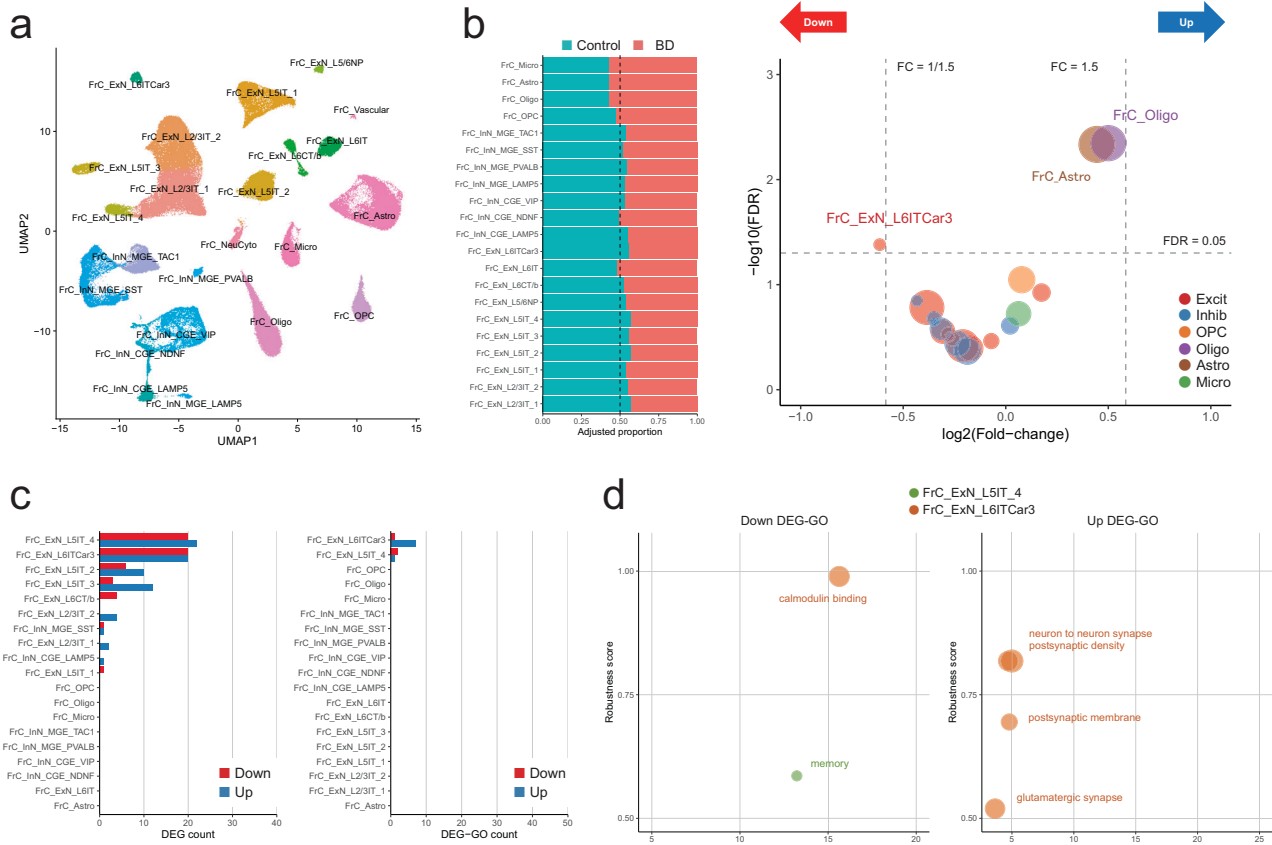

**Fig. 6 | Compositional and transcriptional alterations in the cortex. a** UMAP projection displaying cortical clusters. **b** Left: Proportional distribution of cortical cell clusters, comparing BD and control. Total nuclei counts are normalized to 10,000, and the total nuclei count in BD and control is adjusted to an equivalent sum. Right: Compositional changes estimated by sccomp with log2(Fold-change) on the x-axis and −log10(FDR) on the y-axis. Circle size reflects the number of nuclei per cluster, and color denotes major cell classes. **c** DEG and DEG-GO count across cortical clusters. **d** Downregulated (left) and upregulated (right) robust DEG-GOs. The x-axis denotes fold-change, and the y-axis reflects the robustness score. Circle size represents the number of DEGs within the GO term, and color indicates the respective cell cluster. Among robust DEGs, only enriched GO terms among possible DEGs are displayed for fold-change calculation. Source data are provided as a Source Data file.

with schizophrenia[33] and major depressive disorder[59], while showing weaker or nonsignificant enrichment for other psychiatric conditions: attention-deficit/hyperactivity disorder, post-traumatic stress disorder, obsessive-compulsive disorder, substance use disorder, eating disorder, panic disorder, and autism spectrum disorder (Fig. 8a, details in Supplementary Data 15). These enrichment comparisons should be interpreted with caution, as differences in statistical power across the original GWAS datasets may influence the observed patterns.

We examined which cell types most expressed putative BD risk genes. Thalamic excitatory neurons, including PVT neurons, showed the highest expression of the putative BD risk genes concerning the number of expressed genes and their average expression rate (Fig. 8b). For example, *CACNA1C* and *SHISA9* exhibited relatively higher expression in thalamic excitatory neurons than in cortical excitatory neurons (Fig. 8c). However, the differences were modest, and both genes showed the highest expression in Tha_ExN_PVALB_1, suggesting that additional factors likely contribute to the transcriptional alterations observed in PVT neurons, as discussed later.

We examined the expression patterns of BD-associated genes identified through rare variant analyses, specifically *HECTD2* and *AKAP11*[3,60]. Neither gene was differentially expressed in any cell type clusters in the thalamus or cortex at an FDR threshold of <0.1. The expression levels of *HECTD2* and *AKAP11* were not notably elevated in thalamic excitatory neurons, and their spatial expression patterns showed no specificity to the PVT region (Supplementary Fig. 8a, b). These findings may reflect distinct mechanisms underlying BD-

associated genes identified through rare variants compared to those implicated by common variant analyses.

In parallel with region-specific variations, significant parallels were identified between the thalamus and cortex. First, a recurring trend of compositional neuronal reduction coupled with compositional increases in oligodendrocytes and astrocytes were observed (Figs. 3b, 6b). These findings suggest potential brain-wide neuronal damage in BD. Second, probable DEGs identified in PVT neurons (Tha_ExN_CALB) overlapped with those in deep-layer cortical excitatory neurons (FrC_ExN_L5IT_4 and FrC_ExN_L6Car3), exhibiting similar patterns of upregulation and downregulation (Fig. 8d). Notably, *KCNQ3*, *SLC8A3*, and *ATXN1* emerged among the commonly downregulated genes. They were highlighted as key candidates in the DEG-GO analyses of both PVT neurons and cortical deep-layer excitatory neurons, underscoring their potential as markers of widespread pathology. *KCNQ3* encodes a subunit of voltage-gated potassium channels integral to neuronal excitability regulation. *KCNQ3* was a core DEG in PVT neurons, and its broad downregulation extended to the cortex was consistent with Kaminsky et al.'s report of *KCNQ3* downregulation in bulk brain tissues from patients with BD[61]. *SLC8A3*, a transmembrane sodium/calcium exchanger, is essential for maintaining $Ca^{2+}$ homeostasis in both neuronal and cardiac cells, while *ATXN1*, a DNA-binding protein, is associated with spinocerebellar ataxia, a neurodegenerative disease. Although these genes were downregulated across the thalamus and cortex, their functional impact on PVT neurons might be more pronounced, given the unique

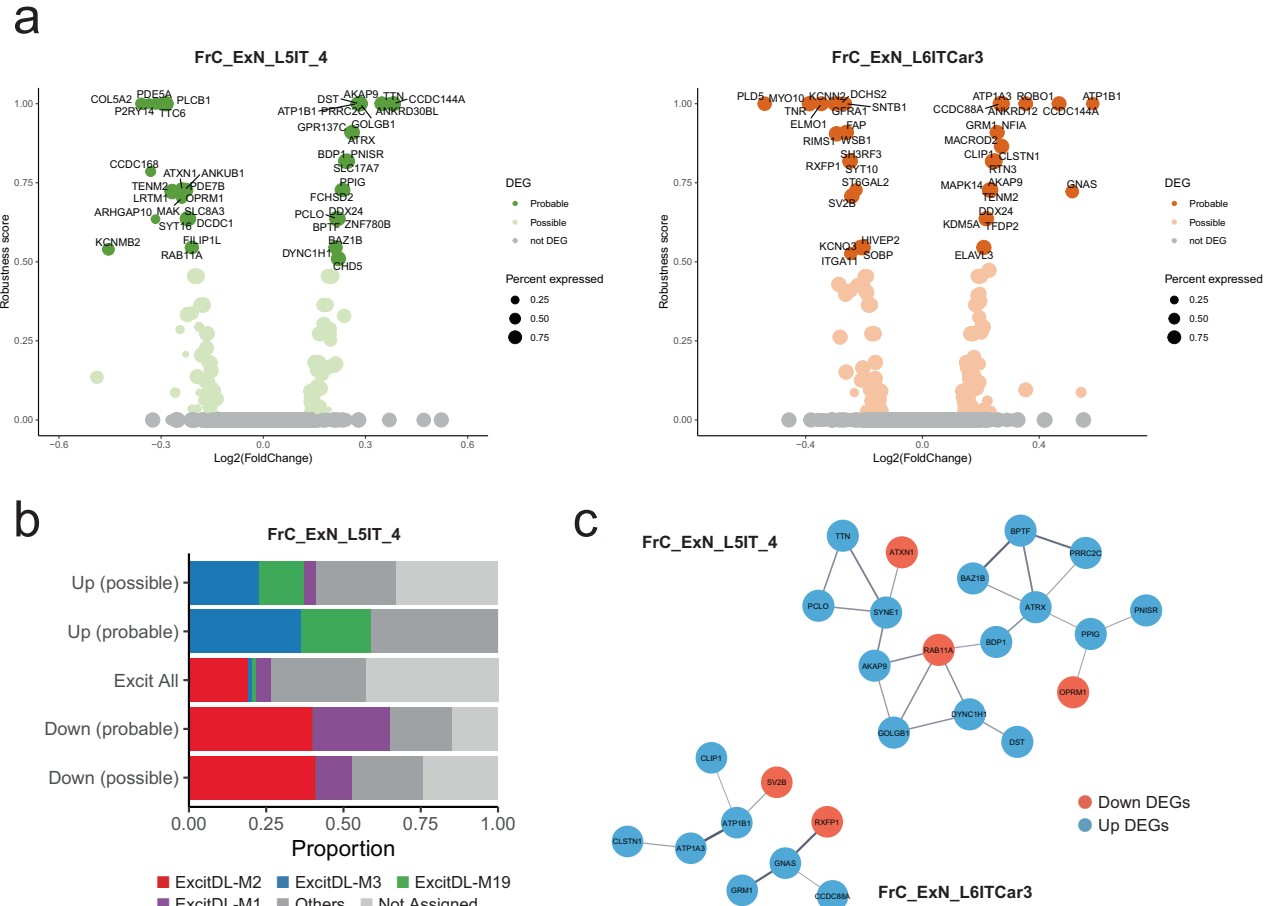

**Fig. 7 | Transcriptional disturbances in deep-layer cortical excitatory neurons.** **a** Volcano plots of the genes expressed in FrC_ExN_L5IT_4 and FrC_ExN_L6ITCar3, displaying log2(Fold-change) on the x-axis and DEG robustness score on the y-axis. The probable DEGs were labeled by their gene names. **b** Proportion of co-expression modules among downregulated and upregulated DEGs in FrC_ExN_-L5IT_4. **c** PPI networks highlighting downregulated and upregulated probable DEGs in FrC_ExN_L5IT_4 (unadjusted one-sided PPI enrichment $P$-value calculated by STRING = 7.66 × 10⁻⁶) and FrC_ExN_L6ITCar3 (unadjusted one-sided PPI enrichment $P$-value calculated by STRING = 0.00522). Node color indicates the direction of gene regulation (Red, down; blue, up). Source data are provided as a Source Data file.

characteristics of PVT neurons identified in previous analyses (see Fig. 2f).

Third, Excit-Micro meta-gene programs were also identified in the cortex (Fig. 8e). Besides, *SYNDIG1*, which exhibited the largest downregulation in the thalamic microglia, also exhibited the most notable downregulation in cortical microglia (Fig. 8f). Despite a low robustness score and a lack of significant downregulation of cortical Excit-Micro meta-gene programs in BD (Fig. 8e), the observed downregulation of *SYNDIG1* in cortical microglia mirrors findings in thalamic microglia, suggesting possible broad disturbances of Excit-Micro meta-gene programs in BD. This hypothesis warrants further validation with more cell nuclei counts and additional samples. We applied GeneNMF on the integrated dataset encompassing both thalamic and cortical regions to examine trans-regional gene pro-grams. While this analysis did not reveal any discernible trans-regional meta-gene programs, the thalamic and cortical Excit-Micro meta-gene programs were still identified in the thalamus-cortex integrated dataset, further validating the robustness of our approach (Supplementary Fig. 8c).

Although we did not identify trans-regional cortico-thalamic meta-gene programs, we extended the LIANA analysis to explore potential interactions between the cortex and thalamus, focusing specifically on connections between cortical deep-layer excitatory neurons (ExN_FrC_DL) and thalamic excitatory neurons (ExN_Tha)

(Fig. 1b). Among the top 20 ligand-receptor pairs in the cortex-to-thalamus direction, eight involved DEGs in either ExN_Tha or ExN_FrC_DL (Supplementary Fig. 9), and four DEG-containing pairs were detected in the thalamus-to-cortex direction. These results sug-gest potential disruptions in cortico-thalamic connectivity mediated by altered ligand-receptor interactions. However, this analysis is exploratory and hypothesis-generating; further studies are required to more definitively characterize these connections.

**Medication signature**

Multiple medications and substances were detected in most post-mortem brain samples at the time of death (Supplementary Fig. 10a). As subgrouping based on substances at death or medication history was not feasible in our dataset, we assessed the potential overlap between medication signatures (previously reported DEGs induced by specific medications/substances) and the DEGs identified in our study. In the four neuronal clusters with the largest numbers of DEGs, there was generally no substantial overlap with medication signatures for alcohol, antipsychotics, antidepressants, lithium, or valproic acid, except in a few instances (Fig. 8g, Supplementary Data 16). The upre-gulated genes in Tha_ExN_CALB significantly overlapped with genes upregulated by olanzapine in medium spiny neurons (MSNs)[62] (Bon-ferroni-adjusted $P$ = 0.00355, $n$ of tests = 18 per cell type). A similar, though weaker, overlap was observed in Tha_ExN_PVALB_1 (adjusted

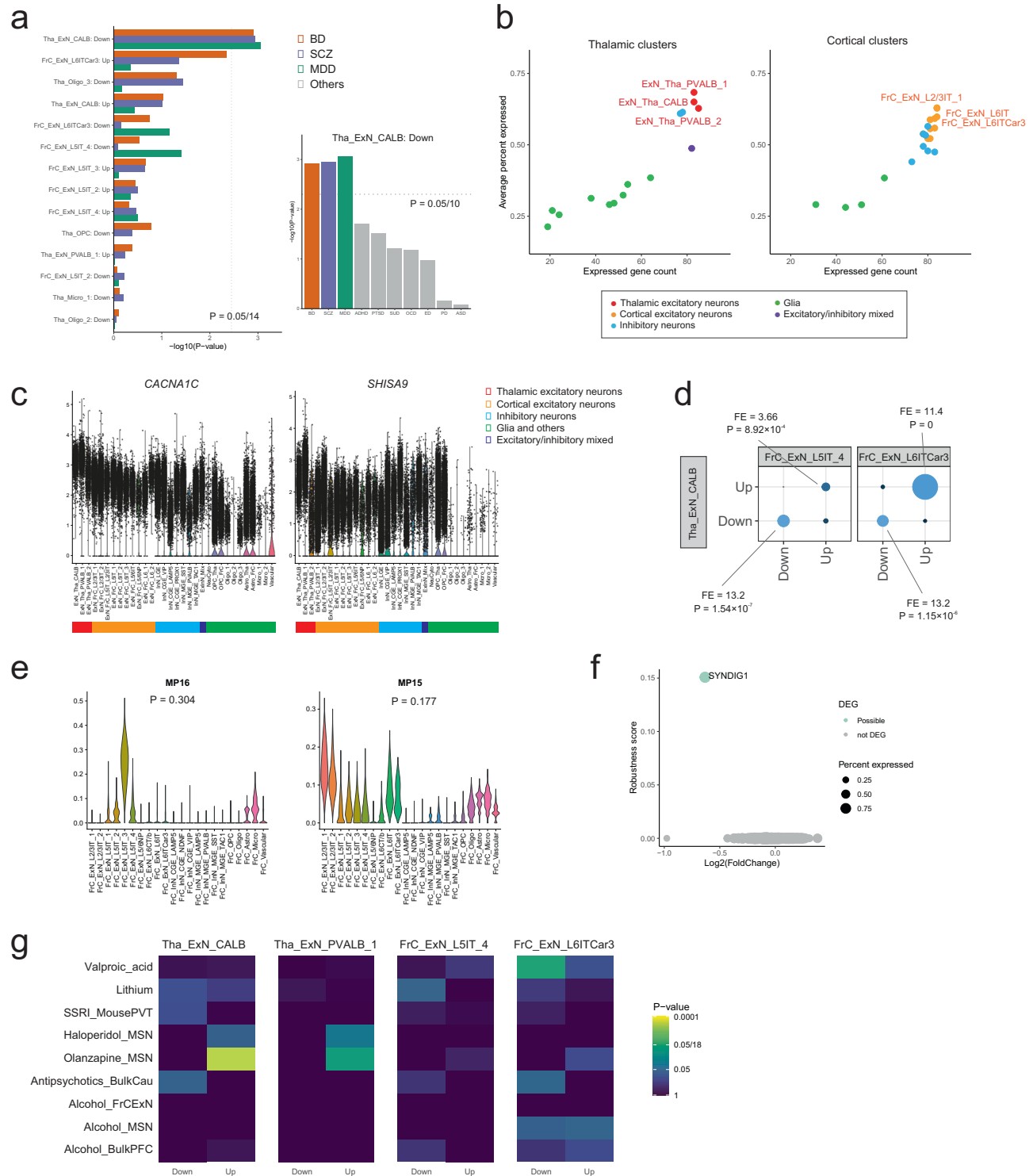

$P = 0.0772$). Another finding was the overlap between downregulated genes in FrC_ExN_L6ITCar3 and genes downregulated by valproic acid in forebrain organoids[63] (adjusted $P = 0.0573$). Although MSNs represent a distinct neuronal subtype from thalamic excitatory neurons, the consistent overlap between Tha_ExN_CALB/Tha_ExN_PVALB_1 and olanzapine-treated MSNs for upregulated genes suggests that these transcriptional changes could be influenced by medication exposure. However, no notable overlaps were observed for other combinations. Importantly, the downregulated genes in Tha_ExN_CALB—which were a central focus of our results—were unlikely to be attributable to medication or substance effects in this analysis.

## Single-molecule fluorescence in situ hybridization for *SYNDIG1* in microglia

The unexpected detection of *SYNDIG1* in thalamic and cortical microglia prompted further validation of its expression and spatial distribution using RNAscope. *SYNDIG1* transcripts were detected in *CX3CR1*-positive cells, consistent with a microglial identity (Fig. 9a). *CX3CR1/SYNDIG1* double-positive cells—representing *SYNDIG1*-expressing microglia—were preferentially localized to the gray matter of the frontal cortex (Fig. 9b). In the thalamus, these cells appeared more abundant in medial regions, although this pattern was not definitive. Comparison of representative thalamic male samples from five BD

**Fig. 8 | Comparative analysis between the thalamus and cortex. a** Left: MAGMA-based genomic risk enrichment of DEGs for BD, schizophrenia (SCZ), and major depressive disorder (MDD). Right: MAGMA-based genomic risk enrichment of downregulated DEGs in PVT neurons for BD, SCZ, MDD, attention-deficit/hyperactivity disorder (ADHD), post-traumatic stress disorder (PTSD), substance use disorder (SUD), obsessive-compulsive disorder (OCD), eating disorder (ED), panic disorder (PD), and autism spectrum disorder (ASD). Unadjusted two-sided *P*-values from MAGMA are plotted on the x-axis (left panel) and y-axis (right panel). Dotted lines indicate Bonferroni-corrected significance thresholds for each analysis. **b** Expression of putative BD risk genes across cell nuclei clusters in the thalamus and cortex set (see Figs. 3a, 6a). The x-axis represents the count of expressed putative BD risk genes, and the y-axis denotes their average expression rates (percent expressed) among the respective clusters. The genes are limited to protein-coding genes with expression rates >0.1 within each cluster. **c** Relative expression of *CACNA1C* and *SHISA9* across all clusters in the thalamus-cortex integration dataset. **d** Overlap of DEGs in Tha_ExN_CALB (PVT neurons) and

FrC_ExN_L5IT_4/FrC_ExN_L6ITCar3 (Deep-layer cortical excitatory neurons). Color intensity corresponds to the fold-enrichment (FE), and dot size is proportional to -log10(unadjusted two-sided *P*-value) calculated by hypergeometric tests. **e** Trans-cellular Excit-Micro meta-gene programs in the frontal cortex. Unadjusted two-sided *P*-values for score changes in BD by GLMM are described. **f** Volcano plot of the genes expressed in FrC_Micro_1, displaying log2 (Fold-change) on the x-axis and DEG robustness score on the y-axis. No probable DEGs (robustness score >0.5) were detected. **g** Heatmap of unadjusted two-sided *P*-values from hypergeometric tests for the overlap between DEGs in each cell type and previously reported DEGs induced by specific medications/substances (medication signatures). Color indicates -log10(unadjusted two-sided P-value), with Bonferroni-corrected significance threshold for each panel (0.05/18) shown in the color bar. SSRI selective serotonin reuptake inhibitor, MSN medium spiny neuron, Bulk bulk RNA-seq data, Cau caudate, FrCExN frontal cortex excitatory neuron, PFC prefrontal cortex. Source data are provided as a Source Data file.

cases and five controls showed relatively lower *SYNDIG1* expression in BD microglia (raw FC = 0.801, Fig. 9c). While generalized linear mixed model (GLMM) analysis with a negative binomial distribution did not reach statistical significance, it indicated a trend toward reduction (GLMM *P* = 0.153, estimated FC = 0.783).

## Discussion

In this study, we investigated cellular and molecular pathologies associated with BD using snRNA-seq data from 41 individuals. By comparing thalamic and cortical samples from the same cohort, we aimed to elucidate regional differences while minimizing confounding variables such as age, sex, and sample conditions—challenges that often complicate human studies. Among various cell types in the thalamus and cortex, PVT neurons exhibited the most pronounced transcriptional and compositional alterations in BD. The observed compositional reduction in PVT neurons was more substantial than the reduction in cortical neurons, which was previously reported[64–66] and replicated in this study. These results underscore a pivotal role of PVT neurons in BD pathology, with dysregulated molecular networks highlighting genes such as *SHISA9*, *CACNA1C*, and *KCNQ3* as potentially critical hubs.

The observed reduction in thalamic excitatory neurons, including PVT neurons, may be associated with the reduced thalamus volume reported in BD[19], although our analysis was limited to the medial thalamus. Our sccomp and scCODA analysis consistently suggested a widespread reduction across thalamic excitatory neuron subtypes. This suggested a global reduction across thalamic excitatory neurons beyond PVT neurons. Given that variability in tissue dissection is a potential confounder, future studies encompassing the entire thalamus will be required to determine whether this reduction is specific to the medial thalamus or extends to the whole structure. It also remains an open question whether the present findings can account for the volumetric reductions of the thalamus observed in BD.

Among all the thalamic and cortical cell types in our study, PVT neurons exhibited the highest number of DEGs in BD. Notably, downregulated DEGs in PVT neurons were more biologically interpretable than upregulated ones and included putative BD risk genes. Among the PVT neuron DEGs, *SHISA9*, *CACNA1C*, and *KCNQ3* stood out as particularly relevant to BD pathophysiology, as they were centrally positioned in the PPI network and consistently highlighted in GO, GSEA, and SynGO analyses. *SHISA9* is one of the 36 credible genes identified in the largest BD GWAS. *CACNA1C* has repeatedly ranked among the top BD-associated loci by *P*-value across PGC1-4 studies[1,2,50,51] and was implicated in our FUSION-based TWAS analysis. *CACNA1C* has additionally been associated with schizophrenia in prior GWAS[33], suggesting a shared genetic architecture between BD and schizophrenia. Consistently, downregulated DEGs in PVT neurons were enriched for schizophrenia polygenic risk as assessed by

MAGMA. While *KCNQ3* is not classified as a putative BD risk gene, its downregulation was reported in bulk brain tissue from patients with BD[61]. *KCNQ3* may represent a BD-associated gene identified through transcriptomic rather than genomic approaches.

Our prior findings showed that regulation of neurotransmission in the PVT via tetanus toxin or Designer Receptors Exclusively Activated by Designer Drugs (DREADDs) induced recurrent depression-like behavioral changes even in wildtype mice[24,25]. These results suggest that PVT plays a causative role in BD pathophysiology. Prevention or reversal of reduced thalamic volume by lithium[19] supports this postulated causality. PVT neurons are essential for emotional processing and salience, projecting to regions such as the nucleus accumbens, amygdala, and mPFC in mice[21–23]. PVT is also among the few brain regions enriched with both serotonin and noradrenaline in mice[26], underscoring its relevance in mood regulation and responsiveness to psychotropic modulation. In alignment with findings in mice, human MRI studies report PVT connectivity with the basolateral amygdala[67], brainstem, mPFC, anterior and posterior cingulate cortex, ventral striatum, hippocampus, and amygdala[68]. Using diffusion-weighted MRI tractography, we successfully parcellated the human PVT based on connectivity[69]. In mice, recent work has revealed functional and transcriptomic differences between the anterior and posterior PVT[23,38,40]. In our study, most human PVT neurons corresponded to the posterior PVT neurons in mice. Future investigations examining the entire PVT will be essential to clarify the evolutionary correspondence of PVT neurons between humans and mice.

While human PVT neurons have been defined anatomically through immunohistochemical markers, acetylcholine esterase, calretinin, and calbindin[36,70,71], the molecular characteristics of these neurons have remained largely unexplored. This study provides an in-depth molecular characterization of human PVT neurons, focusing on gene expression patterns associated with dopamine signaling and calcium ion regulation. Although categorized as excitatory, PVT neurons express calcium-binding proteins (CALB1, CALB2) commonly linked with cortical inhibitory neurons. These calcium-binding proteins are critical in buffering rapid fluctuations in intracellular calcium concentrations, which are associated with high-frequency firing. This profile may indicate a predisposition to pathological hyperexcitability and consequent cellular damage, as postulated in BD[72]. Potassium channels, such as *KCNQ3*—a gene identified as a key DEG in BD—are pivotal regulators of high-frequency firing. *KCNQ3* encodes the Kv7.3 subunit, which assembles with Kv7.2 to form heteromeric channels mediating the non-inactivating, slowly activating M-current near the resting membrane potential[73]. The M-current regulates neuronal excitability by modulating input resistance, action potential threshold, and the medium afterhyperpolarization, thereby influencing firing frequency[74]. Inhibition of the M-current lowers the action potential threshold, enhances afterdepolarization, and depolarizes the

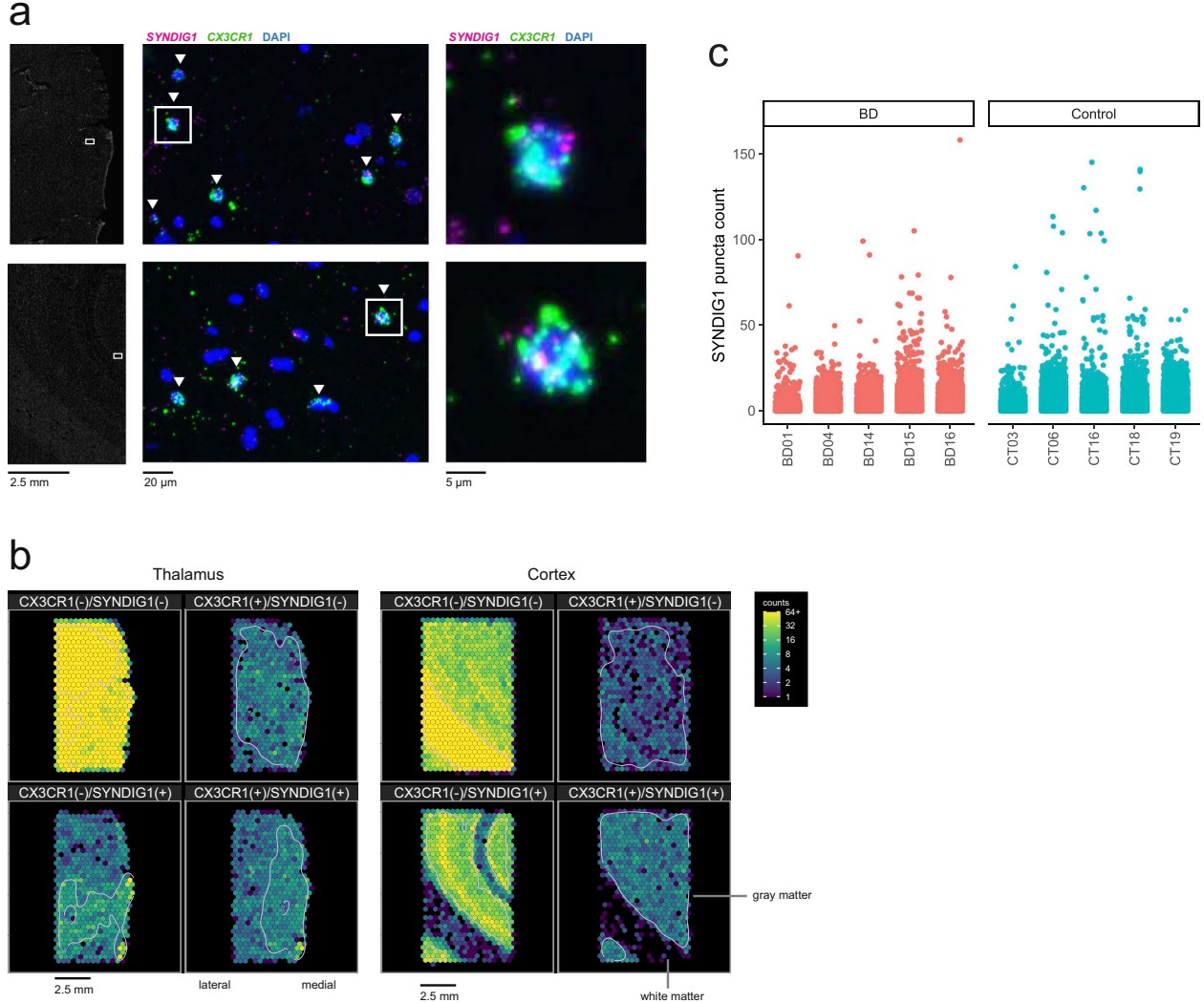

**Fig. 9 | Single-molecule fluorescence in situ hybridization of *SYNDIG1* transcript. a** RNAscope validation of *SYNDIG1* expression in *CX3CR1*-positive cells within the thalamus and cortex. Left panels: Low-magnification view of the thalamus and cortex with DAPI staining. White rectangles denote the regions magnified in the middle panels. Middle panels: Magnified views highlighting *CX3CR1*-positive cells (likely microglia), indicated by white arrowheads. Magenta signals represent *SYNDIG1* expression, which varies among *CX3CR1*-positive cells. The white square indicates the region further magnified in the right panels. Right panels: Higher magnification of *CX3CR1/SYNDIG1* double-positive cells, corresponding to CX3CR1(+)/SYNDIG1(+) cells in panel b. This experiment was performed in one control sample (CT11). **b** Spatial distribution of cell types classified by CX3CR1/ SYNDIG1 expression. CX3CR1(+)/SYNDIG1(+) cells are considered *SYNDIG1*-expressing microglia. CX3CR1(+)/SYNDIG1(−) cells represent microglia with little or no *SYNDIG1* expression. CX3CR1(−)/SYNDIG1(+) cells likely correspond to non-microglial *SYNDIG1*-expressing cells, such as excitatory neurons. CX3CR1(−)/SYN-DIG1(−) cells are classified as other cell types. Cell counts per hexagonal bin are assigned a color code at log2 scale. White contours are the median of cell density for each cell classification. **c** Quantification of *SYNDIG1* puncta in *CX3CR1*-positive cells (likely microglia) across 10 representative male samples (five BD cases and five controls). Each dot represents an individual cell. Source data are provided as a Source Data file.

axonal resting potential, resulting in increased excitability and a propensity for high-frequency firing[75]. Accordingly, reduced Kv7.3 expression may contribute to neuronal hyperexcitability. Additionally, the observed enrichment of genes related to dopamine signaling (e.g., *SNCA*, *PRKN*) implies potential susceptibilities to neurodegeneration. PVT neurons are one of the top cell clusters expressing the putative BD risk genes, although the differences in BD risk gene expression between PVT neurons and other cell clusters were not markedly distinct, as illustrated in Fig. 8b. Perturbed expression of these putative risk genes in patients with BD may interact with the physiological vulnerabilities of PVT neurons, accelerating neuronal damage and dysfunction, as demonstrated in our study. Indeed, the characteristic expression of dopamine-related and calcium-regulating genes in PVT neurons (Fig. 2f) aligns well with the BD-associated biological processes revealed by the largest GWAS to date[1].

In addition to PVT neurons, other thalamic excitatory neurons express *CALB1*, *CALB2*, and/or *PVALB*, all of which encode calcium-binding proteins. The widespread expression of these proteins may underlie the unexpected transcriptomic similarity observed between thalamic excitatory neurons and cortical inhibitory neurons. Calcium-binding proteins are critical for buffering rapid fluctuations in intracellular calcium concentrations associated with high-frequency neuronal firing. Kim et al. reported that thalamic excitatory neurons originate from radial glia derived from prosomere 2 of the embryonic diencephalon[35], whereas cortical inhibitory neurons arise from the ganglionic eminence, highlighting their distinct developmental origins. The observed transcriptomic similarity is unlikely to reflect shared embryonic origins but may instead suggest functional convergence, particularly in intracellular calcium regulation. Interestingly, both excitatory and inhibitory thalamic neurons showed preferential

expression of genes typically associated with cardiac muscle. This observation may reflect a shared physiological trait—frequent and rhythmic intracellular calcium regulation—between thalamic neurons and cardiac myocytes.

Beyond cell-type-based analysis, we identified trans-cellular meta-gene programs between excitatory neurons and microglia (Excit-Micro meta-gene programs), likely reflecting physiological synaptic interactions between these cell types. Microglia are known to modulate synaptic plasticity and neuronal circuitry through dynamic interactions with neurons[76]. Notably, such meta-gene programs in the thalamus are disrupted in BD, further characterized by a significant decrease in synaptic gene expression in both PVT neurons and microglia. Ling et al. reported downregulation of the synaptic neuron-astrocyte program in schizophrenia[10]; our findings highlight downregulation of the synaptic genes in the thalamic neuron and microglia in BD. Nonetheless, these meta-gene programs remain data-driven constructs. Future studies employing animal and in vitro cellular models will be crucial for elucidating these potential interactions.

Although BD is also associated with reduced cortical volume, the transcriptional and compositional changes in the cortex were less pronounced than those in the thalamus. The frontal cortex has been frequently analyzed in psychiatric research, not only due to its suspected involvement in disease pathology but also because of its relative accessibility. This may reflect a form of the "streetlight effect," wherein research tends to focus on more accessible regions. However, higher accessibility does not necessarily equate to higher biological significance. Our findings underscore the importance of extending investigations to subcortical regions, which may harbor critical yet underexplored components of BD pathophysiology. It is noteworthy that the cortical region analyzed in this study was BA10. Results may differ in other cortical regions, such as the anterior cingulate cortex, emphasizing the need for broader sampling across diverse cortical areas. Integration of GWAS data with brain-wide snRNA-seq indicated heritability enrichment in cortical neuronal populations, including interneurons and deep-layer intra-telencephalic neurons[1]. While such genetic insights are informative, caution is warranted when inferring cell-type-specific dysfunction based solely on heritability enrichment. Genetic enrichment does not necessarily guarantee transcriptional dysregulation. Moreover, technical variability in defining cell-type-specific gene expression profiles can influence enrichment outcomes and lead to divergent interpretations[77]. Therefore, direct observations of transcriptional and compositional changes remain essential to elucidate disease-associated cellular alterations.

Our study has several limitations. Despite efforts to maximize thalamic sample collection from the paraventricular region, the sample size remains relatively limited. Additionally, inter- and intra-individual variability in the anatomical location of PVT presents a significant methodological challenge, particularly concerning compositional analysis. Although an effort was made to standardize the sampling location, variability in the anatomical positioning of the specimens was unavoidable among the 41 individuals. Nevertheless, we validated the compositional changes by immunohistochemistry. Furthermore, the enrichment of DEGs in PVT neurons aligns with the observed compositional changes in PVT neurons, and fewer DEGs in cortical neurons align with less pronounced compositional changes in the cortex. Future research utilizing larger specimens (e.g., 1 cm thickness) and a greater sample size would more effectively control regional variability. While the thalamus was identified as a promising brain region, the pathological mechanisms in other regions, such as the hippocampus, amygdala, and striatum, remain to be elucidated. Further investigation into these regions would help clarify the underlying mechanisms of BD. The sample condition differences between patients with BD and controls—particularly in terms of cause of death—may introduce potential biases into our findings. However, if such sample condition disparities were the main factors driving the

observed distinctions within the thalamus, similar differences would likely appear in cortical analyses. As our cohort is composed exclusively of individuals of European ancestry, the generalizability of these findings to diverse ethnic groups is uncertain. The potential impact of medication effects cannot be entirely ruled out, as such factors are inherent to postmortem brain samples, particularly in countries with advanced healthcare systems. Our DEG analysis was restricted to protein-coding genes for the available comprehensive GO and PPI annotations; nonetheless, non-coding genes also hold considerable biological significance. As annotation resources for non-coding genes expand, future studies will likely yield a more comprehensive understanding of their roles. SnRNA-seq does not capture cytoplasmic or dendritic RNA. Future advances in large-scale spatial transcriptomics and emerging technologies may help address this constraint further. Our modest sample size also precluded an analysis of sex differences in BD, as demonstrated in major depression by Maitra et al.[14]. While the differences between BD subtypes (BD-I and BD-II) are of interest, the limited number of BD-II cases ($N = 5$) in our cohort also precludes a reliable assessment of subtype-specific effects. In this study, numerous experimental and sequencing batches were required due to limited resources. Future work would benefit from automated library preparation and concurrent sequencing of pooled libraries on high-throughput platforms to further control batch effects. Because the exact time of death for the donors was unavailable, the PMI in our dataset was not completely accurate. Given that circadian rhythms can influence both gene expression and disease-associated patterns[78], precise determination of the time of death would facilitate the investigation of circadian rhythm-associated gene expression changes.

Despite these limitations, this study indicates the disturbances of PVT neurons as a core pathology in BD, providing a promising foundation for developing new diagnostic and therapeutic strategies. Neuroimaging techniques targeting the PVT, such as MRI and PET, could enhance clinical assessment for BD. Integrating multi-modal methodologies, encompassing cellular, molecular, and neuroimaging techniques, will provide a more comprehensive understanding of the pathological contributions of PVT in BD. PVT neurons represent a promising target for the development of advanced diagnostic and therapeutic strategies aimed at improving patient outcomes.

## Methods

### Post-mortem brain samples

Postmortem brain specimens were sourced from the Douglas–Bell Canada Brain Bank (www.douglasbrainbank.ca). Frozen samples were obtained from 21 patients with BD and 21 matched controls (Supplementary Data 1), with written informed consent provided by the next of kin. No compensation was provided to donor families. Due to the limited availability of postmortem thalamic tissue, no formal sample size calculation was performed; instead, we included all eligible samples available at the time of collection, ensuring sex matching between BD cases and controls. Each group comprised 14 males and 7 females prior to quality control. Sex was determined from coroner files; gender identity information was not available for deceased donors. Analyses were not stratified by sex because of the limited sample size within each diagnostic group. The average age at death was significantly higher in the control group than in the BD group (52.5 vs. 44.3 years; two-sided t test, $P = 0.0472$), reflecting our prioritization of paraventricular thalamic region availability and sex matching over age matching. The cause of death was significantly skewed toward suicide in the BD group (two-sided Fisher's exact test, $P = 2.33 \times 10^{-8}$), consistent with the elevated suicide risk in BD and the characteristics of this brain bank. Of the 42 donors, 41 were Caucasian and one Asian. Donors exhibited substantial heterogeneity in substances present at death, precluding stratification into subgroups of more than two individuals. Summary statistics of substances present at death are presented in Supplementary Fig. 10a.

Bulk specimens of the medial thalamus and frontal cortex (BA10) were dissected from archived fresh-frozen slabs and then sectioned at a thickness of 200 μm using a CM3050S cryostat-microtome (Leica Microsystems, Wetzlar, Germany). First, the whole brain was bisected into left and right hemispheres. The left hemisphere was snap-frozen in 2-methylbutane precooled to −40 °C and stored at −80 °C, while the right hemisphere was fixed in 10% neutral buffered formalin (phosphate buffer, pH 7.0; Fisher Scientific, Cat. #SF-100-20) for paraffin embedding and sectioning. Within a half hemisphere, the PVT was macroscopically identified, with its anterior boundary defined by the coronal plane at the anterior edge of the mammillary body and its posterior boundary by the coronal plane at the anterior edge of the posterior commissure, spanning approximately 20 mm along the anterior-posterior axis. A 10-mm-thick coronal slab encompassing the PVT was then selected and excised. Each slab extended anteriorly to the center of the mammillary body and posteriorly to midbrain structures, including the red nucleus and substantia nigra. From each slab, tissue blocks were prepared to include the PVT together with major adjacent landmarks (putamen, caudate, and insular cortex), ensuring accurate anatomical orientation and precise localization of the PVT (Fig. 1a, Supplementary Fig. 10b-d). Dissections were typically performed from the posterior aspect of the slab, with each block carefully inspected to confirm that the slab contained the appropriate rostrocaudal level of the PVT. Although the procedure was designed to target the mid-PVT, unavoidable interindividual variation in gross neuroanatomy occasionally resulted in slight shifts in sampling position along the anterior-posterior axis. The medial region of the thalamic cryosections and the gray matter part of the cortical cryosections were excised for snRNA-seq. Approximately 50 mg of cryosections, corresponding to two to four cryosections, underwent the subsequent cell nuclei isolation. This research was conducted in accordance with the Helsinki Declaration and received approval from the Research Ethics Committee of the Faculty of Medicine, Juntendo University (approval ID: M19-0278).

### Cell nuclei isolation

Cell nuclei were isolated from the prepared specimens, with myelin sheath removal achieved through a four-layer density gradient centrifugation. The tissue was homogenized in 900 μl of homogenization buffer (HB; 250 mM sucrose, 25 mM KCl, 5 mM $MgCl_2$, 20 mM Tris-HCl pH 8.0, 1× cOmplete™, EDTA-free Protease Inhibitor Cocktail (Roche, Basel, Switzerland), 0.1 mM dithiothreitol (Fujifilm, Tokyo, Japan) and 0.04 U μl$^{-1}$ Protector RNase Inhibitor (Roche)) by using approximately 20 strokes of a pellet pestle and then by pipetting 10-times through a 1000-μl low retention fine point tip (Rainin, Oakland, USA). The homogenized tissue was collected as a pellet by centrifugation at $500 \times g$ for 5 min at 4 °C. The pellet was suspended in 1000 μl of HB supplemented with 0.05% polyoxyethylene sorbitan monolaurate (Tween 20, Nakarai, Kyoto, Japan) by pipetting, and then incubated for 12 min with gentle pipetting at times through a 1000-μl low retention fine point tip to dissociate nuclei. The dissociated nuclei were collected as a pellet by centrifugation at $500 \times g$ for 5 min at 4 °C, and then suspended in 900 μl of nuclei suspension buffer (NSB; 1× PBS, 1%(w/v) UltraPure BSA (Thermo Fisher, Waltham, USA) and 0.04 U μl$^{-1}$ Protector RNase Inhibitor). The nuclei suspension was filtered through 70-μm and 40-μm Mini Cell Strainer (Funakoshi, Tokyo, Japan), serially, and then collected as a pellet by centrifugation at $500 \times g$ for 5 min at 4 °C. For density-gradient centrifugation, 50% iodixanol solution was made, which consists of the same composition as HB except where sucrose was replaced with iodixanol using OptiPrep (60% iodixanol, Serumwerk Bernburg AG, Bernburg, Germany), and then mixed with HB to prepare 25%, 29%, and 35% iodixanol solution. The nuclei pellet was suspended in 1000 μl of 25% iodixanol solution. A four-layer discontinuous iodixanol density gradient was assembled by sequentially overlaying 200 μl of 35% iodixanol solution as a cushion, 400 μl of 29%

iodixanol solution, all nuclei suspension in 25% iodixanol solution, and 150 μl of HB. The density-gradient was centrifuged by 2-steps: at $500 \times g$ for 6 min at 4 °C, and then at $5000 \times g$ for 30 min at 4 °C. After 1400 μl of upper layers (partially including 29% iodixanol layer) was removed totally from the top surface of layers, nuclei on 35% iodixanol layer were collected as a suspension in remaining solution. For iodixanol removal, the nuclei suspension was diluted by adding 1000 μl of NSB, and then centrifuged at $500 \times g$ for 30 min at 4 °C to collect nuclei as precipitates. The nuclei were washed with 500 μl of NSB, and then re-precipitated by centrifugation at $500 \times g$ for 10 min at 4 °C. The purified nuclei were suspended in 100 μl of NSB, 30 μl of which was diluted 5-fold with NSB, and then filtered through a 40-μm porosity Flowmi cell strainer (Sigma Aldrich, Burlington, USA). Isolated single nuclei were counted, and if needed, more diluted with NSB to adjust the concentration of nuclei to the recommended concentration for the following library preparation according to the manufacture's guidelines (10x Genomics). All procedures were performed on ice or at 4 °C. Two specimens from one BD patient and one control were processed simultaneously from nuclei isolation to snRNA-seq library preparation to minimize systematic bias due to experimental batches. All procedures were performed by the same experimenter (KI), who was blind to the disease status of the samples, to avoid inter-experimenter differences.

### Single-nucleus RNA-sequencing (snRNA-seq)

Nuclei isolated from the medial thalamus and frontal cortex underwent library preparation using the 10x Chromium 3′ RNA v3.1 dual index protocol, following the manufacturer's instructions (10x Genomics, Pleasanton, USA). The target recovery rate was set at 5000 nuclei, with library amplification achieved through 14 PCR cycles. The snRNA-seq libraries were sequenced on the NextSeq2000 platform (Illumina, San Diego, USA) in accordance with the protocols provided by Illumina and 10x Genomics. To minimize experimental bias, 10x Genomics library preparation was performed exclusively by the same experimenter (KI), with all case–control pairs processed in parallel. Due to the limitations of the NextSeq 2000 platform, which could accommodate only a few single-nucleus RNA-seq libraries per run, the dataset spans multiple sequencing batches, including several salvage runs needed to achieve sufficient read count per nuclei.

The output files were processed with Cell Ranger (v7.0.0, 10x Genomics) using default parameters (reference: GRCh38-A) to generate count data. Sample BD21 thalamus was excluded due to a low recovery rate (<1000 nuclei, target of 5000). We did not process the BD21 frontal cortex to maintain consistency in paired analysis. The final dataset comprised 82 snRNA-seq count datasets from 41 individuals (21 BD cases and 20 controls) across the thalamus and frontal cortex. All samples achieved read depth per nucleus exceeding 50,000, with a median depth of 74,000 reads per nucleus.

### Human Brain Cell Atlas (HBCA) dataset

We analyzed snRNA-seq data from 20 thalamic samples in the HBCA dataset[6] to enhance the generalizability of our findings. The sample size of 20 was selected to match the size of our in-house control thalamic snRNA-seq dataset. The original sample IDs were 10 × 190_3, 10 × 190_4, 10 × 193_7, 10 × 193_8, 10 × 348_3, 10 × 348_4, 10 × 354_5, 10 × 354_6, 10 × 361_3, 10 × 361_4, 10 × 362_1, 10 × 362_2, 10 × 375_3, 10 × 375_4, 10 × 377_1, 10 × 377_2, 10 × 389-5, 10 × 389-6, 10 × 393_1, and 10 × 393_2, referred to as HBCA01-20 in our study (Supplementary Data 17). These 20 samples were selected based on their potential inclusion of PVT regions. Note that the HBCA dataset was derived from only 2–3 individual donors, and the proportions of PVT neurons varied across these samples, because these samples were not specifically dissected for PVT analysis but were prepared to investigate regional differences, as described in the original study[6].

## Integration of snRNA-seq data

Count data from Cell Ranger were decontaminated using the DecontX tool in the celda package[79] (v1.16.1) and subjected to multiplet identification via scDblFinder[80] (v1.14.0). These steps were performed for each count data in our and the HBCA[6] dataset prior to integration. After decontamination and multiplet labeling, only nuclei with >300 and <15,000 detected genes and <5% reads that mapped to mitochondrial genome were retained. The resulting count data were processed using Seurat (v5.0/v5.1)[81] with Harmony[82] (v1.2.0) batch correction. Multiplets were defined as cells labeled by scDblFinder, as well as those within clusters exhibiting a high fraction of scDblFinder-identified multiplets and expressing markers of multiple major cell types (e.g., oligodendrocytes and astrocytes) (Supplementary Fig. 11a, b). After removing multiplets and low-represented clusters, each dataset was re-processed using Seurat with Harmony batch correction. This two-step procedure was consistently applied across the study to minimize the influences of doublets and low-represented clusters. PCA dimensions and clustering resolution parameters were tuned to obtain a reasonable number of clusters for each dataset. Clusters from <75% of samples were removed as low-represented clusters after both the first and second rounds of clustering.

For batch-correction, Harmony was preferred over Seurat RPCA, because the latter failed to clearly separate thalamic and cortical OPCs/astrocytes (Supplementary Fig. 11c), a result misaligned with findings by Siletti et al.[6]. We chose sbDblFinder for multiplet labeling according to ref. 83, and checked its validity as described above. We selected DecontX for ambient RNA removal in preference to SoupX[84], CellBender[85], or analysis performed without any ambient RNA decontamination package (no_decont_package) for the following reasons: SoupX and no_decont_package could not effectively distinguish NeuCyto clusters (details are described later) from oligodendrocytes in the UMAP projection, probably reflecting the effect of ambient RNA to clusters expressing a small number of genes (Supplementary Fig. 11d). CellBender was applied using the expected cell count estimated by Cell Ranger along with other default parameters. However, in several cases, CellBender appeared to fail in clearly distinguishing nuclei-containing droplets from empty droplets (Supplementary Fig. 12a, *External Data*, see *Data Availability*). CellBender-based clustering resulted in atypical clusters that expressed neuronal markers but exhibited extremely low total RNA counts (Supplementary Figs. 12b-g). These clusters, labeled as Tha_ExN_PVALB_X and Tha_InN_X did not correspond to any known thalamic cell types and were not consistent with cytoplasmic droplets, as determined by nuclear fraction and mitochondrial RNA content. Although a definitive ground truth for cell-type identity was not available, we interpreted these clusters as likely artifacts resulting from inappropriate ambient RNA removal, leading to biologically implausible transcriptomic profiles. In contrast, cell-type annotations were consistent across DecontX, SoupX, and the no_decontamination_package, and no unexplained anomalous clusters emerged under these approaches, even if increasing clustering resolution. While the exact cause of this phenomenon remains uncertain, we suspect that the presence of cytoplasmic components—discussed later and supported by NeuCyto clustering—may interfere with CellBender's ability to accurately distinguish between nuclei-containing droplets and empty droplets.

## Cell-type markers and annotation

Neuronal cell clusters in the frontal cortex and glial cell clusters were annotated using cell-type label transfer with Azimuth, using a cortex reference[29]. Azimuth internally employed anchor-based transfer using FindTransferAnchors and TransferData functions in Seurat. The transferred cell-type labels were validated and further annotated by known cell-type markers (Supplementary Data 18)[6,13,34]. Thalamic cell clusters were annotated based on specific cell-type markers. T cells were annotated by Seurat anchor transferring of reference peripheral blood cell datasets (10x Genomics, https://www.10xgenomics.com/).

Clusters designated as NeuCyto were considered neuronal cytoplasmic components from the following properties: low nuclear fraction, high content of mitochondrial RNA (Supplementary Fig. 13a), high expression of neuronal markers, virtually no expression of other glial markers (Supplementary Figs. 1b, 3a, 6a), and distinctive clustering from probable empty droplets (Supplementary Fig. 13b, c). The probable empty droplets, designated as "Empty", were defined as droplets filtered out by Cell Ranger with detected genes of 100 – 300. This range was set by QC procedure and barcode rank plot for UMI counts by Cell Ranger. NeuCyto droplets were robustly detected from 40/41 of our thalamic samples (occupying 1.46% of total cell nuclei) and 40/41 of our cortex samples (occupying 1.46% of total cell nuclei) as well as 20/20 of HBCA thalamic samples (occupying 1.66% of total cell nuclei). The clear separation of VGLUT+ and GAD+ NeuCyto clusters in the thalamus indicated that they were distinct cytoplasmic components (Fig. 3a). We removed NeuCyto clusters for the subsequent analysis as non-nuclei droplets but retained them for UMAP visualization and dot plots as a technical reference. Indeed, in the final nuclei suspension, we observed objects stained with Nonyl Acridine Orange (NAO) but negative for Propidium Iodide (PI) and Hoechst 33342 (Supplementary Fig. 13d), likely representing mitochondria-rich, nucleus-free cytoplasmic fragments corresponding to the NeuCyto clusters.

Hierarchical clustering was performed using the hclust function and the Wald2 test in R. Specific gene expressions for thalamic cell types were identified using the FindMarkers function in Seurat. Gene set enrichment analysis (GSEA) was conducted using the gseGO function in clusterProfiler[86] (v4.10.1). For GSEA input, genes were ranked based on the product of fold-changes (FCs) and expression rates (percentage of cells expressing the gene). The input was restricted to the genes with expression rates greater than 0.1 for each major class or cell type.

## Spatial transcriptomic data of the medial thalamus

We selected one medial thalamus sample (CT06) as a representative specimen for spatial transcriptomic analysis using Visium Spatial Gene Expression for FFPE (10x Genomics). The frozen coronal section of the medial thalamus was cut at 12 µm thickness, subsequently placed on an 11 mm × 11 mm capture area of Visium slide (10x Genomics), and fixed in 100% methanol at −20 °C for 30 min. Hematoxylin and eosin (H&E) staining was performed according to the Visium CytAssist Spatial Gene Expression for Fresh Frozen—Methanol Fixation, H&E Staining, Imaging & Destaining Protocol (10x Genomics). The sample was processed using the Visium CytAssist Spatial Gene Expression Reagent Kit for Human Transcriptome. Libraries were sequenced on DNBSEQ-G400 (MGI), with sequence reads mapped to the human probe set (v2.0) to generate gene-level UMI counts per spot. The fiducial frame was aligned to locate the spatial position of the barcodes on the H&E-stained image using SpaceRanger (v2.0.1). Unbiased clustering of spatial spots was performed by SpaceRanger and visualized using the Loupe Browser (v8.0.0, 10x Genomics). RNA quality was assessed from adjacent sections using the Direct-zol RNA MicroPrep kit (Zymo Research) and High Sensitivity RNA ScreenTape (Agilent Technologies), yielding an RNA integrity number (RIN) of 6.6, within the acceptable range for Visium analysis.

## Immunohistochemistry for calretinin

The anatomical distribution of calretinin, a protein encoded by the *CALB2*, was evaluated through immunohistochemistry using an adjacent frozen section to that employed in the spatial transcriptomic analysis. Fresh-frozen tissue was sectioned at a thickness of 12 µm and incubated in 4% paraformaldehyde at 4 °C overnight after a 5-min drying period at room temperature. Following washes with phosphate-

buffered saline (PBS), the sections were sequentially dehydrated in increasing concentrations of ethanol and then allowed to dry for 5 min at room temperature. Antigen retrieval was performed by boiling the sections in a solution containing 5% urea (pH 9.5) in PBS with 0.05% Tween 20 (PBST) at 80 °C for 20 min. After cooling to room temperature, the sections were blocked with a blocking buffer (0.4% Blockace in PBST) for 1 h at room temperature. Primary antibody incubation was performed overnight at 4 °C in a humidified chamber using a primary anti-calretinin antibody (Spring Bioscience #SP-13, 1:500 dilution in blocking buffer supplemented with Canget A). The following day, sections were rinsed with PBST and treated with 0.3% hydrogen peroxide (diluted in PBS) for 10 min at room temperature. After additional PBST washes, the sections were incubated with secondary antibodies (1:1000 dilution in blocking buffer) for 1.5 h at room temperature in the humidified chamber. Immunostaining was visualized using the DAB Peroxidase Substrate Kit (Vector Laboratories, Newark, USA), and the nuclei were counterstained with Hematoxylin for 2 min. The slides were washed with running water for 30 min, then dehydrated through a graded series of ethanol and xylene. Coverslips were mounted using Entellan New (Merck Millipore, Burlington, USA). Images were captured using a Digital Slide Scanner Nanozoomer2.0-RS (Hamamatsu Photonics, Hamamatsu, Japan) with a 40× objective.

## Deconvolution of spatial transcriptomic data with snRNA-seq cell clusters

We deconvoluted the Visium spatial transcriptomic data using cell-type expression profiles obtained from snRNA-seq via CytoSPACE[39] (v1.1.0) with default parameters to estimate the spatial distribution of each cell type. The reference cell types included excitatory/inhibitory neurons, astrocytes, OPCs, oligodendrocytes, and microglia, as defined in the integrated set comprising the HBCA and our dataset (see Fig. 2a). The low representation of the ExN_PVALB_2 cluster in Fig. 2d was consistent with its low/no proportion in the CT06 snRNA-seq dataset (Supplementary Data 17), supporting the validity of our deconvolution approach. As a robustness check, we repeated the deconvolution using either our control thalamic dataset or CT06-derived data alone, both of which yielded consistent results (Supplementary Fig. 13e, f). The absence of the ExN_PVALB_2 cluster in CT06 is likely due to dissection-related variability rather than true biological absence. In fact, ExN_PVALB_2 clusters exceeded 1% of total cells in only 11 of 20 HBCA samples and 12 of 20 of our samples. Given their inconsistent representation and presumed off-target origin, we considered ExN_PVALB_2 clusters as off-target cell populations in our analysis.

For reference, we also mapped the location of thalamic NeuCyto clusters. The NeuCyto clusters within the thalamus could be further categorized into three sub-clusters based on marker expression (Supplementary Fig. 13g). Spatial deconvolution revealed that NeuCyto_VGLUT2_CALB was relatively localized in the PVT region, while NeuCyto_VGLUT2_PVALB was sparsely distributed across other areas (Supplementary Fig. 13h). NeuCyto_GAD clusters seemed relatively located in proximity to inhibitory neurons. The three NeuCyto subclusters likely represented neuronal cytoplasm from excitatory and inhibitory neurons.

## Matching mouse PVT neurons to the human PVT neurons

To identify cell types in our human snRNA-seq data that correspond most closely to mouse PVT neurons, we performed anchor-based label transfer using the Seurat functions FindTransferAnchors and TransferData. As reference datasets of mouse PVT neurons, we used the snRNA-seq data from ref. 40 and the single-cell RNA-seq data from ref. 23. The mouse gene name was converted to the corresponding human gene name in the Ensemble database. Prior to anchor transfer, each dataset was independently processed using Seurat, including

normalization, scaling, and principal component analysis (PCA). For the Gao et al. dataset, we followed the authors' published analytical pipeline to identify PVT subtypes, with the exception of PCA dimensions to improve clustering resolution, and annotated PVT subtypes based on the marker genes presented in Fig. 3 of ref. 40. For the Shima et al. dataset, we used their original cell type annotations.

## Cellular compositional analysis

To enhance resolution and precision in observing BD-related changes in specific cell clusters, we integrated thalamic and frontal cortex data separately, utilizing samples from the same cohort of individuals, 21 BD cases and 20 controls.

As compositional analysis methods rely on specific assumptions, we applied two methods of compositional analysis for single-cell data, sccomp[42] (v2.1.6) and scCODA[43] (v0.1.9), to compare nuclei counts across cell clusters between BD and control groups. The sccomp employs a sum-constrained beta-binomial model, while the scCODA employs a Bayesian model using a reference cluster. The threshold of inclusion probability in scCODA was set as 0.95. We considered the analysis to have failed when the observed inclusion probability was 1.0 for all clusters within a given covariate, indicating a lack of model convergence or informativeness. In cases where raw counts could not be used successfully (cortex analysis and PMI analysis), total nuclei counts were normalized to 10,000 per individual, and cluster-specific counts were rounded to integers to reduce inter-individual variability in total nuclei counts. The compositional analysis was limited to the clusters in six major classes: excitatory/inhibitory neurons, oligodendrocytes, OPCs, astrocytes, and microglia.

## Covariate selection in compositional analysis

Initially, we considered a wide range of potential covariates: diagnosis (Dx), sex, age, postmortem interval (PMI), brain pH (pH), race, cause of death, refrigerator delay, substances at death, experimental batch, and sequencing batch. Given our modest sample size ($N = 41$), covariates should be selected to minimize the risk of overfitting and multicollinearity. Race was excluded from statistical modeling because the cohort was heavily biased toward Caucasian individuals, with only one Asian sample. Cause of death was excluded to reduce multicollinearity, because it exhibited a significant imbalance with suicide overrepresented among BD cases (two-sided Fisher's exact test, $P = 2.33 \times 10^{-8}$). The refrigerator delay contained three missing values, and RD was correlated with PMI (Pearson's correlation, $P = 0.0181$). We prioritized PMI as a relatively more relevant potential covariate. Due to the wide heterogeneity of substances at death and missing information in several donors, subgrouping based on substances at death was not feasible in our dataset. Regarding experimental and sequencing batches, we had 21 batches for nuclei isolation and 16 sequencing batches in thalamic analysis. Modeling batch effects was unsuccessful due to the large number of batches relative to the sample size ($N = 41$). Although experimental and sequencing batches could not be incorporated into the subsequent statistical modeling due to their high dimensionality relative to sample size, we confirmed that the identified cell clusters were broadly and consistently represented across batches (Supplementary Fig. 14a, Supplementary Data 19, 20).

The remaining candidate covariates were Dx, sex, age, PMI, and pH. Among them, pH values obtained from cerebellum samples, may not directly represent the specimen conditions of the thalamus and frontal cortex; given probable regional variability, we treated pH as an indirect indicator for this study. Notably, pH and PMI were associated with Dx, with pH showing a stronger signal (Supplementary Fig. 14b, c). These associations likely reflect differences in cause of death, as BD samples predominantly originated from individuals who died by suicide, unlike control samples. Nonetheless, we tested statistical models with all five covariates (Dx, Sex, Age, PMI, and pH), as commonly used covariates in postmortem brain studies.

For the compositional analysis, model fit was evaluated using Akaike Information Criterion (AIC) from generalized linear models (GLM) employing negative binomial distribution (glm.nb in MASS [v7.3-60] package), because the sccomp and scCODA packages do not provide scores indicating model fit like AIC. In the GLM, total nuclei counts were normalized to 10,000, with each cluster's count rounded to integers to reduce variability in total nuclei counts across individuals. We compared multiple candidate models and ranked them by summed AIC rankings across both thalamic and cortical datasets (see rankSum in Supplementary Data 21). The model including Dx, age, and sex (Dx + Age + Sex) showed the lowest overall AIC, indicating the best model fit. While the contributions of PMI and pH differed between the thalamic and cortical analyses, we selected PMI as the secondary covariate because pH showed a stronger correlation with diagnosis and represents a less direct measure of thalamus and cortex tissues. Based on these findings, we selected "Dx + Age + Sex" as the primary covariate combination for compositional modeling, and "Dx + Age + Sex + PMI" as the secondary covariate combination. The relative contribution of each covariate was assessed by Type III likelihood ratio chi-square tests (Anova(type = 3) from the car package, Supplementary Data 22).

As support for the above covariate selection, adding PMI did not increase the model fitness in the sccomp analysis, without changes of elpd_diff/se_diff > 5. More than five of elpd_diff/se_diff change is suggested for significant model improvement according to the developers' guideline[42].

## Cell density of excitatory neurons and oligodendrocytes

We assessed the cell density of excitatory neurons and oligodendrocytes in BD and control by immunohistochemistry using antibodies for VGLUT2 and SOX10. Formalin-fixed paraffin-embedded (FFPE) medial thalamus tissues were obtained from 19 BD cases (11 males and 8 females) and 17 controls (13 males and 4 females) by Douglas–Bell Canada Brain Bank. Among these, 28 samples overlapped with those used in the snRNA-seq analysis, and 8 additional samples were included for this analysis. We had two different sampling batches: 21 samples were formalin-fixed and paraffin-embedded in Canada (batch A), while the remaining 15 samples were formalin-fixed at the Douglas–Bell Canada Brain Bank and paraffin-embedded in Japan (batch B). Although samples in batch B exhibited poorer staining, probably due to different fixation procedures, they were included to enhance the sample size for the immunohistochemical assessment.

The FFPE sections were cut at a thickness of 4 μm. Immunostaining was performed using a Ventana BenchMark GX automated slide stainer (Roche Diagnostics), with a slightly modified procedure based on the calretinin protocol outlined above. FFPE slides were first heated on a hot plate at 60 °C for 10 min, followed by deparaffinization with xylene and dehydration through a series of decreasing ethanol concentrations. For batch A, lipid removal was performed by incubating the samples overnight in a 50% chloroform and 50% ethanol mixture (equal volumes) following deparaffinization. After PBS washes, antigen retrieval was carried out in Cell Conditioning (CC1) buffer (pH 8.4) at 120 °C for 5 min. The slides were cooled for 20 min and then incubated with reaction buffer. The reaction buffer, EZ Prep buffer, and Inhibitor solutions were used according to the manufacturer's protocol for IHC detection systems. Mouse anti-SOX10 antibody (Santa Cruz #sc-365692A-2) and rabbit anti-VGLUT2 (Synaptic Systems #135403) were applied at dilutions of 1:250 and 1:750, respectively, for 32 min at 37 °C. Antibody reactions were visualized using the Ultraview Universal DAB detection kit and Universal Alkaline Phosphatase Red Detection kit (Roche Diagnostics), followed by rinsing in EZ buffer. SOX10 was visualized as brown, while VGLUT2 was visualized as red. The nuclei were counterstained manually with Hematoxylin solution for 2 min. After a 5-min wash in tap

water, the slides were dried for 20 min at room temperature and coverslipped with Entellan New (Merck Millipore).

Cell counts were performed by Definiens Tissue Studio 3.6 (Definiens AG), using the IHC Dual Brown/Red chromogen package. Before analysis, image analysis algorithms were tested and optimized. The optimized parameters were applied to all images. The tissue was selected from the background of each scanned image by setting the initialization magnification to 0.3 units. The region of analysis was manually limited to the medial thalamic paraventricular region, 8.5 mm from the third ventricle, corresponding to the area analyzed in the snRNA-seq. The section was segmented into geographic regions based on VGLUT2 distribution, excluding areas without cells, such as blood vessels or tissue cavities. Semiautomatic segmentation was then performed. The following parameters were applied for all analysis runs: Brown marker threshold for nucleus detection set to 0.3, typical nucleus size selected as 50 μm$^2$, Red marker threshold for cytoplasmic detection set to 0.08, and typical cellular size set over 100 μm$^2$. Data on the region of interest (ROI) area, and the count of nucleus and cytoplasm-positive cells were exported for further analysis.

Cell density was calculated as the cell count per 100 mm$^2$ (10 × 10 mm). The integer-rounded cell density values were analyzed using a generalized linear model (GLM) with a negative binomial distribution (glm.nb in MASS [v7.3-60]), incorporating covariates for Dx, sex, age, and sampling batch. In this dataset, no effect of age, sex, PMI, and pH for Dx was observed ($P > 0.20$ by simple regression).

## Differentially expressed gene analysis

Differentially expressed genes (DEGs) were analyzed within each cell cluster in BD compared to controls using zlm function in MAST[44] (v1.28.0), a statistical package for single-cell RNA-seq DEG analysis. The model included Dx, age, sex, and cngeneson (a covariate for cell quality) as fixed effects, and sample ID as a random effect. Sample-level covariates were selected based on the prior compositional analysis and further validated by evaluating AIC values from GLMs fitted to log-transformed gene expression values. In this GLM-based model to select covariates, gene expression was aggregated at the specimen level and restricted to genes expressed in at least 10% of cells within each cluster, consistent with the downstream DEG filtering criteria. These GLMs were used solely to assess covariate impact. Among the tested covariate combinations, "Dx + Age + Sex" yielded the lowest overall AIC across gene-cluster pairs, followed by "Dx + Age + Sex + PMI", as determined by cumulative AIC rank sums (*External Data*, see *Data Availability*). These results support the use of "Dx + Age + Sex" as the primary model and "Dx + Age + Sex + PMI" as a secondary model. The relative contribution of each covariate was further quantified using Type III chi-square statistics (Anova(type = 3) from the car package), as reported in *External Data*. Following MAST guidelines, we incorporated cngeneson, a variable of scaled gene expression counts, to adjust for cell nuclei quality, where higher gene detection rates suggest better nuclei quality within the same cluster.

DEG analysis was limited to protein-coding genes with Entrez IDs to streamline downstream analyses. Clusters with fewer than 1000 nuclei were excluded to ensure sufficient counts per individual. All the *P*-values from the MAST analysis were adjusted for multiple comparisons using the Benjamini-Hochberg method to control the false discovery rate (FDR). The raw P-values and FDRs are provided in Supplementary Data 10. Before DEG analysis, we observed no significant differences in gene count or log-transformed UMI count per nucleus between BD and control samples (Supplementary Fig. 14d), supporting the feasibility of this case-control comparison.

Given the sensitivity of DEG analysis to preset thresholds such as fold-changes (FC) and false discovery rates (FDR), we assessed DEG robustness by testing 1368 threshold combinations to mitigate arbitrary thresholding. The 1368 combinations = 21 expression rate patterns (0.10−0.30) × 11 FC patterns (1.10−1.20) × 6 FDR patterns

(0.05–0.10). A robustness score for each gene in a cell cluster was defined based on the frequency it was classified as a DEG across the 1368 threshold combinations. Genes with scores above 0.5 were designated as probable DEGs, and those with scores above 0 as possible DEGs. Naturally, genes with high expression rates, low FDRs, and elevated FCs were prioritized.

We performed gene ontology (GO) enrichment analysis across the 1368 threshold combinations using enrichGO function in clusterProfiler[86] (v4.10.1) with the FDR threshold of 0.05, setting the backgrounds as the genes expressed in >10% of the cluster. Each GO term's robustness score was derived from the number of times it appeared as a DEG-GO across the 1368 threshold combinations. The resulting GO terms were summarized using the simplifyGO function in clusterProfiler, and terms scoring above 0.5 were considered robust DEG-GOs. GO analyses were conducted separately for downregulated and upregulated DEGs, and only GO terms with at least three associated genes were retained for further interpretation. The reference GO terms were restricted to those with 20–500 genes to avoid too narrow or broad biological pathways. To get insight into synaptic genes, SynGO[45] enrichment analysis was added to the clusters with robust DEG-GOs, using the enricher function in clusterProfiler with the FDR threshold of 0.2 and gene set size of 5–100 genes. SynGO terms with at least two genes were retained to accommodate the smaller number of target genes compared to the above DEG-GO analysis. As we set FDR < 0.2, the SynGO enrichment should be regarded as a potential one. We also performed FC-based GSEA separately for the downregulated and upregulated gene sets to further evaluate the robustness of the identified biological signals, using gseGO function in clusterProfiler with the enrichmentScore threshold of 0.7.

## Assessment of BD risk genes

To evaluate the enrichment of SNP-based genomic risk for BD, we first applied MAGMA[46] to roughly assess the overlap between BD GWAS signals and probable DEGs, using multi-ancestry PGC4 BD summary statistics[1]. Enrichment analyses were conducted for cell clusters containing five or more probable DEGs ($n = 14$ clusters). To assess specificity, we extended the analysis to GWAS summary statistics from other recent Psychiatric Genomics Consortium (PGC) datasets, including schizophrenia (SCZ)[33], major depressive disorder (MDD)[59], attention-deficit/hyperactivity disorder (ADHD)[87], post-traumatic stress disorder (PTSD)[88], obsessive-compulsive disorder (OCD)[89], substance use disorder (SUD)[90], eating disorder (ED)[91], panic disorder (PD)[92], and autism spectrum disorder (ASD)[93] (https://pgc.unc.edu/).

Beyond assessing SNP-based genomic enrichment, we further prioritized specific BD risk genes. As MAGMA alone provides only preliminary gene-level association signals, we adopted a multi-step approach to define "putative BD risk genes". First, we designated the "36 credible genes" reported in the PGC4 BD study[1] as the most reliable BD-associated genes currently available. Given the highly polygenic architecture of BD, it is unlikely that only 36 genes account for all the BD genetic basis. Additional BD risk gene candidates were defined as those overlapping between MAGMA-significant genes and those harboring fine-mapped SNPs identified by PolyFun+SuSiE/FINEMAP[47–49], using the multi-ancestry PGC4 BD summary statistics[1]. MAGMA identified 167 genes associated with BD at Bonferroni-corrected significance (adjusted P < 0.05, based on 17,487 genes with Entrez IDs). Fine-mapping was conducted following the developer's pipeline, using UKBB_LD as the reference panel. Fine-mapped SNPs were annotated using promoter regions (PsychENCODE TSS ± 2 kb; file: tss.sites.codingOnly.gencode.-v19.annotation.bed), enhancer regions (PsychENCODE GRN2; file: INT-14_ElasticNet_Filtered_Cutoff_0.1_cross_assembly_GRN_2.csv), and SnpEff-4.3 for coding consequences. We then defined "putative BD risk genes" as the union of the 36 credible genes and 75 MAGMA-based genes containing fine-mapped SNPs in promoters, enhancers, or as missense variants, resulting in a total of 107 genes (four overlapped,

Supplementary Data 23). To explore regional and cell-type specificity, we evaluated the expression of these 107 genes across thalamic and cortical cell clusters using two metrics: (1) the number of putative BD risk genes expressed per cluster (expression rate > 0.1), and (2) the mean expression level of those genes.

To further support gene prioritization, we conducted transcriptome-wide association studies (TWAS) using FUSION[94] with precomputed weights from the GTEx v8 brain datasets (http://gusevlab.org/projects/fusion/). Due to the absence of thalamic samples in GTEx, the TWAS results were used only for complementary reference and were not included in the definition of putative BD risk genes. TWAS-significant genes were identified using Bonferroni correction for each precomputed weight set (adjusted P < 0.05, across 2557–7732 tested genes per dataset). Given that GTEx represents bulk RNA-seq data from multiple brain regions (not including thalamus), we interpreted these results broadly, without strict anatomical specificity.

## Co-expression network analysis

Gene co-expression networks were constructed using hdWGCNA[52] (v.0.3.03), a weighted gene co-expression network analysis (WGCNA) package[53,95] for high-dimensional data. We applied hdWGCNA separately to the datasets from the thalamus and cortex, following the recommended workflow provided by the developers. To optimize the analysis, we grouped genes by major cell classes, as the sparse count matrices for specific subtypes (e.g., Tha_ExN_CALB) did not yield co-expression modules. For the enrichment analysis of DEGs within specific modules, we employed hypergeometric tests, limiting the analysis to genes with a cell expression rate greater than 0.1 in the relevant cell clusters. Additionally, GSEA of module genes was performed, with genes ordered based on their module scores to capture the GO enrichment within the co-expression network.

## Protein-protein interaction network

We used STRING[54] plugin (v2.1.1.60) in Cytoscape[96] (v3.10.2) for protein-protein interaction (PPI) analysis. PPI with three or more nodes is shown in the figures. The P-value of PPI enrichment was calculated using the web version of STRING (https://string-db.org/, analysis date = 2024-09-30).

## Extended analysis incorporating PMI as an additional covariate for robustness check

As described above, we extensively analyzed compositional and transcriptional changes by incorporating a covariate of PMI into the main covariates (Dx, age, and sex) to check the robustness of our results. This extended analysis yielded results consistent with our initial findings (Supplementary Figs. 15, 16). Specifically, we observed a compositional decrease of thalamic excitatory neurons in BD, including PVT neurons (FDR = 0.0115, FC = 0.582), and FrC_ExN_L6ITCar3 (FDR = 0.0257, FC = 0.578) (Supplementary Fig. 15a, Supplementary Data 24). Dx, sex, and age affected the cellular compositions, while PMI did not (Supplementary Fig. 15b). The scCODA model did not perform reliably when PMI was included alongside Dx, age, and sex, yielding inclusion probabilities of 1.0 for all variables across all clusters, even if using total cell count normalization. This likely reflects overparameterization due to the number of covariates relative to our sample size.

In this extended analysis, Dx, sex, and age prominently affected cell composition and gene expression, while PMI less affected them (Supplementary Fig. 15b and Supplementary Data 25). The DEGs by age or PMI were defined by those with absolute value of FC > 1.1 for the period of two standard deviations in our dataset (i.e., 26.7 years for age, and 38.7 h for PMI). Notably, PMI showed no detectable impact on either cell composition or gene expression in the PVT neurons (Tha_ExN_CALB), suggesting our key findings in PVT neurons are unlikely to be confounded by PMI.

When incorporating PMI into the statistical models, the Dx-based DEG-GOs were again most observed in PVT neurons (Supplementary Fig. 16a). The DEGs in PVT neurons (Supplementary Fig. 16b, Supplementary Data 26) continued to exhibit enrichment in synaptic and neuronal GO terms with the same top GOs—transporter complex (GO:1990351) and monoatomic ion channel complex (GO:0034702)—including *SHISA9*, *CACNA1C*, and *KCNQ3* (Supplementary Fig. 16c, Supplementary Data 27). Again, the downregulated DEGs were linked to the co-expression module Excit-M4, while upregulated DEGs were linked to Excit-M11 and Excit-M9 (Supplementary Fig. 16d). Among the DEGs identified, 43 of 64 downregulated DEGs (67.2%), and 100 of 157 upregulated DEGs (63.7%) from the primary analysis were also classified as DEGs in this extended analysis (Supplementary Data 26). Notably, downregulated BD risk genes—*SHISA9*, *CACNA1C*, and *CTNN5*—remained detectable as downregulated DEGs and constituted core PPI networks in this extended analysis (Supplementary Fig. 16e). *SHISA9* was also detected as a downregulated DEGs in Tha_ExN_P-VALB_1 in the extended analysis. The SNP-based BD genomic risk was again most enriched in the downregulated genes in PVT neurons ($P = 0.00419$), whereas upregulated genes included no BD risk genes despite a greater total number of DEGs (69 downregulated vs. 110 upregulated). The DEG-GOs in the frontal cortex cell clusters were virtually not observed, consistent with the primary analysis (Supplementary Fig. 16f). Finally, GSEA performed on DEGs from this extended analysis yielded results similar to those of the primary analysis (Supplementary Data 28), with downregulated genes in PVT neurons again enriched for synaptic and neuronal GO terms.

## Extended analysis to the samples with low PMI

While PMI had no apparent effect on our findings of PVT neurons, we analyzed the samples with relatively low PMI (<50 h), consisting of 12 BD cases and 9 controls, to check the robustness of our findings for PVT neurons and thalamic microglia. The covariates were set as the same to the primary analysis: Dx, age, and sex. PVT neurons and thalamic excitatory neurons robustly exhibited compositional decreases (FC = 0.563, FDR = 0.0223 for PVT neurons; FC = 0.541, FDR = 0.0193 for thalamic excitatory neurons) by sccomp analysis. DEGs in PVT neurons were also consistent with the main analysis, showing comparable FCs (Supplementary Data 29). For instance, *KCNQ3*, *SHISA9*, and *CACNA1C* showed FC values of 0.755, 0.840, and 0.843, respectively. GSEA also yielded similar results, with the top enriched term being potassium ion transmembrane transporter activity (GO:0015079), including *KCNQ3* (Supplementary Data 30). In this analysis, we employed GSEA instead of DEG-based GO enrichment, as the smaller sample size (12 BD cases vs. 9 controls) reduced the statistical power and FDR control.

## Transcellular meta-gene program

Meta-gene programs (MPs), including trans-cellular MPs, were inferred using GeneNMF[55] (v0.6.0), a package designed to decompose a count matrix (cells × genes) into two matrices (cells × MPs and MPs × genes) via non-negative matrix factorization. The identified MPs represent coordinated gene expression patterns across various cell types. When MPs spanned two or more major cell classes, these were designated as potential trans-cellular MPs. We implemented GeneNMF according to the developer's guidelines, with the following parameters: variable features set to 5000 genes, PCA dimensions ranging from 20 to 30, and a maximum of 20 MPs and 300 MP genes, while retaining all other parameters at the default settings. The nuclei count was downsampled to 2,000 to observe the contribution of each cluster unbiasedly. To compare MP scores between BD and control across cell clusters, we applied a generalized linear mixed model (GLMM) using glmer in lme4 (v1.1-35), incorporating Dx, age, and sex as fixed effects, and sample ID as a random effect. To investigate potential ligand-receptor interactions facilitating cell-to-cell communication within identified

transcellular MPs, we used LIANA[58] (v0.1.13) with default settings, following the developer's workflow. The results of LIANA are presented for the top 20 ligand-receptor pairs.

## Medication signature

We evaluated the potential influence of medications and substances on DEGs by assessing their overlap with established medication signatures. A medication signature was defined as the set of genes reported in previous studies to be differentially expressed in response to a given medication or substance. Target medications/substances included those detected in more than 20% of BD donors and at least twice as frequently as in controls (alcohol[97–99], antipsychotics[62,100], and antidepressants[101]; Supplementary Fig. 10a), as well as medications commonly prescribed for BD (lithium[102] and valproic acid[63]). The reference data are summarized in Supplementary Data 31. Because thalamic snRNA-seq data were unavailable for these medications/substances, we instead used bulk RNA-seq data derived from postmortem brain tissue, human induced neurons generated from induced pluripotent stem cells, or animal brain samples, selecting the most appropriate publicly available dataset for each medication/substance. We used possible DEGs for this analysis to increase statistical power. This approach has inherent limitations due to brain region and cell-type differences, and the results should be interpreted as referential.

## Single-molecule fluorescence in situ hybridization (smFISH)

We performed smFISH using the RNAscope Multiplex Fluorescent Reagent Kit v2 (Advanced Cell Diagnostics [ACD], Newark, USA) to detect *SYNDIG1* and *CX3CR1* transcripts in fresh-frozen human frontal cortex (FrC) and thalamus (Tha) sections from CT11 as well as additional representative thalamic sections from five BD male cases and five male controls. *SYNDIG1* served as the primary target, while *CX3CR1* was used to identify microglia. Fresh-frozen 12-µm-thick sections were cut onto SuperFrost Plus Gold slides and post-fixed overnight in 4% formaldehyde prepared in 0.1 M PBS at 4 °C. The sections were washed with 1× PBS, and dehydrated through a graded ethanol series (30% EtOH in 1× PBS, 50% EtOH in 0.5× PBS, 70% EtOH in water, and 100% EtOH twice), baked at 60 °C for 30 min, and circumscribed with a hydrophobic barrier. Sections were treated with hydrogen peroxide and Protease IV, then hybridized with prewarmed RNAscope probes complementary to *SYNDIG1* (C3; #804391-C3/ACD) and *CX3CR1* (C1; #411251/ACD) transcripts for 2 h at 40 °C. After washing, sections were photobleached in 5× SSC solution for 60 min by using the autofluorescence Quenching illuminator TiYO (Nepa Gene, Chiba, Japan)[103]. Signal was amplified via AMP1–AMP3 hybridization, and sequentially developed per probe by HRP-C#/fluorophore-tyramide (TSA Vivid, 1:750) reactions and HRP blocking. Sections were counterstained with DAPI, mounted in ProLong Gold Antifade, and cured at RT for 48 h. Imaging was performed on a NanoZoomer 2.0-RS (Hamamatsu Photonics) after brightfield and fluorescence calibration each day. While correcting a focal plane (z = 0 µm) every 12 min by autofocus on brightfield, wide-field (~8 mm×12 mm) images were captured at a high-resolution (227 nm/pixel) as multi-channel (4 channels; brightfield, Cy5 for TSA Vivid 650, TRITC for TSA Vivid 570, and DAPI) z-stacks (3 slices; 0 ± 1.5 µm) in NDPI/NDPIS format. The signal quality of smFISH on RNAscope was confirmed per specimen with positive (#320861/ACD) and negative (#320871/ACD) control probes. *CX3CR1* and *SYNDIG1* probes were visualized with TSA Vivid 570 and TSA Vivid 650, respectively. For case–control sets, one case and one control were processed in parallel to minimize batch effects.

Image preprocessing was performed on multi-channel z-stacks using Fiji/ImageJ[104] (v2.16.0/1.54p). To prevent out-of-memory errors, each multi-channel z-stack was split into four z-stacks per channel without importing the entire image, by editing NDPIS files. NDPIS files

are text-formatted lists that specify the order in which individual channel images (NDPI files) are read and reconstructed into a multi-channel z-stack. The brightfield z-stack was excluded from subsequent analyses. The remaining three fluorescent z-stacks (monochrome, 8-bit grayscale) were sequentially imported via the Bio-Formats Importer, underwent background subtraction by the sliding paraboloid algorithm (radius: 300 pixels, disable smoothing: true), converted into maximum intensity z-projection images (MIzPs), and saved to disk to free memory. The fluorescent MIzPs were then assembled into a multi-channel stack with pseudo-color.

Object detection and quantification were performed by using QuPath[105] (v0.5.1). A consistent fluorescent signal threshold of 45 in 8-bit (0–255) grayscale was applied to each channel across all specimens. Cells were detected using the WatershedCellDetection plugin, defined as areas containing a nucleus (DAPI) and a peripheral region within 3-µm of the nucleus. By the SubcellularDetection plugin, the smFISH signals of *SYNDIG1* (TSA Vivid 650, Cy5 channel) and *CX3CR1* (TSA Vivid 570, TRITC channel) were detected as puncta or clusters in cells, and counted as the number of spots estimated, assuming that single spots are approximately $1\,\mu m^2$ (range: $0.5\,\mu m^2$ to $3.0\,\mu m^2$). Additional details are available at https://github.com/msk240/BDsnRNAseq.

Data aggregation was conducted in R (v4.1.0). Positive cells were defined as those containing three or more spots corresponding to the target gene, based on the number of spots estimated from smFISH signals. *SYNDIG1* expression in *CX3CR1*-positive cells, representing putative microglia, was assessed using a generalized linear mixed model (GLMM) implemented in the glmmTNB package (v1.1.12), with diagnosis (Dx) and age as fixed effects and sample ID as a random effect, employing the nbinom2 parameterization of the negative binomial distribution. Sex was not included as a covariate, as all samples were from male donors.

### Reporting summary

Further information on research design is available in the Nature Portfolio Reporting Summary linked to this article.

## Data availability

The count data generated in this study have been deposited in the Gene Expression Omnibus (GEO) database under accession code GSE306819. The data generated in this study are provided in the Supplementary Data files. The External Data generated in this study are provided in figshare (https://figshare.com/articles/dataset/BDsnRNAseq/29876237). The HBCA data used in this study are available in the HBCA database [https://data.nemoarchive.org/biccn/]. The Penzo et al. data used in this study are available in the GEO under accession code GSE208707. The Shima et al. data used in this study are available in the DNA Data Bank of Japan (DDBJ) Genomic Expression Archive (GEA) database under accession code E-GEAD-626. The GWAS summary statistics data used in this study are available in the Psychiatric Genomics Consortium (PGC) database [https://pgc.unc.edu/for-researchers/download-results/]. The PyschENCODE reference data used in this study are available in the PyschENCODE database [http://resource.psychencode.org/Datasets/Integrative/]. Source data are provided with this paper.

## Code availability

All software used in this study is publicly available. The analysis scripts are accessible at https://github.com/msk240/BDsnRNAseq (https://doi.org/10.5281/zenodo.17845882)[106].

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

## Acknowledgements

We thank Kanako Mori for technical assistance, Akira Watanabe from CyberomiX for assistance with spatial transcriptomics, Shima Yasuyuki for anatomical information of mouse PVT neurons, and Naotake Yanagisawa for statistical assistance. We thank Taro Kato and Momoe Kassai from Sumitomo Pharmaceutical, and all the members of the Department of Psychiatry at Juntendo University for their valuable discussion. We thank Sapporo General Pathology Laboratory Co., Ltd., and all members of the Laboratory of Morphology and Image Analysis, Biomedical Research Core Facilities, Juntendo University Graduate School of Medicine, for technical assistance with histopathology. We used ChatGPT (https://chatgpt.com/) to improve language and readability. This work is supported by the following funding agencies: AMED under Grant Number JP24wm0425006 (T.K.), JP24tm0424224 (T.K.), JP25wm0625323 (T.K.), JP24tm0424229 (M.N.); JSPS KAKENHI under Grant Number JP22H00468 (T.K.), JP25H01047 (T.K.), JP23K07021 (M.N.), JP21K07528 (M.S.-K.); JST under Grant Number JPMJFR231W (M.N.); Subsidies for Current Expenditures to Private Institutions of Higher Education from the Promotion and Mutual Aid Corporation for

Private Schools of Japan (T.K.); Takeda Science Foundation (M.N.). The samples were provided by the Douglas Brain Bank (RRID: SCR_025991. G.T. and N.M.). The Douglas Brain Bank receives financial support from Brain Canada and from the Fonds de recherche du Quebec—Sante (FRQS).

## Author contributions

M.N., M.S.-K., and T.K. conceptualized the study design. N.M. and G.T. facilitated donor recruitment for postmortem brain samples and managed brain bank operations, including sample storage and data management. M.S.-K., Y.H., and N.M. prepared brain samples by sectioning and assessing regions for targeted sampling. K.I. conducted library preparation. M.N. and K.T. performed sequencing and bioinformatics analyses. M.S.-K. and Y.H. performed immunohistochemistry. M.S.-K., K.I., Y.H., M.I., and R.I. performed RNAscope and imaging analysis. M.N. drafted the initial manuscript, with M.S.-K. and T.K. providing critical review and revisions. M.N. and T.K. jointly supervised this project. All authors reviewed and approved the final manuscript.

## Competing interests

M.N., M.S.-K., and T.K. are affiliated with the Department of Molecular Pathology of Mood Disorders at Juntendo University Graduate School of Medicine, with funding provided by Sumitomo Pharma Co. Ltd. Sumitomo Pharma Co. Ltd. had no control over the interpretation, writing, or publication of this work. K.I. is also affiliated with TechnoPro, Inc., a research assistance company. TechnoPro had no control over the interpretation, writing, or publication of this work.
