## [Transparent Peer Review file · Nature Communications]

Disturbances of paraventricular thalamic nucleus neurons in bipolar disorder revealed by single-nucleus analysis

Corresponding Author: Dr Masaki Nishioka

Version 0:

Reviewer comments:

Reviewer #1

(Remarks to the Author)

Nishioka et al. submit an interesting manuscript that investigates postmortem brain samples from bipolar disorder (BD) patients using single-nucleus RNA-seq. The authors analyzed a cohort of >80 samples of the prefrontal cortex and thalamus from ~40 subjects to identify new insights into brain dysfunction in BD. The authors showed that gene expression and cell composition changes are predominant in the thalamus, and much lower in the cortex. This led them to focus most of their analyses on the paraventricular thalamic nucleus. Subsequent analyses showed that neurons of the paraventricular thalamic nucleus had large decrease in cell number in BD and exhibited the largest changes in gene expression. In particular, genes involved in synaptic processes were downregulated and few dysregulated hub genes were proposed. Using tools predicting cell-cell interaction changes, the authors proposed that dysfunction of microglia might mediate impairment of synaptic function in BD paraventricular thalamic nucleus neurons. Overall, the study is novel for the field and provides good insights into specialization of thalamic excitatory neurons and their dysfunction in BD. Nevertheless, few major issues, mainly concerning data analysis, should be addressed before accepting the publication.

Major issues:

1. Covariate control needs significant improvement.
 - a. There have been multiple batches of sequencing, however, it is not described how many batches, how many samples per batch, etc. Given that sequencing batch effect is one of the major confounders in snRNA-seq data, the authors should provide thorough analysis of sequencing batch impact on their clustering, cell type annotation, differential gene expression and cell composition. Supplementary Table 1 should also contain information regarding sequencing batch.
 - b. Medication is another large confounder in such analyses, since BD brains are usually exposed to a variety of drugs for a long period of life vs neurotypical brains. Potential impact of medication should be addressed, such as tests whether different medications within their BD cohort have impact on gene expression, e.g. by comparing subgroups of patient samples, comparing to other snRNA-seq studies, or to medication signatures identified in previous animal/patient studies.
 - c. Some samples that were analyzed have high PMI, >24hours, and there are many studies showing the impact of high PMI on the mRNA content, including benchmarking studies for RNA-seq experiments. The authors should perform a thorough analysis for potential confounding impact of PMI and if necessary exclude samples that are too different from low PMI samples and interfere with data analysis.
 - d. Additionally, the authors should statistically analyze potential impact of all metadata variables, including age, sex, drug exposure etc.
2. Although authors did include filter of ambient RNA, it should be improved.
 - a. There is some ambiguity with the analysis of detection and removal of ambient RNA. In Methods Line 565: DecontX was chosen over SoupX and no ambient RNA decontamination
Do the authors refer to "no ambient RNA decontamination" as a tool or package? The authors should be more specific.
 - b. Overall, given potential contamination with ambient RNA, more robust analysis is necessary. We propose to use CellBender and provide the following data in Supplementary figures:
 - Sample-wise expression of ambient RNAs
 - CellBender ELBO (test and train) plots
 - CellBender cell calling curves displaying cell probabilities of droplets
3. Data integration

The authors used iterative integration with Seurat and Harmony. There is no justification provided for why Seurat-only or Harmony-only integration are not enough. Simple comparison plots will help the reader to understand the rationale for choosing iterative integration.

4. Differential gene expression

- a. We could not find information whether p values for differentially expressed genes were adjusted. The authors should outline how they did the adjustment.
- b. While GO analysis of DE genes is often used in snRNA-seq studies, it works better for larger gene lists than the authors obtained in this study. GSEA is better applicable here, since GSEA implements all genes, not only short list of significantly changed DE genes.

5. Integration with genetics is rather preliminary and more sophisticated tools can be implemented for integration with the latest GWAS BD study. This will help validating the findings in the study for their relevance to BD. The thalamus and cortex should be analyzed separately by integration with GWAS data to show potential regional specialization with respect to BD-linked brain dysfunction.

Finally, at the moment, it seems that most association of DE genes in this study with BD risk factors is based only on 1 gene from 36 credible genes from the last GWAS study - SHISA9. We would caution the authors for not to overinterpret such association.

6. Compositional analysis

While it is justified to use sccomp package for compositional analysis, the package does not control for inter-dependency of proportional changes, and the authors admit it when comparing to immunohistochemical assessment, Lines 169-172: Note that immunohistochemistry directly assessed cell density, while compositional analysis using snRNA-seq data assessed relative proportional changes. The immunohistochemical assessment indicated that the compositional increase in oligodendrocytes and astrocytes in BD should be mainly derived from prominent reductions of excitatory neurons, not reflecting absolute cell density increases.

To avoid such issue, there are packages that allow for independent compositional analysis of cell proportions, such as scCODA. The authors should use one of such packages to cross-compare snRNA-seq with immunohistochemistry.

7. Cell-cell interaction

- a. Based on the text lines 270-283, it sounds as the authors propose that SHISA9 protein on neurons interacts with SYNDIG1 on microglia? If yes, then some literature search should help confirming this potential interaction. Otherwise, we suggest concentrating interpretation on general discussion on how microglia modulate synaptic processes in excitatory neurons via AMPA receptor complex. Additionally, LIANA analysis should help with such discussion, however the data is just shown in Figure 5e, without any interpretation.
- b. One missing opportunity in the study is analysis of changes in cortico-thalamic connectivity in BD. This can be done using similar cell-cell interaction tools.

8. Such snRNA-seq studies usually require validation of some findings by single-molecular FISH. We propose selecting the most interesting finding in gene expression and validate it by RNAscope or similar technologies.

9. The cortex does not show large changes in gene expression that is why the authors argue to focus on the thalamus. Nevertheless, this is surprising given previous role of the cortex that has been proposed in brain dysfunction in BD. There is a lot of focus in the Discussion section on the thalamus, but such surprisingly minor changes in the cortex are not discussed. We think these findings are very important, since the study can pave the way for future analyses of postmortem brain tissue in psychiatric disorders. Currently, almost all studies focus on the prefrontal cortex and maybe in the future the thalamus should be the major region for such snRNA-seq analyses.

Minor

1. Discussion lacks similarity of BD and schizophrenia, although CACNA1C is one of the major genes in the current study and it has been identified as genetic risk factor by schizophrenia GWAS and not BD.
2. In general, although Discussion is quite long, some points are extensively discussed, whereas others are missing. For instance, close relation of thalamic excitatory neurons with GABAergic neurons is interesting. Is it related to cell fate or neuronal specialization? Similarly, cardiac tissue similarity can be moved from Results to Discussion section.
3. Lines 137-141 compare data from the current study to previously published organoid data. This comparison is completely separated from all other analyses in the study and in the current state should be removed from the manuscript.
4. In Figure 3b – it will help to add direction of "increased/decreased" for BD.
5. We are not sure why the authors decided to include reference to NeuCyto in Lines 158-159. If this "cell type" is ambient RNA contamination, then it is difficult to make conclusions based on such "cell type".
6. Please correct Lines 255-256:
related to synaptic function, although microglia are primarily regarded as an immune-related cell type
Microglia has well known role in synaptic plasticity and modification of neuronal connectivity, which is also one of their primary roles in the brain.

7. Figure 8a: show p-adjusted line on the figure

8. Lines 452-460 repeat lines 463-472.

9. Correct misprint:

Fig1a Neruon-microglia synaptic interaction in BD (Figure 5)

Reviewer #2

(Remarks to the Author)

Nishioka et al. present a snRNA-seq study of PVT in 21 bipolar disorder (BD) patients and 20 controls. BD is a severe mental health condition marked by recurring manic and depressive episodes, requiring new treatments due to resistance, adverse effects, and high suicide rates. Recent research has identified the thalamus, particularly the paraventricular thalamic nucleus (PVT), as a crucial brain region for mood regulation, with significant differences in BD patients compared to controls. Authors report that thalamus samples from the BD patients they studied exhibited a 50% reduction in PVT neurons, which also showed notable downregulation in genes related to synaptic function and ion channel regulation. Additionally, disruptions in interactions between PVT neurons and microglia, particularly in AMPA receptor signaling pathways, were observed. These findings suggest that targeting AMPA signaling in PVT neurons could pave the way for innovative treatments and diagnostics aimed at improving outcomes for BD patients.

This paper has several strengths: Well written, addresses important questions, focuses on an understudied brain region, uses mostly state-of-the-art methods.

The biggest concern has to do with overinterpretation of results, with too little consideration for confounding and other limitations. There is also insufficient methodological detail in several areas.

Concerns in order of importance:

- Anatomical definition of PVT
- CALB1 and CALB2 were used to define PVT, but they are not exclusive to PVT; CALB1 is broadly expressed in all 'matrix' nuclei; CALB2 is expressed in all midline and all intralaminar nuclei
- Needs more detailed description (ideally with illustration) how the "medial thalamus" was dissected; e.g. more anterior part bordering anterior thalamic nuclei or more posterior part bordering mediodorsal thalamus
- Gao et al. and Shima et al. have identified major differences within PVT, especially along anterior-posterior axis; the author only included a general scoring of cells relative to these mouse datasets, not specific to PVT subdivisions
- Consideration of alternative explanations for differences in cell type composition
- Dissection likely influences which excitatory populations were captured. How are cell populations represented across the 40 individuals?
- Spatial transcriptomics shows Exc_PVALB3, but no Exc_PVALB1/2. Why?
- Is neuronal loss in BD limited to PVT (as defined by Exc_CALB cluster) or only detectable there due to more consistent dissection of the midline?
- Overinterpretation of mechanistic link between excitatory neurons and microglia
- SHISA9 and SYNDIG1 are both AMPAR auxiliary subunits, but an interaction between them is not known; SYNDIG1 encodes a transmembrane protein that is known to interact with AMPAR in neurons. Its function in microglia is not known. More evidence is needed of any claimed interactions in microglia.
- Discussion would benefit from a more nuanced handling of the data, given its limitations, and more attention to areas where evidence is less clear and multiple conclusions are possible, e.g.,
- Findings in PVT may not apply to the rest of the thalamus
- Avoid anthropomorphic statements such as "Mice with recurrent depressive episodes"
- Claims about high-frequency firing and role of KCNQ3 in potential response to high firing are too speculative; citations would be needed to support such a claim for this gene
- Lack of relevance of thalamic organoid integration analysis
- Authors did not sample the entire thalamus to claim that organoids recapitulate properties of PVT
- This analysis is not really related to the rest of the paper
- Insufficient methodological detail on Quality Control
- Authors should describe QC and filtering process in more detail
- No vascular or ependymal cells seen in the final annotated cell types: were they filtered due to low quality or not included because too few cells were represented?
- Insufficient methodological detail on anchor-based integration
- Was CCA or RPCA approach used, and why?
- Insufficient exploration of confounding variables for DGE (and cellular composition)
- Why were the covariates selected as described? The authors did not appear to have investigated other quality-related variables and explore variance they explained.
 - o Defining BD risk genes
 - o Reliance on MAGMA-based gene level scores from GWAS has significant limitations
 - o What about genes implicated by rare lof alleles in exome studies?
 - o Developmental origin of thalamic inhibitory neurons
 - o Only MGE, CGE, and LGE were considered, but it is not convincing that the observed thalamic inhibitory neurons were derived from LGE
 - o Existing literature points to midbrain-derived (SOX14+) inhibitory neurons
 - o Comparison with Siletti et al. is unclear, esp. regarding sample subject variation

Reviewer #4

(Remarks to the Author)

This manuscript investigates cellular and molecular alterations in bipolar disorder (BD) through single-nucleus RNA-seq (snRNA-seq) on postmortem brain tissue from BD patients and controls, focusing on the medial thalamus and frontal cortex. The authors first identify a subpopulation of thalamic excitatory neurons that represent neurons of the paraventricular thalamic nucleus (PVT), and provide a detailed molecular and spatial characterization of this cell population. Subsequent analyses reveal changes in cell type compositions and gene expression in BD. Intriguingly, these changes are more substantial in the thalamus compared to the cortex, highlighting the importance of this region and PVT neurons in particular.

Overall, this study makes important contributions to our understanding of both the cytoarchitecture of the human PVT and alterations in BD at the cellular compositional and transcriptional levels. In addition, the analyses of the snRNA-seq data are thorough and I commend the authors for their careful assessments of QC metrics and the robustness of their results. There are several considerations that if addressed would strengthen the study.

Major comments:

- (1) Dysregulation of circadian rhythms is a feature of BD. Were any associations observed between the transcriptional changes in BD and patients' time of death?
- (2) The gene set enrichment analyses in this study use terms from the Gene Ontology (GO) database. Given that synaptic dysfunction is implicated in BD pathology, I recommend the authors also perform these analyses using the curated gene sets from the SynGO database (Koopmans et al. 2019), which were also used in O'Connell et al. (2025).
- (3) The cohort includes both BDI and BDII subtype cases. Is the dataset sufficiently powered to stratify analyses by subtype? I understand that the sample size of the BDII cases may preclude such analyses; if so, it would be worth mentioning this in the limitations section of the Discussion (together with sex differences in lines 449-451).
- (4) In Fig. 8a, downregulated genes in Tha_ExN_CALB are enriched in SNP-based genomic risk of all 3 phenotypes tested. Is this observed for other psychiatric disorders (such as post-traumatic stress disorder, attention deficit-hyperactivity disorder, borderline personality disorder, and autism spectrum disorder)?
- (5) In addition to the analyses of common variant-based BD genomic risk, it would also be helpful to comment on how these findings relate to the rare loss-of-function variants recently described in Thorgeirsson et al. (2025). For example, which cell types in thalamus and cortex express HECTD2 and AKAP11, and how does their expression in BD patients compare to controls?
- (6) While the data that support potential disruptions in PVT-microglia trans-cellular interactions in BD are compelling (MP7, Figs. 5b-c), it was unclear to me why the authors focused on SHISA9 in PVT neurons as a representative MP7 gene throughout the manuscript. Although SHISA9 is part of the core PPI network of BD-associated downregulated genes in PVT neurons (Fig. 5d), it appears to be expressed in only a small fraction of Tha_ExN_CALB (PVT) neurons, while is expressed in most Tha_ExN_PVALB_1 neurons (Figs. 5d and 3a). I suggest that the authors either characterize this SHISA9-positive subpopulation of PVT neurons further (for example, are these detected in all thalamic samples?), or reduce/remove references to SHISA9 in PVT neurons throughout the text. These include parts of the section on "Disturbed interaction between PVT neurons and microglia in BD" (lines 251-284), the Discussion (lines 415-417), as well as the following sentence in the abstract: "This interaction was further exemplified by the concurrent downregulation of SYNDIG1 in microglia and SHISA9 in PVT neurons, both of which are involved in AMPA receptor signaling pathways" (top of pg. 3, line numbers not available).

Minor comments:

- (1) Please provide details on the cohort in the Methods, including information on how individuals were selected for inclusion and the approval of the study protocol. Details are in the Reporting Summary, but must also be provided in the manuscript.
- (2) The final two paragraphs of the Discussion appear to be redundant.

Version 1:

Reviewer comments:

Reviewer #1

(Remarks to the Author)

All my comments have been thoroughly addressed, the authors have done an excellent job during the revision, and I look forward to seeing this manuscript published.

Reviewer #2

(Remarks to the Author)

The authors have done an impressive job in exhaustively addressing the reviewers comments and revising their manuscript accordingly. While the work is now substantially improved, a few minor concerns remain:

Authors have provided more information on the dissection and integration with mouse PVT. Suggest authors avoid referring to CALB1 and CALB2 as "markers for PVT" since particularly CALB1 is not very specific. More heterogeneity may exist within human PVT than these markers suggest.

The cell type proportions are now more comprehensively assessed, including removal of outlier samples. Suggest the authors discuss whether neuronal loss extends beyond PVT; the scCODA results seem more consistent with a global reduction across thalamic excitatory neurons. Could the absence of PVALB_2 in CT06 be due to dissection variability?

While authors have addressed the previous use of anthropomorphic and superlative language, they still tend to read too much into individual differentially expressed genes. It is appropriate to point out potential relevance, but the proposed mechanistic speculations about individual DEGs appear to be exaggerated. For example, SYNDIG1's role in microglia and SHISA9 interaction is still overinterpreted; the authors should either cut or tone such statements down further and not refer to SYNDIG1 as an AMPA receptor auxiliary subunit since it was shown to be a more general regulator of excitatory synapse number (e.g., 10.1371/journal.pone.0066171); SHISA9 does seem to be an AMPA receptor auxiliary subunit in neurons (e.g., 10.1016/j.neuron.2014.07.004), but it is not clear that SHISA9 interacts with SYNDIG1 beyond regulation of excitatory synapse number.

Authors have addressed the error concerning inclusion of non-thalamic samples. While the authors refer to HBCA data as consisting of 20 samples, it should be made clear that those samples were derived from only 2-3 individual donors.

Reviewer #4

(Remarks to the Author)

Nishioka and colleagues have prepared a thoughtful revision, incorporating additional analyses and experimental validation of their PVT neuron findings by RNAscope, along with a more nuanced interpretation of their results. The authors have thoroughly addressed my comments on their initial submission. I have no further comments.

Point-by-point response

We would like to express our sincere gratitude to the reviewers for their encouraging and constructive comments. Below is our point-by-point response to each reviewer's specific feedback. The reviewers' comments are presented in *blue italics*. The revised sections of our manuscript are highlighted in *red*. The rectangles indicate the revisions made to the manuscript. The references and tables have been renumbered to reflect the changes in the revised submission.

We list the Supplementary materials as follows. We have deposited our count data to Gene Expression Omnibus (GEO) with accession number GSE306819. At this moment, only reviewers and editors can access these data with the following reviewer token and access URL.

Supplementary Tables

Supplementary Table 1. Sample information and sequencing metrics

Supplementary Table 2. GO enrichment analysis by GSEA.

Supplementary Table 3. Compositional changes in BD.

Supplementary Table 4. Differentially expressed genes in BD.

Supplementary Table 5. Robust DEG-GOs in BD (FDR < 0.05).

Supplementary Table 6. Downregulated genes in Tha_ExN_CALB and Tha_Micro_1 with biological interpretability

Supplementary Figures

Supplementary Figure 1. Cell-type annotation of thalamic and cortical cell clusters.

Supplementary Figure 2. Characterization of thalamic excitatory neurons and PVT neurons.

Supplementary Figure 3. Cell-type annotation in the thalamus.

Supplementary Figure 4. Co-expression modules in PVT neurons.

Supplementary Figure 5. Thalamic meta-gene programs estimated by geneNMF.

Supplementary Figure 6. Cell-type annotation in the cortex.

Supplementary Figure 7. Co-expression modules in cortical excitatory neurons.

Supplementary Figure 8. HECTD2 and AKAP11 expression and thalamus-cortex meta-gene programs estimated by geneNMF.

Supplementary Figure 9. Thalamus-cortex communication estimated by LIANA.

Supplementary Figure 10. Sample status and anatomical dissection.

Supplementary Figure 11. Quality control (QC) for cell clustering.

Supplementary Figure 12. CellBender-based ambient RNA decontamination.

Supplementary Figure 13. NeuCyto clusters and additional spatial deconvolution.

Supplementary Figure 14. Quality control (QC) for cell clustering and comparative analysis.

Supplementary Figure 15. sccomp analyses incorporating PMI as an additional covariate.

Supplementary Figure 16. DEG analyses incorporating PMI as an additional covariate.

Supplementary Data

01.DataS01.ExN_Tha_specific_genes.txt

02.DataS02.InN_Tha_specific_genes.txt

03.DataS03.PVT_Tha_specific_genes.txt

04.DataS04.scCODA.txt
05.DataS05.IHC_count_data.txt
06.DataS06.IHC_GLM.txt
07.DataS07.DEG_GSEA.txt
08.DataS08.PVT_module_GSEA_top10_GOs.txt
09.DataS09.MAGMA_PsychiatricDisorders.txt
10.DataS10.medication_signatures.txt
11.DataS11.HBCA.profiles.txt
12.DataS12.Celltype_markers.txt
13.DataS13.thalamus.exp_seq_batches.cellcomposition.txt
14.DataS14.cortex.exp_seq_batches.cellcomposition.txt
15.DataS15.CompositionalAnalysis.GLM.AIC.txt
16.DataS16.CompositionalAnalysis.GLM.Anova_ChiSq.txt
17.DataS17.PutativeBDriskgenes.txt
18.DataS18.Scomp_PMI.txt
19.DataS19.DEG_by_other_covariates.txt
20.DataS20.DEG_Dx_withPMI.txt
21.DataS21.DEG-GO_Dx_withPMI.txt
22.DataS22.DEG_GSEA_withPMI.txt
23.DataS23.lowPMI.DEG.txt
24.DataS24.lowPMI.GSEA.txt
25.DataS25.Mediation_signatures.references.txt

External Data (figshare)

Reviewer access: <https://figshare.com/s/dba4d20d4d7d02a67f9f>

BDsnRNAseq.thalamus.allclusters.logLM_Anova_CR01.model_comparison.txt.gz

BDsnRNAseq.cortex.allclusters.logLM_Anova_CR01.model_comparison.txt.gz

CellBender.output_pdf.zip

CellBender.removedRNA.zip

Repository data (GEO/SRA)

GEO: GSE306819 (reviewer token: izenmomqlhihpqd)

SRA: PRJNA1309835 (reviewer access:
<https://dataview.ncbi.nlm.nih.gov/object/PRJNA1309835?reviewer=t8fe31h5nf0gtpj229q7a66d02>)

Review Materials

ThaFrC.CTRL.merge.harmony.QC.IDrenamed.rds (Seurat Object)

thalamus.HBCAmerge.harmony.QC.IDrenamed.rds (Seurat Object)

thalamus.merge.harmony.QC.IDrenamed.rds (Seurat Object)

cortx.merge.harmony.QC.IDrenamed.rds (Seurat Object)

Dissection_PVT.wmv (video)

Reviewer #1

Nishioka et al. submit an interesting manuscript that investigates postmortem brain samples from bipolar disorder (BD) patients using single-nucleus RNA-seq. The authors analyzed a cohort of >80 samples of the prefrontal cortex and thalamus from ~40 subjects to identify new insights into brain dysfunction in BD. The authors showed that gene expression and cell composition changes are predominant in the thalamus, and much lower in the cortex. This led them to focus most of their analyses on the paraventricular thalamic nucleus. Subsequent analyses showed that neurons of the paraventricular thalamic nucleus had large decrease in cell number in BD and exhibited the largest changes in gene expression. In particular, genes involved in synaptic processes were downregulated and few dysregulated hub genes were proposed. Using tools predicting cell-cell interaction changes, the authors proposed that dysfunction of microglia might mediate impairment of synaptic function in BD paraventricular thalamic nucleus neurons. Overall, the study is novel for the field and provides good insights into specialization of thalamic excitatory neurons and their dysfunction in BD. Nevertheless, few major issues, mainly concerning data analysis, should be addressed before accepting the publication.

We thank the reviewer for the thoughtful and encouraging assessment of our manuscript and for recognizing the novelty and significance of our findings on thalamic excitatory neuron dysfunction in BD. We appreciate the constructive feedback on data analysis and have addressed all major points in the revised manuscript through methodological clarifications, additional analyses, and new supplementary materials. We believe these revisions have strengthened the rigor and clarity of our study.

Major issues:

1. Covariate control needs significant improvement.

We appreciate the reviewer's insightful comment and apologize for omitting the detailed rationale for covariate selection. In response, we have now added a comprehensive description of the covariate selection process and thoroughly revised the relevant section in the Methods.

Covariate Selection Strategy

Initially, we considered a wide range of potential covariates: diagnosis (Dx), Sex, Age, Postmortem interval (PMI), brain pH (pH), race, cause of death, refrigerator delay, substances at death, experimental/sequencing batch. Given our modest sample size (N = 41), covariates should be selected to minimize the risk of overfitting and multicollinearity.

- Race: The cohort was heavily biased toward Caucasian individuals, with only one Asian sample. Thus, race was excluded from statistical modeling due to lack of variability.
- Cause of death: A significant imbalance was observed, with suicide overrepresented among cases (Fisher's exact test, $P = 2.33 \times 10^{-8}$). To reduce multicollinearity, this covariate was excluded.
- Refrigerator delay (RD): RD contained three missing values, which is substantial for a

sample of 41. Furthermore, RD was significantly correlated with PMI (Pearson's correlation, $P = 0.0181$). We prioritized PMI as a relatively more relevant potential covariate.

- Medication: Due to the wide heterogeneity of substances at death and missing information in several donors, subgrouping was not feasible with the available sample size. The details are described later.
- Experimental and sequencing batches: We had 21 batches for nuclei isolation and 16 sequencing batches in thalamic analysis. Modeling batch effects failed due to the large number of batches relative to sample size ($N = 41$). The details are described later.

The remaining candidate covariates were Dx, sex, age, PMI, and pH. Among them, pH was particularly correlated with Dx (GLM $P = 0.0074$; see revised **Supplementary Fig. 14b**), suggesting potential collinearity. Nonetheless, we tested models with all five covariates (Dx, Sex, Age, PMI, and pH), as these are commonly used in postmortem brain studies.

Compositional Analysis

For the compositional analysis, model fit was evaluated using Akaike Information Criterion (AIC) from generalized linear models (GLM) employing negative binomial distribution (glm.nb, MASS package), given that the scomp package does not provide scores indicating model fit like AIC. We compared multiple candidate models and ranked them by summed AIC rankings (rankSum) across both thalamic and cortical datasets. The thalamic microglia cluster was excluded in this GLM-based calculation because the cell count distribution of this cluster did not fit well with the negative binomial distribution.

The model including Dx, age, and sex (Dx + Age + Sex) exhibited the lowest AIC by rankSum, indicating the best overall fit (**Supplementary Data 15**). The contributions of PMI and pH differed between the thalamic and cortical analyses. We selected PMI as the secondary covariate because pH showed a stronger correlation with diagnosis and represents a less direct measure of tissue integrity for both the thalamus and cortex (as described in the original manuscript, Line 672). Based on these findings, we selected Dx + Age + Sex as the primary covariate combination for compositional modeling, and Dx + Age + Sex + PMI as the secondary covariate combination.

In line with the reviewer's suggestion, we also assessed the relative contribution of each covariate using Type III likelihood ratio chi-square tests (Anova(type = 3) from the car package). The results are provided in **Supplementary Data 16**. Dx, Sex, and Age consistently demonstrated significant contributions to compositional variation.

Differential Expression Analysis

In the differentially expressed gene (DEG) analysis, we adopted the same covariate combination used in the compositional analysis. In response to the reviewer's comments, we calculated the AIC of GLMs for each gene's log-transformed expression to specifically evaluate model fit when incorporating PMI and/or pH, as MAST does not provide scores indicating model fit like AIC. Gene expression was aggregated at the specimen level and limited to genes expressed in at least 10% of cells within each cluster, consistent with the DEG criteria. This GLM-based modeling was performed solely to assess the impact of covariates; the DEG analysis itself was conducted using zlm in the MAST package.

Among the tested models, Dx + Age + Sex yielded the lowest total AIC across genes by clusters, followed by Dx + Age + Sex + PMI. This supports the use of Dx + Age + Sex as the primary covariate combination and Dx + Age + Sex + PMI as a secondary model. In addition, we assessed the relative contribution of each covariate using Type III chi-square statistics (Anova(type = 3) from the car package) for both models. As these data are too huge to submit for the journal web system, we deposited these data in figshare as **External Data**.

We have revised the Methods section to incorporate these details. The updated description enhances the clarity and transparency of our statistical approach.

Line 897 in the revised manuscript

Covariate selection in compositional analysis

Initially, we considered a wide range of potential covariates: diagnosis (Dx), sex, age, postmortem interval (PMI), brain pH (pH), race, cause of death, refrigerator delay, substances at death, experimental batch, and sequencing batch. Given our modest sample size ($N = 41$), covariates should be selected to minimize the risk of overfitting and multicollinearity. Race was excluded from statistical modeling because the cohort was heavily biased toward Caucasian individuals, with only one Asian sample. Cause of death was excluded to reduce multicollinearity, because it exhibited a significant imbalance with suicide overrepresented among BD cases (two-sided Fisher's exact test, $P = 2.33 \times 10^{-8}$). Refrigerator delay contained three missing values, and RD was correlated with PMI (Pearson's correlation, $P = 0.0181$). We prioritized PMI as a relatively more relevant potential covariate. Due to the wide heterogeneity of substances at death and missing information in several donors, subgrouping based on substances at death was not feasible in our dataset. Regarding experimental and sequencing batches, we had 21 batches for nuclei isolation and 16 sequencing batches in thalamic analysis. Modeling batch effects was unsuccessful due to the large number of batches relative to the sample size ($N = 41$). Although experimental and sequencing batches could not be incorporated into the subsequent statistical modeling due to their high dimensionality relative to sample size, we confirmed that the identified cell clusters were broadly and consistently represented across batches (**Supplementary Fig. 14a, Supplementary Data 13 and 14**).

The remaining candidate covariates were Dx, sex, age, PMI, and pH. Among them, pH values obtained from cerebellum samples, may not directly represent the specimen conditions of the thalamus and frontal cortex; given probable regional variability, we treated pH as an indirect indicator for this study. Notably, pH and PMI were associated with Dx, with pH showing a stronger signal (**Supplementary Fig. 14b-c**). These associations likely reflect differences in cause of death, as BD samples predominantly originated from individuals who died by suicide, unlike control samples. **Nonetheless, we tested statistical models with all five covariates (Dx, Sex, Age, PMI, and pH), as commonly used covariates in postmortem brain studies.**

For the compositional analysis, model fit was evaluated using Akaike Information Criterion (AIC) from generalized linear models (GLM) employing negative binomial distribution (glm.nb in MASS [v7.3-60] package), because the sccomp and scCODA packages do not provide scores indicating model fit like AIC. In the GLM, total nuclei counts were normalized to 10,000, with each cluster's count rounded to integers to reduce variability in total nuclei counts across

individuals. We compared multiple candidate models and ranked them by summed AIC rankings across both thalamic and cortical datasets (see rankSum in **Supplementary Data 15**). The model including Dx, age, and sex (Dx + Age + Sex) showed the lowest overall AIC, indicating the best model fit. While the contributions of PMI and pH differed between the thalamic and cortical analyses, we selected PMI as the secondary covariate because pH showed a stronger correlation with diagnosis and represents a less direct measure of thalamus and cortex tissues. Based on these findings, we selected “Dx + Age + Sex” as the primary covariate combination for compositional modeling, and “Dx + Age + Sex + PMI” as the secondary covariate combination. The relative contribution of each covariate was assessed by Type III likelihood ratio chi-square tests (Anova(type = 3) from the car package, **Supplementary Data 16**).

As support for the above covariate selection, adding PMI did not increase the model fitness in the sccomp analysis, without changes of elpd_diff/se_diff > 5. More than five of elpd_diff/se_diff change is suggested for significant model improvement according to the developers’ guideline⁴².

Line 985 in the revised manuscript

Differentially expressed gene analysis

Differentially expressed genes (DEGs) were analyzed within each cell cluster in BD compared to controls using glm function in MAST⁴⁴ (v1.28.0), a statistical package for single-cell RNA-seq DEG analysis. The model included Dx, age, sex, and cngeneson (a covariate for cell quality) as fixed effects, and sample ID as a random effect. Sample-level covariates were selected based on the prior compositional analysis and further validated by evaluating AIC values from GLMs fitted to log-transformed gene expression values. In this GLM-based model to select covariates, gene expression was aggregated at the specimen level and restricted to genes expressed in at least 10% of cells within each cluster, consistent with the downstream DEG filtering criteria. These GLMs were used solely to assess covariate impact. Among the tested covariate combinations, “Dx + Age + Sex” yielded the lowest overall AIC across gene-cluster pairs, followed by “Dx + Age + Sex + PMI”, as determined by cumulative AIC rank sums (**External Data**, see **Data Availability**). These results support the use of “Dx + Age + Sex” as the primary model and “Dx + Age + Sex + PMI” as a secondary model. The relative contribution of each covariate was further quantified using Type III chi-square statistics (Anova(type = 3) from the car package), as reported in **External Data**. Following MAST guidelines, we incorporated cngeneson, a variable of scaled gene expression counts, to adjust for cell nuclei quality, where higher gene detection rates suggest better nuclei quality within the same cluster.

Supplementary Data

15.DataS15.CompositionalAnalysis.GLM.AIC.txt

Akaike information criterion (AIC) assessment for the GLM in compositional analysis.

16.DataS16.CompositionalAnalysis.GLM.Anova_ChiSq.txt

Analysis of variance (ANOVA) Chi-square assessment for the GLM in compositional analysis.

External Data (figshare)

BDsnRNAseq.thalamus.allclusters.logLM_Anova_CR01.model_comparison.txt.gz

AIC and ANOVA Chi-square assessments for covariate combinations in the GLM of log-normalized summed gene expression in each thalamic cluster.

BDsnRNAseq.cortex.allclusters.logLM_Anova_CR01.model_comparison.txt.gz

AIC and ANOVA Chi-square assessments for covariate combinations in the GLM of log-normalized summed gene expression in each cortical cluster.

a. There have been multiple batches of sequencing, however, it is not described how many batches, how many samples per batch, etc. Given that sequencing batch effect is one of the major confounders in snRNA-seq data, the authors should provide thorough analysis of sequencing batch impact on their clustering, cell type annotation, differential gene expression and cell composition. Supplementary Table 1 should also contain information regarding sequencing batch.

We appreciate the reviewer's emphasis on addressing sequencing batch effects and apologize for the omission of these details in the original manuscript. While we attempted to evaluate the potential influence of experimental and sequencing batches for differential gene expression and cell composition, the number of batches was too high to allow robust statistical modeling. As shown in the revised **Supplementary Table 1**, the thalamic analysis involved 21 batches for nuclei isolation and 16 sequencing batches, including salvage sequencing. As a result, GLM-based approaches failed to estimate the effects of these batch variables. Similarly, sccomp and scCODA did not perform when incorporating these batch covariates, and the zlm function in MAST either failed to converge or arbitrarily dropped variables, rendering estimation of batch effects infeasible.

To mitigate experimental variability, we limited nuclei isolation and 10x library preparation to a single experimenter and consistently processed case-control pairs in parallel. We recognized that nuclei isolation and library preparation were likely sources of bias and therefore prioritized strict procedural consistency at these stages. Due to the limited throughput of our sequencing platform (Illumina NextSeq 2000), which allowed only at most 3-4 libraries per run using the P3 reagent (as the P4 reagent was not available at the time), the dataset comprises many sequencing batches, including additional salvage runs to obtain sufficient read count per nuclei. Although pooling and sequencing all libraries together on a NovaSeq 6000 or others would have been ideal for batch control, we opted against this approach due to the pace of our library preparation and concerns about potential degradation during prolonged storage. In hindsight, a NovaSeq-based approach could have improved batch uniformity, although it would not have addressed upstream variability introduced during nuclei isolation, a process inherently difficult to standardize due to its low-throughput, manual nature.

Given the limitations in incorporating batch variables into our statistical models, we instead evaluated potential biases in cell clustering and annotation across experimental and sequencing batches. As shown in **Supplementary Fig. 14a** and **Supplementary Data 13/14**, all identified cell types were well represented across nearly all batches, indicating that batch effects did not drive cell-type clustering.

In addition to revising the covariate selection strategy (as described above), we have now included: Cell-type representation per experimental and sequencing batch, a detailed explanation of our experimental design aimed at minimizing batch bias, and a discussion of batch effects as a limitation of the study.

Line 618 in the revised manuscript

In this study, numerous experimental and sequencing batches were required due to limited resources. Future work would benefit from automated library preparation and concurrent sequencing of pooled libraries on high-throughput platforms to further control batch effects.

Line 712 in the revised manuscript

Single-nucleus RNA-sequencing (snRNA-seq)

Nuclei isolated from the medial thalamus and frontal cortex underwent library preparation using the 10X Chromium 3' RNA v3.1 dual index protocol, following the manufacturer's instructions (10x Genomics, Pleasanton, USA). The target recovery rate was set at 5,000 nuclei, with library amplification achieved through 14 PCR cycles. The snRNA-seq libraries were sequenced on the NextSeq2000 platform (Illumina, San Diego, USA) in accordance with the protocols provided by Illumina and 10X Genomics. To minimize experimental bias, 10x Genomics library preparation was performed exclusively by the same experimenter (KI), with all case-control pairs processed in parallel. Due to the limitations of the NextSeq 2000 platform, which could accommodate only a few single-nucleus RNA-seq libraries per run, the dataset spans multiple sequencing batches, including several salvage runs needed to achieve sufficient read count per nuclei.

Line 908 in the revised manuscript

Regarding experimental and sequencing batches, we had 21 batches for nuclei isolation and 16 sequencing batches in thalamic analysis. Modeling batch effects was unsuccessful due to the large number of batches relative to the sample size ($N = 41$). Although experimental and sequencing batches could not be incorporated into the subsequent statistical modeling due to their high dimensionality relative to sample size, we confirmed that the identified cell clusters were broadly and consistently represented across batches (Supplementary Fig. 14a, Supplementary Data 13 and 14).

Supplementary Figure 14. Quality control (QC) for cell clustering and comparative analysis.

a) Proportions of thalamic and cortical cell clusters across experimental batches.

Supplementary Table 1 (abridged)

Add experiment and sequencing batch information.

Sequencing metrics										Frontal Cortex snRNA-seq data before QC	
Individual ID	Medial Thalamus snRNA-seq data before QC	Experimental batch (date)	Sequence batch (date)	Number of Reads	Estimated Number of Cells	Mean Reads per Cell	Median UMI Counts per Cell	Median Genes per Cell	Total Genes Detected	Experimental batch (date)	Sequence batch (date)
BD01	2023.01.18	2023.02.01		536,394,116	4,565	117,501	7,977	3,366	32,240	2024.01.17	2024.02.09, 2024.04.10
BD02	2023.01.25	2023.02.01		408,852,506	6,755	60,526	6,526	2,918	32,444	2024.01.24	2024.02.09
BD03	2022.11.16	2022.11.22		355,123,785	5,616	65,571	5,428	2,750	32,574	2023.10.18	2023.10.27
BD04	2022.11.30	2023.01.06		443,680,194	6,074	73,046	6,048	2,796	32,801	2023.12.20	2024.01.11
BD05	2022.10.12	2022.10.25		291,489,151	5,900	52,998	7,078	2,921	32,082	2023.07.26	2023.08.16, 2024.03.29
BD06	2022.11.02	2022.11.24		295,121,742	5,822	50,691	6,158	2,796	32,534	2023.11.29	2023.12.27
BD07	2022.12.14	2023.01.06, 2024.01.19		726,083,267	11,699	62,064	3,947	2,209	33,011	2024.01.31	2024.02.20
BD08	2023.03.08	2023.04.06		343,189,318	4,554	75,195	6,274	2,770	31,940	2023.06.28	2023.07.05
BD09	2023.03.15	2023.04.06		312,728,159	3,354	93,241	7,062	2,967	31,152	2024.03.13	2024.03.29, 2024.04.10
BD10	2022.09.28	2022.10.11		338,263,339	5,795	58,372	5,432	2,536	31,781	2023.12.13	2024.01.11
BD11	2022.09.14	2022.09.21		268,382,353	1,990	134,866	3,404	1,693	30,159	2024.02.07	2024.03.29
BD12	2022.10.26	2022.11.24		247,944,212	3,502	70,801	5,672	2,666	31,531	2023.10.04	2023.10.27
BD13	2023.03.01	2023.03.07		302,622,446	5,204	58,152	4,857	2,184	31,536	2023.06.21	2023.07.05, 2024.04.10
BD14	2022.10.19	2022.10.25		408,788,669	4,770	85,700	6,271	2,786	32,487	2023.11.08	2023.12.27
BD15	2022.10.05	2022.10.11		296,375,442	2,951	100,432	2,994	1,750	31,117	2024.02.28	2024.03.29
BD16	2022.12.07	2023.01.06		503,252,045	3,440	146,294	4,898	2,402	32,410	2023.09.27	2023.10.10
BD17	2023.02.28	2023.03.08		375,224,628	3,339	112,376	4,048	1,924	30,008	2023.08.16	2023.08.31
BD18	2023.05.11	2023.02.15		470,911,524	5,094	92,444	7,994	3,147	32,834	2024.03.06	2024.03.26
BD19	2022.12.21	2023.01.10		526,582,775	6,032	96,304	6,371	2,932	33,005	2023.08.30	2023.10.10
BD20	2023.02.15	2023.03.08		316,704,401	3,005	105,392	5,804	2,614	30,174	2023.08.09	2023.08.31, 2024.04.10
BD21	2023.02.08	2023.03.07		315,948,313	3,469	91,078	5,759	2,708	31,196	2023.08.02	2023.08.16, 2024.03.26
CT01	2023.02.15	2023.03.08		357,136,667	3,757	95,059	6,454	2,894	31,335	2023.07.26	2023.08.16, 2024.03.29
CT02	2022.10.26	2022.11.22		303,580,536	4,697	74,098	6,189	2,867	32,410	2023.08.30	2023.10.10
CT03	2022.11.30	2023.01.16		280,331,512	4,506	62,219	3,935	1,906	30,712	2024.01.17	2024.02.09
CT04	2023.02.08	2023.03.07		370,361,482	2,660	139,234	5,802	2,606	30,791	2023.08.16	2023.08.31, 2024.04.10
CT05	2022.09.14	2022.09.21		267,773,954	3,342	80,124	4,580	2,282	31,785	2024.01.24	2024.02.09
CT06	2022.10.05	2022.10.11		328,601,156	5,359	61,318	4,690	2,311	32,243	2023.09.27	2023.10.10
CT07	2022.11.16	2022.11.22		340,267,380	5,687	66,890	5,507	2,653	32,350	2024.01.31	2024.02.20
CT08	2023.01.18	2023.02.01		383,971,900	5,368	71,530	6,018	2,730	32,330	2023.10.18	2023.10.27, 2024.04.10
CT09	2022.12.07	2023.01.10		416,944,185	4,374	95,323	6,042	2,860	32,609	2023.10.04	2023.10.27, 2024.04.10
CT10	2022.12.21	2023.03.16, 2024.01.19		347,275,217	3,319	104,632	7,373	3,176	32,301	2024.02.07	2024.02.20, 2024.04.10
CT11	2022.10.19	2022.10.25		382,229,632	2,620	145,889	6,985	2,894	31,654	2024.02.28	2024.03.26, 2024.04.10
CT12	2023.02.01	2023.03.07		299,220,986	2,948	101,500	5,258	2,556	31,080	2023.06.28	2023.07.05
CT13	2023.01.25	2023.02.15		435,958,528	3,714	117,582	9,792	3,864	32,257	2023.11.29	2023.12.27
CT14	2022.11.02	2022.11.22		350,735,124	3,636	96,462	7,692	3,282	32,045	2023.11.08	2023.12.27
CT15	2023.02.28	2023.03.08		270,321,755	3,440	78,582	9,452	3,897	31,386	2023.06.21	2023.07.05
CT16	2023.01.11	2023.02.15		470,000,427	4,793	98,060	8,150	3,333	32,609	2023.12.20	2024.01.11, 2024.04.10
CT17	2023.03.15	2023.04.06		304,534,156	2,694	113,042	5,758	2,670	30,504	2023.08.09	2023.08.31
CT18	2022.12.14	2023.01.10		426,760,514	5,334	80,008	6,555	2,786	32,451	2024.03.06	2024.03.26, 2024.04.10
CT19	2022.09.28	2022.10.11		338,697,368	4,795	70,636	5,710	2,636	32,431	2023.12.13	2024.01.11
CT20	2022.10.12	2022.10.25, 2024.01.19		333,619,723	5,171	64,517	4,846	2,504	32,440	2023.08.02	2023.08.16, 2024.02.20
CT21 (not analyz	2023.03.08	2023.04.06		390,100,472	837	466,070	2,785	1,435	25,292	2024.03.13	NA

Supplementary Data

13.DataS13.thalamus.exp_seq_batches.cellcomposition.txt

Cell cluster composition by experimental and sequencing batches in the thalamus integration set.

14.DataS14.cortex.exp_seq_batches.cellcomposition.txt

Cell cluster composition by experimental and sequencing batches in the cortex integration set.

b. Medication is another large confounder in such analyses, since BD brains are usually exposed to a variety of drugs for a long period of life vs neurotypical brains. Potential impact of medication should be addressed, such as tests whether different medications within their BD cohort have impact on gene expression, e.g. by comparing subgroups of patient samples, comparing to other snRNA-seq studies, or to medication signatures identified in previous animal/patient studies.

We agree with the reviewer that medication exposure is an important potential confounder in postmortem studies of psychiatric disorders. Before addressing the scientific concerns, we would like to apologize for including individual-level medication data in the original submission. In accordance with Canadian regulations on donor privacy, such individual data should not have been included for publication. As Coroner data in Canada, where the brain tissue specimens were collected are nominal and public, there is a theoretical risk of deanonymizing the information by providing individual level data.

In the revised manuscript, we have instead included aggregated summary data, including a summary of medication exposure (new **Supplementary Table 1** and **Supplementary Fig. 10a**). Given that the “Last 3 months medication” information reflects prescribed records rather than actual adherence, we have instead used “Substance at death” to represent medication status for biological studies. In fact, the substances at death were not consistent with the record-based last 3 months medication.

Even if individual-level “Substance at death” data were to be used, the high heterogeneity across donors substantially limited the feasibility of subgroup analyses in our sample (N = 41). Among BD cases, no substance combination category included more than two donors, precluding meaningful group-level comparisons. We also attempted compositional and differential expression analyses using only donors without any reported medication or substance exposure; however, only one BD donor met this criterion, rendering such stratification infeasible. As a result, detailed analysis of medication effects was not possible within this dataset.

As an alternative approach, we cross-referenced our DEGs with transcriptomic signatures from psychotropic medication studies in both human and animal models by examining the overlap between our DEGs and transcriptional changes induced by target psychotropic medications/substances. The target medications/substances included those detected in more than 20% of BD donors and at least twice as frequently as in controls (antipsychotics, antidepressants, and alcohol), as well as medications commonly prescribed for BD (lithium and valproic acid). Because snRNA-seq data derived from the human thalamus were not available for these medications/substances, we instead used bulk RNA-seq data from postmortem brain samples, human induced neurons derived from induced pluripotent stem cells, or animal brain samples, selecting the most appropriate report for each medication/substance. While this comparison is not ideal, it represents the most feasible approach given current data constraints.

As a result, we observed minimal overlap in most of the combinations, suggesting that the observed transcriptional alterations are unlikely to be solely attributable to medication/substance effects, except for the effect of olanzapine on the upregulated genes in thalamic excitatory neurons. At least, downregulated genes in PVT neurons, our focus in this study, are unlikely to be caused by psychotropic medication/substances in this analysis.

We have added a summary of this analysis in the revised Discussion and Methods sections, a new **Fig.8g**, and **Supplementary Data 10**

Line 441 in the revised manuscript

Medication signature

Multiple medications and substances were detected in most postmortem brain samples at the time of death (**Supplementary Fig. 10a**). As subgrouping based on substances at death or lifetime medication history was not feasible in our dataset, we assessed the potential overlap between medication signatures and the DEGs identified in our study. In the four neuronal clusters with the largest numbers of DEGs, there was generally no substantial overlap with medication signatures for alcohol, antipsychotics, antidepressants, lithium, or valproic acid, except in a few instances (**Fig. 8g, Supplementary Data 10**). The upregulated genes in Tha_ExN_CALB significantly overlapped with genes upregulated by olanzapine in medium spiny neurons (MSNs)⁶² (Bonferroni-adjusted P = 0.00355, n of tests = 18 per cell type). A similar, though weaker, overlap was observed in Tha_ExN_PVALB_1 (adjusted P = 0.0772). Another finding was the overlap between downregulated genes in FrC_ExN_L6ITCar3 and

genes downregulated by valproic acid in forebrain organoids⁶³ (adjusted $P = 0.0573$). Although MSNs represent a distinct neuronal subtype from thalamic excitatory neurons, the consistent overlap between Tha_ExN_CALB/Tha_ExN_PVALB_1 and olanzapine-treated MSNs for upregulated genes suggests that these transcriptional changes could be influenced by lifetime medication exposure. However, no notable overlaps were observed for other combinations. Importantly, the downregulated genes in Tha_ExN_CALB—which were a central focus of our results—were unlikely to be attributable to medication or substance effects in this analysis.

Line 646 in the revised manuscript

Donors exhibited substantial heterogeneity in substances present at death, precluding stratification into subgroups of more than two individuals. Summary statistics of substances present at death are presented in **Supplementary Fig. 10a**.

Line 1155 in the original manuscript

Medication signature

We evaluated the potential influence of medications and substances on DEGs by assessing their overlap with established medication signatures. A medication signature was defined as the set of genes reported in previous studies to be differentially expressed in response to a given medication or substance. Target medications/substances included those detected in more than 20% of BD donors and at least twice as frequently as in controls (alcohol⁹⁸⁻¹⁰⁰, antipsychotics^{62,101}, and antidepressants¹⁰²; **Supplementary Fig. 10a**), as well as medications commonly prescribed for BD (lithium¹⁰³ and valproic acid⁶³). The reference data are summarized in **Supplementary Data 25**. Because thalamic snRNA-seq data were unavailable for these medications/substances, we instead used bulk RNA-seq data derived from postmortem brain tissue, human induced neurons generated from induced pluripotent stem cells, or animal brain samples, selecting the most appropriate publicly available dataset for each medication/substance. We used possible DEGs for this analysis to increase statistical power. This approach has inherent limitations due to brain region and cell-type differences, and the results should be interpreted as referential.

Figure 8. Comparative analysis between the thalamus and cortex.

g) Heatmap of P-values from hypergeometric tests for the overlap between DEGs in each cell type and previously reported DEGs induced by specific medications/substances (medication

signatures). Color indicates $-\log_{10}(P\text{-value})$ with Bonferroni-corrected significance threshold for each panel (0.05/18). SSRI, selective serotonin reuptake inhibitor; MSN, medium spiny neuron; Bulk, bulk RNA-seq data; Cau, caudate; FrCEXN, frontal cortex excitatory neuron; PFC, prefrontal cortex.

Supplementary Table 1 (abridged)

Supplementary Table 1. Sample information and sequencing metrics											
Sample information											
Group	Sex	Age*	PMI (h)*	pH*	Race	Cause of death	Refrigerator Delay (h)*	Subtype	Medication at time of death	Substance at time of death	
BD (N = 21)	14 males (66.7%)	44.3 ± 13.9	44.1 ± 17.5	6.54 ± 0.21	20 Caucasians 1 Asian	Suicide (17)	7.8 ± 5.0 with 3 missing values	Bipolar II disorder (9; 42.8%) Bipolar I disorder (5; 23.8%) Bipolar disorder without subtype information (7; 33.3%)	lithium (2; 9.5%) anticonvulsant (3; 14.3%) antipsychotics (5; 23.8%) antidepressant (7; 33.3%) benzodiazepines (2; 9.5%) opioid (5; 23.8%)	psychostimulant (3; 14.3%) alcohol (7; 33.3%) cannabinoid (2; 9.5%) cyanide (1; 4.8%)	
	7 females (33.3%)					Accidental (3) Natural (1)					
Control (N = 21)	14 males (66.7%)	52.5 ± 11.8	59.9 ± 21.7	6.37 ± 0.16	21 Caucasians	Accidental (7) Natural (12) Undetermined (2)	11.9 ± 9.2 with 0 missing values	-	anticonvulsant (1; 4.8%) antipsychotics (0) antidepressant (3; 14.3%) opioid (4; 19.0%)	psychostimulant (2; 9.5%) alcohol (1; 4.8%) cannabinoid (1; 4.8%) cyanide (0)	
	7 females (33.3%)										
P value	1	0.0472	0.0153	0.00866	1	2.33×10^{-8}	0.0986	-	-	-	
test type (two-sided)	Fisher exact test	Welch's t test	Welch's t test	Welch's t test	Fisher's exact test	Fisher's exact test	Welch's t test	-	-	-	

* Data are presented as the mean ± s.e.m.

Supplementary Figure 10. Sample status and anatomical dissection.

a) Number of donors with detected substances at the time of death among 20 BD cases and 19 controls, excluding those with missing data.

Supplementary Table 1

REMOVED: individual data including medication/substance information

c. Some samples that were analyzed have high PMI, >24hours, and there are many studies showing the impact of high PMI on the mRNA content, including benchmarking studies for RNA-seq experiments. The authors should perform a thorough analysis for potential confounding impact of PMI and if necessary exclude samples that are too different from low PMI samples and interfere with data analysis.

We thank the reviewer for raising this important point. Prior to analysis, we also considered PMI to be a critical covariate, particularly as the majority of our samples had relatively high PMI values (>24 hours). Somewhat unexpectedly, however, incorporating PMI into the

statistical models increased AIC values in both compositional and DEG analyses. Based on these model fit metrics, we selected Dx + Age + Sex as our primary covariate combination, as previously described. Nonetheless, given the known importance of PMI in postmortem studies and the fact that Dx + Age + Sex + PMI consistently ranked as the second-best model, we designated it as our secondary covariate set. Although we cannot provide a definitive explanation for this result, one possible interpretation is that the impact of PMI may be more pronounced in samples with shorter postmortem intervals (e.g., PMI < 10 hours), where enzymatic degradation and other rapid postmortem processes are most active.

In response to the reviewer's suggestion, we have added a more detailed assessment of the impact of Dx, Age, Sex, and PMI in our statistical models (**Supplementary Data 15/16, and External Data**). The External Data is provided via figshare due to its large file sizes. The effects of Age, Sex, and PMI on cell composition are summarized in **Supplementary Fig. 15b** and **Supplementary Data 18**. The DEG results associated with each of these covariates are newly listed in **Supplementary Data 19** (in addition to the original **Supplementary Fig. 16**), enabling evaluation of covariate-specific effects.

As shown in these data, PMI had no significant effect on cell composition in the scomp analysis, while Dx, Age, and Sex were associated with changes in several cell types. The scCODA models incorporating PMI failed to converge reliably. In DEG analysis, Age and Sex exhibited considerable effects on gene expression, while PMI had less impact. Notably, PMI had no observable effect on either cell composition or gene expression in the PVT neuron cluster (Tha_ExN_CALB), suggesting that our principal findings in PVT neurons are unlikely to be driven by PMI-related confounding.

To further confirm the robustness of our results, we performed an additional analysis restricted to samples with relatively lower PMI values (<50 hours; n = 21). In the context of our sample set, a PMI of <50 hours represents a comparatively short interval, although we are aware that this value is not considered as low in other studies. Nevertheless, both the compositional shifts and DEG patterns remained consistent with our original results, further supporting the validity of our conclusions.

While the observed effects of Age and Sex on brain transcriptomes are intriguing, a more detailed investigation is beyond the scope of the current study. We are currently pursuing this direction in a follow-up project.

Line 1088 in the revised manuscript

Extended analysis incorporating PMI as an additional covariate for robustness check

As described above, we extendedly analyzed compositional and transcriptional changes by incorporating a covariate of PMI into the main covariates (Dx, age, and sex) to check the robustness of our results. This extended analysis yielded results consistent with our initial findings (**Supplementary Fig. 15 and 16**). Specifically, we observed a compositional decrease of thalamic excitatory neurons in BD, including PVT neurons (FDR = 0.0115, FC = 0.582), and FrC_ExN_L6ITCar3 (FDR = 0.0257, FC = 0.578) (**Supplementary Fig. 15a, Supplementary Data 18**). Dx, sex, and age affected the cellular compositions, while PMI did not (**Supplementary Fig. 15b**). **The scCODA model did not perform reliably when PMI was included alongside Dx,**

age, and sex, yielding inclusion probabilities of 1.0 for all variables across all clusters, even if using total cell count normalization. This likely reflects overparameterization due to the number of covariates relative to our sample size.

In this extended analysis, Dx, sex, and age prominently affected cell composition and gene expression, while PMI less affected them (**Supplementary Fig. 15b** and **Supplementary Data 19**). The DEGs by age or PMI were defined by those with absolute value of FC > 1.1 for the period of two standard deviations in our dataset (i.e. 26.7 years for age, and 38.7 h for PMI). Notably, PMI showed no detectable impact on either cell composition or gene expression in the PVT neurons (Tha_ExN_CALB), suggesting our key findings in PVT neurons are unlikely to be confounded by PMI.

When incorporating PMI into the statistical models, the Dx-based DEG-GOs were again most observed in PVT neurons (**Supplementary Fig. 16a**). The DEGs in PVT neurons (**Supplementary Fig. 16b**, **Supplementary Data 20**) continued to exhibit enrichment in synaptic and neuronal GO terms with the same top GOs—transporter complex (GO:1990351) and monoatomic ion channel complex (GO:0034702)—including *SHISA9*, *CACNA1C*, and *KCNQ3* (**Supplementary Fig. 16c**, **Supplementary Data 21**). Again, the downregulated DEGs were linked to the co-expression module Excit-M4, while upregulated DEGs were linked to Excit-M11 and Excit-M9 (**Supplementary Fig. 16d**). Among the DEGs identified, 43 of 64 downregulated DEGs (67.2%), and 100 of 157 upregulated DEGs (63.7%) from the primary analysis were also classified as DEGs in this extended analysis (**Supplementary Data 20**). Notably, downregulated BD risk genes—*SHISA9*, *CACNA1C*, and *CTNN5*—remained detectable as downregulated DEGs and constituted core PPI networks in this extended analysis (**Supplementary Fig. 16e**). *SHISA9* was also detected as a downregulated DEGs in Tha_ExN_PVALB_1 in the extended analysis. The SNP-based BD genomic risk was again most enriched in the downregulated genes in PVT neurons ($P = 0.00419$), whereas upregulated genes included no BD risk genes despite a greater total number of DEGs (69 downregulated vs. 110 upregulated). The DEG-GOs in the frontal cortex cell clusters were virtually not observed, consistent with the primary analysis (**Supplementary Fig. 16f**). Finally, GSEA performed on DEGs from this extended analysis yielded results similar to those of the primary analysis (**Supplementary Data 22**), with downregulated genes in PVT neurons again enriched for synaptic and neuronal GO terms.

Line 1125 in the revised manuscript

Extended analysis to the samples with low PMI

While PMI had no apparent effect on our findings of PVT neurons, we analyzed the samples with relatively low PMI (< 50 h), consisting of 12 BD cases and 9 controls, to check the robustness of our findings for PVT neurons and thalamic microglia. The covariates were set as the same to the primary analysis: Dx, age, and sex. PVT neurons and thalamic excitatory neurons robustly exhibited compositional decreases (FC = 0.563, FDR = 0.0223 for PVT neurons; FC = 0.541, FDR = 0.0193 for thalamic excitatory neurons) by sccomp analysis. DEGs in PVT neurons were also consistent with the main analysis, showing comparable FCs (**Supplementary Data 23**). For instance, *KCNQ3*, *SHISA9*, and *CACNA1C* showed FC values of 0.755, 0.840, and 0.843, respectively. GSEA also yielded similar results, with the top enriched term being potassium ion transmembrane transporter activity (GO:0015079), including *KCNQ3* (**Supplementary Data 24**). In this analysis, we employed GSEA instead of DEG-based GO enrichment, as the smaller sample size (12 BD cases vs. 9 controls) reduced the statistical power and FDR control.

Supplementary Figure 15. scomp analyses incorporating PMI as an additional covariate.

a) Compositional changes of cell clusters in the thalamus (left) and the cortex (right), with log₂(Fold-change) on the x-axis and -log₁₀(FDR) on the y-axis. Circle size corresponds to the nuclei count within each cluster, and color represents major cell classes. b) Effect (c_effect) of each covariate on cell clusters, as estimated by scomp. The endpoints of the lines represent the lower and upper bounds of the c_effect estimate. Clusters with a false discovery rate (FDR) < 0.05 are indicated by red points and lines.

Supplementary Figure 16. DEG analyses incorporating PMI as an additional covariate.

a) DEG and DEG-GO count across thalamic and cortical clusters. b) Volcano plot of the genes expressed in Th_ExN_CALB, displaying $\log_2(\text{Fold-change})$ on the x-axis and DEG robustness score on the y-axis. c) Downregulated (left) and upregulated (right) robust DEG-GOs in the thalamus, plotted with fold-change on the x-axis and robustness score on the y-axis. Circle size reflects the DEG count within each GO, with colors indicating respective cell clusters. Among robust DEGs, only enriched GO terms among possible DEGs are displayed for fold-change calculation. d) Proportions of co-expression modules in downregulated and upregulated DEGs in Th_ExN_CALB. e) PPI network of downregulated possible DEGs in Th_ExN_CALB. Colored nodes indicate overlap with DEGs from the primary analysis. f) Upregulated robust DEG-GOs in the cortex, plotted with fold-change on the x-axis and robustness score on the y-axis. Circle size reflects the DEG count within each GO, with colors indicating respective cell clusters. No downregulated robust DEG-GO was observed in this analysis.

Supplementary Data

15.DataS15.CompositionalAnalysis.GLM.AIC.txt

Akaike information criterion (AIC) assessment for the GLM in compositional analysis.

16.DataS16.CompositionalAnalysis.GLM.Anova_ChiSq.txt

Analysis of variance (ANOVA) Chi-square assessment for the GLM in compositional analysis.

18.DataS18.Sccomp_PMI.txt

sccomp analysis including postmortem interval (PMI) as an additional covariate.

19.DataS19.DEG_by_other_covariates.txt

DEGs by sex, age, and PMI, using PMI as an additional covariate.

20.DataS20.DEG_Dx_withPMI.txt

DEGs by disease status, using PMI as an additional covariate.

21.DataS21.DEG-GO_Dx_withPMI.txt

Gene ontology (GO) enrichment of DEGs by disease status, using PMI as an additional covariate.

22.DataS22.DEG_GSEA_withPMI.txt

GSEA results for thalamic and cortical DEGs incorporating PMI as an additional covariate.

23.DataS23.lowPMI.DEG.txt

DEGs identified using only low-PMI samples.

24.DataS24.lowPMI.GSEA.txt

GSEA results using only low-PMI samples.

External Data (figshare).

BDsnRNAseq.thalamus.allclusters.logLM_Anova_CR01.model_comparison.txt.gz

AIC and ANOVA Chi-square assessments for covariate combinations in the GLM of log-normalized summed gene expression in each thalamic cluster.

BDsnRNAseq.cortex.allclusters.logLM_Anova_CR01.model_comparison.txt.gz

AIC and ANOVA Chi-square assessments for covariate combinations in the GLM of log-normalized summed gene expression in each cortical cluster.

d. Additionally, the authors should statistically analyze potential impact of all metadata variables, including age, sex, drug exposure etc.

We thank the reviewer for raising this important issue. As described earlier, we have added analyses to evaluate the potential impact of age, sex, and PMI. The effects of these covariates on cell composition are summarized in **Supplementary Table 3** and **Supplementary Data 16**, and their effects on gene expression are also detailed in **Supplementary Data 19**. As the files for the relative contribution of each covariate for gene expression are too large, we have prepared them in figshare as **External Data**.

Although we recognize the significance of medication as a potential confounder, the wide variety of substance use across samples and incomplete information made it infeasible to statistically model medication effects within our dataset. Instead, as noted above, we investigated the potential influence of pharmacological exposure by comparing our transcriptional findings to known expression signatures of major psychiatric medications. Please, refer to the above responses.

Line 935 in the revised manuscript

The relative contribution of each covariate was assessed by Type III likelihood ratio chi-square tests (Anova(type = 3) from the car package, **Supplementary Data 16**).

Line 997 in the revised manuscript

The relative contribution of each covariate was further quantified using Type III chi-square statistics (Anova(type = 3) from the car package), as reported in **External Data** (figshare).

Line 1099 in the revised manuscript

As described above, we extendedly analyzed compositional and transcriptional changes by incorporating a covariate of PMI into the main covariates (Dx, age, and sex) to check the robustness of our results. This extended analysis yielded results consistent with our initial findings (**Supplementary Fig. 15 and 16**). Specifically, we observed a compositional decrease of thalamic excitatory neurons in BD, including PVT neurons (FDR = 0.0115, FC = 0.582), and FrC_ExN_L6ITCar3 (FDR = 0.0257, FC = 0.578) (**Supplementary Fig. 15a, Supplementary Data 17**). Dx, sex, and age affected the cellular compositions, while PMI did not (**Supplementary Fig. 15b**).

Line 1101 in the revised manuscript

In this extended analysis, Dx, sex, and age prominently affected cell composition and gene expression, while PMI less affected them (**Supplementary Fig. 15b and Supplementary Data 19**). The DEGs by age or PMI were defined by those with absolute value of FC > 1.1 for the period of two standard deviations in our dataset (i.e. 26.7 years for age, and 38.7 h for PMI). Notably, PMI showed no detectable impact on either cell composition or gene expression in the PVT neurons (Tha_ExN_CALB), suggesting our key findings in PVT neurons are unlikely to be confounded by PMI.

Supplementary Table 3 (abridged)

Line 759 in the revised manuscript

We selected DecontX for ambient RNA removal in preference to SoupX⁸⁵, CellBender⁸⁶, or analysis performed without any ambient RNA decontamination package (no_decont_package) from the following reasons. SoupX and no_decont_package could not effectively distinguish NeuCyto clusters (details are described later) from oligodendrocytes in the UMAP projection, probably reflecting the effect of ambient RNA to clusters expressing a small number of genes (Supplementary Fig. 11d).

Supplementary Figure 11. Quality control (QC) for cell clustering.

d) UMAP projection displaying thalamic nuclei clusters processed with SoupX decontamination and without any ambient RNA decontamination package (no_decont_package).

b. Overall, given potential contamination with ambient RNA, more robust analysis is necessary.

We propose to use CellBender and provide the following data in Supplementary figures:

- *Sample-wise expression of ambient RNAs*
- *CellBender ELBO (test and train) plots*
- *CellBender cell calling curves displaying cell probabilities of droplets*

We appreciate this comment. CellBender has been reported as one of the most effective tools for ambient RNA removal, as demonstrated in Janssen et al. (Genome Biology 2023) and Caglayan et al. (Neuron 2022).

However, Caglayan et al. (“Neuronal ambient RNA contamination causes misinterpreted and masked cell types in brain single-nuclei datasets”, *Neuron* 110:4043-4056, 2022) indicated that neuronal cell clusters are difficult to decontaminate even with CellBender, as follows.

“Strikingly, CellBender did not reduce ambient RNA contamination from the neurons (Figures S6C and S6E). As expected, non-nuclear ambient RNA markers NRG1 and CHN1 levels were higher in more contaminated neuronal nuclei, whereas nuclear-retained MALAT1 levels were lower (Figures S6D and S6F). Contamination patterns were similar among the previously annotated neuronal subtypes of NSD1, indicating that ambient RNA contamination in neurons is a cell-type agnostic problem and, unlike glia, is not accounted for by CellBender (Figures S7A and S7B). Other ambient RNA contamination removal tools were also more effective in glia than neurons, indicating a general deficiency in the current methods to remove ambient RNA contamination from the dominant cell type in the tissue (Figures S7C and S7D).” (p.4049)

Thus, there is currently no definitive method for ambient RNA decontamination in snRNA-seq datasets focusing on neurons. In our analysis, we heuristically selected the most appropriate decontamination approach based on clustering results, acknowledging that there is no absolute ground truth for ambient RNA removal in our study.

We initially applied CellBender to our dataset, recognizing its potential advantages for ambient RNA removal. We have deposited “Sample-wise expression of ambient RNAs”, “CellBender ELBO (test and train) plots”, and “CellBender cell calling curves displaying cell probabilities of droplets” in **External Data**. Unexpectedly, it did not perform well with our thalamic samples. In several cases, CellBender appeared to fail in clearly distinguishing nuclei-containing droplets from empty droplets in several samples, as illustrated in **Supplementary Fig. 12a** and files in **External Data**.

The downstream integration of CellBender-processed data resulted in atypical clusters (Tha_ExN_PVALB_X and Tha_InN_X), which were annotated as putative excitatory or inhibitory neurons based on marker gene expression but exhibited abnormally low total RNA counts (**Supplementary Fig. 12b-g**). These atypical clusters are not interpreted as cytoplasmic components. In contrast, no such atypical clusters were observed in other pipelines using DecontX, SoupX, or no RNA decontamination packages, even if increasing cluster resolution. We interpreted these atypical clusters in the CellBender-based pipeline as likely artifacts introduced by miscalculation during ambient RNA removal.

This suboptimal performance of CellBender might stem from the presence of cytoplasmic fragment-containing droplets in our dataset. Based on the presence of NeuCyto clusters and probable cytoplasmic components in microscopic observation, we classify droplets into three categories:

- (A) droplets containing intact nuclei,
- (B) droplets containing cytoplasmic components without nuclei
- (C) droplets containing only liquid ambient RNA without nuclei or cytoplasmic components.

Although type B droplets are not ideal for nuclear transcriptomics, they seemed to exist in our samples and HBCA dataset, likely due to the fragility of postmortem human brain tissue—

particularly in thalamic samples. In fact, TapeStation QC during library preparation also indicated greater RNA degradation in thalamic than cortical tissue. To preserve nuclear integrity, we employed a gentler homogenization protocol using 0.05% Tween-20 instead of the more commonly used 0.1% Triton X-100. This adjustment improved RNA integrity, yielding higher RIN scores during pilot testing. Additionally, to avoid laser-induced RNA damage, we relied on centrifugation rather than fluorescence-activated cell sorting (FACS) for nuclei purification. While these protocols helped preserve RNA quality, they may also contribute to the retention of cytoplasmic fragments (type B droplets), which correspond to NeuCyto clusters in our analysis. While microscopic observations were limited to our samples, we noticed NeuCyto clusters in HBCA samples. This implies that cytoplasmic components may also exist in other snRNA-seq datasets derived from the human brain.

CellBender's underlying model assumes a binary classification between nuclei-containing (A) and empty (C) droplets. Consequently, type B droplets—particularly those enriched in neuronal cytoplasmic RNA—may be difficult to resolved to A or C. This difficulty could bias ambient RNA estimation and result in the removal of biologically meaningful transcripts. Although this provides a plausible explanation for CellBender's suboptimal performance, we acknowledge that the exact cause of the aberrant clusters remains uncertain.

Given this uncertainty at the time of initial submission, we did not include CellBender results in the original manuscript. Additionally, we had limited opportunities to optimize CellBender parameters due to the availability of minimal GPU resources. Processing a single snRNA-seq library sometimes required many hours on conventional CPUs, making parameter tuning impractical without adequate GPU resources.

In response to the reviewer's comments, we have now included these results in the revised version. While cytoplasmic fragments likely contributed to CellBender's limitations in our dataset, we acknowledge that this interpretation remains speculative and have therefore only minimally mentioned it in the main text.

Line 764 in the revised manuscript

CellBender was applied using the expected cell count estimated by Cell Ranger along with other default parameters. However, in several cases, CellBender appeared to fail in clearly distinguishing nuclei-containing droplets from empty droplets (**Supplementary Fig. 12a, External Data** [figshare]). CellBender-based clustering resulted in atypical clusters that expressed neuronal markers but exhibited extremely low total RNA counts (**Supplementary Figs. 12b-g**). These clusters, labeled as Tha_ExN_PVALB_X and Tha_InN_X did not correspond to any known thalamic cell types and were not consistent with cytoplasmic droplets, as determined by nuclear fraction and mitochondrial RNA content. Although a definitive ground truth for cell-type identity was not available, we interpreted these clusters as likely artifacts resulting from inappropriate ambient RNA removal, leading to biologically implausible transcriptomic profiles. In contrast, cell-type annotations were consistent across DecontX, SoupX, and the no_decontamination_package, and no unexplained anomalous clusters emerged under these approaches, even if increasing clustering resolution. While the exact cause of this phenomenon remains uncertain, we suspect that the presence of cytoplasmic components—discussed later and supported by NeuCyto clustering—may interfere with

CellBender's ability to accurately distinguish between nuclei-containing droplets and empty droplets.

Supplementary Figure 12. CellBender-based ambient RNA decontamination.

a) Representative plots illustrating unsuccessful cell probability inference by CellBender, with barcodes ranked by total UMI counts. b) UMAP projection of thalamic cell nuclei clustered following CellBender-based ambient RNA removal. c) Zoomed-in view of excitatory neuron clusters from panel b, highlighting Tha_ExN_PVALB_X using distinct coloring. d) Zoomed-in view of inhibitory neuron clusters from panel b, highlighting Tha_InN_X. e) Dot plot showing cell-type marker expression across thalamic clusters identified using CellBender. Dot size indicates the proportion of expressing nuclei, and color intensity represents average expression levels. f) Nuclear fractions and mitochondrial RNA fractions across CellBender-defined clusters. g) Number of detected genes per cluster across four RNA decontamination

methods: CellBender, DecontX, SoupX, and no decontamination. Red and blue triangles indicate Tha_ExN_PVALB_X and Tha_InN_X, respectively.

External Data

CellBender.output_pdf.zip

CellBender output PDF files for all samples.

CellBender.removedRNA.zip

CellBender removed RNA data for all samples.

In addition to the arguments for CellBender, we have revised the supplementary figures illustrating the possible presence of cytoplasmic fragments (presumably neuronal in origin).

Line 789 in the revised manuscript

Clusters designated as NeuCyto were considered neuronal cytoplasmic components from the following properties: low nuclear fraction, high content of mitochondrial RNA (**Supplementary Fig. 13a**), high expression of neuronal markers, virtually no expression of other glial markers (**Supplementary Fig. 1a, 3a, and 6a**), and distinctive clustering from probable empty droplets (**Supplementary Fig. 13b-c**). The probable empty droplets, designated as “Empty”, were defined as droplets filtered out by Cell Ranger with detected genes of 100 – 300. This range was set by QC procedure and barcode rank plot for UMI counts by Cell Ranger. NeuCyto droplets were robustly detected from 40/41 of our thalamic samples (occupying 1.46% of total cell nuclei) and 40/41 of our cortex samples (occupying 1.46% of total cell nuclei) as well as 20/20 of HBCA thalamic samples (occupying 1.66% of total cell nuclei). The clear separation of VGLUT+ and GAD+ NeuCyto clusters in the thalamus indicated that they were distinct cytoplasmic components (**Fig. 3a**). We removed NeuCyto clusters for the subsequent analysis as non-nuclei droplets but retained for UMAP visualization and dot plots as a technical reference. Indeed, in the final nuclei suspension, we observed objects stained with Nonyl Acridine Orange (NAO) but negative for Propidium Iodide (PI) and Hoechst 33342 (**Supplementary Fig. 13d**), likely representing mitochondria-rich, nucleus-free cytoplasmic fragments corresponding to the NeuCyto clusters.

Line 861 in the revised manuscript

For reference, we also mapped the location of thalamic NeuCyto clusters. The NeuCyto clusters within the thalamus could be further categorized into three sub-clusters based on marker expression (**Supplementary Fig. 13g**). Spatial deconvolution revealed that NeuCyto_VGLUT2_CALB was relatively localized in the PVT region, while NeuCyto_VGLUT2_PVALB was sparsely distributed across other areas (**Supplementary Fig. 13h**). NeuCyto_GAD clusters seemed relatively located in proximity to inhibitory neurons. The three NeuCyto sub-clusters likely represented neuronal cytoplasm from excitatory and inhibitory neurons.

Supplementary Figure 13. NeuCyto clusters and additional spatial deconvolution.

a) Nuclear fractions and mitochondrial RNA fractions across samples in the thalamic integration dataset (see Fig. 3a). b) UMAP projection illustrating “Empty” clusters (ambient RNA, 1,000 droplets per sample) and canonical cell nuclei clusters shown in Fig. 3a. c) UMAP projection of the same clusters as in panel b, with distinct coloring to highlight Tha_NeuCyto_VGLUT, Tha_NeuCyto_GAD, and Empty clusters. d) Microscopic images of cytoplasmic fragments (white arrowheads) in two representative spots. Cytoplasmic fragments are indicated by NAO staining (green) in the absence of PI or Hoechst 33342 signals. Cell nuclei appear magenta due to combined PI and Hoechst 33342 staining. e) CytoSPACE deconvolution based solely on our snRNA-seq dataset (excluding the HBCA dataset). f) CytoSPACE deconvolution using only the CT06 snRNA-seq dataset. g) UMAP projection

displaying three NeuCyto sub-clusters within the thalamus. h) Predicted spatial distribution of the three NeuCyto sub-clusters. Color intensity in the side bars reflects the estimated number of cells per spatial spot. For panels g and h, only control samples were used to prevent potential bias from BD-associated alterations in spatial deconvolution.

3. Data integration

The authors used iterative integration with Seurat and Harmony. There is no justification provided for why Seurat-only or Harmony-only integration are not enough. Simple comparison plots will help the reader to understand the rationale for choosing iterative integration.

We apologize for the lack of clarity in our original description. Our approach, referred to as “iterative integration” in the original manuscript, consists of the following two-step process:

1. We first merged the target data and performed Seurat normalization, scaling, and PCA using all nuclei, including doublets and low-represented clusters (i.e., clusters detected in <75% of samples). Clustering and doublet identification are then conducted on this dataset after Harmony batch-correction. At this stage, the scaled expression values still reflect contributions from doublets and potentially artifactual clusters.
2. After removing doublets and low-represented clusters, we re-normalized, re-scaled, and re-clustered the dataset using Seurat and performed Harmony batch-correction again. This second round ensures that scaling and downstream analyses are no longer biased by doublet-contaminated or artifactual clusters.

We adopted this two-step procedure because omitting the second round would allow doublets and low-represented clusters to influence scaling and clustering, potentially distorting batch correction and downstream analyses such as differential expression.

In retrospect, we acknowledge that the terms “iterative” and “integration” were ambiguous. Our intention was not to imply simultaneous use of both Seurat and Harmony batch correction methods. Rather, we used only Harmony for batch correction in both rounds, and the word “integration” referred to the consecutive processing of merging, normalization, scaling, PCA, and batch correction. To improve clarity, we have revised the terminology in the manuscript: “iterative integration” has been removed. A word “batch-correction” is now used to describe more specifically to describe how Harmony was used.

Line 741 in the revised manuscript

Integration of snRNA-seq data

Count data from Cell Ranger were decontaminated using the DecontX tool in the celda package⁸⁰ (v1.16.1) and subjected to multiplet identification via scDbfFinder⁸¹ (v1.14.0). These steps were performed for each count data in our and the HBCA⁶ dataset prior to integration. After decontamination and multiplet labeling, only nuclei with > 300 and < 15,000 detected genes and < 5% reads that mapped to mitochondrial genome were retained. The resulting count data were **processed by** Seurat (v5.0/v5.1)⁸² **with** Harmony⁸³ (v1.2.0) **batch-correction**. Multiplets were defined as cells labeled by scDbfFinder, as well as those within clusters exhibiting a high fraction of scDbfFinder-identified multiplets and expressing markers of

multiple major cell types (e.g., oligodendrocytes and astrocytes) (**Supplementary Fig. 11a-b**). After removing multiplets and low-represented clusters, each dataset was re-processed using Seurat with Harmony batch-correction. This two-step procedure was consistently applied across the study to minimize the influences of doublets and low-represented clusters. PCA dimensions and clustering resolution parameters were tuned to obtain a reasonable number of clusters for each dataset. Clusters from < 75% of samples were removed as low-represented clusters after both the first and second rounds of clustering.

The details of the scripts are available at Github repository.

Provided that the word “integration” in the reviewer’s comment means “batch-correction”, the integration (batch-correction) method in our main pipeline is Harmony-only. Seurat-only integration (batch-correction) would correspond to RPCA-based batch-correction using Seurat. The results of this procedure were described in the original manuscript. RPCA (reciprocal principal component analysis) batch-correction is a native algorithm in Seurat package without other external package. A simple comparison was described in the main text and **Supplementary Fig. 11**.

Line 756 in the original manuscript

For batch-correction, Harmony was preferred over Seurat RPCA, because the latter failed to clearly separate thalamic and cortical OPCs/astrocytes (**Supplementary Fig. 11c**), a result misaligned with findings by Siletti et al.⁶

Supplementary Figure (not changed from the original manuscript set)

Supplementary Figure 11. Quality control (QC) for cell clustering.

c) UMAP projection of major cell types (left) and their respective brain region origins (right) for the thalamus-cortex integration dataset, using Seurat RPCA integration.

This is contrasted to Fig. 1b (below)

Figure 1b

4. Differential gene expression

a. We could not find information whether *p* values for differentially expressed genes were adjusted. The authors should outline how they did the adjustment.

We apologize for the omission and confusion caused by our original description. In the initial manuscript, we reported only false discovery rate (FDR) values for differentially expressed genes (DEGs) in the main text. All *p*-values from the differential expression analysis using MAST were adjusted for multiple comparisons using the Benjamini-Hochberg method to control the FDR. These FDR-adjusted values and the original non-adjusted *P*-values are provided in the original version of **Supplementary Table 4** (column M and J). To improve clarity, we have now added a description of the FDR calculation method in the revised manuscript.

Line 1002 in the revised manuscript

DEG analysis was limited to protein-coding genes with Entrez IDs to streamline downstream analyses. Clusters with fewer than 1,000 nuclei were excluded to ensure sufficient counts per individual. **All the *P*-values from the MAST analysis were adjusted for multiple comparisons using the Benjamini-Hochberg method to control the false discovery rate (FDR). The raw *P*-values and FDRs are provided in **Supplementary Table 4**.**

Supplementary Table 4 (abridged, not changed from the original manuscript set)

J	K	L	M
P value	CI (high)	CI (low)	FDR (ER > 0.1)
4.89E-05	-0.296307859	-0.767248982	0.29244194
0.000156634	-0.047313579	-0.181291343	0.120399546
6.51E-05	-0.230797192	-0.576405625	0.15914844
7.43E-05	-0.137050537	-0.415813179	0.15914844
0.000187753	-0.157879791	-0.390448292	0.15914844
0.000249648	-0.088387724	-0.249476438	0.15914844
0.000320717	-0.053293168	-0.239742981	0.15914844
0.000404596	-0.055215433	-0.236732783	0.161060182
0.000340116	-0.063799	-0.221369908	0.15914844
0.000167374	-0.01819091	-0.172748915	0.066181847
0.000479561	-0.035949383	-0.160885956	0.096650075
3.09E-05	0.192504864	0.085848887	0.043325274
0.00029489	0.362705345	-0.059599625	0.07777369
6.23E-05	0.175277153	0.063987136	0.043325274
0.000163582	0.185838942	0.042285811	0.066181847
0.000282179	0.180868696	0.070063402	0.07777369
0.000333785	0.206767038	0.054672787	0.081350409
0.000198796	0.223629279	-0.012021975	0.066181847
0.000205055	0.161370572	0.050301715	0.066181847
0.000296846	0.16993503	0.040666642	0.07777369
0.000198309	0.309836391	0.12184621	0.066181847
6.99E-05	0.238252831	0.042445018	0.043325274

b. While GO analysis of DE genes is often used in snRNA-seq studies, it works better for larger gene lists than the authors obtained in this study. GSEA is better applicable here, since GSEA implements all genes, not only short list of significantly changed DE genes.

We agree that GSEA is generally a powerful and widely used method for DEGs. We initially considered using GSEA, but opted for a GO enrichment approach based on the following rationales.

1. Primary objective and DEG quantification

One of the primary aims of this study was to identify cell types exhibiting substantial transcriptional changes, as represented by the number of DEGs per cluster. This required us to define and count DEGs explicitly. As DEG calling inherently depends on thresholding, we sought to minimize arbitrariness by adopting the approach used in our original manuscript: applying a range of thresholds and evaluating the robustness of DEG classification based on the frequency with which genes are identified as DEGs across thresholds. Given that DEGs were defined as discrete entities through the above procedure, a conventional over-representation analysis using GO terms was a natural and direct downstream analysis. This allowed us to interrogate the biological functions of explicitly defined gene sets.

2. Limitations of single-metric ranking in GSEA:

Gene Set Enrichment Analysis (GSEA) requires ranking genes based on a single metric, typically fold change (FC). However, in our dataset, FC alone may not reliably capture gene relevance due to substantial inter-individual variability and our moderate sample size (N = 41), which

makes the analysis susceptible to outlier effects. In addition, genes with low expression rates (e.g., ER = 0.1) can exhibit unstable or spurious FC values, further complicating interpretation. While we considered composite ranking approaches that integrate FC, FDR, and ER (e.g., via multiplication), we ultimately opted against them due to the inherent subjectivity and arbitrariness involved in selecting weighting schemes.

Despite the above considerations, we acknowledge that GSEA remains a robust and informative approach and can serve as a valuable complement to traditional GO enrichment analysis. In response to the reviewer's suggestion, we have incorporated GSEA based on FC as a complementary analysis. Notably, the GSEA results for the downregulated genes in PVT neurons are relatively consistent with the DEG-GO enrichment findings, further supporting the robustness of our results. In contrast, the GSEA results for the upregulated genes in PVT neurons are less consistent with the DEG-GO enrichment findings, indicating a difference in biological interpretability. We again thank the reviewer for this insightful suggestion.

These GSEA results are provided in **Supplementary Data 7** and summarized in the revised Results section. To illustrate the notable genes identified by DEG-GO and GSEA analyses, we have revised Supplementary Table 5 and created a new **Supplementary Table 6**. In response to the Reviewer 4's suggestion, we have also incorporated SynGO enrichment analysis. We have also performed GSEA for secondary analysis.

Line 193 in the revised manuscript

This analysis revealed 56 downregulated and 69 upregulated probable DEGs in the medial thalamus (**Fig. 3e, Supplementary Table 4**). Tha_ExN_CALB (PVT neurons) displayed the highest DEG count, with 22 downregulated and 43 upregulated probable DEGs (64 downregulated and 157 upregulated possible DEGs). Tha_ExN_PVALB_1 exhibited the second-largest number of upregulated DEGs, with 81.4% of these genes overlapping with those upregulated in Tha_ExN_CALB. There were 29 downregulated and 67 upregulated robust DEG-GOs in the thalamus, detected only in Tha_ExN_CALB, Tha_ExN_PVALB_1, and Tha_Micro_1 (**Fig. 3f, Supplementary Table 5**). Tha_ExN_CALB also demonstrated the highest number of DEG-GOs, 24 downregulated and 58 upregulated. The robust DEG-GOs were predominantly enriched in neuronal and synaptic processes, and several DEG-GOs demonstrated overlap between downregulated and upregulated categories in Tha_ExN_CALB, such as the regulation of trans-synaptic signaling (GO:0099177) and modulation of chemical synaptic transmission (GO:0050804). **Given the enrichment of synaptic terms in the DEG-GOs, we additionally performed SynGO enrichment analysis⁴⁵. The downregulated DEGs in the Tha_ExN_CALB cluster again showed the strongest SynGO enrichment, including three terms related to synaptic structure: integral component of postsynaptic membrane (GO:0099055), integral component of presynaptic membrane (GO:0099056), and integral component of postsynaptic density membrane (GO:0099061) (**Supplementary Table 5**).**

Complementary GSEA based on fold changes supported the enrichment of neuronal and synaptic processes among downregulated genes in Tha_ExN_CALB (Supplementary Data 7**). GSEA identified potassium channel activity, including *KCNQ3*, *KCND2*, and *KCNC2* as the top GO term for the downregulated genes in Tha_ExN_CALB. The overlap between DEG-GO and GSEA results (GSEA-GO) was more pronounced for downregulated than upregulated genes (9/24 vs. 6/58; two-sided Fisher's exact test, $P = 0.00931$, odds ratio = 5.07) (**Supplementary****

Table 5). Notably, the top three DEG-GO terms among downregulated genes—**transporter complex** (GO:1990351), **monatomic ion channel complex** (GO:0034702), and **main axon** (GO:0044304)—were also identified by GSEA. The downregulated genes in Tha_ExN_CALB exhibit greater biological coherence and interpretability across enrichment methods. **Supplementary Table 6** summarizes the downregulated probable DEGs indicated by both DEG-GO and GSEA as biologically interpretable genes for BD. Among them, *KCNQ3*, *CACNA1C*, *SHISA9*, *KCNC2*, *LRR7*, *ROBO2*, and *CNTNAP4* were consistently enriched across all three analytical approaches (DEG-GO, GSEA, and SynGO).

Line 269 in the revised manuscript

In PVT neurons, the downregulated genes showed consistent GO term enrichment in both GO analysis and GSEA, and the MAGMA-based BD genomic risk was relatively prominent for downregulated genes. Thus, we focused on the downregulated genes. When extending the PPI analysis to possible DEGs, the downregulated possible DEGs in PVT neurons also demonstrated robust PPI networks (enrichment $P < 1.0 \times 10^{-16}$), encompassing genes within the top three DEG-GOs as biological processes (BP) or cellular components (CC) (**Fig. 4d**). These key genes, including *CACNA1C* and *SHISA9* as putative BD risk genes, were identified as core DEGs based on their PPI and GO enrichment. Particularly, *KCNQ3*, *GRIN2A*, and *GRM5* intersected with the top three BP GOs—modulation of chemical synaptic transmission (GO:0050804), regulation of trans-synaptic signaling (GO:0099177), and regulation of monoatomic ion transmembrane transport (GO:0034765). *KCNQ3* and *KCNC2* intersected with the top three CC GOs—transporter complexes (GO:1990351), monoatomic ion channel complexes (GO:0034702), and main axon (GO:0044304). These synaptic and ion-channel-related genes as PPI network hubs aligned with the biological pathways indicated by BD genomic studies^{1,2,4} and our DEG-GO/GSEA/SynGO results. Particularly, the overlap of *KCNQ3* in both BP and CC top GOs underscores its significant relevance to BD, besides putative BD risk genes.

Line 311 in the revised manuscript

Supplementary Table 6 summarizes the downregulated genes in Tha_ExN_CALB and Tha_Micro_1 that were enriched in one or more GO terms in both DEG-GO and GSEA analyses. Notably, these genes belonged to the same co-expression modules in each cluster.

Line 1030 in the revised manuscript

We also performed FC-based GSEA separately for the downregulated and upregulated gene sets to further evaluate the robustness of the identified biological signals, using gseGO function in clusterProfiler with the enrichmentScore threshold of 0.7.

Line 1121 in the revised manuscript

Finally, GSEA performed on DEGs from this extended analysis yielded results similar to those of the primary analysis (**Supplementary Data 22**), with downregulated genes in PVT neurons again enriched for synaptic and neuronal GO terms.

Line 1133 in the revised manuscript

GSEA also yielded similar results, with the top enriched term being potassium ion transmembrane transporter activity (GO:0015079), including *KCNQ3* (**Supplementary Data**

24). In this analysis, we employed GSEA instead of DEG-based GO enrichment, as the smaller sample size (12 BD cases vs. 9 controls) reduced the statistical power and FDR control.

Supplementary Table 5 (abridged)

Supplementary Table 5. Robust DEG-GOs in BD (FDR < 0.05). Add GSEA overlap information (column G).

A	B	C	D	E	F	G	H	I
Supplementary Table 5. Robust DEG-GOs in BD (FDR < 0.05).								
Region	Cluster	Direction	ONTOLOGY	GO ID	Description	GSEA.overlap (enrichmentScore)	Robustness	Fold-change
Medial thalamus	Tha_ExN_CALB	Down	CC	GO:1990351	transporter complex	0.708501553	0.839105339	5.904394299
Medial thalamus	Tha_ExN_CALB	Down	CC	GO:0034702	monoatomic ion channel complex	0.706822224	0.777777778	8.285833333
Medial thalamus	Tha_ExN_CALB	Down	CC	GO:0044304	main axon	0.791081515	0.767676768	18.55037313
Medial thalamus	Tha_ExN_CALB	Down	BP	GO:0050804	modulation of chemical synaptic transmission	.	0.727272727	8.091232931
Medial thalamus	Tha_ExN_CALB	Down	BP	GO:0099177	regulation of trans-synaptic signaling	.	0.727272727	8.074720211
Medial thalamus	Tha_ExN_CALB	Down	BP	GO:0034765	regulation of monoatomic ion transmembrane transport	0.702095932	0.721500722	6.033465966
Medial thalamus	Tha_ExN_CALB	Down	BP	GO:0031346	positive regulation of cell projection organization	.	0.716450216	6.001363017
Medial thalamus	Tha_ExN_CALB	Down	CC	GO:0097060	synaptic membrane	0.723487465	0.670995671	10.27822837
Medial thalamus	Tha_ExN_CALB	Down	BP	GO:0010975	regulation of neuron projection development	.	0.655122655	4.703054903
Medial thalamus	Tha_ExN_CALB	Down	CC	GO:0099572	postsynaptic specialization	0.704427824	0.647186147	10.77637283
Medial thalamus	Tha_ExN_CALB	Down	BP	GO:0019932	second-messenger-mediated signaling	0.739723111	0.636363636	7.854318418
Medial thalamus	Tha_ExN_CALB	Down	BP	GO:0034329	cell junction assembly	.	0.604617605	5.459279618
Medial thalamus	Tha_ExN_CALB	Down	CC	GO:0043204	perikaryon	0.720988025	0.591630592	12.51216443
Medial thalamus	Tha_ExN_CALB	Down	CC	GO:0031256	leading edge membrane	.	0.585858586	6.943435754
Medial thalamus	Tha_ExN_CALB	Down	CC	GO:0031252	cell leading edge	.	0.556277056	6.372798463
Medial thalamus	Tha_ExN_CALB	Down	MF	GO:0098632	cell-cell adhesion mediator activity	.	0.545454545	17.78461538
Medial thalamus	Tha_ExN_CALB	Down	MF	GO:0008013	beta-catenin binding	0.734692675	0.53030303	18.13333333
Medial thalamus	Tha_ExN_CALB	Down	BP	GO:0007409	axonogenesis	.	0.53030303	na
Medial thalamus	Tha_ExN_CALB	Down	BP	GO:0050808	synapse organization	.	0.52958153	6.931476658
Medial thalamus	Tha_ExN_CALB	Down	BP	GO:1901888	regulation of cell junction assembly	.	0.525252525	8.573375738
Medial thalamus	Tha_ExN_CALB	Down	BP	GO:0008038	neuron recognition	.	0.525252525	20.29032258
Medial thalamus	Tha_ExN_CALB	Down	BP	GO:0050769	positive regulation of neurogenesis	.	0.523088023	7.545987737
Medial thalamus	Tha_ExN_CALB	Down	CC	GO:0032279	asymmetric synapse	.	0.519480519	na
Medial thalamus	Tha_ExN_CALB	Down	BP	GO:0051962	positive regulation of nervous system development	.	0.502164502	7.346496107
Medial thalamus	Tha_Micro_1	Down	CC	GO:0043197	dendritic spine	.	0.952380952	11.20464278
Medial thalamus	Tha_Micro_1	Down	CC	GO:0044309	neuron spine	.	0.952380952	11.07731729

Supplementary Table 6. Downregulated genes in Tha_ExN_CALB and Tha_Micro_1 with biological interpretations

Supplementary Table 6. Downregulated genes in Tha_ExN_CALB and Tha_Micro_1 with biological interpretations								
Cluster	Gene	DEG-GO term	GSEA term	Robustness	Fold-change	WGICNA		
Tha_ExN_CALB	KCNQ3	transporter complex, monoatomic ion channel complex, main axon	potassium channel activity, voltage-gated potassium channel activity, main axon	1.00	0.80	Excit-M4		
Tha_ExN_CALB	ROBO1	positive regulation of cell projection organization, regulation of neuron projection development	cardiac ventricle development	1.00	0.82	Excit-M4		
Tha_ExN_CALB	PDE10A	second-messenger-mediated signaling	second-messenger-mediated signaling	1.00	0.83	Excit-M4		
Tha_ExN_CALB	LRRC7	main axon, positive regulation of cell projection organization, synaptic membrane	main axon, synaptic membrane, postsynaptic membrane	0.98	0.79	Excit-M4		
Tha_ExN_CALB	CACNA1C*	transporter complex, monoatomic ion channel complex, regulation of monoatomic ion transmembrane transport	regulation of muscle contraction, voltage-gated monoatomic ion channel activity, voltage-gated channel activity	0.76	0.84	Excit-M4		
Tha_ExN_CALB	SHISA9*	transporter complex, monoatomic ion channel complex, modulation of chemical synaptic transmission	synaptic membrane, postsynaptic membrane, postsynaptic specialization membrane	0.73	0.85	Excit-M4		
Tha_ExN_CALB	PTPRJ	second-messenger-mediated signaling, cell junction assembly, leading edge membrane	second-messenger-mediated signaling, beta-catenin binding, calcium-mediated signaling	0.73	0.85	Excit-M4		
Tha_ExN_CALB	PDE4D	transporter complex, monoatomic ion channel complex, regulation of monoatomic ion transmembrane transport	regulation of muscle contraction, second-messenger-mediated signaling, regulation of blood circulation	0.73	0.85	Excit-M4		
Tha_ExN_CALB	CUX2	modulation of chemical synaptic transmission, regulation of trans-synaptic signaling, positive regulation of cell projection organization	modulation of excitatory postsynaptic potential, regulation of postsynaptic membrane potential, excitatory postsynaptic potential	0.73	0.85	Excit-M4		
Tha_ExN_CALB	KCNC2	transporter complex, monoatomic ion channel complex, main axon	potassium channel activity, voltage-gated potassium channel activity, regulation of potassium ion transmembrane transport	0.67	0.82	Excit-M4		
Tha_ExN_CALB	CNTN4	modulation of chemical synaptic transmission, regulation of trans-synaptic signaling, cell-cell adhesion mediator activity	cell adhesion mediator activity	0.64	0.86	Excit-M4		
Tha_ExN_CALB	CNTN5	synaptic membrane, cell junction assembly, cell-cell adhesion mediator activity	cell adhesion mediator activity, synaptic membrane, sensory perception of sound	0.55	0.87	Excit-M4		
Tha_ExN_CALB	AUTS2	positive regulation of cell projection organization, axonogenesis	neuron migration	0.55	0.87	Excit-M4		
Tha_Micro_1	SYNDIG1	dendritic spine, neuron spine, regulation of synapse organization	postsynaptic membrane	1.00	0.70	Micro-M11		
Tha_Micro_1	CYFIP1	dendritic spine, neuron spine, regulation of synapse organization	regulation of axon extension	1.00	0.79	Micro-M11		
Tha_Micro_1	TIAM1	dendritic spine, neuron spine, regulation of synapse organization	activation of GTPase activity, regulation of postsynapse organization, positive regulation of chemotaxis	0.95	0.78	Micro-M11		
Tha_Micro_1	ABR	dendritic spine, neuron spine	activation of GTPase activity	0.59	0.82	Micro-M11		
Tha_Micro_1	DIP2B	regulation of nervous system development	regulation of axon extension	0.55	0.87	Micro-M11		

Downregulated probable DEGs in Tha_ExN_CALB or Tha_Micro_1 enriched in one or more GO terms in both DEG-GO and GSEA analyses
 * Putative BD risk genes

Supplementary Data
07.DataS7.DEG_GSEA.txt

Gene set enrichment analysis (GSEA) results for thalamic and cortical differentially expressed

genes (DEGs).

22.DataS22.DEG_GSEA_withPMI.txt

GSEA results for thalamic and cortical DEGs incorporating PMI as an additional covariate.

24.DataS24.lowPMI.GSEA.txt

GSEA results using only low-PMI samples.

In contrast, cortical clusters showed less enrichment by GSEA.

Line 349 in the revised manuscript

Our DEG analysis revealed 55 downregulated and 73 upregulated probable DEGs in the frontal cortex (**Fig. 6c, Supplementary Table 4**). FrC_ExN_L5IT_4 and FrC_ExN_L6Car3 exhibited the highest numbers of probable DEGs (**Supplementary Table 4**) and 11 robust DEG-GOs (**Supplementary Table 5**). While the DEG-GOs primarily pertained to synaptic and neuronal functions, their number in the frontal cortex was lower than that observed in the thalamus (**Fig. 6d vs. Fig. 3d**). **Among cortical DEG-GO, only GO:0045211 (postsynaptic membrane), an upregulated GO term in FrC_ExN_L6ITCar3, was identified by FC-based GSEA, and no SynGO enrichment was observed. These findings suggest that cortical DEGs exhibit lower biological interpretability compared to thalamic DEGs.**

5. Integration with genetics is rather preliminary and more sophisticated tools can be implemented for integration with the latest GWAS BD study. This will help validating the findings in the study for their relevance to BD. The thalamus and cortex should be analyzed separately by integration with GWAS data to show potential regional specialization with respect to BD-linked brain dysfunction.

Finally, at the moment, it seems that most association of DE genes in this study with BD risk factors is based only on 1 gene from 36 credible genes from the last GWAS study - SHISA9. We would caution the authors for not to overinterpret such association.

We appreciate the reviewer's suggestion to strengthen the genetic integration. As correctly noted, interpreting GWAS results based solely on MAGMA is preliminary. The recent PGC4 study (O'Connell Nature 2025) identified 36 "credible genes" through integrative approaches, of which only SHISA9 overlapped with the genes identified in our analysis. We acknowledge that only one gene among the 36 credible genes was associated with our results. We should be cautious in interpreting this result.

However, given the polygenic nature of BD, it is unlikely that only these 36 genes are involved in its pathophysiology. Although no definitive method currently exists for exhaustive gene prioritization, it remains important to consider a broader set of BD-associated gene candidates and assess their overlap with DEGs. Our original intent in using MAGMA was to provide a preliminary estimate of SNP-based genomic risk, and we now clarify this motivation in the revised manuscript.

In line with the reviewer's comment that MAGMA alone is insufficient for robust gene prioritization, we have now incorporated additional integrative approaches to complement the MAGMA findings:

- Fine-mapping using PolyFun+SuSiE/FINEMAP
- Transcriptome-wide association study (TWAS) using FUSION

We used fine-mapping to narrow down the putative BD risk genes from MAGMA analysis. TWAS data was not used to narrow down the putative BD risk genes, but only to support the findings of *CACNA1C*, because TWAS data is not comprehensive across all brain regions, lacking thalamic data. These methods were employed to refine candidate BD-associated genes. Nevertheless, we emphasize that the results from these approaches remain putative, and the identified genes should be interpreted with appropriate caution, given the inherent challenges of mapping disease-associated genes from GWAS.

Accordingly, we have removed the previously described list of "164 BD risk genes" derived from MAGMA. While we acknowledge the preliminary nature of MAGMA, we have retained its results to provide a rough estimate of SNP-based genomic risk and to address Reviewer #4's suggestion regarding the extension of MAGMA analyses to other psychiatric disorder GWAS datasets. Finally, as further recommended, we have conducted and visualized separate analyses for the thalamus and cortex to improve anatomical specificity.

We have revised the following points. Throughout the manuscript, we replaced "BD risk genes" with "putative BD risk genes."

Line 223 in the revised manuscript

Upon closer inspection of probable DEGs, MAGMA-based BD genomic risk^{1,46} was most enriched in the downregulated DEGs in *Tha_ExN_CALB* ($P = 0.00123$). Additional fine-mapping by PolyFun+SuSiE/FINEMAP⁴⁷⁻⁴⁹ further supported *CACNA1C* and *SHISA9* as putative BD risk genes within the downregulated set. Especially, *SHISA9* is one of the 36 credible genes for BD in the largest GWAS to date¹. While *CACNA1C* is not listed in these 36 credible genes, *CACNA1C* has consistently ranked among the top genes by p-value in prior PGC1-4 studies^{1,2,50,51}, and was also implicated by our FUSION-based TWAS integration. In contrast, the upregulated probable DEGs in *Tha_ExN_CALB* exhibited weaker enrichment for MAGMA-based BD genomic risk ($P = 0.0916$), including no putative BD risk genes despite the greater number of upregulated genes ($n = 22$ for downregulated vs. 43 for upregulated, **Supplementary Table 4**).

Line 387 in the revised manuscript

We examined which cell types most expressed putative BD risk genes. Thalamic excitatory neurons, including PVT neurons, showed the highest expression of the putative BD risk genes concerning the number of expressed genes and their average expression rate (**Fig. 8b**). For example, *CACNA1C* and *SHISA9* exhibited relatively higher expression in thalamic excitatory neurons than in cortical excitatory neurons (**Fig. 8c**). However, the overall differences were modest, and both genes showed the highest expression in *Tha_ExN_PVALB_1*, suggesting that additional factors likely contribute to the transcriptional alterations observed in PVT neurons, as discussed later.

Line 1034 in the revised manuscript

Assessment of BD risk genes

To evaluate the enrichment of SNP-based genomic risk for BD, we first applied MAGMA⁴⁶ to roughly assess the overlap between BD GWAS signals and probable DEGs, using multi-ancestry PGC4 BD summary statistics¹. Enrichment analyses were conducted for cell clusters containing five or more probable DEGs (n = 14 clusters). To assess specificity, we extended the analysis to GWAS summary statistics from other recent Psychiatric Genomics Consortium (PGC) datasets, including schizophrenia (SCZ)³³, major depressive disorder (MDD)⁵⁹, attention-deficit/hyperactivity disorder (ADHD)⁸⁸, post-traumatic stress disorder (PTSD)⁸⁹, obsessive-compulsive disorder (OCD)⁹⁰, substance use disorder (SUD)⁹¹, eating disorder (ED)⁹², panic disorder (PD)⁹³, and autism spectrum disorder (ASD)⁹⁴ (<https://pgc.unc.edu/>).

Beyond assessing SNP-based genomic enrichment, we further prioritized specific BD risk genes. As MAGMA alone provides only preliminary gene-level association signals, we adopted a multi-step approach to define “putative BD risk genes”. First, we designated the “36 credible genes” reported in the PGC4 BD study¹ as the most reliable BD-associated genes currently available. Given the highly polygenic architecture of BD, it is unlikely that only 36 genes account for all the BD genetic basis. Additional BD risk gene candidates were defined as those overlapping between MAGMA-significant genes and those harboring fine-mapped SNPs identified by PolyFun+SuSiE/FINEMAP⁴⁷⁻⁴⁹, using the multi-ancestry PGC4 BD summary statistics¹. MAGMA identified 167 genes associated with BD at Bonferroni-corrected significance (adjusted $P < 0.05$, based on 17,487 genes with Entrez IDs). Fine-mapping was conducted following the developer’s pipeline, using UKBB_LD as the reference panel. Fine-mapped SNPs were annotated using promoter regions (PsychENCODE TSS \pm 2kb; file: [tss.sites.codingOnly.gencode.v19.annotation.bed](https://www.encodeproject.org/track-hubs/psychencode/track-hub/psychencode.v19.annotation.bed)), enhancer regions (PsychENCODE GRN2; file: [INT-14_ElasticNet_Filtered_Cutoff_0.1_cross_assembly_GRN_2.csv](https://www.encodeproject.org/track-hubs/psychencode/track-hub/psychencode.v19.grn2.csv)), and SnpEff-4.3 for coding consequences. We then defined “putative BD risk genes” as the union of the 36 credible genes and 75 MAGMA-based genes containing fine-mapped SNPs in promoters, enhancers, or as missense variants, resulting in a total of 107 genes (four overlapped, **Supplementary Data 17**). To explore regional and cell-type specificity, we evaluated the expression of these 107 genes across thalamic and cortical cell clusters using two metrics: (1) the number of putative BD risk genes expressed per cluster (expression rate > 0.1), and (2) the mean expression level of those genes.

To further support gene prioritization, we conducted transcriptome-wide association studies (TWAS) using FUSION⁹⁵ with precomputed weights from the GTEx v8 brain datasets (<http://gusevlab.org/projects/fusion/>). Due to the absence of thalamic samples in GTEx, the TWAS results were used only for complementary reference and were not included in the definition of putative BD risk genes. TWAS-significant genes were identified using Bonferroni correction for each precomputed weight set (adjusted $P < 0.05$, across 2,557-7,732 tested genes per dataset). Given that GTEx represents bulk RNA-seq data from multiple brain regions (not including thalamus), we interpreted these results broadly, without strict anatomical specificity.

Figure 4. Transcriptional disturbances in PVT neurons.

c) PPI networks highlighting downregulated and upregulated probable DEGs in PVT neurons. Node color indicates the direction of gene regulation (Red, down; blue, up). Rectangles indicate BD risk genes. PPI enrichment P-value $< 1.0 \times 10^{-16}$. **d)** PPI networks highlighting downregulated possible DEGs in PVT neurons. Rectangles represent BD risk genes, and colors denote the genes in top DEG-GOs. Font size indicates DEG-GO enrichment. Note that the two PPI networks are identical but have different focuses on the genes. PPI enrichment P-value $< 1.0 \times 10^{-16}$.

Figure 8. Comparative analysis between the thalamus and cortex.

b) Expression of putative BD risk genes across cell nuclei clusters in the thalamus and cortex set (see Fig. 3a and 6a). The x-axis represents the count of expressed putative BD risk genes, and the y-axis denotes their average expression rates (percent expressed) among the respective clusters. The genes are limited to protein-coding genes with expression rates > 0.1 within each cluster.

Supplementary Data

17.DataS17.PutativeBDriskgenes.txt

Putative bipolar disorder risk genes identified in this study and their sources.

We have also revised the relevant phrasing (from “and” to “and/or”) to reflect the putative

nature of BD risk genes, acknowledging that *KCNQ3* is not included among the putative BD risk genes.

Abstract in the revised manuscript

Notably, PVT neurons showed significant downregulation of genes associated with synaptic function and ion channel regulation, including *SHISA9*, *CACNA1C*, and *KCNQ3*—genes linked to **putative** BD risk and/or implicated as critical hubs in the downregulated molecular networks.

Line 40 in the revised manuscript

In PVT neurons, *SHISA9*, *CACNA1C*, and *KCNQ3* exhibited notable changes as **putative** BD risk genes and/or critical hubs in the downregulated synaptic/neuronal gene networks.

6. Compositional analysis

*While it is justified to use *scomp* package for compositional analysis, the package does not control for inter-dependency of proportional changes, and the authors admit it when comparing to immunohistochemical assessment, Lines 169-172:*

*Note that immunohistochemistry directly assessed cell density, while compositional analysis using *snRNA-seq* data assessed relative proportional changes. The immunohistochemical assessment indicated that the compositional increase in oligodendrocytes and astrocytes in BD should be mainly derived from prominent reductions of excitatory neurons, not reflecting absolute cell density increases.*

*To avoid such issue, there are packages that allow for independent compositional analysis of cell proportions, such as *scCODA*. The authors should use one of such packages to cross-compare *snRNA-seq* with immunohistochemistry.*

We agree with the reviewer's point and have now conducted a parallel compositional analysis using *scCODA*. As correctly noted, compositional methods rely on specific assumptions, and applying multiple complementary methods is important to ensure robustness. We have now included the *scCODA* results in the revised manuscript. To define an appropriate reference, we first selected oligodendrocyte precursor cells (OPCs), as they exhibited no significant case-control differences in both the thalamus and cortex and appeared relatively stable across samples. We also tested oligodendrocytes as the reference to compare *snRNAseq* to immunohistochemistry (IHC) data.

Notably, the *scCODA* results were concordant with those obtained using *scomp* in the thalamus, while not concordant in the cortex. In cortex analysis, total nuclei counts were normalized to 10,000 per individual for *scCODA*, because *scCODA* did not work well with raw counts. This supports our argument that the thalamus exhibited more notable BD-associated changes than the cortex. We include these *scCODA* results in a new Supplementary Data file and have updated the Results and Discussion sections accordingly. Unfortunately, *scCODA* did not perform reliably when PMI was included alongside diagnosis, age, and sex, yielding inclusion probabilities of 1.0 for all variables across all clusters, even if using normalized counts. This unlikely reflects the true phenomena, which might be due to too many covariates for *scCODA* analysis with our sample size.

Line 160 in the revised manuscript

Complementary analysis using scCODA⁴³ also supported a consistent compositional reduction in thalamic excitatory neurons, including PVT neurons (**Supplementary Data 4**). In the scCODA analysis, Tha_ExN_CALB and Tha_ExN_PVALB_1 exhibited comparable compositional decreases (FC = 0.686 and 0.663).

Line 182 in the revised manuscript

If we set Tha_Oligo_1 as a reference cluster in scCODA analysis, assuming that the absolute counts of oligodendrocytes were comparable between BD and control from IHC results, we consistently observed compositional decreases in thalamic excitatory neurons (**Supplementary Data 4**).

Line 339 in the revised manuscript

The scCODA analysis was not consistent to the sccomp analysis, exhibiting the compositional decreases of FrC_ExN_L2/3IT_1 (FC = 0.797), FrC_ExN_L5IT_4 (FC = 0.866), and FrC_InN_CGE_LAMP5 (FC = 0.848). In contrast to findings in the thalamic excitatory neurons, the cortical neurons showed **inconsistent results and** smaller reductions (**Fig. 6b vs. Fig. 3b**).

Line 885 in the revised manuscript

As compositional analysis methods rely on specific assumptions, we applied two methods of a compositional analysis for single-cell data, sccomp⁴² (v2.1.6) and scCODA⁴³ (v0.1.9), to compare nuclei counts across cell clusters between BD and control groups. The sccomp employs a sum-constrained beta-binomial model, while the scCODA employs a Bayesian model using a reference cluster. The threshold of inclusion probability in scCODA was set as 0.95. We considered the analysis to have failed when the observed inclusion probability was 1.0 for all clusters within a given covariate, indicating a lack of model convergence or informativeness. In cases where raw counts could not be used successfully (cortex analysis and PMI analysis), total nuclei counts were normalized to 10,000 per individual, and cluster-specific counts were rounded to integers to reduce inter-individual variability in total nuclei counts. The compositional analysis was limited to the clusters in six major classes: excitatory/inhibitory neurons, oligodendrocytes, OPCs, astrocytes, and microglia.

Line 1095 in the revised manuscript

The scCODA model did not perform reliably when PMI was included alongside Dx, age, and sex, yielding inclusion probabilities of 1.0 for all variables across all clusters, even if using total cell count normalization. This likely reflects overparameterization due to the number of covariates relative to our sample size.

Supplementary Data

04.DataS4.scCODA.txt

scCODA results for thalamic and cortical clusters, including estimated fold-changes and posterior probabilities for each covariate.

a. Based on the text lines 270-283, it sounds as the authors propose that SHISA9 protein on neurons interacts with SYNDIG1 on microglia? If yes, then some literature search should help confirming this potential interaction. Otherwise, we suggest concentrating interpretation on general discussion on how microglia modulate synaptic processes in excitatory neurons via AMPA receptor complex. Additionally, LIANA analysis should help with such discussion, however the data is just shown in Figure 5e, without any interpretation.

As correctly pointed out, there is currently no direct evidence in the literature that SHISA9 and SYNDIG1 proteins physically interact, and the original manuscript contained an overinterpretation in retrospect. Studies such as Kalashnikova et al. (Neuron 2010) and Khodosevich et al. (Neuron 2014) describe SHISA9 and SYNDIG1 proteins as auxiliary membrane proteins associated with AMPA receptor (AMPA) complexes. While both proteins are involved in regulating synaptic function, there is no published evidence indicating a direct molecular binding between them. We have now revised this section to accurately reflect current knowledge. The revised statement emphasizes that microglia might modulate synaptic processes in excitatory neurons through pathways involving the AMPA receptor complex, without suggesting a direct interaction between SHISA9 and SYNDIG1.

Abstract in the original manuscript

~~REMOVED: This interaction was further exemplified by the concurrent downregulation of SYNDIG1 in microglia and SHISA9 in PVT neurons, both of which are involved in AMPA receptor signaling pathways.~~

Line 279 in the original manuscript

~~REMOVED: The coordinated downregulation of SYNDIG1 in microglia and SHISA9 in excitatory neurons exemplifies a probable disturbance in microglia-neuron interactions for synapses in the BD thalamus, indicating a plausible pathological mechanism of SHISA9, one of the 36 putative BD-causal genes¹.~~

Line 415 in the original manuscript

~~REMOVED: The downregulation of SYNDIG1 in microglia and SHISA9 in PVT neurons might be an example of dysregulated molecular interactions within the Excit-Micro meta-gene programs associated with BD.~~

Line 314 in the revised manuscript

At the gene level, SYNDIG1 and SHISA9 encode auxiliary subunits for AMPA receptor (AMPA) complex^{56,57}, with both genes' robustness scores exceeding 0.7 (Supplementary Table 4). Notably, SYNDIG1 was prominently expressed in microglia, while SHISA9 was primarily expressed in excitatory neurons (Fig. 5d). SHISA9 consisted of core PPI networks of BD-associated downregulated genes in PVT neurons (Fig. 4d) and overlapped in the downregulated genes in Tha_ExN_PVALB_1, as one of the 36 putative BD-causal genes¹. Beyond microglia and excitatory neurons, thalamic T cells were included in this meta-gene program. Fig. 5e illustrates potential ligand-receptor interactions between neurons and microglia/T cells, as analyzed through LIANA⁵⁸. Although we did not detect concurrently downregulated DEGs as ligand-receptor pairs, KCND2 in Tha_ExN_CALB, LAMB1 in Tha_ExN_PVALB_2, and KCNQ1 in Tha_Micro_1 were identified as downregulated DEGs

among the LIANA-based ligand-receptor pairs. These genes encode ion channels or extracellular matrix proteins that may potentially contribute to Excit–Micro interactions. However, as this analysis is an inference, the precise ligand–receptor mechanisms remain to be elucidated in future studies.

Line 556 in the revised manuscript

Beyond cell-type-based analysis, we identified trans-cellular meta-gene programs between excitatory neurons and microglia (Excit-Micro meta-gene programs), likely reflecting physiological synaptic interactions between these cell types. **Microglia are known to modulate synaptic plasticity and neuronal circuitry through dynamic interactions with neurons**⁷⁶. Notably, such meta-gene programs in the thalamus are disrupted in BD, further characterized by a significant decrease in synaptic gene expression in both PVT neurons and microglia. Ling et al. reported downregulation of the synaptic neuron and astrocyte program in schizophrenia¹⁰; our findings highlight downregulation of the synaptic **genes in the thalamic neuron and microglia in BD. The concurrent downregulation of *SYNDIG1* in microglia and *SHISA9* in excitatory neurons suggests disruptions of AMPAR signaling, as both genes encode auxiliary subunits of the AMPAR complex^{56,57}. *TIAM1*, another downregulated gene in microglia, has also been implicated in regulating synaptic AMPAR function⁷⁷. The preferential localization of *SYNDIG1*-positive microglia in cortical gray matter is consistent with this interpretation. Nonetheless, these meta-gene programs remain data-driven constructs, and the roles of *SYNDIG1* in microglia—as well as their mechanistic links to *SHISA9* in PVT neurons—are currently unknown. Future studies employing animal and in vitro cellular models will be crucial for elucidating these mechanisms.**

Additionally, LIANA does not support a SHISA9-SYNDIG1 interaction. In the manuscript, LIANA was employed to investigate potential microglia-excitatory neuron interactions suggested by NMF-based module analysis. However, no ligand-receptor pair was identified in which both the ligand and receptor genes were downregulated.

Nevertheless, several DEGs were detected that may participate in intercellular signaling as either the ligand or receptor: *KCND2* in Tha_ExN_CALB, *LAMB1* in Tha_ExN_PVALB_2, and *KCNQ1* in Tha_Micro_1. These genes encode ion channels or extracellular matrix proteins that may indirectly contribute to cell-cell interactions. However, the precise ligand-receptor mechanisms remain unresolved, and we now explicitly state that cell-cell communication analysis remains a subject for future investigation.

Line 320 in the revised manuscript

Fig. 5e illustrates potential ligand-receptor interactions between neurons and microglia/T cells, as analyzed through LIANA⁵⁸. **Although we did not detect concurrently downregulated DEGs as ligand–receptor pairs, *KCND2* in Tha_ExN_CALB, *LAMB1* in Tha_ExN_PVALB_2, and *KCNQ1* in Tha_Micro_1 were identified as downregulated DEGs among the LIANA-based ligand-receptor pairs. These genes encode ion channels or extracellular matrix proteins that may potentially contribute to Excit–Micro interactions. However, as this analysis is an inference, the precise ligand–receptor mechanisms remain to be elucidated in future studies.**

b. One missing opportunity in the study is analysis of changes in cortico-thalamic connectivity in BD. This can be done using similar cell-cell interaction tools.

Thank you for your excellent suggestion. In the original manuscript, we refrained from presenting cell-cell communication between cortical and thalamic clusters because we did not identify meta-gene programs supporting such cortico-thalamic interactions. Nevertheless, interactions between cortical and thalamic cells could be of interest to readers.

Although our snRNA-seq approach does not permit direct inference of long-range connectivity, LIANA can provide indirect evidence by evaluating potential ligand-receptor interactions between cortical and thalamic cells. Given the large number of possible cortical-thalamic cell-type combinations, we restricted our analysis to deep-layer cortical excitatory neurons and thalamic excitatory neurons, the most plausible candidate cell types for cortex-thalamus connectivity. These findings are now presented in **Supplementary Fig. 9**.

Line 431 in the revised manuscript

Although we did not identify trans-regional cortico-thalamic meta-gene programs, we extended the LIANA analysis to explore potential interactions between the cortex and thalamus, focusing specifically on connections between cortical deep-layer excitatory neurons (ExN_FrC_DL) and thalamic excitatory neurons (ExN_Tha) (**Fig. 1b**). Among the top 20 ligand-receptor pairs in the cortex-to-thalamus direction, eight involved DEGs in either ExN_Tha or ExN_FrC_DL (**Supplementary Fig. 9**), and four DEG-containing pairs were detected in the thalamus-to-cortex direction. These results suggest potential disruptions in cortico-thalamic

connectivity mediated by altered ligand-receptor interactions. However, this analysis is exploratory and hypothesis-generating; further studies are required to more definitively characterize these connections.

Supplementary Figure 9. Thalamus-cortex communication estimated by LIANA.

Predicted ligand-receptor interactions between thalamic excitatory neurons (ExN_Tha) and cortical deep-layer excitatory neurons (ExN_FrC_DL). Dot plots represent expression magnitudes and interaction specificities, with interactions labeled as [ligand gene] -> [receptor gene]. Blue or red stars indicate ligand-receptor pairs in which the ligand or receptor is a DEG. Top: ligands expressed in ExN_FrC_DL targeting receptors in ExN_Tha. Bottom: ligands expressed in ExN_Tha targeting receptors in ExN_FrC_DL.

8. Such snRNA-seq studies usually require validation of some findings by single-molecular FISH. We propose selecting the most interesting finding in gene expression and validate it by RNAscope or similar technologies.

We appreciate the reviewer's suggestion to validate key findings using single-molecule fluorescence in situ hybridization (smFISH). In response, we performed RNAscope Multiplex Fluorescent in situ hybridization to validate one of our most notable gene expression observations—the unexpected expression of *SYNDIG1* in microglia. Using probes for *SYNDIG1* and *CX3CR1* (a microglial marker), we confirmed *SYNDIG1* expression in *CX3CR1*-positive cells within both the thalamus and cortex in representative specimens (new **Fig. 9**). We further mapped the spatial distribution of *SYNDIG1*-expressing microglia in these regions and quantified *SYNDIG1* puncta within *CX3CR1*-positive cells in representative cases and controls.

CX3CR1/SYNDIG1 double-positive cells were primarily localized to the gray matter of the frontal cortex, consistent with our hypothesis—derived from geneNMF analysis (**Fig. 5b** and **Fig. 8e**)—that *SYNDIG1*-positive microglia might interact with excitatory neurons. In the thalamus, *SYNDIG1*-positive microglia appeared enriched in ventricular regions, although this trend was not definitive. Quantification of *SYNDIG1* expression in ten specimens revealed a tendency toward reduced mean expression in BD compared with controls. Although sample size was limited due to the scarcity of available specimens (many of which had been used), these results support and extend our snRNA-seq findings, providing orthogonal single-cell resolution validation.

While *SNCA* expression in PVT neurons represents another noteworthy observation relevant to the basic understanding of the human brain, this has already been validated by Schulmann et al. (bioRxiv, doi: <https://doi.org/10.1101/2024.09.03.611112>). We therefore did not replicate this experiment in our study.

Line 461 in the revised manuscript

Single-molecule fluorescence in situ hybridization for *SYNDIG1* in microglia

The unexpected detection of *SYNDIG1* in thalamic and cortical microglia prompted further validation of its expression and spatial distribution using RNAscope. *SYNDIG1* transcripts were detected in *CX3CR1*-positive cells, consistent with a microglial identity (**Fig. 9a**). *CX3CR1/SYNDIG1* double-positive cells—representing *SYNDIG1*-expressing microglia—were preferentially localized to the gray matter of the frontal cortex (**Fig. 9b**). In the thalamus, these cells appeared more abundant in medial regions, although this pattern was not definitive. Comparison of representative thalamic male samples from five BD cases and five controls showed relatively lower *SYNDIG1* expression in BD microglia (raw FC = 0.801, **Fig. 9c**). While generalized linear mixed model (GLMM) analysis with a negative binomial distribution did not reach statistical significance, it indicated a trend toward reduction (GLMM P = 0.153, estimated FC = 0.783).

Line 1170 in the revised manuscript

Single-molecule fluorescence in situ hybridization (smFISH)

We performed smFISH using the RNAscope Multiplex Fluorescent Reagent Kit v2 (Advanced Cell Diagnostics [ACD], Newark, USA) to detect *SYNDIG1* and *CX3CR1* transcripts in fresh-frozen human frontal cortex (FrC) and thalamus (Tha) sections from CT11 as well as additional representative thalamic sections from five BD male cases and five male controls. *SYNDIG1* served as the primary target, while *CX3CR1* was used to identify microglia. Fresh-frozen 12-

μm -thick sections were cut onto SuperFrost Plus Gold slides and post-fixed overnight in 4% formaldehyde prepared in 0.1 M PBS at 4 °C. The sections were washed with 1 \times PBS, and dehydrated through graded ethanol series (30% EtOH in 1 \times PBS, 50% EtOH in 0.5 \times PBS, 70% EtOH in water, and 100% EtOH twice), baked at 60 °C for 30 min, and circumscribed with a hydrophobic barrier. Sections were treated with hydrogen peroxide and Protease IV, then hybridized with prewarmed RNAscope probes complementary to *SYNDIG1* (C3; #804391-C3 / ACD) and *CX3CR1* (C1; #411251 / ACD) transcripts for 2 h at 40 °C. After washing, sections were photobleached in 5 \times SSC solution for 60 min by using the autofluorescence Quenching illuminator TiYO (Nepa Gene, Chiba, Japan)¹⁰⁴. Signal was amplified via AMP1–AMP3 hybridization, and sequentially developed per probe by HRP-C#/fluorophore-tyramide (TSA Vivid, 1:750) reactions and HRP blocking. Sections were counterstained with DAPI, mounted in ProLong Gold Antifade, and cured at RT for 48 h. Imaging was performed on a NanoZoomer 2.0-RS (Hamamatsu Photonics) after brightfield and fluorescence calibration each day. While correcting a focal plane ($z = 0 \mu\text{m}$) every 12 min by autofocus on brightfield, wide-field ($\sim 8 \text{ mm} \times 12 \text{ mm}$) images were captured at a high-resolution (227 nm/pixel) as multi-channel (4 channels; brightfield, Cy5 for TSA Vivid 650, TRITC for TSA Vivid 570, and DAPI) z-stacks (3 slices; $0 \pm 1.5 \mu\text{m}$) in NDPI/NDPIS format. The signal quality of smFISH on RNAscope was confirmed per specimen with positive (#320861 / ACD) and negative (#320871 / ACD) control probes. *CX3CR1* and *SYNDIG1* probes were visualized with TSA Vivid 570 and TSA Vivid 650, respectively. For case-control sets, one case and one control were processed in parallel to minimize batch effects.

Image preprocessing was performed on multi-channel z-stacks using Fiji/ImageJ¹⁰⁵ (v2.16.0/1.54p). To prevent out-of-memory errors, each multi-channel z-stack was split into four z-stacks per channel without importing the entire image, by editing NDPIS files. NDPIS files are text-formatted lists that specify the order in which individual channel images (NDPI files) are read and reconstructed into a multi-channel z-stack. The brightfield z-stack was excluded from subsequent analyses. The remaining three fluorescent z-stacks (monochrome, 8-bit grayscale) were sequentially imported via the Bio-Formats Importer, underwent background subtraction by the sliding paraboloid algorithm (radius: 300 pixels, disable smoothing: true), converted into maximum intensity z-projection images (MlZPs), and saved to disk to free memory. The fluorescent MlZPs were then assembled into a multi-channel stack with pseudo-color.

Object detection and quantification were performed by using QuPath¹⁰⁶ (v0.5.1). A consistent fluorescent signal threshold of 45 in 8-bit (0–255) grayscale was applied to each channel across all specimens. Cells were detected using the WatershedCellDetection plugin, defined as areas containing a nucleus (DAPI) and a peripheral region within 3- μm of the nucleus. By the SubcellularDetection plugin, the smFISH signals of *SYNDIG1* (TSA Vivid 650, Cy5 channel) and *CX3CR1* (TSA Vivid 570, TRITC channel) were detected as puncta or clusters in cells, and counted as the number of spots estimated, assuming that single spots are approximately 1 μm^2 (range: 0.5 μm^2 to 3.0 μm^2). Additional details are available at <https://github.com/msk240/BDsnRNAseq>

Data aggregation was conducted in R (v4.1.0). Positive cells were defined as those containing three or more spots corresponding to the target gene, based on the number of spots estimated from smFISH signals. *SYNDIG1* expression in *CX3CR1*-positive cells, representing putative microglia, was assessed using a generalized linear mixed model (GLMM) implemented in the glmmTNB package (v1.1.12), with diagnosis (Dx) and age as fixed effects and sample ID as a

random effect, employing the nbinom2 parameterization of the negative binomial distribution. Sex was not included as a covariate, as all samples were from male donors.

Figure 9. Single-molecule fluorescence in situ hybridization of *SYNDIG1* transcript

a) RNAscope validation of *SYNDIG1* expression in *CX3CR1*-positive cells within the thalamus and cortex. Left panels: Low-magnification view of the thalamus and cortex with DAPI staining. White rectangles denote the regions magnified in the middle panels. Middle panels: Magnified views highlighting *CX3CR1*-positive cells (likely microglia), indicated by white arrowheads. Magenta signals represent *SYNDIG1* expression, which varies among *CX3CR1*-positive cells. The white square indicates the region further magnified in the right panels. Right panels: Higher magnification of *CX3CR1*/*SYNDIG1* double-positive cells, corresponding to *CX3CR1*(+)/*SYNDIG1*(+) cells in panel b. **b)** Spatial distribution of cell types classified by *CX3CR1*/*SYNDIG1* expression. *CX3CR1*(+)/*SYNDIG1*(+) cells are considered *SYNDIG1*-expressing microglia. *CX3CR1*(+)/*SYNDIG1*(-) cells represent microglia with little or no *SYNDIG1* expression. *CX3CR1*(-)/*SYNDIG1*(+) cells likely correspond to non-microglial *SYNDIG1*-expressing cells, such as excitatory neurons. *CX3CR1*(-)/*SYNDIG1*(-) cells are classified as other cell types. Cell counts per hexagonal bin are assigned a color code at log₂ scale. White contours are the median of cell density for each cell classification. **c)** Quantification of *SYNDIG1* puncta in *CX3CR1*-positive cells (likely microglia) across 10 representative male samples (five BD cases

and five controls). Each dot represents an individual cell.

9. The cortex does not show large changes in gene expression that is why the authors argue to focus on the thalamus. Nevertheless, this is surprising given previous role of the cortex that has been proposed in brain dysfunction in BD. There is a lot of focus in the Discussion section on the thalamus, but such surprisingly minor changes in the cortex are not discussed. We think these findings are very important, since the study can pave the way for future analyses of postmortem brain tissue in psychiatric disorders. Currently, almost all studies focus on the prefrontal cortex and maybe in the future the thalamus should be the major region for such snRNA-seq analyses.

We thank the reviewer for highlighting this important point. In the revised Discussion, we have added a dedicated paragraph addressing the unexpectedly subtle changes observed in the cortex, in contrast to the well-established cortical emphasis in prior BD studies. While we acknowledge the significance of cortical pathology, we also note that some degree of “streetlight effect” (i.e., the tendency to investigate areas that are more accessible or better characterized) may have contributed to the predominant focus on the cortex in previous research.

Our findings suggest that the thalamus may play a more central role in the pathophysiology of BD than previously recognized. This is especially notable given the relative inaccessibility of subcortical regions in human postmortem research. We believe that this shift in focus could help guide future transcriptomic and neuroimaging investigations in psychiatric disorders, potentially revealing underexplored yet biologically critical brain regions.

Line 572 in the revised manuscript

Although BD is also associated with reduced cortical volume, the transcriptional and compositional changes in the cortex were less pronounced than those in the thalamus. The frontal cortex has been frequently analyzed in psychiatric research, not only due to its suspected involvement in disease pathology but also because of its relative accessibility. This may reflect a form of the “streetlight effect,” wherein research tends to focus on more accessible regions. However, higher accessibility does not necessarily equate to higher biological significance. Our findings underscore the importance of extending investigations to subcortical regions, which may harbor critical yet underexplored components of BD pathophysiology. It is noteworthy that the cortical region analyzed in this study was BA10. Results may differ in other cortical regions, such as the anterior cingulate cortex, emphasizing the need for broader sampling across diverse cortical areas. Integration of GWAS data with brain-wide snRNA-seq indicated heritability enrichment in cortical neuronal populations, including interneurons and deep-layer intra-telencephalic neurons¹. While such genetic insights are informative, caution is warranted when inferring cell-type-specific dysfunction based solely on heritability enrichment. Genetic enrichment does not necessarily guarantee transcriptional dysregulation. Moreover, technical variability in defining cell-type-specific gene expression profiles can influence enrichment outcomes and lead to divergent interpretations⁷⁸. Therefore, direct observations of transcriptional and compositional changes remain essential to elucidate disease-associated cellular alterations.

Minor

1. Discussion lacks similarity of BD and schizophrenia, although *CACNA1C* is one of the major genes in the current study and it has been identified as genetic risk factor by schizophrenia GWAS and not BD.

We thank the reviewer for this important point. In the revised Discussion, we now highlight the shared genetic architecture between bipolar disorder (BD) and schizophrenia, with particular emphasis on *CACNA1C*, which has been consistently implicated in both disorders. Although *CACNA1C* is not included among the 36 causal genes for BD listed in the PGC4 study (O'Connell et al., *Nature* 2025), it has remained one of the top loci associated with BD across multiple GWAS phases based on p-value rankings:

PGC1: ranked 1st in Sklar et al., *Nat Genet* 2011 (Table 3)

PGC2: ranked 4th in Stahl et al., *Nat Genet* 2019 (Table S4)

PGC3: ranked 3rd in Mullins et al., *Nat Genet* 2021 (Table S4)

PGC4: ranked 6th in O'Connell et al., *Nature* 2025 (based on our in-house MAGMA analysis)

The exclusion of *CACNA1C* from the 36 causal gene list in PGC4 may reflect methodological factors. Causal gene identification remains largely inferential and is influenced by analytical parameters and reference datasets, rather than definitive experimental validation. Notably, our own analyses using fine-mapping and TWAS integration based on previous BD GWAS—an approach comparable to that used in the PGC4 study—also identified *CACNA1C* as one of promising BD-associated genes. This convergence might support the hypothesis that BD and schizophrenia share molecular mechanisms, including dysregulation of calcium channel signaling, particularly within thalamic excitatory neurons. We have expanded the revised Discussion to address this potential overlap in pathophysiology.

We have also added the relative expression profiles of *CACNA1C* alongside *SHISA9* to highlight these genes as probable DEGs in PVT neurons (**Fig. 8**). Both genes show the highest expression in thalamic excitatory neurons within the thalamus-cortex integration set.

Line 227 in the revised manuscript

While *CACNA1C* is not listed in these 36 credible genes, *CACNA1C* has consistently ranked among the top genes by p-value in prior PGC1-4 studies^{1,2,50,51}, and was also implicated by our FUSION-based TWAS integration.

Line 489 in the revised manuscript

Among all the thalamic and cortical cell types in our study, PVT neurons exhibited the highest number of DEGs in BD. Notably, downregulated DEGs in PVT neurons were more biologically interpretable than upregulated ones and included putative BD risk genes. Among the PVT neuron DEGs, *SHISA9*, *CACNA1C*, and *KCNQ3* stood out as particularly relevant to BD pathophysiology, as they were centrally positioned in the PPI network and consistently highlighted in GO, GSEA, and SynGO analyses. *SHISA9* is one of the 36 credible genes identified in the largest BD GWAS. Although *CACNA1C* is not among these 36, it has repeatedly ranked among the top BD-associated loci by p-value across PGC1-4 studies^{1,2,50,51} and was also

implicated in our FUSION-based TWAS analysis. *CACNA1C* has additionally been associated with schizophrenia in prior GWAS³³, suggesting a shared genetic architecture between BD and schizophrenia. Consistently, downregulated DEGs in PVT neurons were enriched for schizophrenia polygenic risk as assessed by MAGMA. While *KCNQ3* is not classified as a putative BD risk gene, its downregulation was reported in bulk brain tissue from BD patients⁶¹. *KCNQ3* may represent a BD-associated gene identified through transcriptomic rather than genomic approaches.

Figure 8. Comparative analysis between the thalamus and cortex.

c) Relative expression of *CACNA1C* and *SHISA9* across all clusters in the thalamus-cortex integration dataset.

2. In general, although Discussion is quite long, some points are extensively discussed, whereas others are missing. For instance, close relation of thalamic excitatory neurons with GABAergic neurons is interesting. Is it related to cell fate or neuronal specialization? Similarly, cardiac tissue similarity can be moved from Results to Discussion section.

We appreciate the reviewer’s insightful comment. In response, we have expanded the Discussion to address the transcriptional similarity between thalamic excitatory neurons and cortical inhibitory neurons. We now suggest that this resemblance may reflect a unique subtype specialization potentially related to electrophysiological function, rather than indicating a shared developmental origin. In line with the reviewer’s suggestion, we have also relocated the discussion of cardiac gene expression profiles in thalamic neurons from the Results section to the Discussion section.

Line 72 in the original manuscript

REMOVED: When compared to ExN_Fc, ExN_Tha showed a preferential expression of genes associated with cardiac muscle development, such as *SHOX2*, *PROX1*, *HCN4*, and *CACNA1G*, as determined by gene set enrichment analysis (GSEA) (Fig. 1e, Supplementary Table 2, Supplementary Data 1). ~~The unexpected enrichment of genes associated with cardiac muscle,~~

~~particularly atrial muscle cells, may reflect a shared characteristic of frequent and rhythmic Ca^{2+} regulation between thalamic neurons and cardiac muscle. *CACNA1G*, which encodes a subunit of T-type calcium channels involved in electrical pace-making activity, is specifically expressed in the thalamus and the heart³⁰, and has been identified as a core gene associated with schizophrenia³¹. Among the ExN_Tha-specific genes, *CASQ2* was highly characteristic to ExN_Tha, with virtually no expression in other cell clusters (**Supplementary Fig. 2a**). *CASQ2* encodes calsequestrin-2, a high-capacity calcium-binding protein particularly expressed in cardiac muscle cells³², further underscoring the commonality between thalamic excitatory neurons and cardiac muscle cells, probably reflecting the frequent and rhythmic activities with intra-cellular Ca^{2+} changes.~~

Line 102 in the original manuscript

~~REMOVED: These findings may suggest shared characteristics of frequent and rhythmic activities with intra-cellular Ca^{2+} changes between thalamic neurons and cardiac muscle cells.~~

Line 543 in the revised manuscript

In addition to PVT neurons, other thalamic excitatory neurons express *CALB1*, *CALB2*, and/or *PVALB*, all of which encode calcium-binding proteins. The widespread expression of these proteins may underlie the unexpected transcriptomic similarity observed between thalamic excitatory neurons and cortical inhibitory neurons. Calcium-binding proteins are critical for buffering rapid fluctuations in intracellular calcium concentrations associated with high-frequency neuronal firing. Kim et al. reported that thalamic excitatory neurons originate from radial glia derived from prosomere 2 of the embryonic diencephalon³⁵, whereas cortical inhibitory neurons arise from the ganglionic eminence, highlighting their distinct developmental origins. The observed transcriptomic similarity is unlikely to reflect shared embryonic origins but may instead suggest functional convergence, particularly in intracellular calcium regulation. Interestingly, both excitatory and inhibitory thalamic neurons showed preferential expression of genes typically associated with cardiac muscle. This observation may reflect a shared physiological trait—frequent and rhythmic intracellular calcium regulation—between thalamic neurons and cardiac myocytes.

3. Lines 137-141 compare data from the current study to previously published organoid data. This comparison is completely separated from all other analyses in the study and in the current state should be removed from the manuscript.

We agree that this comparison was not sufficiently integrated with the main narrative. Accordingly, we have removed this section, including supplementary figures, from the manuscript to maintain focus and cohesion.

Line 137 in the original manuscript

~~REMOVED: To explore opportunities for simulating human PVT neurons in vitro, we investigated whether human thalamic organoids contain PVT-like neuronal populations. PVT-like neurons were identified in the thalamic organoids created by Kiral et al.⁴⁴ (**Supplementary Fig. 2d-e**). These neurons expressed PVT marker genes, including *SLC17A6*, *CALB2*, and *SNCA*. Their expression profile suggests the feasibility of modeling human PVT neurons in vitro.~~

Line 463 in the original manuscript

~~REMOVED:~~ Despite these limitations, this study indicates the disturbances of PVT neurons as a core pathology in BD, providing a promising foundation for developing new diagnostic and therapeutic strategies. Neuroimaging techniques targeting the PVT, such as MRI and PET, could enhance clinical assessment for BD. ~~PVT-like neurons in thalamic organoids underscore the feasibility of modeling PVT neurons from patient derived peripheral samples, which can accelerate the discovery of novel pharmacological interventions to the molecular mechanisms underlying BD.~~ Integrating multi-modal methodologies, encompassing cellular, molecular, and neuroimaging techniques, will provide a more comprehensive understanding of the pathological contributions of PVT in BD. PVT neurons represent a promising target for the development of advanced diagnostic and therapeutic strategies aimed at improving patient outcomes.

Line 833 in the original manuscript

~~REMOVED: Kiral et al. data can be obtained from GEO with accession number GSE210720.~~

Supplementary Figure

~~REMOVED: Supplementary Figure S2d-e in the original manuscript~~

4. In Figure 3b, it will help to add direction of "increased/decreased" for BD.

We thank the reviewer for this helpful suggestion. We have now added directional annotations in **Figs. 3b/5b** and **Supplementary Fig.15a** to clarify compositional changes for readers.

Figure 3b

b

Figure 5b

b

Supplementary Figure 15a

5. We are not sure why the authors decided to include reference to NeuCyto in Lines 158-159. If this “cell type” is ambient RNA contamination, then it is difficult to make conclusions based on such “cell type”.

We appreciate this observation. The NeuCyto cluster represents cytoplasmic components derived from neuronal populations and was used solely as a technical reference. Due to their indirectness, it was excluded from quantitative analyses. We have revised the text to clarify this point and prevent any misinterpretation.

Line 157 in the original manuscript

~~REMOVED: For reference, a decrease in Tha_NeuCyto_VGLUT2 (putative excitatory neuron cytoplasm, see Methods) indirectly supported these reductions (Supplementary Fig. 3c).~~

Nonetheless, the presence of such cluster would be interesting for snRNA-seq communities. Thus, we have revised the method sections and **Supplementary Fig. 13** describing NeuCyte for clarity.

Line 789 in the revised manuscript

Clusters designated as NeuCyto were considered neuronal cytoplasmic components from the following properties: low nuclear fraction, high content of mitochondrial RNA (**Supplementary Fig. 13a**), high expression of neuronal markers, virtually no expression of other glial markers (**Supplementary Fig. 1a, 3a, and 6a**), and distinctive clustering from probable empty droplets (**Supplementary Fig. 13b-c**). The probable empty droplets, designated as “Empty”, were defined as droplets filtered out by Cell Ranger with detected genes of 100 – 300. This range was set by QC procedure and barcode rank plot for UMI counts by Cell Ranger. NeuCyto droplets were robustly detected from 40/41 of our thalamic samples (occupying 1.46% of total cell nuclei) and 40/41 of our cortex samples (occupying 1.46% of total cell nuclei) as well as 20/20 of HBCA thalamic samples (occupying 1.66% of total cell nuclei). The clear separation of VGLUT+ and GAD+ NeuCyto clusters in the thalamus indicated that they were distinct

cytoplasmic components (Fig. 3a). We removed NeuCyto clusters for the subsequent analysis as non-nuclei droplets but retained for UMAP visualization and dot plots as a technical reference. Indeed, in the final nuclei suspension, we observed objects stained with Nonyl Acridine Orange (NAO) but negative for Propidium Iodide (PI) and Hoechst 33342 (Supplementary Fig. 13d), likely representing mitochondria-rich, nucleus-free cytoplasmic fragments corresponding to the NeuCyto clusters.

Supplementary Figure 13. NeuCyto clusters and additional spatial deconvolution.

a) Nuclear fractions and mitochondrial RNA fractions across samples in the thalamic integration dataset (see Fig. 3a). b) UMAP projection illustrating “Empty” clusters (ambient RNA, 1,000 droplets per sample) and canonical cell nuclei clusters shown in Fig. 3a. c) UMAP

projection of the same clusters as in panel b, with distinct coloring to highlight Tha_NeuCyto_VGLUT, Tha_NeuCyto_GAD, and Empty clusters. d) Microscopic images of cytoplasmic fragments (white arrowheads) in two representative spots. Cytoplasmic fragments are indicated by NAO staining (green) in the absence of PI or Hoechst 33342 signals. Cell nuclei appear magenta due to combined PI and Hoechst 33342 staining. e) CytoSPACE deconvolution based solely on our snRNA-seq dataset (excluding the HBCA dataset). f) CytoSPACE deconvolution using only the CT06 snRNA-seq dataset. g) UMAP projection displaying three NeuCyto sub-clusters within the thalamus. h) Predicted spatial distribution of the three NeuCyto sub-clusters. Color intensity in the side bars reflects the estimated number of cells per spatial spot. For panels g and h, only control samples were used to prevent potential bias from BD-associated alterations in spatial deconvolution.

6. Please correct Lines 255-256:

related to synaptic function, although microglia are primarily regarded as an immune-related cell type

Microglia has well known role in synaptic plasticity and modification of neuronal connectivity, which is also one of their primary roles in the brain.

We thank the reviewer for this helpful suggestion. In response, we have revised the relevant sentence (moved to the Discussion) to better reflect the established role of microglia in regulating synaptic plasticity and neuronal circuitry. The revised sentence strengthens our interpretation and supports the overall argument.

Line 254 in the original manuscript

~~REMOVED:~~ Furthermore, all five DEG-GOs identified within Tha_Micro_1 were related to synaptic function, ~~although microglia are primarily regarded as an immune-related cell type~~ (Supplementary Table 5).

Line 556 in the revised manuscript

Beyond cell-type-based analysis, we identified trans-cellular meta-gene programs between excitatory neurons and microglia (Excit-Micro meta-gene programs), likely reflecting physiological synaptic interactions between these cell types. **Microglia are known to modulate synaptic plasticity and neuronal circuitry through dynamic interactions with neurons⁷⁶.**

7. Figure 8a: show p-adjusted line on the figure

We have updated **Fig. 8a** to include a line indicating the Bonferroni-corrected significance threshold. This improves visual interpretation of the statistical results.

Figure 8a

8. Lines 452-460 repeat lines 463-472.

Thank you for pointing this out. We have removed the redundant text to improve clarity and flow.

Line 452 in the original manuscript

~~REMOVED: Despite the limitations of this study, our findings suggest that disturbances in PVT neurons constitute a central pathology in BD. Neuroimaging techniques targeting the PVT can enhance clinical assessments of BD. The presence of PVT-like neurons in thalamic organoids highlights the feasibility of modeling PVT neurons from patient-derived peripheral samples. This approach could facilitate the discovery of pharmacological interventions that target the molecular mechanisms underlying BD. The integration of multi-modal methodologies, including cellular, molecular, and neuroimaging techniques, will offer a more comprehensive understanding of the pathological role of PVT neurons in BD. PVT neurons emerge as a promising target for the development of novel diagnostic and therapeutic strategies for patients with BD.~~

9. Correct misprint:

Fig1a Neruon-microglia synaptic interaction in BD (Figure 5)

Thank you for pointing this out. We have corrected the typographical error. In addition, we recognized that the original legend for **Fig. 1a** did not adequately explain the figure. We have therefore revised the legend of **Fig. 1a** to improve clarity.

Reviewer #2

Nishioka et al. present a snRNA-seq study of PVT in 21 bipolar disorder (BD) patients and 20 controls. BD is a severe mental health condition marked by recurring manic and depressive episodes, requiring new treatments due to resistance, adverse effects, and high suicide rates. Recent research has identified the thalamus, particularly the paraventricular thalamic nucleus (PVT), as a crucial brain region for mood regulation, with significant differences in BD patients compared to controls. Authors report that thalamus samples from the BD patients they studied exhibited a 50% reduction in PVT neurons, which also showed notable downregulation in genes related to synaptic function and ion channel regulation. Additionally, disruptions in interactions between PVT neurons and microglia, particularly in AMPA receptor signaling pathways, were observed. These findings suggest that targeting AMPA signaling in PVT neurons could pave the way for innovative treatments and diagnostics aimed at improving outcomes for BD patients. This paper has several strengths: Well written, addresses important questions, focuses on an understudied brain region, uses mostly state-of-the-art methods. The biggest concern has to do with overinterpretation of results, with too little consideration for confounding and other limitations. There is also insufficient methodological detail in several areas.

We sincerely thank the reviewer for their thoughtful and constructive comments. In response, we have carefully revised several sections to avoid overinterpretation and to better reflect the limitations of our data. Specifically, we have removed or revised speculative statements concerning SYNDIG1 and SHISA9 proteins.

The Methods section has also been expanded to explicitly address potential confounders, including disease status (Dx), age, sex, postmortem interval (PMI), and pH. Additionally, we now provide a clear rationale for excluding certain other potential confounders, such as experimental/sequencing batch, substances at death, race, and cause of death, based on their minimal impact or the risk of introducing bias due to incomplete metadata. We have further revised the Methods to enhance clarity and reproducibility, particularly in describing our thalamic sampling strategy.

The reviewer's comment led us to identify and correct an error in our integration of the Human Brain Cell Atlas (HBCA) dataset (Siletti et al., Science 2023). We had inadvertently included hypothalamic samples instead of the intended epithalamic ones. Specifically:

- Incorrect samples used:

10X190_3, 10X190_4, 10X193_7, 10X193_8, 10X348_3, 10X348_4, 10X354_5, 10X354_6, 10X361_3, 10X361_4, 10X362_1, 10X362_2, 10X375_3, 10X375_4, 10X377_1, 10X377_2, **10X389-1, 10X389-2**, 10X393_1, 10X393_2

- Correct samples:

10X190_3, 10X190_4, 10X193_7, 10X193_8, 10X348_3, 10X348_4, 10X354_5, 10X354_6, 10X361_3, 10X361_4, 10X362_1, 10X362_2, 10X375_3, 10X375_4, 10X377_1, 10X377_2, **10X389-5, 10X389-6**, 10X393_1, 10X393_2

The samples 10X389-1 and 10X389-2 are derived from hypothalamus tissue, whereas 10X389-5 and 10X389-6 are derived from the epithalamus, as originally intended. We apologize for this

oversight.

To correct this, we re-performed the integration using the correct epithalamus-derived samples, followed by the full downstream analyses. The original integration, which included hypothalamic samples, was enriched in Splatter neuron clusters due to the nature of hypothalamic tissues, as illustrated in Siletti et al. 2023. As expected, the revised integration showed a reduction in Splatter neuron clusters and yielded clearer cluster distinction. Moreover, we adjusted the clustering resolution accordingly, since the previous resolution was overly high. Given that our aim is to characterize PVT neurons, we optimized the resolution to better distinguish PVT neurons from other thalamic excitatory populations.

Subsequently, we revised the downstream analyses based on the corrected HBCA-integrated dataset. As a result, the characteristics of CytoSPACE-based deconvolution, the correspondence to mouse PVT neurons, and the gene set enrichment analysis (GSEA) of PVT-neuron-specific genes became clearer and more interpretable, as detailed in the revised sections. We thank the reviewer for helping us identify and address this issue, which improved the robustness and clarity of our dataset and analyses. Notably, the excluded datasets (10X389-1 and 10X389-2) were not critical to our main analyses and appeared to reflect off-target contamination. Their exclusion does not affect the validity of the original results.

Line 733 in the revised manuscript

The original sample IDs were 10X190_3, 10X190_4, 10X193_7, 10X193_8, 10X348_3, 10X348_4, 10X354_5, 10X354_6, 10X361_3, 10X361_4, 10X362_1, 10X362_2, 10X375_3, 10X375_4, 10X377_1, 10X377_2, **10X389-5, 10X389-6**, 10X393_1, and 10X393_2, referred to as HBCA01-20 in our study (**Supplementary Data 11**).

Figure 2. Characterization of human PVT neurons.

a) UMAP projection displaying cell clusters identified in the HBCA-integrated thalamic dataset. **88.6%** of ExN_Tha_CALB cells in **Fig. 1b** are included within the ExN_CALB cluster here. **b) Top:** Representative paraventricular thalamus with a dashed black rectangle indicating the target region for Visum analysis. **Bottom:** Illustrated version with topographical orientation. D, dorsal. V, ventral. M, medial. L, lateral. **c) Left:** Spatial transcriptomic data delineating 15 distinct clusters, each represented by a different color, illustrating the spatial organization of gene expression across the paraventricular thalamus. Right: Calretinin immunohistochemistry staining of the adjacent section of Visum analysis. **d)** Spatial localization of cell clusters as predicted by CytoSPACE deconvolution, illustrating their estimated anatomical distribution within the thalamus. Color intensity in the sidebars indicates the estimated number of cells per spot, providing a quantitative measure of cell density across regions. **e) Merged view of the spatial clusters from panel d for comparison with panel c.** **f)** Matching scores of each cell cluster to mouse PVT neurons. **g)** Proportions of mouse PVT neuron subtypes from Gao et al. and Shima et al. corresponding to human ExN_CALB clusters. All PVT subtypes refer to mouse

PVT neurons; human PVT neurons were not subclassified in this study. aPVT, anterior PVT. The annotated PVT subtypes in pPVT, posterior PVT. AL, anterior lateral. AM, anterior medial. PL, posterior lateral. PM, posterior medial. PVT# in Gao et al. denotes the original subtype classification. **h**) GO network visualization for PVT-enriched genes, illustrating major functional pathways related to dopamine metabolism and calcium/ion channels. GO terms with enrichment scores > 0.5 and network connectivity > 2 are shown; core genes are highlighted. Core genes involved in these pathways are highlighted. **i**) Spatial expression profiles of key PVT marker genes, with darker red indicating higher expression. *PVALB* serves as a negative control to demonstrate spatial specificity.

Concerns in order of importance:

- *Anatomical definition of PVT*
- *CALB1 and CALB2 were used to define PVT, but they are not exclusive to PVT; CALB1 is broadly expressed in all 'matrix' nuclei; CALB2 is expressed in all midline and all intralaminar nuclei*
- *Needs more detailed description (ideally with illustration) how the "medial thalamus" was dissected; e.g. more anterior part bordering anterior thalamic nuclei or more posterior part bordering mediodorsal thalamus*

We agree with the reviewer that *CALB1* and *CALB2* are not exclusive markers of the paraventricular thalamic nucleus (PVT). In the revised manuscript, we have clarified that these genes were initially used as putative markers for PVT neurons. This preliminary identification was subsequently refined through anatomical localization using spatial transcriptomic data and anchor-based transfer from mouse PVT neuron data. Our anatomical definition of the PVT follows Uroz et al. 2004.

Our workflow for identifying human PVT neurons proceeded as follows. First, excitatory neuron clusters co-expressing *CALB1* and *CALB2* were identified as putative PVT neuron candidates. These were then validated by (i) spatial localization using CytoSPACE deconvolution and (ii) transcriptomic correspondence to mouse PVT neurons. While the mouse PVT is anatomically well-defined, the human PVT lacks a sharp boundary with surrounding regions. This anatomical ambiguity is also reflected in transcriptomic space, where PVT clusters form a continuum with adjacent excitatory populations, consistent with observations in Schulmann et al. (bioRxiv 2024). We have revised the text to more clearly reflect this stepwise identification process.

To further clarify our dissection protocol, we have added a new Supplementary Figure (**Supplementary Fig. 10b-d**) illustrating how blocks were cut from coronal thalamic slabs containing the mammillary body, targeting the mid-PVT. A more detailed description of the dissection strategy has also been incorporated into the Methods section. In addition, we have attached a demonstration video illustrating the dissection of slabs containing the PVT from the human postmortem brain, narrated by Tadafumi Kato. This video was made for the brain cutting in this project in 2013, as an internal communication between Japan and Canada. As we have no copyright for the photo in this video, this video is not to be public. We attach this video only for review purposes.

In retrospect, “a relatively anterior side of PVT” in the original manuscript was inappropriate. This description was derived from the CT06 section for Visium analysis. We aimed at the mid-PVT and inherent anatomical variation resulted in slight differences in sampling positions along the anterior-posterior axis. As anatomical studies of the human PVT remain limited, we hope that our findings, including candidate molecular markers, may contribute to future efforts to better characterize this region.

Line 483 in the original manuscript

~~REMOVED: We selected slabs including a relatively anterior side of PVT and cut the specimen with enough margins from PVT in the coronal section (Fig. 1a).~~

Line 102 in the revised manuscript

Cytoarchitecture of PVT neurons

Among the thalamic excitatory neurons, ExN_Tha_CALB cells in **Fig. 1b** were putatively supposed to be PVT neurons, as characterized by the expression of *CALB1* and *CALB2*—markers for PVT neurons in humans³⁶ and rodents^{37,38}—while these two markers were also expressed in other clusters. These putative PVT neurons were also robustly identified in the Human Brain Cell Atlas (HBCA) dataset⁶, as demonstrated by the integration of our and HBCA datasets (**Fig. 2a**). ExN_CALB in **Fig. 2a** corresponded to ExN_Tha_CALB in **Fig. 1b**. The putative PVT neurons (ExN_CALB cells) were detected from all 20 our samples and 20 HBCA samples (see **Methods**). This integrated dataset was used for subsequent analyses to enhance generalizability.

As *CALB1* and *CALB2* expression alone was insufficient to confirm that the ExN_CALB clusters represented PVT neurons, we validated this classification using two approaches: (1) spatial transcriptomic deconvolution and (2) anchor-based label transfer using mouse PVT neuron reference data. We obtained spatial transcriptomic data of the medial thalamus from one control sample as a representative specimen (**Fig. 2b**). The PVT was characterized through unbiased clustering of spatial transcriptomic data and partially corresponded to the areas immunostained by calretinin, a protein coded by *CALB2*³⁶ (**Fig. 2c**). Deconvolution of thalamic spatial transcriptomic data using the integrated cell clusters by CytoSPACE³⁹ indicated that ExN_CALB cells were predominantly localized in the paraventricular regions of the thalamus with a clear demarcation from other excitatory neuron populations (**Fig. 2d-e**), aligning with the anatomical definition of the PVT by Uroz et al.³⁶. Oligodendrocytes were concentrated in the lateral side of the thalamus, while astrocytes were primarily found in the medial side. Oligodendrocyte precursor cells (OPCs) and microglia displayed a diffuse distribution throughout the thalamus. Expectedly, ExN_CALB corresponded most closely with mouse PVT neurons when employing anchor transfer with published single-cell datasets from mice^{23,40} (**Fig. 2f**). These data confirmed the validity of ExN_CALB as PVT neurons. Upon closer examination of PVT neuron subtypes, the majority of human PVT neurons, particularly those from our dataset, corresponded to mouse posterior PVT neurons (**Fig. 2g**).

Line 649 in the revised manuscript

Bulk specimens of the medial thalamus and frontal cortex (BA10) were dissected from archived fresh-frozen slabs and then sectioned at a thickness of 200 μm using a CM3050S cryostat-microtome (Leica Microsystems, Wetzlar, Germany). First, the whole brain was bisected into left and right hemispheres. The left hemisphere was snap-frozen in 2-methylbutane precooled

to -40 °C and stored at -80 °C, while the right hemisphere was fixed in 10% neutral buffered formalin (phosphate buffer, pH 7.0; Fisher Scientific, Cat. #SF-100-20) for paraffin embedding and sectioning. Within a half hemisphere, the PVT was macroscopically identified, with its anterior boundary defined by the coronal plane at the anterior edge of the mammillary body and its posterior boundary by the coronal plane at the anterior edge of the posterior commissure, spanning approximately 20 mm along the anterior-posterior axis. A 10-mm-thick coronal slab encompassing the PVT was then selected and excised. Each slab extended anteriorly to the center of the mammillary body and posteriorly to midbrain structures including the red nucleus and substantia nigra. From each slab, tissue blocks were prepared to include the PVT together with major adjacent landmarks (putamen, caudate, and insular cortex), ensuring accurate anatomical orientation and precise localization of the PVT (**Fig. 1a**, **Supplementary Fig. 10b-d**). Dissections were typically performed from the posterior aspect of the slab, with each block carefully inspected to confirm that the slab contained the appropriate rostrocaudal level of the PVT. Although the procedure was designed to target the mid-PVT, unavoidable interindividual variation in gross neuroanatomy occasionally resulted in slight shifts in sampling position along the anterior-posterior axis. The medial region of the thalamic cryosections and the gray matter part of the cortical cryosections were excised for snRNA-seq. Approximately 50 mg of cryosections, corresponding to two to four cryosections, underwent the subsequent cell nuclei isolation.

Supplementary Figure 10. Sample information and anatomical dissection.

b) Schematic illustration of the targeted thalamic region in the left hemisphere. The 10-mm-thick slab selected for PVT dissection is indicated by violet lines. The center of the mammillary body and the red nucleus/substantia nigra, used as anatomical landmarks, are indicated by red and green circles, respectively. c) Representative coronal brain slab encompassing the medial thalamus (black dashed rectangle), the region of interest in this study. d) Representative thalamic tissue block for cryosectioning.

Review material

Dissection_PVT.wmv (video):

Demonstration video illustrating the dissection of slabs containing the PVT from the human postmortem brain.

- *Gao et al. and Shima et al. have identified major differences within PVT, especially along anterior-posterior axis; the author only included a general scoring of cells relative to these mouse datasets, not specific to PVT subdivisions*

We thank the reviewer for this insightful suggestion. Our anatomical dissection did not explicitly distinguish between anterior and posterior subregions in the human PVT due to limitations in tissue availability. In response to this comment, we performed a new analysis by mapping our identified ExN_CALB cluster (human PVT neurons) onto the Gao et al. and Shima et al. mouse datasets using an anchor-based integration approach. We used the corrected HBCA-integrated dataset for this new analysis.

This analysis revealed that the ExN_CALB population in our dataset aligns more closely with posterior PVT (pPVT) neurons in both reference datasets. Notably, despite methodological differences (Gao et al. used single-nucleus RNA-seq with the 10x Chromium platform, whereas Shima et al. employed single-cell RNA-seq with Quartz-Seq2), the observed correspondence was consistent across datasets. These findings have been added to the Results and Methods sections, and the mapping results are presented in **Fig. 2g**.

Line 123 in the revised manuscript

Expectedly, ExN_CALB corresponded most closely with mouse PVT neurons when employing anchor transfer with published single-cell datasets from mice^{23,40} (**Fig. 2f**). These data confirmed the validity of ExN_CALB as PVT neurons. **Upon closer examination of PVT neuron subtypes, the majority of human PVT neurons, particularly those from our dataset, corresponded to mouse posterior PVT neurons (Fig. 2g).**

Line 513 in the revised manuscript

In mice, recent work has revealed functional and transcriptomic differences between the anterior and posterior PVT^{23,38,40}. In our study, most human PVT neurons corresponded to the posterior PVT neurons in mice. Future investigations examining the entire PVT will be essential to clarify the evolutionary correspondence of PVT neurons between humans and mice.

Line 869 in the revised manuscript

Matching mouse PVT neurons to the human PVT neurons

To identify cell types in our human snRNA-seq data that correspond most closely to mouse PVT neurons, we performed anchor-based label transfer using the Seurat functions FindTransferAnchors and TransferData. As reference datasets of mouse PVT neurons, we used the snRNA-seq data from Gao et al. (2023)⁴⁰ and the single-cell RNA-seq data from Shima et al. (2023)²³. The mouse gene name was converted to the corresponding human gene name in the Ensemble database. Prior to anchor transfer, each dataset was independently processed using Seurat, including normalization, scaling, and principal component analysis (PCA). For the Gao et al. dataset, we followed the authors' published analytical pipeline to identify PVT subtypes, with the exception of PCA dimensions to improve clustering resolution, and annotated PVT subtypes based on the marker genes presented in Figure 3 of Gao et al. (2023). For the Shima et al. dataset, we used their original cell type annotations.

- *Consideration of alternative explanations for differences in cell type composition*
- *Dissection likely influences which excitatory populations were captured. How are cell populations represented across the 40 individuals?*

We fully agree with the reviewer that dissection variability may influence the observed compositional differences. To address this concern, we systematically examined the distribution of major cell types and excitatory neuron subpopulations across all 41 individual samples, as presented in the newly added **Supplementary Fig. 3b-c**. The Tha_ExN_CALB cluster was consistently detected across samples, with two outliers: BD13 and CT10. Importantly, even when these two samples were excluded, the compositional reduction of Tha_ExN_CALB remained, supporting the robustness of our findings.

To further address the potential influence of dissection variability or anatomical outliers, we performed an additional analysis using repeated random subsampling. Specifically, we randomly selected 10 BD and 10 control samples from the full dataset (21 BD vs. 20 controls) and assessed disease-associated fold-changes (FC) using sccomp, while adjusting for Age and Sex. This approach allowed us to test whether the originally observed Dx-based FCs might have been driven by a small number of anatomical outliers.

We repeated this procedure 1,000 times, and the resulting FC distributions are presented in the newly added **Supplementary Fig. 3e**. As shown, over 97.5% of the random subsamples yielded Dx-based FCs < 0.79 for both Tha_ExN_CALB and thalamic excitatory neurons in total,

suggesting that the observed compositional reductions are unlikely to be artifacts driven by a small number of outlier samples with atypical anatomical positions.

In addition, we have added scCODA analysis as suggested by Reviewer 1.

Line 141 in the revised manuscript

Cellular compositional changes in the medial thalamus of BD

We subsequently examined the cellular compositional and transcriptional alterations in the medial thalamus and frontal cortex between 21 BD patients and 20 controls. The thalamus and frontal cortex were analyzed separately to enhance precision. Cellular compositions refer to the relative proportions of cell clusters within the total population of cell nuclei.

We obtained 146,643 cell nuclei in 18 unique cell clusters and 2,139 droplets in two NeuCyto clusters after QC (**Fig. 3a** and **Supplementary Fig. 3a**). Two NeuCyto clusters were removed as non-nuclei from the subsequent analysis. Tha_ExN_CALB in **Fig. 3a** corresponded to PVT neurons. **All major cell classes were consistently detected across the 41 samples (Supplementary Fig. 3b).** The Tha_ExN_CALB cluster was also robustly identified in all samples among thalamic excitatory neurons with two outliers, BD13 and CT10 (**Supplementary Fig. 3c**).

Significant compositional reductions were observed exclusively in thalamic excitatory neurons (FDR < 0.05, FC = 0.54 – 0.59), including PVT neurons (Tha_ExN_CALB, FDR = 0.00488, FC = 0.55), as determined by sccomp⁴² (**Fig. 3b, Supplementary Table 3**). In contrast, five oligodendrocytes and astrocytes clusters showed compositional increases (FDR < 0.05, FC = 1.22 - 1.55). **The reduction of Tha_ExN_CALB remained significant even after excluding the two outlier samples, BD13 and CT10 (FDR = 0.011, FC = 0.592).** In examining six major cell classes, we observed a significant compositional reduction in excitatory neurons (FDR = 0.00525, FC = 0.58) and compositional increases in oligodendrocytes and astrocytes in BD (FDR = 0.0168 and 0.004, FC = 1.29 and 1.44, respectively) (**Supplementary Fig. 3d**). **Complementary analysis using scCODA⁴³ also supported a consistent compositional reduction in thalamic excitatory neurons, including PVT neurons (Supplementary Data 4).** In the scCODA analysis, Tha_ExN_CALB and Tha_ExN_PVALB_1 exhibited comparable compositional decreases (FC = 0.686 and 0.663).

Variability in tissue dissection may also influence cell-type composition; however, such variability is difficult to control during sampling and to explicitly model statistically. To address this potential confound, we randomly selected 10 BD and 10 control samples from the full dataset (21 BD vs. 20 controls) and used sccomp to evaluate BD-associated FCs across 1,000 random subsamples. Notably, over 97.5% of these subsamples yielded diagnosis-associated FCs < 0.79 for both Tha_ExN_CALB and the broader thalamic excitatory neuron population (**Supplementary Fig. 3e**), suggesting that the observed compositional reductions are unlikely to be driven by a small number of anatomically atypical outlier samples.

Supplementary Figure 3. Cell-type annotation in the thalamus.

b) Proportions of major cell classes across individual thalamic samples. **c)** Proportions of excitatory neuron subtypes across individual thalamic samples. **d)** Compositional changes of major cell classes estimated by sscmp, with $\log_2(\text{Fold-change})$ on the x-axis and $-\log_{10}(\text{FDR})$ on the y-axis. Circle size corresponds to the nuclei count within each cluster, and color represents major cell classes. **e)** Distribution of fold-change estimates from 1,000 random sub-samplings (10 BD cases vs. 10 controls) for PVT neurons (left) and thalamic excitatory neurons (right), as determined by sscmp. Dashed bold lines indicate the median, and dotted thin lines represent the 2.5th and 97.5th percentiles.

• *Spatial transcriptomics shows Exc_PVALB3, but no Exc_PVALB1/2. Why?*

We appreciate this important question. As described above, we corrected the HBCA integration and used the revised dataset for the CytoSPACE analysis. In the original analysis, the CytoSPACE deconvolution was performed at an excessively high resolution, partly due to the erroneous inclusion of hypothalamic samples containing numerous Splatter clusters. In the updated integration, only ExN_CALB, ExN_PVALB_1, and ExN_PVALB_2 were observed as thalamic excitatory neurons. The original ExN_PVALB_1 and ExN_PVALB_3 clusters were highly similar and are now indistinguishable in the revised integration, consistent with the case-control thalamus analysis (**Fig. 3**). In retrospect, distinguishing ExN_PVALB_1 from ExN_PVALB_3 in the original **Fig. 2** was inadequate, given our focus on PVT neurons.

In the revised **Fig. 2**, ExN_PVALB_1 was primarily detected outside the PVT area. The ExN_PVALB_2 subtype was not detected, consistent with our original submission. This finding aligns with the absence of ExN_PVALB_2 in the CT06 snRNA-seq dataset, thereby supporting

the validity of the CytoSPACE-based spatial deconvolution. In the HBCA integration dataset, the CT06 dataset comprised 4,510 nuclei in total, including 127 ExN_CALB, 364 ExN_PVALB_1, and 0 ExN_PVALB_2, as shown in **Supplementary Data 11**. The absence of ExN_PVALB_2 in CT06 is likely attributable to the anatomical position of the cryosection.

To corroborate this finding, we performed additional CytoSPACE deconvolution using two snRNA-seq datasets: an integrated set comprising our 20 controls (excluding HBCA samples) and the single CT06 snRNA-seq dataset. Both analyses yielded cellular distributions similar to those observed in the original deconvolution (**Supplementary Fig. 13**).

Line 851 in the revised manuscript

Deconvolution of spatial transcriptomic data with snRNA-seq cell clusters

We deconvoluted the Visium spatial transcriptomic data using cell-type expression profiles obtained from snRNA-seq via CytoSPACE³⁹ (v1.1.0) with default parameters to estimate the spatial distribution of each cell type. The **reference** cell types included excitatory/inhibitory neurons, astrocytes, OPCs, oligodendrocytes, and microglia, as defined in the integrated set comprising the HBCA and our dataset (see **Fig. 2a**). The low representation of the ExN_PVALB_2 cluster in **Fig. 2d** was consistent with its low/no proportion in the CT06 snRNA-seq dataset (**Supplementary Data 11**), supporting the validity of our deconvolution approach. As a robustness check, we repeated the deconvolution using either our control thalamic dataset or CT06-derived data alone, both of which yielded consistent results (**Supplementary Fig. 13e-f**).

Figure 2. Characterization of human PVT neurons.

d) Spatial localization of cell clusters as predicted by CytoSPACE deconvolution, illustrating their estimated anatomical distribution within the thalamus. Color intensity in the sidebars indicates the estimated number of cells per spot, providing a quantitative measure of cell density across regions. **e)** Merged view of the spatial clusters from panel d for comparison with panel c.

Supplementary Figure 13. NeuCyto clusters and additional spatial deconvolution.

e) CytoSPACE deconvolution based solely on our snRNA-seq dataset (excluding the HBCA dataset). **h)** CytoSPACE deconvolution using only the CT06 snRNA-seq dataset.

Supplementary Data

11.DataS11.HBCA.profiles.txt

Sample profiles of HBCA data used in this study, including cell cluster counts.

- *Is neuronal loss in BD limited to PVT (as defined by ExN_CALB cluster) or only detectable there due to more consistent dissection of the midline?*

While the prominent reduction was observed in the ExN_CALB population (PVT neurons in our study), other thalamic excitatory neuron subtypes (Tha_ExN_PVALB_1 and Tha_ExN_PVALB_2) also exhibited comparable compositional reductions, as illustrated in **Fig. 3b**. In response to Reviewer 1's suggestion, we additionally applied scCODA, which supported these findings and revealed a slightly more pronounced decrease in Tha_ExN_PVALB_1.

Immunohistochemistry (IHC) further confirmed a reduction in excitatory neurons across the medial thalamus. This analysis was not restricted to the PVT neurons but encompassed all thalamic excitatory neurons. Given that PVT neurons represent only a small fraction of total excitatory neurons (16.2%), the observed IHC reduction cannot be attributed solely to changes within the PVT. These findings suggest that the loss of PVALB-positive neurons also contributes to the broader excitatory neuron decline, indicating a more generalized neuronal vulnerability within the medial thalamus.

As the central focus of our study is PVT, tissue sampling was anatomically centered on this region. While neighboring nuclei, such as the mediodorsal (MD) nucleus, are of significant

interest, they were not explicitly targeted in this analysis. Nonetheless, the observed compositional changes may implicate reductions in excitatory neuron populations extending into adjacent nuclei, potentially including the MD. A more definitive characterization of these changes will require future studies with targeted sampling of these adjacent regions.

We have revised the compositional analysis section to present the results more clearly.

Line 141 in the revised manuscript

Cellular compositional changes in the medial thalamus of BD

We subsequently examined the cellular compositional and transcriptional alterations in the medial thalamus and frontal cortex between 21 BD patients and 20 controls. The thalamus and frontal cortex were analyzed separately to enhance precision. Cellular compositions refer to the relative proportions of cell clusters within the total population of cell nuclei.

We obtained 146,643 cell nuclei in 18 unique cell clusters and 2,139 droplets in two NeuCyto clusters after QC (**Fig. 3a** and **Supplementary Fig. 3a**). Two NeuCyto clusters were removed as non-nuclei from the subsequent analysis. Tha_ExN_CALB in **Fig. 3a** corresponded to PVT neurons. **All major cell classes were consistently detected across the 41 samples (Supplementary Fig. 3b).** The Tha_ExN_CALB cluster was also robustly identified in all samples among thalamic excitatory neurons with two outliers, BD13 and CT10 (**Supplementary Fig. 3c**).

Significant compositional reductions were observed exclusively in thalamic excitatory neurons ($FDR < 0.05$, $FC = 0.54 - 0.59$), including PVT neurons (Tha_ExN_CALB, $FDR = 0.00488$, $FC = 0.55$), as determined by *sccomp*⁴² (**Fig. 3b, Supplementary Table 3**). In contrast, five oligodendrocytes and astrocytes clusters showed compositional increases ($FDR < 0.05$, $FC = 1.22 - 1.55$). **The reduction of Tha_ExN_CALB remained significant even after excluding the two outlier samples, BD13 and CT10 ($FDR = 0.011$, $FC = 0.592$).** In examining six major cell classes, we observed a significant compositional reduction in excitatory neurons ($FDR = 0.00525$, $FC = 0.58$) and compositional increases in oligodendrocytes and astrocytes in BD ($FDR = 0.0168$ and 0.004 , $FC = 1.29$ and 1.44 , respectively) (**Supplementary Fig. 3d**). **Complementary analysis using scCODA⁴³ also supported a consistent compositional reduction in thalamic excitatory neurons, including PVT neurons (Supplementary Data 4).** In the scCODA analysis, Tha_ExN_CALB and Tha_ExN_PVALB_1 exhibited comparable compositional decreases ($FC = 0.686$ and 0.663).

Variability in tissue dissection may also influence cell-type composition; however, such variability is difficult to control during sampling and to explicitly model statistically. To address this potential confound, we randomly selected 10 BD and 10 control samples from the full dataset (21 BD vs. 20 controls) and used *sccomp* to evaluate BD-associated FCs across 1,000 random subsamples. Notably, over 97.5% of these subsamples yielded diagnosis-associated FCs < 0.79 for both Tha_ExN_CALB and the broader thalamic excitatory neuron population (**Supplementary Fig. 3e**), suggesting that the observed compositional reductions are unlikely to be driven by a small number of anatomically atypical outlier samples.

To validate this finding, we compared the cell density of thalamic excitatory neurons and oligodendrocytes by immunohistochemical staining of VGLUT2 and SOX10, calculating cell density from 548,423 excitatory neurons and 1,819,912 oligodendrocytes in 36 thalamic samples (**Supplementary Data 5**). This immunohistochemical assessment revealed significant reductions of the thalamic excitatory neurons in BD ($P = 0.00781$, $FC = 0.53$) and no significant

differences in the oligodendrocytes ($P = 0.252$, $FC = 0.91$) (**Fig. 3c-d**). If incorporating PMI and/or pH to the GLM models, the results remained consistent (**Supplementary Data 6**). Note that immunohistochemistry directly assessed cell density, while compositional analysis using snRNA-seq data assessed relative proportional changes. The immunohistochemical assessment indicated that the compositional increase in oligodendrocytes and astrocytes in BD should be mainly derived from prominent reductions of excitatory neurons, not reflecting absolute cell density increases. **If we set *Tha_Oligo_1* as a reference cluster in scCODA analysis, assuming that the absolute counts of oligodendrocytes were comparable between BD and control from IHC results, we consistently observed compositional decreases in thalamic excitatory neurons (Supplementary Data 4).**

Figure 3. Compositional and transcriptional alterations in the thalamus.

b) Left: Proportional distribution of thalamic cell clusters, comparing BD and control. Total nuclei counts are normalized to 10,000, and the total nuclei count in BD and control is adjusted to an equivalent sum. Right: Compositional changes estimated by sccomp with $\log_2(\text{Fold-change})$ on the x-axis and $-\log_{10}(\text{FDR})$ on the y-axis. Circle size reflects the number of nuclei per cluster, and color denotes major cell classes.

Supplementary Data

04.DataS4.scCODA.txt

scCODA results for thalamic and cortical clusters, including estimated fold-changes and posterior probabilities for each covariate.

05.DataS05.IHC_count_data.txt

Immunohistochemical cell count data for thalamic excitatory neurons and oligodendrocytes.

06.DataS06.IHC_GLM.txt

Generalized linear model (GLM) results for immunohistochemistry of thalamic excitatory

neurons and oligodendrocytes.

- *Overinterpretation of mechanistic link between excitatory neurons and microglia*
- *SHISA9 and SYNDIG1 are both AMPAR auxiliary subunits, but an interaction between them is not known; SYNDIG1 encodes a transmembrane protein that is known to interact with AMPAR in neurons. Its function in microglia is not known. More evidence is needed of any claimed interactions in microglia.*
- *Discussion would benefit from a more nuanced handling of the data, given its limitations, and more attention to areas where evidence is less clear and multiple conclusions are possible, e.g.,*

We fully agree with the reviewer's comment and acknowledge that our previous interpretation overstated the relationship between SYNDIG1 and SHISA9 proteins. As correctly noted, both proteins are AMPA receptor (AMPA) auxiliary subunits, but there is no evidence supporting a direct protein-protein interaction between them. To our knowledge, the mechanistic link between excitatory neurons and microglia regarding SYNDIG1 and SHISA9 proteins remains unknown. We have thoroughly revised the description of this point in the manuscript and the relevant section of the Discussion accordingly.

In response to Reviewer #1's comment, we have incorporated RNAscope validation of *SYNDIG1* expression in microglia. The preferential localization of *SYNDIG1*-positive microglia in cortical gray matter aligns with our meta-gene programs, which implicate interactions between microglia and excitatory neurons. Nonetheless, we recognize that this data-driven construct warrants further experimental validation.

Abstract in the original manuscript

~~REMOVED: This interaction was further exemplified by the concurrent downregulation of SYNDIG1 in microglia and SHISA9 in PVT neurons, both of which are involved in AMPA receptor signaling pathways.~~

Line 279 in the original manuscript

~~REMOVED: The coordinated downregulation of SYNDIG1 in microglia and SHISA9 in excitatory neurons exemplifies a probable disturbance in microglia-neuron interactions for synapses in the BD thalamus, indicating a plausible pathological mechanism of SHISA9, one of the 36 putative BD-causal genes¹.~~

Line 415 in the original manuscript

~~REMOVED: The downregulation of SYNDIG1 in microglia and SHISA9 in PVT neurons might be an example of dysregulated molecular interactions within the Excit-Micro meta-gene programs associated with BD.~~

Line 314 in the revised manuscript

At the gene level, *SYNDIG1* and *SHISA9* encode auxiliary subunits for AMPA receptor (AMPA) complex^{56,57}, with both genes' robustness scores exceeding 0.7 (**Supplementary Table 4**). Notably, *SYNDIG1* was prominently expressed in microglia, while *SHISA9* was primarily

expressed in excitatory neurons (**Fig. 5d**). *SHISA9* consisted of core PPI networks of BD-associated downregulated genes in PVT neurons (**Fig. 4d**) and overlapped in the downregulated genes in *Tha_ExN_PVALB_1*, as one of the 36 putative BD-causal genes¹.

Line 556 in the revised manuscript

Beyond cell-type-based analysis, we identified trans-cellular meta-gene programs between excitatory neurons and microglia (Excit-Micro meta-gene programs), likely reflecting physiological synaptic interactions between these cell types. **Microglia are known to modulate synaptic plasticity and neuronal circuitry through dynamic interactions with neurons⁷⁶.** Notably, such meta-gene programs in the thalamus are disrupted in BD, further characterized by a significant decrease in synaptic gene expression in both PVT neurons and microglia. Ling et al. reported downregulation of the synaptic neuron and astrocyte program in schizophrenia¹⁰; our findings highlight downregulation of the synaptic **genes in the thalamic neuron and microglia in BD. The concurrent downregulation of *SYNDIG1* in microglia and *SHISA9* in excitatory neurons suggests disruptions of AMPAR signaling, as both genes encode auxiliary subunits of the AMPAR complex^{56,57}. *TIAM1*, another downregulated gene in microglia, has also been implicated in regulating synaptic AMPAR function⁷⁷.** The preferential localization of *SYNDIG1*-positive microglia in cortical gray matter is consistent with this interpretation. Nonetheless, these meta-gene programs remain data-driven constructs, and the roles of *SYNDIG1* in microglia—as well as their mechanistic links to *SHISA9* in PVT neurons—are currently unknown. Future studies employing animal and in vitro cellular models will be crucial for elucidating these mechanisms.

- *Findings in PVT may not apply to the rest of the thalamus*

As pointed out, the findings from the PVT may not be directly applicable to the entire thalamus. Other areas outside of the medial thalamus need further investigations. We have added this argument in the discussion.

Line 483 in the revised manuscript

The observed reduction in thalamic excitatory neurons, including PVT neurons, **may be associated with the reduced thalamus volume reported in BD¹⁹, although our analysis was limited to the medial thalamus. Given that variability in tissue dissection represents a potential confounder, future studies encompassing the entire thalamus will be required to determine whether this reduction is specific to the medial thalamus or extends to the whole structure. It remains unclear whether the present findings alone can account for the volumetric reductions observed in BD.**

- *Avoid anthropomorphic statements such as “Mice with recurrent depressive episodes”*
- *Claims about high-frequency firing and role of *KCNQ3* in potential response to high firing are too speculative; citations would be needed to support such a claim for this gene*

We have revised the Discussion section to adopt a more cautious tone. Specifically, we

removed anthropomorphic phrases such as “mice with recurrent depressive episodes” and added a detailed explanation for *KCNQ3* and potential hyper-excitability. We appreciate the reviewer’s emphasis on scientific rigor and nuance.

KCNQ3 encodes the Kv7.3 subunit, which forms heteromers with Kv7.2 to generate the non-inactivating, slowly activating M-current near resting membrane potential (Jentsch. Nature Reviews Neuroscience 2000, 1:21-30; Wang et al. Science 1998, 282:1890-1893). These channels modulate neuronal excitability by affecting input resistance, the action potential threshold, and the medium afterhyperpolarization (mAHP), thereby controlling firing frequency (Yue and Yaari. Journal of Neurophysiology 2006, 95:3480-95). Suppression of the M-current lowers the action potential threshold, increases afterdepolarization, and depolarizes the axonal resting potential, leading to enhanced excitability and a predisposition to high-frequency firing (Brown and Adams. Nature 1998, 283:673-676; Delmas and Brown. Nature Reviews Neuroscience 2005, 6: 850-862; Selyanko and Brown. Neuron 1996, 16:151-162).

Kv7.2/7.3 channels are enriched at the axon initial segment and nodes of Ranvier, where they stabilize the resting potential. This stabilization, which is distinct from the inhibitory effect of M current at the soma and dendrites, inhibits Na⁺ channel inactivation by preventing excessive depolarization, thereby maintaining their availability for reliable action potential initiation and propagation and preserving normal, rather than excessive, excitability (Shah et al. PNAS 2008, 105:7869-7874; Battenfeld et al. Journal of Neuroscience 2004, 34:3719-3732). Thus, downregulation of Kv7.3 could be associated with excessive excitability.

As the above explanation is too lengthy for this manuscript, we have added the shorter version in the main manuscript.

Line 502 in the revised manuscript

Our prior findings showed that regulation of neurotransmission in the PVT via tetanus toxin or Designer Receptors Exclusively Activated by Designer Drugs (DREADDs) induced **recurrent depression-like behavioral changes** even in wildtype mice^{24,25}.

Line 526 in the revised manuscript

Potassium channels, such as *KCNQ3*—a gene identified as a key DEG in BD—are **pivotal regulators of high-frequency firing. *KCNQ3* encodes the Kv7.3 subunit, which assembles with Kv7.2 to form heteromeric channels mediating the non-inactivating, slowly activating M-current near the resting membrane potential⁷³. The M-current regulates neuronal excitability by modulating input resistance, action potential threshold, and the medium afterhyperpolarization, thereby influencing firing frequency⁷⁴. Inhibition of the M-current lowers the action potential threshold, enhances afterdepolarization, and depolarizes the axonal resting potential, resulting in increased excitability and a propensity for high-frequency firing⁷⁵. Accordingly, reduced Kv7.3 expression may contribute to neuronal hyperexcitability.**

- *Lack of relevance of thalamic organoid integration analysis*
- *Authors did not sample the entire thalamus to claim that organoids recapitulate properties*

of PVT

- *This analysis is not really related to the rest of the paper*
- *Insufficient methodological detail on Quality Control*
- *Authors should describe QC and filtering process in more detail*

We agree with the reviewer that the organoid analysis, while interesting, was peripheral to the central findings. Another reviewer also pointed out that this section is irrelevant and should be removed. Initially, we had incorporated this part for the final discussion. In retrospect, this part is not essential for our manuscript. We have removed the organoid comparison from the main text and supplementary figures.

Line 137 in the original manuscript

~~REMOVED: To explore opportunities for simulating human PVT neurons in vitro, we investigated whether human thalamic organoids contain PVT-like neuronal populations. PVT-like neurons were identified in the thalamic organoids created by Kiral et al.⁴¹ (Supplementary Fig. 2d-e). These neurons expressed PVT marker genes, including SLC17A6, CALB2, and SNCA. Their expression profile suggests the feasibility of modeling human PVT neurons in vitro.~~

Line 463 in the original manuscript

~~REMOVED: Despite these limitations, this study indicates the disturbances of PVT neurons as a core pathology in BD, providing a promising foundation for developing new diagnostic and therapeutic strategies. Neuroimaging techniques targeting the PVT, such as MRI and PET, could enhance clinical assessment for BD. PVT-like neurons in thalamic organoids underscore the feasibility of modeling PVT neurons from patient-derived peripheral samples, which can accelerate the discovery of novel pharmacological interventions to the molecular mechanisms underlying BD.~~ Integrating multi-modal methodologies, encompassing cellular, molecular, and neuroimaging techniques, will provide a more comprehensive understanding of the pathological contributions of PVT in BD. PVT neurons represent a promising target for the development of advanced diagnostic and therapeutic strategies aimed at improving patient outcomes.

Line 833 in the original manuscript

~~REMOVED: Kiral et al. data can be obtained from GEO with accession number GSE210720.~~

Supplementary Figure

~~REMOVED: Supplementary Figure S2d-e in the original manuscript~~

- *No vascular or ependymal cells seen in the final annotated cell types: were they filtered due to low quality or not included because too few cells were represented?*

Vascular cells were annotated in the integrated datasets shown in **Fig. 2a**, **Fig. 3a**, and **Fig. 6a**, but with low proportions. Specifically, Endo (endothelial cells) and Peri (pericytes) in **Fig. 2a** comprised 0.70% of total nuclei; the Tha_Vascular cluster in **Fig. 3a** accounted for 0.216%; and the FrC_Vascular cluster in **Fig. 6a** represented 0.196% of total nuclei in each respective integration. Although further subclassification into endothelial cells and pericytes would be

possible at higher resolution, we did not pursue this, as our primary focus was the characterization of PVT neurons.

The integration of our thalamic samples with HBCA thalamic samples (**Fig. 2a**) contained a higher proportion of vascular cells than the integration of our thalamic samples alone (**Fig. 3a**). This likely reflects a technical limitation, as low-abundance cell types such as vascular cells are more susceptible to loss during clustering and integration procedures.

Vascular cells were not retained in the final annotated dataset (**Fig. 1b**), likely due to the integration of a large number of clusters from both thalamus and cortex, which comprise distinct cell types. Such integration may have lacked the resolution needed to preserve low-abundance populations, including vascular cells.

Ependymal cells were not annotated in our dataset, likely because they are difficult to isolate using our nuclear extraction protocol. These cells are embedded within a dense extracellular matrix, and our protocol employed a relatively mild detergent concentration (0.05% Tween 20) to preserve nuclear integrity and RNA quality in long-stored postmortem brain samples, including those with a postmortem interval (PMI) exceeding 50 hours. This relatively mild detergent condition may also have reduced the yield of vascular cells, as vascular tissue is enriched in collagen fibers, which are more resistant to dissociation into single cells. Resistance to detergent-based cell nuclei isolation procedures would be another reason for the low representation of vascular cells in our study.

Figure 2a (Integration of HBCA and our thalamic dataset)

Figure 3a (Integration of our thalamic dataset)

Figure 5a (Integration of our cortical dataset)

- *Insufficient methodological detail on anchor-based integration*
- *Was CCA or RPCA approach used, and why?*

For anchor-based processing, we employed Seurat’s “anchor-based transfer” approach using the FindTransferAnchors and TransferData functions, as described in the official tutorial “Mapping and annotating query datasets”

https://satijalab.org/seurat/articles/integration_mapping

We used the web-based Azimuth platform (<https://azimuth.hubmapconsortium.org/>), which internally uses the same functions. We have revised the manuscript to include this information.

While CCA- or RPCA-based integration could be used to align mouse and human cell types, we opted for anchor-based transfer using FindTransferAnchors and TransferData functions for the following reasons:

1. It allows for explicit, cell-to-cell mapping scores, providing a quantitative measure of how

well a human cell type corresponds to a specific mouse cell type.

2. Mouse and human expression profiles differ substantially, and anchor-based transfer is better suited for cross-species annotation without enforcing artificial alignment.
3. The mouse data from Shima et al. were generated using Quartz-seq2, a platform with different technical characteristics from 10x Genomics Chromium system, further supporting the use of transfer-based methods over direct integration.

Accordingly, we have revised the Methods section and uploaded a script for processing Gao et al. data in our GitHub repository.

Line 782 in the revised manuscript

Neuronal cell clusters in the frontal cortex and glial cell clusters were annotated using cell-type label transfer with Azimuth, using a cortex reference²⁹. **Azimuth internally employed anchor-based transfer using FindTransferAnchors and TransferData functions in Seurat.**

Line 869 in the revised manuscript

Matching mouse PVT neurons to the human PVT neurons

To identify cell types in our human snRNA-seq data that correspond most closely to mouse PVT neurons, we performed anchor-based label transfer using the Seurat functions FindTransferAnchors and TransferData. As reference datasets of mouse PVT neurons, we used the snRNA-seq data from Gao et al. (2023)⁴⁰ and the single-cell RNA-seq data from Shima et al. (2023)²³. The mouse gene name was converted to the corresponding human gene name in the Ensemble database. Prior to anchor transfer, each dataset was independently processed using Seurat, including normalization, scaling, and principal component analysis (PCA). For the Gao et al. dataset, we followed the authors' published analytical pipeline to identify PVT subtypes, with the exception of PCA dimensions to improve clustering resolution, and annotated PVT subtypes based on the marker genes presented in Figure 3 of Gao et al. (2023). For the Shima et al. dataset, we used their original cell type annotations.

- *Insufficient exploration of confounding variables for DGE (and cellular composition)*
- *Why were the covariates selected as described? The authors did not appear to have investigated other quality-related variables and explore variance they explained.*

We appreciate the reviewer's insightful comment and apologize for omitting the detailed rationale for covariate selection. In response, we have now added a comprehensive description of the covariate selection process and thoroughly revised the relevant section in the Methods.

Covariate Selection Strategy

Initially, we considered a wide range of potential covariates: diagnosis (Dx), Sex, Age, Postmortem interval (PMI), brain pH (pH), race, cause of death, refrigerator delay, substances at death, experimental/sequencing batch. Given our modest sample size (N = 41), covariates should be selected to minimize the risk of overfitting and multicollinearity.

- Race: The cohort was heavily biased toward Caucasian individuals, with only one Asian

sample. Thus, race was excluded from statistical modeling due to lack of variability.

- Cause of death: A significant imbalance was observed, with suicide overrepresented among cases (Fisher's exact test, 2.33×10^{-8}). To reduce multicollinearity, this covariate was excluded.
- Refrigerator delay (RD): RD contained three missing values, which is substantial for a sample of 41. Furthermore, RD was significantly correlated with PMI (Pearson's correlation, $P = 0.0181$). We prioritized PMI as a relatively more relevant potential covariate.
- Medication: Due to the wide heterogeneity of substances at death and missing information in several donors, subgrouping was not feasible with the available sample size.
- Experimental and sequencing batches: We had 21 batches for nuclei isolation and 16 sequencing batches in thalamic analysis. Modeling batch effects failed due to the large number of batches relative to sample size ($N = 41$).

The remaining candidate covariates were Dx, sex, age, PMI, and pH. Among them, pH was particularly correlated with Dx (GLM $P = 0.0074$; see revised **Supplementary Fig. 14b**), suggesting potential collinearity. Nonetheless, we tested models with all five covariates (Dx, Sex, Age, PMI, and pH), as these are commonly used in postmortem brain studies.

Compositional Analysis

For the compositional analysis, model fit was evaluated using Akaike Information Criterion (AIC) from generalized linear models (GLM) employing negative binomial distribution (glm.nb, MASS package), given that the scomp package does not provide scores indicating model fit like AIC. We compared multiple candidate models and ranked them by summed AIC rankings (rankSum) across both thalamic and cortical datasets.

The model including Dx, age, and sex (Dx + Age + Sex) consistently exhibited the lowest AIC by rankSum, indicating the best overall fit. The contributions of PMI and pH differed between the thalamic and cortical analyses. We selected PMI as the secondary covariate because pH showed a stronger correlation with diagnosis and represents a less direct measure of tissue integrity for both the thalamus and cortex (as described in the original manuscript, Line 672). Based on these findings, we selected Dx + Age + Sex as the primary covariate combination for compositional modeling, and Dx + Age + Sex + PMI as the secondary covariate combination.

In line with the reviewer's suggestion, we also assessed the relative contribution of each covariate using Type III likelihood ratio chi-square tests (Anova(type = 3) from the car package). The results are provided in **Supplementary Data 16**. Dx, Sex, and Age consistently demonstrated significant contributions to compositional variation.

Differential Expression Analysis

In the differential gene expression (DEG) analysis, we adopted the same covariate combination used in the compositional analysis. In response to the reviewer's comments, we calculated the AIC of generalized linear models (GLMs) for each gene's log-transformed expression [$\log(p + 1)$] to specifically evaluate model fit when incorporating PMI and/or pH, as MAST does not provide scores indicating model fit like AIC. Gene expression was aggregated at the specimen level and limited to genes expressed in at least 10% of cells within each cluster, consistent with the DEG criteria. This GLM-based modeling was performed solely to assess the impact of

covariates; the DEG analysis itself was conducted using `zlm` in the MAST package.

Among the tested models, Dx + Age + Sex yielded the lowest total AIC across genes by clusters, followed by Dx + Age + Sex + PMI. This supports the use of Dx + Age + Sex as the primary covariate combination and Dx + Age + Sex + PMI as a secondary model. In addition, we assessed the relative contribution of each covariate using Type III chi-square statistics (`Anova(type = 3)` from the `car` package) for both models. As these data are too huge to submit for the journal web system, we deposited these data in figshare.

We have revised the Methods section to incorporate these details. The updated description enhances the clarity and transparency of our statistical approach.

Line 897 in the revised manuscript

Covariate selection in compositional analysis

Initially, we considered a wide range of potential covariates: diagnosis (Dx), sex, age, postmortem interval (PMI), brain pH (pH), race, cause of death, refrigerator delay, substances at death, experimental batch, and sequencing batch. Given our modest sample size ($N = 41$), covariates should be selected to minimize the risk of overfitting and multicollinearity. Race was excluded from statistical modeling because the cohort was heavily biased toward Caucasian individuals, with only one Asian sample. Cause of death was excluded to reduce multicollinearity, because it exhibited a significant imbalance with suicide overrepresented among BD cases (two-sided Fisher's exact test, $P = 2.33 \times 10^{-8}$). Refrigerator delay contained three missing values, and RD was correlated with PMI (Pearson's correlation, $P = 0.0181$). We prioritized PMI as a relatively more relevant potential covariate. Due to the wide heterogeneity of substances at death and missing information in several donors, subgrouping based on substances at death was not feasible in our dataset. Regarding experimental and sequencing batches, we had 21 batches for nuclei isolation and 16 sequencing batches in thalamic analysis. Modeling batch effects was unsuccessful due to the large number of batches relative to the sample size ($N = 41$). Although experimental and sequencing batches could not be incorporated into the subsequent statistical modeling due to their high dimensionality relative to sample size, we confirmed that the identified cell clusters were broadly and consistently represented across batches (**Supplementary Fig. 14a, Supplementary Data 13 and 14**).

The remaining candidate covariates were Dx, sex, age, PMI, and pH. Among them, pH values obtained from cerebellum samples, may not directly represent the specimen conditions of the thalamus and frontal cortex; given probable regional variability, we treated pH as an indirect indicator for this study. Notably, pH and PMI were associated with Dx, with pH showing a stronger signal (**Supplementary Fig. 14b-c**). These associations likely reflect differences in cause of death, as BD samples predominantly originated from individuals who died by suicide, unlike control samples. **Nonetheless, we tested statistical models with all five covariates (Dx, Sex, Age, PMI, and pH), as commonly used covariates in postmortem brain studies.**

For the compositional analysis, model fit was evaluated using Akaike Information Criterion (AIC) from generalized linear models (GLM) employing negative binomial distribution (`glm.nb` in MASS [v7.3-60] package), because the `sccomp` and `scCODA` packages do not provide scores indicating model fit like AIC. In the GLM, total nuclei counts were normalized to 10,000,

with each cluster's count rounded to integers to reduce variability in total nuclei counts across individuals. We compared multiple candidate models and ranked them by summed AIC rankings across both thalamic and cortical datasets (see rankSum in **Supplementary Data 15**). The model including Dx, age, and sex (Dx + Age + Sex) showed the lowest overall AIC, indicating the best model fit. While the contributions of PMI and pH differed between the thalamic and cortical analyses, we selected PMI as the secondary covariate because pH showed a stronger correlation with diagnosis and represents a less direct measure of thalamus and cortex tissues. Based on these findings, we selected "Dx + Age + Sex" as the primary covariate combination for compositional modeling, and "Dx + Age + Sex + PMI" as the secondary covariate combination. The relative contribution of each covariate was assessed by Type III likelihood ratio chi-square tests (Anova(type = 3) from the car package, **Supplementary Data 16**).

As support for the above covariate selection, adding PMI did not increase the model fitness in the scomp analysis, without changes of elpd_diff/se_diff > 5. More than five of elpd_diff/se_diff change is suggested for significant model improvement according to the developers' guideline⁴².

Line 985 in the revised manuscript

Differentially expressed gene analysis

Differentially expressed genes (DEGs) were analyzed within each cell cluster in BD compared to controls using zlm function in MAST⁴⁴ (v1.28.0), a statistical package for single-cell RNA-seq DEG analysis. The model included Dx, age, sex, and cngeneson (a covariate for cell quality) as fixed effects, and sample ID as a random effect. Sample-level covariates were selected based on the prior compositional analysis and further validated by evaluating AIC values from GLMs fitted to log-transformed gene expression values [$\log(p + 1)$]. In this GLM-based model to select covariates, gene expression was aggregated at the specimen level and restricted to genes expressed in at least 10% of cells within each cluster, consistent with the downstream DEG filtering criteria. These GLMs were used solely to assess covariate impact. Among the tested covariate combinations, "Dx + Age + Sex" yielded the lowest overall AIC across gene-cluster pairs, followed by "Dx + Age + Sex + PMI", as determined by cumulative AIC rank sums (**External Data**, see **Data Availability**). These results support the use of "Dx + Age + Sex" as the primary model and "Dx + Age + Sex + PMI" as a secondary model. The relative contribution of each covariate was further quantified using Type III chi-square statistics (Anova(type = 3) from the car package), as reported in **External Data**. Following MAST guidelines, we incorporated cngeneson, a variable of scaled gene expression counts, to adjust for cell nuclei quality, where higher gene detection rates suggest better nuclei quality within the same cluster.

Supplementary Data

15.DataS15.CompositionalAnalysis.GLM.AIC.txt

Akaike information criterion (AIC) assessment for the GLM in compositional analysis.

16.DataS16.CompositionalAnalysis.GLM.Anova_ChiSq.txt

Analysis of variance (ANOVA) Chi-square assessment for the GLM in compositional analysis.

External Data (available in figshare)

BDsnRNAseq.thalamus.allclusters.logLM_Anova_CR01.model_comparison.txt.gz

AIC and ANOVA Chi-square assessments for covariate combinations in the GLM of log-normalized summed gene expression in each thalamic cluster.

BDsnRNAseq.cortex.allclusters.logLM_Anova_CR01.model_comparison.txt.gz

AIC and ANOVA Chi-square assessments for covariate combinations in the GLM of log-normalized summed gene expression in each cortical cluster.

o Defining BD risk genes

o Reliance on MAGMA-based gene level scores from GWAS has significant limitations

o What about genes implicated by rare lof alleles in exome studies?

We appreciate the reviewer's suggestion to strengthen the genetic integration. As correctly noted, interpreting GWAS results based solely on MAGMA is preliminary. The recent PGC4 study (O'Connell Nature 2025) identified 36 "credible genes" through integrative approaches, of which only SHISA9 overlapped with the genes identified in our analysis. We acknowledge that only one gene among the 36 credible genes was associated with our results.

However, given the polygenic nature of BD, it is unlikely that only these 36 genes are involved in its pathophysiology. Although no definitive method currently exists for exhaustive gene prioritization, it remains important to consider a broader set of BD-associated gene candidates and assess their overlap with differentially expressed genes (DEGs). Our original intent in using MAGMA was to provide a preliminary estimate of SNP-based genomic risk, and we now clarify this motivation in the revised manuscript.

In line with the reviewer's comment that MAGMA alone is insufficient for robust gene prioritization, we have now incorporated additional integrative approaches to complement the MAGMA findings:

- Fine-mapping using PolyFun+SuSiE/FINEMAP
- Transcriptome-wide association study (TWAS) using FUSION

We used fine-mapping to narrow down the putative BD risk genes from MAGMA analysis. TWAS data was not used to narrow down the putative BD risk genes, but only to support the findings of *CACNA1C*, because TWAS data is not comprehensive across all brain regions, lacking thalamic data. These methods were employed to refine candidate BD-associated genes. Nevertheless, we emphasize that the results from these approaches remain putative, and the identified genes should be interpreted with appropriate caution, given the inherent challenges of mapping disease-associated genes from GWAS.

Accordingly, we have removed the previously described list of "164 BD risk genes" derived from MAGMA. While we acknowledge the preliminary nature of MAGMA, we have retained its results to provide a rough estimate of SNP-based genomic risk and to address Reviewer #4's suggestion regarding the extension of MAGMA analyses to other psychiatric disorder GWAS datasets. Finally, as further recommended, we have conducted and visualized separate analyses for the thalamus and cortex to improve anatomical specificity.

We have revised the following points. In addition, throughout the manuscript, we replaced "BD risk genes" with "putative BD risk genes."

Line 223 in the revised manuscript

Upon closer inspection of probable DEGs, MAGMA-based BD genomic risk^{1,46} was most enriched in the downregulated DEGs in Tha_ExN_CALB (P = 0.00123). Additional fine-mapping by PolyFun+SuSiE/FINEMAP⁴⁷⁻⁴⁹ further supported *CACNA1C* and *SHISA9* as putative BD risk genes within the downregulated set. Especially, *SHISA9* is one of the 36 credible genes for BD in the largest GWAS to date¹. While *CACNA1C* is not listed in these 36 credible genes, *CACNA1C* has consistently ranked among the top genes by p-value in prior PGC1-4 studies^{1,2,50,51}, and was also implicated by our FUSION-based TWAS integration. In contrast, the upregulated probable DEGs in Tha_ExN_CALB exhibited weaker enrichment for MAGMA-based BD genomic risk (P = 0.0916), including no putative BD risk genes despite the greater number of upregulated genes (n = 22 for downregulated vs. 43 for upregulated, **Supplementary Table 4**).

Line 387 in the revised manuscript

We examined which cell types most expressed putative BD risk genes. Thalamic excitatory neurons, including PVT neurons, showed the highest expression of the putative BD risk genes concerning the number of expressed genes and their average expression rate (**Fig. 8b**). For example, *CACNA1C* and *SHISA9* exhibited relatively higher expression in thalamic excitatory neurons than in cortical excitatory neurons (**Fig. 8c**). However, the overall differences were modest, and both genes showed the highest expression in Tha_ExN_PVALB_1, suggesting that additional factors likely contribute to the transcriptional alterations observed in PVT neurons, as discussed later.

Line 1034 in the revised manuscript

Assessment of BD risk genes

To evaluate the enrichment of SNP-based genomic risk for BD, we first applied MAGMA⁴⁶ to roughly assess the overlap between BD GWAS signals and probable DEGs, using multi-ancestry PGC4 BD summary statistics¹. Enrichment analyses were conducted for cell clusters containing five or more probable DEGs (n = 14 clusters). To assess specificity, we extended the analysis to GWAS summary statistics from other recent Psychiatric Genomics Consortium (PGC) datasets, including schizophrenia (SCZ)³³, major depressive disorder (MDD)⁵⁹, attention-deficit/hyperactivity disorder (ADHD)⁸⁸, post-traumatic stress disorder (PTSD)⁸⁹, obsessive-compulsive disorder (OCD)⁹⁰, substance use disorder (SUD)⁹¹, eating disorder (ED)⁹², panic disorder (PD)⁹³, and autism spectrum disorder (ASD)⁹⁴ (<https://pgc.unc.edu/>).

Beyond assessing SNP-based genomic enrichment, we further prioritized specific BD risk genes. As MAGMA alone provides only preliminary gene-level association signals, we adopted a multi-step approach to define “putative BD risk genes”. First, we designated the “36 credible genes” reported in the PGC4 BD study¹ as the most reliable BD-associated genes currently available. Given the highly polygenic architecture of BD, it is unlikely that only 36 genes account for all the BD genetic basis. Additional BD risk gene candidates were defined as those overlapping between MAGMA-significant genes and those harboring fine-mapped SNPs identified by PolyFun+SuSiE/FINEMAP⁴⁷⁻⁴⁹, using the multi-ancestry PGC4 BD summary statistics¹. MAGMA identified 167 genes associated with BD at Bonferroni-corrected significance (adjusted P < 0.05, based on 17,487 genes with Entrez IDs). Fine-mapping was conducted following the developer’s pipeline, using UKBB_LD as the reference panel. Fine-mapped SNPs were annotated using promoter regions (PsychENCODE TSS±2kb; file: tss.sites.codingOnly.gencode.v19.annotation.bed), enhancer regions (PsychENCODE GRN2; file: INT-14_ElasticNet_Filtered_Cutoff_0.1_cross_assembly_GRN_2.csv), and SnpEff-4.3 for

coding consequences. We then defined “putative BD risk genes” as the union of the 36 credible genes and 75 MAGMA-based genes containing fine-mapped SNPs in promoters, enhancers, or as missense variants, resulting in a total of 107 genes (four overlapped, **Supplementary Data 17**). To explore regional and cell-type specificity, we evaluated the expression of these 107 genes across thalamic and cortical cell clusters using two metrics: (1) the number of putative BD risk genes expressed per cluster (expression rate > 0.1), and (2) the mean expression level of those genes.

To further support gene prioritization, we conducted transcriptome-wide association studies (TWAS) using FUSION⁹⁵ with precomputed weights from the GTEx v8 brain datasets (<http://gusevlab.org/projects/fusion/>). Due to the absence of thalamic samples in GTEx, the TWAS results were used only for complementary reference and were not included in the definition of putative BD risk genes. TWAS-significant genes were identified using Bonferroni correction for each precomputed weight set (adjusted $P < 0.05$, across 2,557-7,732 tested genes per dataset). Given that GTEx represents bulk RNA-seq data from multiple brain regions (not including thalamus), we interpreted these results broadly, without strict anatomical specificity.

Figure 4. Transcriptional disturbances in PVT neurons.

c) PPI networks highlighting downregulated and upregulated probable DEGs in PVT neurons. Node color indicates the direction of gene regulation (Red, down; blue, up). Rectangles indicate BD risk genes. PPI enrichment P -value $< 1.0 \times 10^{-16}$. **d)** PPI networks highlighting downregulated possible DEGs in PVT neurons. Rectangles represent BD risk genes, and colors denote the genes in top DEG-GOs. Font size indicates DEG-GO enrichment. Note that the two PPI networks are identical but have different focuses on the genes. PPI enrichment P -value $< 1.0 \times 10^{-16}$.

Supplementary Data

17.DataS17.PutativeBDriskgenes.txt

Putative bipolar disorder risk genes identified in this study and their sources.

We have also revised the relevant phrasing (from “and” to “and/or”) to reflect the putative nature of BD risk genes, acknowledging that *KCNQ3* is not included among the putative BD risk genes.

Abstract in the revised manuscript

Notably, PVT neurons showed significant downregulation of genes associated with synaptic function and ion channel regulation, including *SHISA9*, *CACNA1C*, and *KCNQ3*—genes linked to **putative** BD risk and/or implicated as critical hubs in the downregulated molecular networks.

Line 40 in the revised manuscript

In PVT neurons, *SHISA9*, *CACNA1C*, and *KCNQ3* exhibited notable changes as **putative** BD risk genes and/or critical hubs in the downregulated synaptic/neuronal gene networks.

In response to this comment, we examined the expression patterns of *HECTD2* and *AKAP11* across cell types in both the thalamus and cortex. Both genes exhibited relatively high expression in glial populations in these regions. We did not observe consistent differential expression between BD and control samples for either gene in any cell clusters in this study. This lack of differential expression may reflect distinct mechanisms underlying BD-associated genes identified through rare variants, as opposed to those implicated by common variant analyses. We have added a brief description of these findings to the Results section and included new panels in **Supplementary Fig. 8**.

Line 394 in the revised manuscript

We examined the expression patterns of BD-associated genes identified through rare variant analyses, specifically *HECTD2* and *AKAP11*^{3,60}. Neither gene was differentially expressed in any cell type clusters in the thalamus or cortex at an FDR threshold of < 0.1. The expression levels of *HECTD2* and *AKAP11* were not notably elevated in thalamic excitatory neurons, and their spatial expression patterns showed no specificity to the PVT region (**Supplementary Fig. 8a-b**). These findings may reflect distinct mechanisms underlying BD-associated genes identified through rare variants compared to those implicated by common variant analyses.

Supplementary Figure 8. *HECTD2* and *AKAP11* expression and thalamus-cortex meta-gene programs estimated by geneNMF.

a) Gene expression of *HECTD2* and *AKAP11* across the cell nuclei clusters in the thalamus-cortex integration set. **b)** Spatial gene expression of *HECTD2* and *AKAP11* in the medial thalamus.

o Developmental origin of thalamic inhibitory neurons

o Only MGE, CGE, and LGE were considered, but it is not convincing that the observed thalamic

inhibitory neurons were derived from LGE

o Existing literature points to midbrain-derived (SOX14+) inhibitory neurons

We apologize for the careless omission of discussion regarding thalamic inhibitory neurons during word reduction. As the reviewer correctly noted, thalamic inhibitory neurons are thought to originate from the midbrain and are characterized by *SOX14* and *OTX2* expression, as reported by Kim et al. Our original intention was to discuss lateral ganglionic eminence (LGE) marker expression alongside the existing literature on *SOX14* and *OTX2*, as illustrated in the original **Supplementary Fig. 1c**. However, this discussion was inadvertently removed in the course of final manuscript preparation.

Upon reflection, we recognize that we placed excessive emphasis on the LGE, which was based solely on marker expression and remains highly speculative. In response, we have renamed the cluster originally labeled as “InN_LGE” to “InN_Tha” to more appropriately reflect its characteristics. We have also revised the manuscript to include relevant arguments of *SOX14* and *OTX2* expression.

Line 91 in the revised manuscript

Inhibitory neurons were classified into three distinct classes based on their **source tissues and ganglionic eminence markers** compiled from Schmitz et al.³⁴. Thalamic inhibitory neurons expressed lateral ganglionic eminence (LGE) markers including *ZFHX4*, *FOXP2*, *MEIS2*, and *TSHZ1*³⁴, as well as midbrain-derived inhibitory markers such as *SOX14* and *OTX2*, consistent with the findings from Kim et al.³⁵ (**Supplementary Fig. 1c**). These neurons were annotated as InN_Tha without reference to their developmental origin, which, while intriguing, is beyond the scope of the present study.

Accordingly, we have revised “InN_LGE” to “InN_Tha” throughout the main text and figures (**Fig. 1, Supplementary Figs. 1 and 2**).

C

Supplementary Figure 1. Cell-type annotation of thalamic and cortical cell clusters.

c) Top: Expression of excitatory neuron (ExN) markers and selected inhibitory neuron (InN) markers within ExN clusters. Bottom: Expression of InN markers within InN clusters. NeuCyto cluster is retained as a technical reference.

o Comparison with Siletti et al. is unclear, esp. regarding sample subject variation

In responding to this comment, we realized that two of the HBCA datasets had been misassigned: the correct samples should be 10X398-5 and 10X398-6 (epithalamus), rather than 10X398-1 and 10X398-2 (hypothalamus). We sincerely apologize for this oversight and have revised the relevant figures and corresponding text as described above. Thanks to this correction, the characteristics of PVT neurons become clearer. Please refer to the above corrections for details.

Regarding sample-to-sample variation, we have provided the metadata for the selected 20 samples in the revised **Supplementary Data 11**. In the original manuscript, the left panel of **Fig. 2a** depicted the composition of PVT neurons across our dataset and the HBCA dataset; however, the color coding was unclear. Upon review, we recognized that the composition across different samples was not informative. In the revision, we instead present the thalamic

cellular composition for each sample in our dataset in **Supplementary Data 11** and visualize the case-control comparison of thalamic cellular composition in **Supplementary Figs. 3b-c**, which is more relevant to our study. As the HBCA dataset was not intended for PVT analysis, the cellular composition of PVT neurons in the HBCA samples was highly variable. For this reason, we removed the HBCA panels from the original **Fig. 2a**.

Main Figure

REMOVED: Figure 2a right panel in the original manuscript

Line 146 in the revised manuscript

We obtained 146,643 cell nuclei in 18 unique cell clusters and 2,139 droplets in two NeuCyto clusters after QC (**Fig. 3a** and **Supplementary Fig. 3a**). Two NeuCyto clusters were removed as non-nuclei from the subsequent analysis. Tha_ExN_CALB in **Fig. 3a** corresponded to PVT neurons. All major cell classes were consistently detected across the 41 samples (**Supplementary Fig. 3b**). The Tha_ExN_CALB cluster was also robustly identified in all samples among thalamic excitatory neurons with two outliers, BD13 and CT10 (**Supplementary Fig. 3c**).

Line 852 in the revised manuscript

We deconvoluted the Visium spatial transcriptomic data using cell-type expression profiles obtained from snRNA-seq via CytoSPACE³⁹ (v1.1.0) with default parameters to estimate the spatial distribution of each cell type. The reference cell types included excitatory/inhibitory neurons, astrocytes, OPCs, oligodendrocytes, and microglia, as defined in the integrated set comprising the HBCA and our dataset (see **Fig. 2a**). The low representation of the ExN_PVALB_2 cluster in **Fig. 2d** was consistent with its low/no proportion in the CT06 snRNA-seq dataset (**Supplementary Data 11**), supporting the validity of our deconvolution approach. As a robustness check, we repeated the deconvolution using either our control thalamic dataset or CT06-derived data alone, both of which yielded consistent results (**Supplementary Fig. 13e-f**).

Supplementary Figure 3. Cell-type annotation in the thalamus.

b) Proportions of major cell classes across individual thalamic samples. c) Proportions of excitatory neuron subtypes across individual thalamic samples.

Supplementary Data

11.DataS11.HBCA.profiles.txt

Sample profiles of HBCA data used in this study, including cell cluster counts.

Reviewer #4

This manuscript investigates cellular and molecular alterations in bipolar disorder (BD) through single-nucleus RNA-seq (snRNA-seq) on postmortem brain tissue from BD patients and controls, focusing on the medial thalamus and frontal cortex. The authors first identify a subpopulation of thalamic excitatory neurons that represent neurons of the paraventricular thalamic nucleus (PVT), and provide a detailed molecular and spatial characterization of this cell population. Subsequent analyses reveal changes in cell type compositions and gene expression in BD. Intriguingly, these changes are more substantial in the thalamus compared to the cortex, highlighting the importance of this region and PVT neurons in particular.

Overall, this study makes important contributions to our understanding of both the cytoarchitecture of the human PVT and alterations in BD at the cellular compositional and transcriptional levels. In addition, the analyses of the snRNA-seq data are thorough and I commend the authors for their careful assessments of QC metrics and the robustness of their results. There are several considerations that if addressed would strengthen the study.

We thank the reviewer for their thoughtful and constructive comments, as well as for recognizing the novelty and thoroughness of our study. We appreciate the positive assessment of our analyses, QC measures, and the robustness of our results. We are encouraged that the reviewer finds our work to make important contributions to understanding the human PVT and its alterations in BD.

Major comments:

(1) Dysregulation of circadian rhythms is a feature of BD. Were any associations observed between the transcriptional changes in BD and patients' time of death?

We appreciate the reviewer's comment regarding the potential influence of circadian rhythm on transcriptional states in BD. We agree that circadian rhythm dysregulation is a well-recognized feature of BD. As previously reported in schizophrenia (Seney et al., Nature Communications, 2019, 10:3355), gene expression patterns in BD may also differ from those in controls. While the possible association between transcriptional changes in BD and time of death is intriguing, the time of death in our dataset is only an estimate and is likely to be inaccurate; moreover, the discrepancy between the estimated and actual time of death cannot be determined. This issue is compounded by the fact that our samples were derived from individuals who died by suicide. Given that circadian rhythms can influence gene expression within a 3–4-hour window, reliance on imprecise estimates of time of death could be misleading, rendering any analysis of its association with transcriptional changes in BD virtually infeasible. This limitation also applies to postmortem interval (PMI), and we have now explicitly acknowledged it in the manuscript.

Line 620 in the revised manuscript

Because the exact time of death for the donors was unavailable, the PMI in our dataset was not completely accurate. Given that circadian rhythms can influence both gene expression and

disease-associated patterns⁷⁹, precise determination of the time of death would facilitate the investigation of circadian rhythm-associated gene expression changes.

(2) The gene set enrichment analyses in this study use terms from the Gene Ontology (GO) database. Given that synaptic dysfunction is implicated in BD pathology, I recommend the authors also perform these analyses using the curated gene sets from the SynGO database (Koopmans et al. 2019), which were also used in O'Connell et al. (2025).

We thank the reviewer for this valuable suggestion. We have now performed gene ontology enrichment analysis using the SynGO curated resource (<https://www.syngoportal.org/>, Koopmans et al., Neuron 2019) for differentially expressed genes (DEGs) in the relevant clusters, including PVT neurons. We used more relaxed threshold of FDR (FDR < 0.2) than conventional GO enrichment analysis (FDR < 0.05), because the number of synaptic DEGs in our DEG criteria was limited.

We found six GO terms that were enriched in the thalamic clusters and none in the cortical clusters. Among the six SynGO terms, GO:0099055 (integral component of postsynaptic membrane), GO:0099056 (integral component of presynaptic membrane), and GO:0099061 (integral component of postsynaptic density membrane) were enriched in the downregulated genes in PVT neurons. GO:1905274 (regulation of modification of postsynaptic actin cytoskeleton) was enriched in the downregulated genes in microglia. This analysis further supports our findings of synaptic dysregulation both in PVT neurons and microglia.

These results are included in a new **Supplementary Table 5** and mentioned in the Results and Discussion. In addition to SynGO analysis, we performed GSEA in response to Reviewer #1's comment.

Line 204 in the revised manuscript

Given the enrichment of synaptic terms in the DEG-GOs, we additionally performed SynGO enrichment analysis⁴⁵. The downregulated DEGs in the Tha_ExN_CALB cluster again showed the strongest SynGO enrichment, including three terms related to synaptic structure: integral component of postsynaptic membrane (GO:0099055), integral component of presynaptic membrane (GO:0099056), and integral component of postsynaptic density membrane (GO:0099061) (**Supplementary Table 5**).

Complementary GSEA based on fold changes supported the enrichment of neuronal and synaptic processes among downregulated genes in Tha_ExN_CALB (**Supplementary Data 7**). GSEA identified potassium channel activity, including *KCNQ3*, *KCND2*, and *KCNC2* as the top GO term for the downregulated genes in Tha_ExN_CALB. The overlap between DEG-GO and GSEA results (GSEA-GO) was more pronounced for downregulated than upregulated genes (9/24 vs. 6/58; two-sided Fisher's exact test, P = 0.00931, odds ratio = 5.07) (**Supplementary Table 5**). Notably, the top three DEG-GO terms among downregulated genes—**transporter complex** (GO:1990351), **monatomic ion channel complex** (GO:0034702), and **main axon** (GO:0044304)—were also identified by GSEA. The downregulated genes in Tha_ExN_CALB exhibit greater biological coherence and interpretability across enrichment methods. **Supplementary Table 6** summarizes the downregulated probable DEGs indicated by both DEG-

GO and GSEA as biologically interpretable genes for BD. Among them, *KCNQ3*, *CACNA1C*, *SHISA9*, *KCNC2*, *LRRC7*, *ROBO2*, and *CNTNAP4* were consistently enriched across all three analytical approaches (DEG-GO, GSEA, and SynGO).

Line 285 in the revised manuscript

Disturbed interaction between PVT neurons and microglia in BD

In addition to Tha_ExN_CALB (PVT neurons), we identified downregulated robust DEG-GOs within Tha_Micro_1 (microglia). Notably, these two clusters were the only ones displaying downregulated robust DEG-GOs (**Fig. 3e-f**). Furthermore, all five DEG-GOs identified within Tha_Micro_1 were related to synaptic function (**Supplementary Table 5**). When we assigned the DEGs in Tha_Micro_1 to co-expression network modules, six of ten downregulated probable DEGs (60%) and 32 of the 55 downregulated possible DEGs (58.2%) were assigned to Module-11 of microglia (Micro-M11) ($P = 8.5 \times 10^{-4}$ and 1.20×10^{-14} , $FC = 4.36$ and 4.22 , hypergeometric test) (**Supplementary Table 4**). Among the probable DEGs, *SYNDIG1*, *CYFIP1*, and *TIAM1* emerged as promising synaptic DEGs (**Fig. 5a**), each exhibiting robustness scores exceeding 0.95 and included within the five synaptic DEG-GOs. **Additional GSEA identified synaptic enrichment among the downregulated DEGs in Tha_Micro_1 (Supplementary Data 7)**, again highlighting *SYNDIG1* and *CYFIP1*. SynGO enrichment analysis suggested potential enrichment of GO:1905274 (regulation of modification of the postsynaptic actin cytoskeleton) (**Supplementary Table 5**).

Line 353 in the revised manuscript

Among cortical DEG-GO, only GO:0045211 (postsynaptic membrane), an upregulated GO term in FrC_ExN_L6ITCar3, was identified by FC-based GSEA, and no SynGO enrichment was observed. These findings suggest that cortical DEGs exhibit lower biological interpretability compared to thalamic DEGs.

Line 489 in the revised manuscript

Among all the thalamic and cortical cell types in our study, PVT neurons exhibited the highest number of DEGs in BD. Notably, downregulated DEGs in PVT neurons were more biologically interpretable than upregulated ones and included putative BD risk genes. Among the PVT neuron DEGs, *SHISA9*, *CACNA1C*, and *KCNQ3* stood out as particularly relevant to BD pathophysiology, as they were centrally positioned in the PPI network and consistently highlighted in GO, GSEA, and SynGO analyses.

Line 1025 in the revised manuscript

To get insight into synaptic genes, SynGO⁴⁵ enrichment analysis was added to the clusters with robust DEG-GOs, using enricher function in clusterProfiler with the FDR threshold of 0.2 and gene set size of 5 – 100 genes. SynGO terms with at least two genes were retained to accommodate the smaller number of target genes compared to the above DEG-GO analysis. As we set $FDR < 0.2$, the SynGO enrichment should be regarded as a potential one. We also performed FC-based GSEA separately for the downregulated and upregulated gene sets to further evaluate the robustness of the identified biological signals, using gseGO function in clusterProfiler with the enrichmentScore threshold of 0.7.

Supplementary Table 5 (abridged)

Additional SynGO analysis (FDR < 0.2)					
Region	Cluster	Direction	ONTOLOGY	GO ID	Description
Medial thalamus	Tha_ExN_CALB	Down	SynGO.CC	GO:0099055	integral component of postsynaptic membrane
Medial thalamus	Tha_ExN_CALB	Down	SynGO.CC	GO:0099056	integral component of presynaptic membrane
Medial thalamus	Tha_ExN_CALB	Down	SynGO.CC	GO:0099061	integral component of postsynaptic density membrane
Medial thalamus	Tha_Micro_1	Down	SynGO.BP	GO:1905274	regulation of modification of postsynaptic actin cytoskeleton
Medial thalamus	Tha_ExN_PVALB_1	Up	SynGO.BP	GO:0090128	regulation of synapse maturation
Medial thalamus	Tha_ExN_CALB	Up	SynGO.BP	GO:0140238	presynaptic endocytosis
Frontal cortex	none

(3) The cohort includes both BDI and BDII subtype cases. Is the dataset sufficiently powered to stratify analyses by subtype? I understand that the sample size of the BDII cases may preclude such analyses; if so, it would be worth mentioning this in the limitations section of the Discussion (together with sex differences in lines 449-451).

We agree that differences between BD subtypes may be biologically meaningful. However, our current cohort includes only five individuals with BDII, limiting the statistical power for subtype-stratified analysis. We now explicitly acknowledge this in the revised Discussion (in the limitations paragraph) and have included subtype information in **Supplementary Table 1**.

In addition to addressing the reviewers' comments, we have revised **Supplementary Table 1** to better protect individual privacy. The original version included individual-level demographic and clinical data. Because Coroner data in Canada, where the brain tissue specimens were collected, are nominal and publicly accessible, there is a theoretical risk of re-identification. In the revised manuscript, we now provide aggregated demographic and clinical information, including a summary of subtype information, instead of individual-level data.

Line 616 in the revised manuscript

While the differences between BD subtypes (BD-I and BD-II) are of interest, the limited number of BD-II cases (N = 5) in our cohort also precludes a reliable assessment of subtype-specific effects.

Supplementary Table 1 (abridged)

Supplementary Table 1. Sample information and sequencing metrics										
Sample information										
Group	Sex	Age*	PMI (h)*	pH*	Race	Cause of death	Refrigerator Delay (h)*	Subtype	Medication at time of death	Substance at time of death
BD (N = 21)	14 males (66.7%)	44.3 ± 13.9	44.1 ± 17.5	6.54 ± 0.21	20 Caucasians	Suicide (17) Accidental (3)	7.8 ± 5.0 with 3 missing values	Bipolar II disorder (5; 42.8%) Bipolar I disorder (5; 23.8%) Bipolar disorder without subtype information (7; 33.3%)	lithium (2; 9.5%) anticonvulsant (3; 14.3%) antipsychotics (5; 23.8%) antidepressant (7; 33.3%) benzodiazepines (2; 9.5%) opioid (5; 23.8%)	psychostimulant (3; 14.3%) alcohol (7; 33.3%) cannabinoid (2; 9.5%) cyanide (1; 4.8%)
	7 females (33.3%)				1 Asian	Natural (1)				
Control (N = 21)	14 males (66.7%)	52.5 ± 11.8	59.9 ± 21.7	6.37 ± 0.16	21 Caucasians	Accidental (7) Natural (12) Undetermined (2)	11.9 ± 9.2 with 0 missing values		anticonvulsant (1; 4.8%) antipsychotics (0) antidepressant (3; 14.3%) opioid (4; 19.0%)	psychostimulant (2; 9.5%) alcohol (1; 4.8%) cannabinoid (1; 4.8%) cyanide (0)
	7 females (33.3%)									
P value	1	0.0472	0.0153	0.00866	1	2.33 × 10 ⁻⁹	0.0586	.	.	.
test type (two-sided)	Fisher's exact test	Welch's t test	Welch's t test	Welch's t test	Fisher's exact test	Fisher's exact test	Welch's t test	.	.	.

* Data are presented as the mean ± s.e.m.

(4) In Fig. 8a, downregulated genes in Tha_ExN_CALB are enriched in SNP-based genomic risk of all 3 phenotypes tested. Is this observed for other psychiatric disorders (such as post-traumatic stress disorder, attention deficit-hyperactivity disorder, borderline personality disorder, and autism spectrum disorder)?

We appreciate the reviewer's suggestion. We conducted additional MAGMA enrichment analyses using publicly available GWAS summary statistics for attention-deficit/hyperactivity disorder (ADHD), post-traumatic stress disorder (PTSD), obsessive-compulsive disorder (OCD), substance use disorder (SUD), eating disorder (ED), panic disorder (PD), and autism spectrum disorder (ASD). These analyses revealed no significant enrichment of SNP-based genomic risk in the downregulated genes of PVT neurons for any of these additional disorders. In contrast, enrichment was observed for the GWAS data of bipolar disorder (BD), schizophrenia (SCZ), and major depressive disorder (MDD). While this could be partly derived from the sample size of the original GWAS, where BD, SCZ, and MDD have larger sample sizes, notable enrichment of these associations implicates a possible relevance of PVT neurons to these disorders. To illustrate this clearly, we have revised **Fig. 8**. The full association data are described in **Supplementary Data 9**.

Unfortunately, GWAS summary statistics for borderline personality disorder (BPD) are not yet publicly available. According to the preprint by Streit et al. (medRxiv, 2024, <https://www.medrxiv.org/content/10.1101/2024.11.12.24316957v1>), the summary statistics are expected to be released upon official publication. We will incorporate the BPD analysis once these data become available.

Line 379 in the revised manuscript

This enrichment pattern extended beyond BD, as downregulated DEGs in PVT neurons also showed the highest enrichment for MAGMA-based genomic risk associated with schizophrenia³³ and major depressive disorder⁵⁹, while showing weaker or nonsignificant enrichment for other psychiatric conditions: attention-deficit/hyperactivity disorder, post-traumatic stress disorder, obsessive-compulsive disorder, substance use disorder, eating disorder, panic disorder, and autism spectrum disorder (**Fig. 8a**, details in **Supplementary Data 9**). These enrichment comparisons should be interpreted with caution, as differences in statistical power across the original GWAS datasets may influence the observed patterns.

Line 1038 in the revised manuscript

To assess specificity, we extended the analysis to GWAS summary statistics from other recent Psychiatric Genomics Consortium (PGC) datasets, including schizophrenia (SCZ)³³, major depressive disorder (MDD)⁵⁹, attention-deficit/hyperactivity disorder (ADHD)⁸⁸, post-traumatic stress disorder (PTSD)⁸⁹, obsessive-compulsive disorder (OCD)⁹⁰, substance use disorder (SUD)⁹¹, eating disorder (ED)⁹², panic disorder (PD)⁹³, and autism spectrum disorder (ASD)⁹⁴ (<https://pgc.unc.edu/>).

Figure 8. Comparative analysis between the thalamus and cortex.

a) Left: MAGMA-based genomic risk enrichment in DEGs for BD, schizophrenia (SCZ), and major depressive disorder (MDD). **Right:** MAGMA-based genomic risk enrichment in downregulated DEGs in PVT neurons for BD, SCZ, MDD, attention-deficit/hyperactivity disorder (ADHD), post-traumatic stress disorder (PTSD), substance use disorder (SUD), obsessive-compulsive disorder (OCD), eating disorder (ED), panic disorder (PD), and autism spectrum disorder (ASD).

Supplementary Data

09.DataS09.MAGMA_PsychiatricDisorders.txt

MAGMA results for selected clusters and 10 psychiatric disorders.

(5) In addition to the analyses of common variant-based BD genomic risk, it would also be helpful to comment on how these findings relate to the rare loss-of-function variants recently described in Thorgeirsson et al. (2025). For example, which cell types in thalamus and cortex express HECTD2 and AKAP11, and how does their expression in BD patients compare to controls?

We thank the reviewer for raising this important point. We examined the expression patterns of *HECTD2* and *AKAP11* across cell types in both the thalamus and cortex, as these two genes were associated with BD in Thorgeirsson et al. (Nature Genetics 2025, 57:851-855). Both genes exhibited relatively high expression in glial populations in these regions. We did not observe consistent differential expression between BD and control samples for either gene in any cell

clusters in this study. This lack of differential expression may reflect distinct mechanisms underlying BD-associated genes identified through rare variants, as opposed to those implicated by common variant analyses. We have added a brief description of these findings to the Results section and included new panels in **Supplementary Fig. 8**.

Line 394 in the revised manuscript

We examined the expression patterns of BD-associated genes identified through rare variant analyses, specifically *HECTD2* and *AKAP11*^{3,60}. Neither gene was differentially expressed in any cell type clusters in the thalamus or cortex at an FDR threshold of < 0.1. The expression levels of *HECTD2* and *AKAP11* were not notably elevated in thalamic excitatory neurons, and their spatial expression patterns showed no specificity to the PVT region (**Supplementary Fig. 8a-b**). These findings may reflect distinct mechanisms underlying BD-associated genes identified through rare variants compared to those implicated by common variant analyses.

Supplementary Figure 8. *HECTD2* and *AKAP11* expression and thalamus-cortex meta-gene programs estimated by geneNMF.

a) Gene expression of *HECTD2* and *AKAP11* across the cell nuclei clusters in the thalamus-cortex integration set. **b)** Spatial gene expression of *HECTD2* and *AKAP11* in the medial thalamus.

(6) While the data that support potential disruptions in PVT-microglia trans-cellular interactions in BD are compelling (MP7, Figs. 5b-c), it was unclear to me why the authors focused on SHISA9 in PVT neurons as a representative MP7 gene throughout the manuscript. Although SHISA9 is part of the core PPI network of BD-associated downregulated genes in PVT neurons (Fig. 5d), it appears to be expressed in only in a small fraction of Tha_ExN_CALB (PVT) neurons, while is expressed in most Tha_ExN_PVALB_1 neurons (Figs. 5d and 3a). I suggest that the authors either characterize this SHISA9-positive subpopulation of PVT neurons further (for example, are these detected in all thalamic samples?), or reduce/remove references to SHISA9 in PVT neurons throughout the text. These include parts of the section on "Disturbed interaction between PVT neurons and microglia in BD" (lines 251-284), the Discussion (lines 415-417), as well as the following sentence in the abstract: "This interaction was further exemplified by the concurrent downregulation of SYNDIG1 in microglia and SHISA9 in PVT neurons, both of which are involved in AMPA receptor signaling pathways" (top of pg. 3, line numbers not available).

We thank the reviewer for this careful observation and agree with the recommendation to remove certain overstatements. The previously described “interaction between *SHISA9* and *SYNDIG1*” represented an over-interpretation and has therefore been removed. Instead, we now describe *SHISA9* and *SYNDIG1* more conservatively, noting their roles as auxiliary subunits of AMPA receptors (Kalashnikova et al. Neuron 2010, 65:80-93; Khodosevich et al. Neuron 2014, 83:601-615).

Our focus on *SHISA9* among the downregulated genes in PVT neurons is primarily motivated by its identification as one of the 36 credible BD-associated genes reported in O’Connell et al., Nature (2025). In addition, we noticed that *SYNDIG1*, the top-ranked downregulated gene in microglia, shares the property of encoding auxiliary subunits of the AMPAR complex with *SHISA9*, in parallel with probable interactions between microglia and thalamic excitatory neurons, as estimated by GeneNMF approach. The biological interpretability of *SHISA9* and *SYNDIG1* was further reinforced by GSEA (as suggested by Reviewer #1) and SynGO analysis (as suggested by Reviewer #4). The DEGs as convergence of DEG-GO and GSEA are illustrated in a new **Supplementary Table 6**, which includes *SHISA9* and *SYNDIG1*. In this revisional process, we have also noticed that *TIAM1*, one of downregulated genes in microglia as biologically interpretable genes in **Supplementary Table 6**, might have roles in regulating synaptic AMPAR function (Rao et al. J Neurosci 2019, 39:9306-9315).

We have also revised the visualization of *SHISA9* expression. In the original version of **Fig. 5d**, *SHISA9* and *SYNDIG1* were displayed together. Due to limitations in Seurat’s default color scaling, we were unable to independently adjust the color range for each gene, which led to a misleading color annotation for *SHISA9*. In the revised manuscript, *SHISA9* and *SYNDIG1* are presented in separate panels (**Fig. 5d**). As shown in the updated figure, *SHISA9* was broadly expressed in Tha_ExN_CALB (PVT neurons), while the strongest expression was observed in Tha_ExN_PVALB_1. Additionally, we included detailed expression profiles of *SHISA9*, alongside *CACNA1C*, to highlight that *SHISA9* is highly expressed in thalamic excitatory neurons relative to cortical excitatory populations in **Fig. 8c**. This figure further illustrates high expression of *SHISA9* in Tha_ExN_CALB (PVT neurons).

We selected *SHISA9* and *SYNDIG1* for the reasons outlined above. Although the meta-gene program encompassing thalamic excitatory neurons and microglia requires further experimental validation, the concurrent downregulation of several genes involved in AMPAR regulation represents a noteworthy finding. In the original manuscript, the visualization of *SHISA9* expression in PVT neurons was misleading; however, *SHISA9* is in fact highly expressed in these neurons. We apologize for this earlier misrepresentation.

Abstract in the original manuscript

~~REMOVED: This interaction was further exemplified by the concurrent downregulation of *SYNDIG1* in microglia and *SHISA9* in PVT neurons, both of which are involved in AMPA receptor signaling pathways.~~

Line 279 in the original manuscript

~~REMOVED: The coordinated downregulation of *SYNDIG1* in microglia and *SHISA9* in excitatory neurons exemplifies a probable disturbance in microglia-neuron interactions for synapses in~~

~~the BD thalamus, indicating a plausible pathological mechanism of *SHISA9*, one of the 36 putative BD-causal genes[‡].~~

Line 415 in the original manuscript

~~REMOVED: The downregulation of *SYNDIG1* in microglia and *SHISA9* in PVT neurons might be an example of dysregulated molecular interactions within the Excit-Micro meta-gene programs associated with BD.~~

Line 219 in the revised manuscript

Supplementary Table 6 summarizes the downregulated probable DEGs indicated by both DEG-GO and GSEA as biologically interpretable genes for BD. Among the downregulated probable DEGs, *KCNQ3*, *CACNA1C*, *SHISA9*, *KCNC2*, *LRR7*, *ROBO2*, and *CNTNAP4* were consistently enriched across all three analytical approaches: DEG-GO, SynGO, and GSEA.

Line 311 in the revised manuscript

Supplementary Table 6 summarizes the downregulated genes in Tha_ExN_CALB and Tha_Micro_1 that were enriched in one or more GO terms in both DEG-GO and GSEA analyses. Notably, these genes belonged to the same co-expression modules in each cluster.

Line 314 in the revised manuscript

At the gene level, *SYNDIG1* and *SHISA9* encode auxiliary subunits for AMPA receptor (AMPA) complex^{56,57}, with both genes' robustness scores exceeding 0.7 (**Supplementary Table 4**). Notably, *SYNDIG1* was prominently expressed in microglia, while *SHISA9* was primarily expressed in excitatory neurons (**Fig. 5d**). *SHISA9* consisted of core PPI networks of BD-associated downregulated genes in PVT neurons (**Fig. 4d**) and overlapped in the downregulated genes in Tha_ExN_PVALB_1, as one of the 36 putative BD-causal genes¹. Beyond microglia and excitatory neurons, thalamic T cells were included in this meta-gene program. **Fig. 5e** illustrates potential ligand-receptor interactions between neurons and microglia/T cells, as analyzed through LIANA⁵⁸. Although we did not detect concurrently downregulated DEGs as ligand-receptor pairs, *KCND2* in Tha_ExN_CALB, *LAMB1* in Tha_ExN_PVALB_2, and *KCNQ1* in Tha_Micro_1 were identified as downregulated DEGs among the LIANA-based ligand-receptor pairs. These genes encode ion channels or extracellular matrix proteins that may potentially contribute to Excit-Micro interactions. However, as this analysis is an inference, the precise ligand-receptor mechanisms remain to be elucidated in future studies.

Line 556 in the revised manuscript

Beyond cell-type-based analysis, we identified trans-cellular meta-gene programs between excitatory neurons and microglia (Excit-Micro meta-gene programs), likely reflecting physiological synaptic interactions between these cell types. Microglia are known to modulate synaptic plasticity and neuronal circuitry through dynamic interactions with neurons⁷⁶. Notably, such meta-gene programs in the thalamus are disrupted in BD, further characterized by a significant decrease in synaptic gene expression in both PVT neurons and microglia. Ling et al. reported downregulation of the synaptic neuron and astrocyte program in schizophrenia¹⁰; our findings highlight downregulation of the synaptic genes in the thalamic neuron and microglia in BD. The concurrent downregulation of *SYNDIG1* in microglia and

SHISA9 in excitatory neurons suggests disruptions of AMPAR signaling, as both genes encode auxiliary subunits of the AMPAR complex^{56,57}. *TIAM1*, another downregulated gene in microglia, has also been implicated in regulating synaptic AMPAR function⁷⁷. The preferential localization of *SYNDIG1*-positive microglia in cortical gray matter is consistent with this interpretation. Nonetheless, these meta-gene programs remain data-driven constructs, and the roles of *SYNDIG1* in microglia—as well as their mechanistic links to *SHISA9* in PVT neurons—are currently unknown. Future studies employing animal and in vitro cellular models will be crucial for elucidating these mechanisms.

Figure 5. Disturbed interaction between microglia and PVT neurons in the thalamus.
d) UMAP projection illustrating the expression patterns of *SYNDIG1* and *SHISA9*.

Figure 8. Comparative analysis between the thalamus and cortex.

c) Relative expression of *CACNA1C* and *SHISA9* across all clusters in the thalamus-cortex integration dataset.

Supplementary Table 6. Downregulated genes in Tha_ExN_CALB and Tha_Micro_1 with biological interpretations

Supplementary Table 6. Downregulated genes in Tha_ExN_CALB and Tha_Micro_1 with biological interpretations						
Cluster	Gene	DEG-GO term	GSEA term	Robustness	Fold-change	WGCNA
Tha_ExN_CALB	KCNQ3	transporter complex, monoatomic ion channel complex, main axon	potassium channel activity, voltage-gated potassium channel activity, main axon	1.00	0.80	Excit-M4
Tha_ExN_CALB	ROBO1	positive regulation of cell projection organization, regulation of neuron projection development	cardiac ventricle development	1.00	0.82	Excit-M4
Tha_ExN_CALB	PDE10A	second-messenger-mediated signaling	second-messenger-mediated signaling	1.00	0.83	Excit-M4
Tha_ExN_CALB	LRRC7	main axon, positive regulation of cell projection organization, synaptic membrane	main axon, synaptic membrane, postsynaptic membrane	0.98	0.79	Excit-M4
Tha_ExN_CALB	CACNA1C*	transporter complex, monoatomic ion channel complex, regulation of monoatomic ion transmembrane transport	regulation of muscle contraction, voltage-gated monoatomic ion channel activity, voltage-gated channel activity	0.76	0.84	Excit-M4
Tha_ExN_CALB	SHISA9*	transporter complex, monoatomic ion channel complex, modulation of chemical synaptic transmission	synaptic membrane, postsynaptic membrane, postsynaptic specialization membrane	0.73	0.85	Excit-M4
Tha_ExN_CALB	PTPRJ	second-messenger-mediated signaling, cell junction assembly, leading edge membrane	second-messenger-mediated signaling, beta-catenin binding, calcium-mediated signaling	0.73	0.85	Excit-M4
Tha_ExN_CALB	PDE4D	transporter complex, monoatomic ion channel complex, regulation of monoatomic ion transmembrane transport	regulation of muscle contraction, second-messenger-mediated signaling, regulation of blood circulation	0.73	0.85	Excit-M4
Tha_ExN_CALB	CUX2	modulation of chemical synaptic transmission, regulation of trans-synaptic signaling, positive regulation of cell projection organization	modulation of excitatory postsynaptic potential, regulation of postsynaptic membrane potential, excitatory postsynaptic potential	0.73	0.85	Excit-M4
Tha_ExN_CALB	KCNC2	transporter complex, monoatomic ion channel complex, main axon	potassium channel activity, voltage-gated potassium channel activity, regulation of potassium ion transmembrane transport	0.67	0.82	Excit-M4
Tha_ExN_CALB	CNTN4	modulation of chemical synaptic transmission, regulation of trans-synaptic signaling, cell-cell adhesion mediator activity	cell adhesion mediator activity	0.64	0.86	Excit-M4
Tha_ExN_CALB	CNTN5	synaptic membrane, cell junction assembly, cell-cell adhesion mediator activity	cell adhesion mediator activity, synaptic membrane, sensory perception of sound	0.55	0.87	Excit-M4
Tha_ExN_CALB	AUTS2	positive regulation of cell projection organization, axonogenesis	neuron migration	0.55	0.87	Excit-M4
Tha_Micro_1	SYNDIG1	dendritic spine, neuron spine, regulation of synapse organization	postsynaptic membrane	1.00	0.70	Micro-M11
Tha_Micro_1	CYFIP1	dendritic spine, neuron spine, regulation of synapse organization	regulation of axon extension	1.00	0.79	Micro-M11
Tha_Micro_1	TIAM1	dendritic spine, neuron spine, regulation of synapse organization	activation of GTPase activity, regulation of postsynapse organization, positive regulation of chemotaxis	0.95	0.78	Micro-M11
Tha_Micro_1	ABR	dendritic spine, neuron spine	activation of GTPase activity	0.59	0.82	Micro-M11
Tha_Micro_1	DIP2B	regulation of nervous system development	regulation of axon extension	0.55	0.87	Micro-M11
Downregulated probable DEGs in Tha_ExN_CALB or Tha_Micro_1 enriched in one or more GO terms in both DEG-GO and GSEA analyses						
* Putative BD risk genes						

Minor comments:

(1) Please provide details on the cohort in the Methods, including information on how individuals were selected for inclusion and the approval of the study protocol. Details are in the Reporting Summary, but must also be provided in the manuscript.

We have revised the Methods section to include additional information including cohort selection, inclusion criteria, and ethics approval.

Line 634 in the revised manuscript

Post-mortem brain samples

Postmortem brain specimens were sourced from the Douglas–Bell Canada Brain Bank (www.douglasbrainbank.ca). Frozen samples were obtained from 21 BD patients and 21 matched controls (**Supplementary Table 1**), with written informed consent provided by the next of kin. Due to the limited availability of postmortem thalamic tissue, no formal sample size calculation was performed; instead, we included all eligible samples available at the time of collection, ensuring sex matching between BD cases and controls. Each group comprised 14 males and 7 females prior to quality control. The average age at death was significantly higher in the control group than in the BD group (52.5 vs. 44.3 years; two-sided t-test, $P = 0.0472$), reflecting our prioritization of paraventricular thalamic region availability and sex matching over age matching. The cause of death was significantly skewed toward suicide in the BD group (two-sided Fisher's exact test, $P = 2.33 \times 10^{-8}$), consistent with the elevated suicide risk in BD

and the characteristics of this brain bank. Of the 42 donors, 41 were Caucasian and one Asian. Donors exhibited substantial heterogeneity in substances present at death, precluding stratification into subgroups of more than two individuals. Summary statistics of substances present at death are presented in **Supplementary Fig. 10a**.

Line 670 in the revised manuscript

This research was conducted in accordance with the Helsinki Declaration and received approval from the Research Ethics Committee of the Faculty of Medicine, Juntendo University (approval ID: M19-0278).

(2) The final two paragraphs of the Discussion appear to be redundant.

Thank you for pointing this out. We have removed the final redundant paragraph of the Discussion and streamlined the concluding section to avoid repetition.

Line 452 in the original manuscript

~~REMOVED: Despite the limitations of this study, our findings suggest that disturbances in PVT neurons constitute a central pathology in BD. Neuroimaging techniques targeting the PVT can enhance clinical assessments of BD. The presence of PVT-like neurons in thalamic organoids highlights the feasibility of modeling PVT neurons from patient-derived peripheral samples. This approach could facilitate the discovery of pharmacological interventions that target the molecular mechanisms underlying BD. The integration of multi-modal methodologies, including cellular, molecular, and neuroimaging techniques, will offer a more comprehensive understanding of the pathological role of PVT neurons in BD. PVT neurons emerge as a promising target for the development of novel diagnostic and therapeutic strategies for patients with BD.~~

For all the reviewers and editors

We have revised **Supplementary Table 1** to better protect individual privacy. The original version included individual-level demographic and clinical data. Because Coroner data in Canada, where the brain tissue specimens were collected, are nominal and publicly accessible, there is a theoretical risk of re-identification. In the revised manuscript, we now provide aggregated demographic and clinical information, including a summary of medication exposure, instead of individual-level data.

Supplementary Table 1 (abridged)

Supplementary Table 1. Sample information and sequencing metrics										
Sample information										
Group	Sex	Age*	PMI (h)*	pH*	Race	Cause of death	Refrigerator Delay (h)*	Subtype	Medication at time of death	Substance at time of death
BD (N = 21)	14 males (66.7%)	44.3 ± 13.9	44.1 ± 17.5	6.54 ± 0.21	20 Caucasians 1 Asian	Suicide (17) Accidental (3) Natural (1)	7.8 ± 5.0 with 3 missing values	Bipolar II disorder (9; 42.8%) Bipolar I disorder (5; 23.8%) Bipolar disorder without subtype information (7; 33.3%)	lithium (2; 9.5%) anticonvulsant (3; 14.3%) antipsychotics (5; 23.8%) antidepressant (7; 33.3%) benzodiazepines (2; 9.5%) opioid (5; 23.8%)	psychostimulant (3; 14.3%) alcohol (7; 33.3%) cannabinoid (2; 9.5%) cyanide (1; 4.8%)
	7 females (33.3%)									
Control (N = 21)	14 males (66.7%) 7 females (33.3%)	52.5 ± 11.8	59.9 ± 21.7	6.37 ± 0.16	21 Caucasians	Accidental (7) Natural (12) Undetermined (2)	11.9 ± 9.2 with 0 missing values	.	anticonvulsant (1; 4.8%) antipsychotics (0) antidepressant (3; 14.3%) opioid (4; 19.0%)	psychostimulant (2; 9.5%) alcohol (1; 4.8%) cannabinoid (1; 4.8%) cyanide (0)
P value	1	0.0472	0.0153	0.00866	1	2.33 × 10 ⁻⁶	0.0986	.	.	.
test type (two-sided)	Fisher exact test	Welch's t test	Welch's t test	Welch's t test	Fisher's exact test	Fisher's exact test	Welch's t test	.	.	.

* Data are presented as the mean ± s.e.m.

In responding to Reviewer #1's comment, we recognized that the original legend for **Fig. 1d** was insufficient, and we have revised it to improve clarity.

Figure 1. Characterization of thalamic excitatory neurons.
d) Dot plot displaying **selected neuronal** markers across 20 neuronal cell clusters (see **Supplementary Fig. 1** for details). Dot size represents the proportion of cells expressing the marker, and color intensity indicates average expression. **Colored bars indicate major cell classes as in panel c.**

We have also revised the following points according to Nature Communications style

- Revising “hour” to “h” as SI style description throughout the main text.
- Add company information such as “Illumina, San Diego, USA” and “10x Genomics, Pleasanton, USA”.

Point-by-point response

We sincerely thank the reviewers for their constructive and encouraging comments. Below, we provide our point-by-point responses to each reviewer's feedback. The reviewers' comments are presented in *blue italics*. Revisions to the manuscript are highlighted in red (including editorial changes requested by the journal). Rectangles in the annotated PDF indicate the specific locations of the revisions.

Reviewer #1:

All my comments have been thoroughly addressed, the authors have done an excellent job during the revision, and I look forward to seeing this manuscript published.

We sincerely thank the reviewer for the thoughtful evaluation and encouraging comments. We are grateful for the insightful feedback provided throughout the review process, which has greatly improved the quality and clarity of our manuscript. We appreciate the reviewer's support and are pleased to hear that the revised manuscript meets expectations.

Reviewer #2:

The authors have done an impressive job in exhaustively addressing the reviewers comments and revising their manuscript accordingly. While the work is now substantially improved, a few minor concerns remain:

We thank the reviewer for the thoughtful and constructive comments. We have carefully revised the manuscript to address all remaining concerns. Below we provide point-by-point responses, with corresponding textual changes indicated.

- 1. Authors have provided more information on the dissection and integration with mouse PVT. Suggest authors avoid referring to CALB1 and CALB2 as “markers for PVT” since particularly CALB1 is not very specific. More heterogeneity may exist within human PVT than these markers suggest.*

We agree with the reviewer that *CALB1* and *CALB2* are not specific markers for the PVT and that greater cellular heterogeneity is likely present. We have revised the wording to avoid describing them as “markers for PVT” and instead refer to them as follows.

Line 102 in the revised manuscript.

Cytoarchitecture of PVT neurons

Among the thalamic excitatory neurons, ExN_Tha_CALB cells in **Fig. 1b** were putatively supposed to be PVT neurons, as characterized by the expression of *CALB1* and *CALB2*. **We used these genes as guidance for locating PVT neuron clusters because they have been employed in previous studies as indicators of PVT neurons in both humans³⁶ and rodents^{37,38}. However, note that *CALB1* and *CALB2* expressions are not exclusive to PVT neurons, and additional heterogeneity likely exists within the human PVT beyond what can be defined by these genes.** These putative PVT neurons were also robustly identified in the Human Brain Cell Atlas (HBCA) dataset⁶, as demonstrated by the integration of our and HBCA datasets (**Fig. 2a**). ExN_CALB in **Fig. 2a** corresponded to ExN_Tha_CALB in **Fig. 1b**. The putative PVT neurons (ExN_CALB cells) were detected from all 20 our samples and 20 HBCA samples (see **Methods**). This integrated dataset was used for subsequent analyses to enhance generalizability.

- 2. The cell type proportions are now more comprehensively assessed, including removal of outlier samples. Suggest the authors discuss whether neuronal loss extends beyond PVT; the scCODA results seem more consistent with a global reduction across thalamic excitatory neurons. Could the absence of PVALB_2 in CT06 be due to dissection variability?*

We appreciate this insightful comment. We have expanded the Discussion to address the possibility that neuronal reductions may extend beyond the PVT, as suggested by the broader decrease across thalamic excitatory neuron clusters in our sccomp and scCODA analysis. We also note that the absence of the PVALB_2 cluster in sample CT06 may reflect dissection variability rather than a biological phenomenon in the main text.

Line 487 in the revised manuscript.

The observed reduction in thalamic excitatory neurons, including PVT neurons, may be associated with the reduced thalamus volume reported in BD¹⁹, although our analysis was limited to the medial thalamus. **Our scomp and scCODA analysis consistently suggested a widespread reduction across thalamic excitatory neuron subtypes. This suggested a global reduction across thalamic excitatory neurons beyond PVT neurons.** Given that variability in tissue dissection represents a potential confounder, future studies encompassing the entire thalamus will be required to determine whether this reduction is specific to the medial thalamus or extends to the whole structure. **It also remains an open question** whether the present findings can account for the volumetric reductions observed in BD.

Line 865 in the revised manuscript.

The absence of the ExN_PVALB_2 cluster in CT06 is likely due to dissection-related variability rather than true biological absence. In fact, ExN_PVALB_2 clusters exceeded 1% of total cells in only 11 of 20 HBCA samples and 12 of 20 of our samples. Given their inconsistent representation and presumed off-target origin, we considered ExN_PVALB_2 clusters as off-target cell populations in our analysis.

3. While authors have addressed the previous use of anthropomorphic and superlative language, they still tend to read too much into individual differentially expressed genes. It is appropriate to point out potential relevance, but the proposed mechanistic speculations about individual DEGs appear to be exaggerated. For example, SYNDIG1's role in microglia and SHISA9 interaction is still overinterpreted; the authors should either cut or tone such statements down further and not refer to SYNDIG1 as an AMPA receptor auxiliary subunit since it was shown to be a more general regulator of excitatory synapse number (e.g., 10.1371/journal.pone.0066171); SHISA9 does seem to be an AMPA receptor auxiliary subunit in neurons (e.g., 10.1016/j.neuron.2014.07.004), but it is not clear that SHISA9 interacts with SYNDIG1 beyond regulation of excitatory synapse number.

We thank the reviewer for this correction and have cut or toned down speculative statements. We no longer describe SYNDIG1 as an AMPA receptor auxiliary subunit, and we avoid implying direct interaction between SYNDIG1 and SHISA9.

Line 556 in the original manuscript.

DELETED: Beyond cell-type-based analysis, we identified trans-cellular meta-gene programs between excitatory neurons and microglia (Excit-Micro meta-gene programs), likely reflecting physiological synaptic interactions between these cell types. Microglia are known to modulate synaptic plasticity and neuronal circuitry through dynamic interactions with neurons⁷⁶. Notably, such meta-gene programs in the thalamus are disrupted in BD, further characterized by a significant decrease in synaptic gene expression in both PVT neurons and microglia. Ling et al. reported downregulation of the synaptic neuron and astrocyte program in schizophrenia¹⁰; our findings highlight downregulation of the synaptic genes in the thalamic neuron and microglia in BD. ~~The concurrent downregulation of SYNDIG1 in microglia and~~

~~SHISA9 in excitatory neurons suggests disruptions of AMPAR signaling, as both genes encode auxiliary subunits of the AMPAR complex^{56,57}. TIAM1, another downregulated gene in microglia, has also been implicated in regulating synaptic AMPAR function⁷⁷. The preferential localization of SYNDIG1 positive microglia in cortical gray matter is consistent with this interpretation.~~ Nonetheless, these meta-gene programs remain data-driven constructs, ~~and the roles of SYNDIG1 in microglia as well as their mechanistic links to SHISA9 in PVT neurons are currently unknown.~~ Future studies employing animal and in vitro cellular models will be crucial for elucidating these mechanisms

Line 563 in the revised manuscript.

Beyond cell-type-based analysis, we identified trans-cellular meta-gene programs between excitatory neurons and microglia (Excit-Micro meta-gene programs), likely reflecting physiological synaptic interactions between these cell types. Microglia are known to modulate synaptic plasticity and neuronal circuitry through dynamic interactions with neurons⁷⁶. Notably, such meta-gene programs in the thalamus are disrupted in BD, further characterized by a significant decrease in synaptic gene expression in both PVT neurons and microglia. Ling et al. reported downregulation of the synaptic neuron and astrocyte program in schizophrenia¹⁰; our findings highlight downregulation of the synaptic genes in the thalamic neuron and microglia in BD. Nonetheless, these meta-gene programs remain data-driven constructs. Future studies employing animal and in vitro cellular models will be crucial for elucidating these **potential interactions**.

Line 317 in the revised manuscript.

At the gene level, **SYNDIG1 regulates excitatory synapse number** and **SHISA9** encode a **regulator of** AMPA receptor (AMPA) complex^{56 57}, with both genes' robustness scores exceeding 0.7 (**Supplementary Table 4**).

4. Authors have addressed the error concerning inclusion of non-thalamic samples. While the authors refer to HBCA data as consisting of 20 samples, it should be made clear that those samples were derived from only 2-3 individual donors.

We appreciate this important clarification and have revised the Methods and Results to state explicitly that the 20 samples correspond to 2-3 donors per region.

Line 734 in the revised manuscript.

Human Brain Cell Atlas (HBCA) dataset

We analyzed snRNA-seq data from 20 thalamic samples in the HBCA dataset⁶ to enhance the generalizability of our findings. The sample size of 20 was selected to match the size of our in-house control thalamic snRNA-seq dataset. The original sample IDs were 10X190_3, 10X190_4, 10X193_7, 10X193_8, 10X348_3, 10X348_4, 10X354_5, 10X354_6, 10X361_3, 10X361_4, 10X362_1, 10X362_2, 10X375_3, 10X375_4, 10X377_1, 10X377_2, 10X389-5, 10X389-6, 10X393_1, and 10X393_2, referred to as HBCA01-20 in our study (**Supplementary Data 11**). These 20 samples were selected based on their potential inclusion of PVT regions. Note that **the HBCA dataset was derived from only 2-3 individual donors, and** the proportions of PVT

neurons varied across these samples, because these samples were not specifically dissected for PVT analysis but were prepared to investigate regional differences, as described in the original study⁶.

Line 856 in the revised manuscript.

Deconvolution of spatial transcriptomic data with snRNA-seq cell clusters

We deconvoluted the Visium spatial transcriptomic data using cell-type expression profiles obtained from snRNA-seq via CytoSPACE³⁹ (v1.1.0) with default parameters to estimate the spatial distribution of each cell type. The reference cell types included excitatory/inhibitory neurons, astrocytes, OPCs, oligodendrocytes, and microglia, as defined in the integrated set comprising the HBCA and our dataset (see **Fig. 2a**). The low representation of the ExN_PVALB_2 cluster in **Fig. 2d** was consistent with its low/no proportion in the CT06 snRNA-seq dataset (**Supplementary Data 11**), supporting the validity of our deconvolution approach. As a robustness check, we repeated the deconvolution using either our control thalamic dataset or CT06-derived data alone, both of which yielded consistent results (**Supplementary Fig. 13e-f**). **The absence of the ExN_PVALB_2 cluster in CT06 is likely due to dissection-related variability rather than true biological absence. In fact, ExN_PVALB_2 clusters exceeded 1% of total cells in only 11 of 20 HBCA samples and 12 of 20 of our samples. Given their inconsistent representation and presumed off-target origin, we considered ExN_PVALB_2 clusters as off-target cell populations in our analysis.**

Reviewer #4:

Nishioka and colleagues have prepared a thoughtful revision, incorporating additional analyses and experimental validation of their PVT neuron findings by RNAscope, along with a more nuanced interpretation of their results. The authors have thoroughly addressed my comments on their initial submission. I have no further comments.

We sincerely thank the reviewer for the careful evaluation of our revised manuscript and for the positive feedback. We appreciate the reviewer's time and thoughtful input, which contributed to improving the quality of our work.